# REACTION GRAPH: TOWARD MODELING CHEMICAL REACTIONS WITH 3D MOLECULAR STRUCTURES

## ABSTRACT

Accurately modeling chemical reactions using Artificial Intelligence (AI) can accelerate discovery and development, especially in fields like drug design and material science. Although AI has made remarkable advancements in single molecule recognition, such as predicting molecular properties, the study of interactions between molecules, particularly chemical reactions, has been relatively overlooked. In this paper, we introduce Reaction Graph (RG), a unified graph representation that encapsulates the 3D molecular structures within chemical reactions. RG integrates the molecular graphs of reactants and products into a cohesive framework, effectively capturing the interatomic relationships pertinent to the reaction process. Additionally, it incorporates the 3D structure information of molecules in a simple yet effective manner. We conduct experiments on a range of tasks, including chemical reaction classification, condition prediction, and yield prediction. RG achieves the highest accuracy across six datasets, demonstrating the effectiveness of the proposed method. The code will be publicly available.

## 1 INTRODUCTION

In recent years, data-driven Artificial Intelligence (AI) methods have made significant strides in chemistry (De Almeida et al., 2019), bioinformatics (Senior et al., 2020; Jumper et al., 2021; Abramson et al., 2024), pharmaceutical (Wang et al., 2023a; Mak et al., 2023), and materials science (Butler et al., 2018), considerably enhancing research efficiency and accuracy, reducing costs and accelerating discovery cycles. In the field of chemistry, AI enables precise predictions of molecular behavior (Batzner et al., 2022; Batatia et al., 2022) and reaction outcomes (Coley et al., 2017), improves the analysis of retrosynthesis (Dong et al., 2022), and streamlines synthetic pathways (Segler et al., 2018). However, most related methods primarily concentrate on recognizing and understanding single molecules, such as predicting their properties or functions (Yang et al., 2019; Zhou et al., 2023). The study of interactions between molecules, particularly chemical reactions, has not garnered as much attention.

Learning accurate representation of chemical reactions is essential for reaction recognition and understanding, benefiting various tasks such as predicting reaction conditions (Gao et al., 2018; Wang et al., 2023b), types (Schwaller et al., 2021a; Lu & Zhang, 2022), and yields (Schwaller et al., 2021b; Yin et al., 2024). As shown in Fig. 1, early works typically employed bit vector representations of reactions, i.e., fingerprints, to predict relevant reaction properties (Gao et al., 2018). With the advent of the Transformer in natural language processing, the string-based Simplified Molecular Input Line Entry System (SMILES) has gained widespread popularity (Wang et al., 2023b; Yin et al., 2024).

Recently, molecular graphs have proven inherently advantageous for various chemical tasks (Fang et al., 2022; Zhou et al., 2023). However, as shown in Fig. 1, most graph-based methods first employ single-molecule modeling to extract individual molecule-level representations for reactants and products, and then combine these representations to form an ensemble reaction representation for the prediction (Kwon et al., 2022a;b; Zhang et al., 2022). However, these methods largely overlook the reaction information itself, relying solely on molecule-level representations, which inevitably complicates reaction recognition and understanding. To mitigate this issue, Rxn Hypergraph (Tavakoli et al., 2022) first learns a hypernode for reactants and another for products, and then merges these two nodes as the representation for the reaction. However, this method still separates reactions, which also poses challenges for deep neural networks in reaction modeling. Moreover, in single-

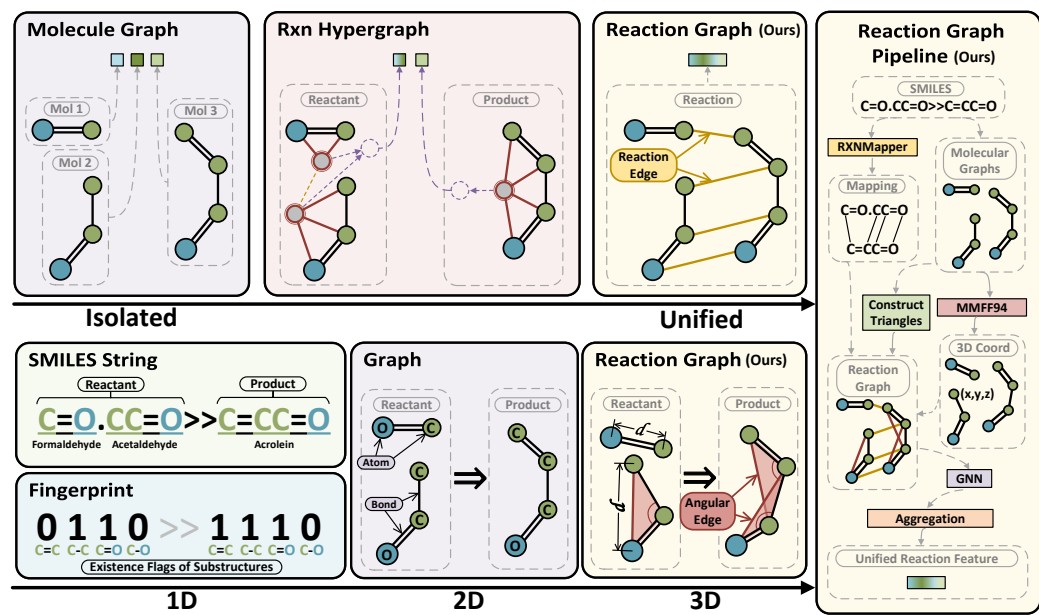

Figure 1: Illustration of Reaction Graph (RG). (1) Existing methods extract isolated representations for reactants and products, and then combine them for prediction, which may fail to effectively model reaction relationship. In contrast, RG unifies the modeling for reactants, products and reactions. (2) Existing 1D- or 2D-based methods may not adequately capture the complexity of molecular structures. RG exploits edge length and an angular edge to implicitly model the 3D structure information. (3) Our method first constructs molecular graph based on SMILES and predicts atomic mapping for creating reaction edges using RXNMapper (Schwaller et al., 2020). Then, 3D atom coordinates are calculated using MMFF94 (Halgren, 1996) and angular edges are constructed for each bond angle. Finally, a GNN is used to extract the unified reaction feature vector based on RG.

molecule modeling, 3D structures have been extensively used because the properties and functions of molecules are intimately connected to their 3D geometric configurations. Yet, this technique has remained unexplored in reaction modeling. This oversight prompts the question of whether incorporating 3D molecular structures could enhance reaction prediction.

In this paper, we propose Reaction Graph (RG) to effectively model chemical reactions. To capture the molecular transformations occurring during reactions, we integrate a reaction edge into graph. This edge connects nodes representing the same atom in both reactants and products, based on atomic mapping relationships, thus allowing graph neural networks (GNNs) to discern molecular independence while assimilating changes in chemical reactions during the message-passing phase. Furthermore, we enhance the graph's capability by embedding 3D spatial information through a new rotationally and translationally invariant approach. Specifically, we utilize edge length and introduce an angular edge to implicitly convey bond angle information by forming shape-stable triangles within the molecular graph. We conduct extensive experiments on a range of reaction-related tasks, including chemical reaction condition prediction, reaction yield prediction and reaction classification. Experimental results indicate that the proposed method is efficient and effective, outperforming existing methods on six datasets. The contributions of this paper are three-fold:

- We propose Reaction Graph, a novel unified graph representation for chemical reactions that allows GNNs to extract reaction transformation related features during the message passing stage.

- We integrate 3D molecular information into reaction modeling. Additionally, we develop a new method to implicitly convey invariant features of bond angles.

- We achieved state-of-the-art accuracy in several tasks, demonstrating the effectiveness of our methods.

## 2 RELATED WORK

**Molecular Representation.** Research interest in molecular representation learning is growing due to its potential in various biochemical tasks like virtual screening and inverse design. To enhance the expressive capabilities of molecular representations, efforts are focused on developing network architectures and training strategies suited to different modalities of molecular input. 1D molecular fingerprint (Morgan, 1965; Durant et al., 2002; Rogers & Hahn, 2010) and SMILES string (Weininger, 1988; O'Boyle & Dalke, 2018; Krenn et al., 2020) are typically processed by language models (Jaeger et al., 2018; Wang et al., 2019; Chithrananda et al., 2020) to extract chemical properties. GNN-based methods (Duvenaud et al., 2015; Kearnes et al., 2016; Xiong et al., 2019) are commonly used to model 2D molecular graphs, which intuitively simulate the relationships between atoms (nodes) and bonds (edges). Recently, the integration of high-dimensional geometric information, including molecular point clouds and 3D molecular graphs (Schütt et al., 2017a; Gasteiger et al., 2020; Atz et al., 2021; Fang et al., 2022; Zhou et al., 2023; Han et al., 2024), has effectively assisted in understanding complex molecular structures.

**Reaction Representation.** Representing chemical reactions is crucial for scientific discovery. A well-designed reaction representation can facilitate the development of various tasks, such as reaction classification (Ghiandoni et al., 2019; Schwaller et al., 2019; Lu & Zhang, 2022), condition recommendation (Gao et al., 2018; Maser et al., 2021; Kwon et al., 2022a; Wang et al., 2023b), and yield prediction (Schwaller et al., 2021b; Kwon et al., 2022b; Yin et al., 2024). To represent chemical reactions, researchers have developed novel fingerprint (Schneider et al., 2015; Probst et al., 2022), graph representation (Varnek et al., 2005; Tavakoli et al., 2022), and deep learning-based methods (Schwaller et al., 2021a; Hou & Dong, 2023). Recently, some studies have introduced strategies such as multi-modal integration (Chen et al., 2024; Zhang et al., 2024) and pre-training (Wen et al., 2022; Shi et al., 2024), providing new insights for constructing reaction representations. However, the reaction representation methods do not pay as much attention to 3D spatial information as molecular representation does. Furthermore, current approaches generally represent reactants and products separately, overlooking the modeling of chemical changes during the reaction process.

## 3 PROPOSED METHOD

In this section, we first briefly review the Molecular Graph (MG) representation. Then, we discuss the potential limitations of MG in reaction modeling and describe the proposed Reaction Graph (RG) in detail. Finally, we incorporate RG into deep neural networks to address multiple chemistry tasks, including reaction condition prediction, yield prediction, and reaction classification.

### 3.1 PRELIMINARY: MOLECULAR GRAPH

In computational ~~and mathematical~~ chemistry, a molecular graph is a representation of a chemical compound's structural formula using graph theory. It is a labeled graph where the vertices represent the compound's atoms and the edges represent chemical bonds. The vertices are labeled with the types of corresponding atoms, while the edges are labeled with the types of bonds.

Specifically, a molecular graph can be represented as $\mathcal{G} = (\boldsymbol{V}, \boldsymbol{E})$, where $\boldsymbol{V} \in \mathbb{R}^{N \times 1}$ denotes the vertices and $\boldsymbol{E} \in \mathbb{R}^{N \times N}$ denotes the edges. Here, $N$ represents the number of vertices. The edge between the $i$-th atom and $j$-th atom is denoted as $e_{ij} \in \{0, 1, 2, 3, 4\}$, with each number corresponding to a specific type of chemical bond:

$$0 : \text{no edge}, \ 1 : \text{single bond}, \ 2 : \text{double bond}, \ 3 : \text{triple bond}, \ 4 : \text{aromatic bond}.$$

Molecular graph provides direct access to the graph underpinning all molecule objects, allowing seamless integration with existing graph functionality.

### 3.2 REACTION GRAPH

When using molecular graphs to model chemical reactions, existing methods typically begin by extracting individual representations for reactants and products, then combine these representations to

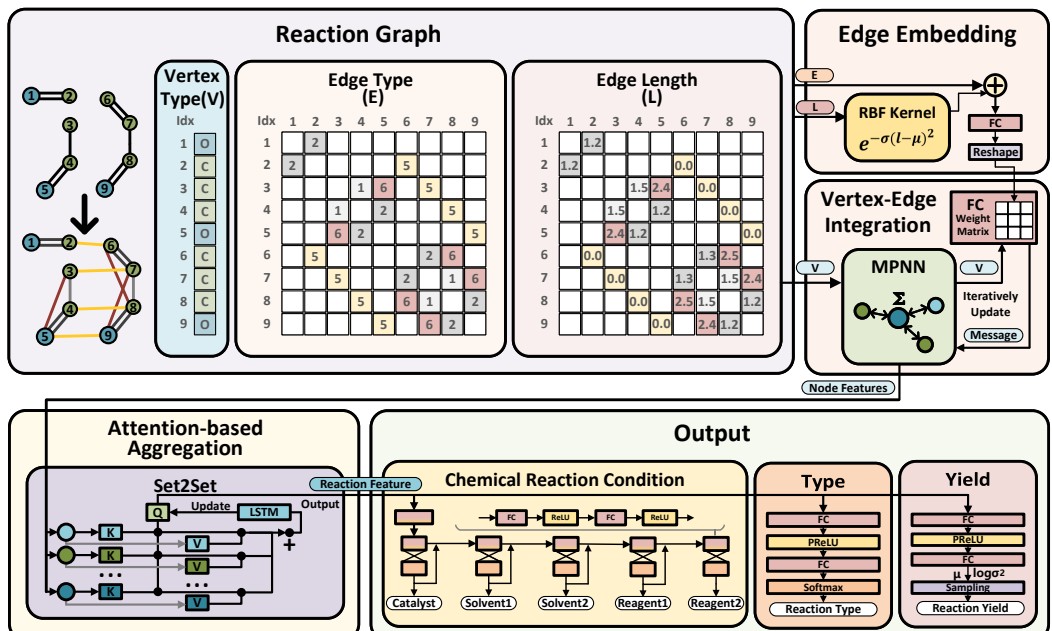

Figure 2: Illustration of the proposed Reaction Graph and the associated model architecture. The input contains the vertex type matrix $V$, the edge type matrix $E$ and the edge length matrix $L$ of Reaction Graph. Model first computes 3D-aware edge embeddings, and then iteratively integrates edge and vertex information into vertex features. Vertex features are aggregated into a unified reaction feature using attention-based method. Finally, task-specific output modules generate prediction results based on reaction features.

form an ensemble reaction representation for prediction. In doing so, these approaches often overlook the reaction information itself, relying exclusively on molecule-level representations. Moreover, the absence of 3D structural information increases the challenge for deep neural networks to effectively model molecules and reactions.

To address these issues, we extend the Molecular Graph into a Reaction Graph (RG). To incorporate reaction modeling, we introduce a reaction edge. This edge links nodes representing the same atom in reactants and products based on atomic mapping relationships, enabling deep neural networks to capture changes in chemical reactions. Additionally, to incorporate 3D spatial structure modeling into RG, we develop a simple yet effective method that is rotationally and translationally invariant. This method utilizes two chemical bond edges and a proposed angular edge to implicitly convey bond angle information by forming stable triangles within molecular graphs. The two bond edges serve as adjacent edges, while the angular edge acts as the diagonal edge. In summary, we extend Molecular Graph to the following Reaction Graph,

$$\mathcal{G} = (\boldsymbol{V}, \boldsymbol{E}, \boldsymbol{L}). \tag{1}$$

In the Reaction Graph, we introduce a new edge attribute, specifically the edge length $\boldsymbol{L} \in \mathbb{R}^{N \times N}$, to represent the 3D structure. We use $l_{ij}$ to denote the length between the $i$-th node and the $j$-node. Additionally, the edge types are expanded to seven categories, i.e., $\boldsymbol{E} \in \{0, 1, 2, 3, 4, 5, 6\}^{N \times N}$, with each number corresponding to a specific type of edge:

0 : no edge, 1 : single bond, 2 : double bond, 3 : triple bond, 4 : aromatic bond,

5 : reaction edge, 6 : angular edge.

If there is no edge between nodes $i$ and $j$, or if the edge type is a reaction edge, the length $l_{ij}$ is defined as 0.

**Edge Embedding.** We use the radial basis function (RBF) kernel to embed the edge length,

$$\boldsymbol{l}_{ij} = \exp(-\boldsymbol{\sigma}(l_{ij} \cdot \boldsymbol{1} - \boldsymbol{\mu})^2), \tag{2}$$

where $\boldsymbol{\sigma}$ and $\boldsymbol{\mu}$ are learnable parameters that transform the scalar edge length into vector representations.

**Vertex-Edge Integration.** To merge the vertex and edge into a unified representation, we follow MPNN (Gilmer et al., 2017) and convert the edge information, including type and length, into a linear projection, which is then applied to the vertex representation as follows,

$$\mathcal{M}_{ij} = \text{Reshape}(\boldsymbol{W}_v \cdot [\boldsymbol{l}_{ij}; \boldsymbol{e}_{ij}]), \quad \boldsymbol{v}_i^{t+1} = \boldsymbol{v}_i^t + \sum_{j \in \mathbb{N}_i} \mathcal{M}_{ij} \cdot \boldsymbol{v}_j^t, \quad \boldsymbol{v}_i' = \boldsymbol{v}_i^{T_1}, \quad (3)$$

where $[\cdot; \cdot]$ denotes concatenation, $\boldsymbol{e}_{ij}$ is the one-hot vector of the edge type for edge $ij$, $\boldsymbol{W}_v$ is the learnable parameters for vertex-edge integration, the $\text{Reshape}(\cdot)$ function reshapes a vector to a matrix, $\mathbb{N}_i$ denotes the set of neighbors of the $i$-th node in RG, $\boldsymbol{v}_j$ represents the representation of the $j$-th vertex, and $T_1$ denotes the total number of iterations. In this way, the vertex representation $\boldsymbol{v}_i'$ becomes edge-related and is able to collect related information from its neighbors.

**Attention-based Aggregation.** To capture the global representation of a Reaction Graph, inspired by Set2Set (Vinyals et al., 2016), we employ an attention-based aggregation method with an LSTM. Specifically, at each iteration of the LSTM, we use the hidden state $\boldsymbol{h}$, initially set to $\boldsymbol{0}$ (the same initialization applies to the cell state $\boldsymbol{c}_0 = \boldsymbol{0}$), to query over all vertices with a softmax-based attention mechanism and collect the most informative clues from these vertices, as described below:

$$\alpha_i^t = \frac{\exp(\boldsymbol{v}_i' \cdot \boldsymbol{h}^t)}{\sum_{j=1}^N \exp(\boldsymbol{v}_j' \cdot \boldsymbol{h}^t)}, \quad \boldsymbol{q}^{t+1} = \sum_{i=1}^N \alpha_i^t \times \boldsymbol{v}_i', \quad \boldsymbol{h}^{t+1}, \boldsymbol{c}^{t+1} = \text{LSTM}(\boldsymbol{q}^{t+1}; \boldsymbol{h}^t, \boldsymbol{c}^t). \quad (4)$$

where $\alpha_i^t$ represents the attention weight of atom $i$ at the $t$-th iteration. After the $T_2$ iteration, the model outputs the reaction global representation vector $\boldsymbol{r}$, where $\boldsymbol{r} = \boldsymbol{W}_r \cdot [\boldsymbol{q}^{T_2}; \boldsymbol{h}^{T_2-1}] + \boldsymbol{b}_r$, with $\boldsymbol{W}_r$ and $\boldsymbol{b}_r$ being the learnable weight matrix and bias, respectively.

**Implementation Details.** As shown in Fig. 1, to construct RG, we first use RXNMapper to predict the atomic mapping, and then employ MMFF94 to calculate atom coordinates. Our method traverses all the angles in molecular graphs to construct angular edges and use the atomic mapping relationship to construct reaction edges, resulting in the final RG.

As shown in Fig. 2, when applying RG to reaction condition prediction, we use an iterative output technique (Gao et al., 2018) to support beam search. Moreover, we employ a two-stage training strategy. Following the joint training in the first stage, the parameters of the neural network are frozen, and the output module's parameters are reinitialized. Then in the second stage, the output module is trained separately. Details can be found in Sec. C and D.

For reaction yield prediction, due to the high noise in yield data, we follow Kwon et al. (2022b) and simultaneously output the mean $y$ and variance $\sigma^2$ of the predicted yield. When the model encounters noise during training, it can increase the predicted variance to keep the output mean relatively stable, thus enhancing training stability. In implementation, we utilize a Multilayer Perceptron (MLP) with Monte Carlo Dropout (Gal & Ghahramani, 2016) technique.

Lastly, the reaction classification module uses a standard three-layer MLP for output.

## 4 EXPERIMENTS

### 4.1 WHAT DOES REACTION GRAPH ADDRESS?

**Attention Weights Visualization.** To illustrate the advantages of Reaction Graph (RG) compared to Molecular Graph (MG), we train a condition prediction model on USPTO[1] and visualize the attention weights $\alpha_i$ of reactions. Attention weights can display the model's focus on different parts of molecules, especially reaction centers, revealing the model's understanding of the reaction mechanism. As shown in Fig. 3, we take 3-Amino-5-bromobenzoic acid ($C_7H_6BrNO_2$) and its two related reactions as examples. The $C_7H_6BrNO_2$ features three active functional groups: bromo, amino, and carboxyl-hydroxyl. In reaction $A$, the bromo group acts as the reaction center, while the carboxyl-hydroxyl group serves this role in reaction $B$.

---

[1] https://figshare.com/articles/dataset/Chemical_reactions_from_US_patents_1976-Sep2016_/5104873

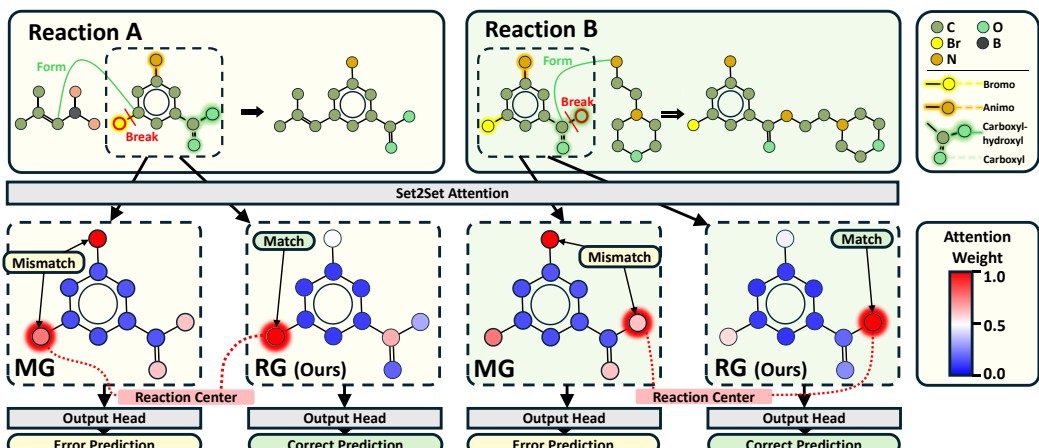

Figure 3: Visualization of attention weights and prediction results for two reactions involving the bromo and carboxyl-hydroxyl groups in $C_7H_6BrNO_2$. The colors of the atoms in the upper diagrams correspond to the types of atoms, while the colors of the atoms in the lower diagrams correspond to the sizes of the atomic attention weights. The model using the Molecular Graph (MG) focuses on atoms that are less relevant to the reaction, thus leading to prediction errors. In contrast, the model equipped with the Reaction Graph (RG) accurately concentrates on the reaction center, and produces the correct prediction results.

As depicted in Fig. 3, we find that the attention distribution remains almost unchanged across different reactions when adopting MG as input. In both reactions, the MG-based model focuses more on the non-reactive amino group and insufficiently on reaction centers, resulting in prediction errors. In contrast, the RG-empowered model pays correct attention to reaction centers and provides reasonable reaction conditions. The experiment results validate our hypothesis: MG, which represents reactants and products independently, struggles to capture atom and bond transformations during the reaction process, while RG helps the model accurately locate the reaction center and extract relevant features of reaction changes. For more visualization results and experiments regarding Attention Weights, please refer to Sec. G.

**Leaving Group Identification Analysis.** We also design the Leaving Group (LvG) identification task to further validate the effectiveness of RG. LvG refers to the atomic group that is present in the reactants and detaches from the products during a chemical reaction, which is closely related to the reaction mechanism (Wang et al., 2023c). LvG identification is a node-level multi-class classification task, where the node label indicates whether an atom belongs to a LvG and specifies its type. This requires the model to focus not only on the features of the molecule itself but also on reaction-related features.

Both models based on MG and RG are trained on the LvG dataset extracted from USPTO. The evaluation metrics include accuracy (ACC), confusion entropy (CEN), and the multi-class Matthews Correlation Coefficient (MCC), and the Macro F1 Score (F1). CEN assesses the misclassification level, while MCC and F1 measure accuracy accounting for the imbalance of sample categories. We report relevant metrics for all atoms and LvG atoms, separately.

As shown in Tab.1, RG outperforms MG on all metrics. Specifically, compared to MG, RG improves the overall ACC, MCC and F1 by 4.7%, 42.4% and 53.9%, respectively. As for LvG atoms, RG achieves an ACC of 94.7%, which is twice that of the MG. The excellent performance of RG on LvG identification task demonstrates its ability to understand the reaction mechanism.

Table 1: Leaving group (LvG) identification results of Molecular Graph (MG) and Reaction Graph (RG) representations, with overall and LvG atom-specific evaluation.

| Rep. | Overall | | | | LvG Atom-Specific | | | |
|------|---------|---------|---------|-------|-------------------|---------|---------|-------|
|      | ACC↑ | CEN↓ | MCC↑ | F1↑ | ACC↑ | CEN↓ | MCC↑ | F1↑ |
| MG | 0.950 | 0.036 | 0.549 | 0.365 | 0.448 | 0.201 | 0.519 | 0.404 |
| RG (ours) | **0.997** | **0.002** | **0.973** | **0.904** | **0.947** | **0.031** | **0.945** | **0.903** |

Table 2: Influence of different types of 3D information on the USPTO-Condition dataset. Experimental groups include no 3D information, bond edge length, bond edge length and bond angle, as well as bond edge length and angular edge length.

| 3D Information | Accuracy |
|---|---|
| - | 0.3133 |
| Bond Edge Length | 0.3165 |
| Bond Edge Length +Bond Angle | 0.3179 |
| Bond Edge Length +Angular Edge Length | **0.3246** |

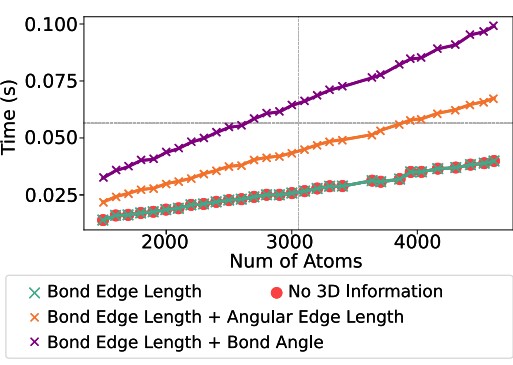

Figure 4: Influence of different methods of 3D structure modeling on running time. Compared to using bond angles, the proposed angular edge method effectively reduces inference time.

## 4.2 THE EFFECT OF 3D INFORMATION

**Settings.** In this section, we explore the effects of incorporating various 3D information in RG. Specifically, we investigate: (1) no 3D information, (2) only bond edge length, (3) bond edge length and bond angle, as well as (4) bond edge length and angular edge length. We conduct experiments on the USPTO-Condition dataset to evaluate the accuracy. To assess computational efficiency, we further design the following experiments. Inspired by bin-packing (Cormen et al., 2022), we select 16 sets of chemical reactions from USPTO-Condition, ensuring that each set contains the same number of atoms, with quantities ranging from 1500 to 4500. Subsequently, the 16 sets of reactions are input into the model for condition prediction, and the average runtime is measured.

**Accuracy Evaluation.** According to Tab. 2, the incorporation of 3D information effectively improves the model's performance. Specifically, the RG equipped with bond edge length and angular edge length achieves the best performance. Results also suggest that the introduction of angular edge length is more effective than directly using bond angle. This is because the angular edge length is integrated into the GNN as part of the graph structure. The geometric consistency helps to more accurately maintain the spatial relationship of the molecule. In contrast, the bond angle needs to be treated separately from the bond edge length, which may distort the original geometric continuity and integrity of the molecule.

**Efficiency Evaluation.** Fig. 4 illustrates the impact of different 3D structure modeling in RG on the running time. The results suggest that incorporating bond length brings almost no extra computational overhead. Besides, compared to bond angle, using angular edge length can significantly reduce the inference time. Moreover, according to the curve steepness, the time cost associated with using bond angle rises more significantly as the number of atoms increases. Hence, when integrating 3D molecular information into RG, we ultimately adopt bond edge length and angular edge length, enhancing accuracy while maintaining model efficiency.

## 4.3 REACTION CONDITION PREDICTION

**Dataset.** The USPTO-Condition dataset is derived from Parrot (Wang et al., 2023b), comprising over 680K samples, divided into 80% for training, 10% for validation, and 10% for testing. Besides, we construct Pistachio-Condition from the Pistachio database by thorough cleaning and filtering. It includes over 560K samples, with a training, validation, and testing split of 8:1:1.

**Evaluation Metrics.** Following Gao et al. (2018); Wang et al. (2023b), we use the top-$k$ accuracy to evaluate the condition prediction performance.

**Comparison Methods.** CRM (Gao et al., 2018) utilizes molecular fingerprints, while Parrot (Wang et al., 2023b) employs SMILES. Both AR-GCN (Maser et al., 2021) and CIMG (Zhang et al., 2022) use MG, whereas D-MPNN (Heid & Green, 2021) leverages the condensed graph of reac-

Table 3: Top-$k$ accuracy of reaction condition prediction on the USPTO-Condition and Pistachio-Condition datasets. (*) indicates that the result is sourced from Wang et al. (2023b).

| Method | USPTO-Condition | | | | | Pistachio-Condition | | | | |
|---|---|---|---|---|---|---|---|---|---|---|
| | Top-1↑ | Top-3↑ | Top-5↑ | Top-10↑ | Top-15↑ | Top-1↑ | Top-3↑ | Top-5↑ | Top-10↑ | Top-15↑ |
| CRM (Gao et al., 2018) | 0.260* | 0.377* | 0.421* | 0.461* | 0.472* | 0.330 | 0.469 | 0.510 | 0.548 | 0.554 |
| Parrot (Wang et al., 2023b) | 0.269* | 0.404* | 0.451* | 0.491* | 0.503* | 0.350 | 0.532 | 0.588 | 0.626 | 0.630 |
| AR-GCN (Maser et al., 2021) | 0.146* | 0.237* | 0.273* | 0.312* | 0.326* | - | - | - | - | - |
| CIMG (Zhang et al., 2022) | 0.184* | 0.271* | 0.303* | 0.339* | 0.353* | - | - | - | - | - |
| D-MPNN (Heid & Green, 2021) | 0.198 | 0.300 | 0.334 | 0.378 | 0.392 | 0.259 | 0.342 | 0.378 | 0.442 | 0.469 |
| Rxn Hypergraph (Tavakoli et al., 2022) | 0.213 | 0.308 | 0.345 | 0.381 | 0.393 | 0.288 | 0.367 | 0.412 | 0.464 | 0.485 |
| Reaction Graph (ours) | **0.325** | **0.434** | **0.472** | **0.506** | **0.518** | **0.392** | **0.557** | **0.604** | **0.638** | **0.643** |

tions (CGR) (Varnek et al., 2005), and Rxn Hypergraph (Tavakoli et al., 2022) employs its own designed graph representation.

**Results.** The performance comparisons are reported in Tab. 3. Our method outperforms all the comparison methods on both datasets, demonstrating the superiority of RG on reaction feature modeling. On USPTO-Condition, compared to domain models with 1D and 2D representations, our method improves the top-1 accuracy by 17.2% and 76.6%, respectively. Compared with graph-based methods, RG improves the top-$k$ accuracy by an average of 39.0%. On Pistachio-Condition, RG also demonstrates its advantage by surpassing other methods by 3.4%-18.8%.

**Ablation Study.** The reaction information and 3D structure are key components of RG. We evaluate their respective effects on modeling chemical reactions. Ablation results in Tab. 4 reveal that both the reaction information and 3D structure in RG effectively enhance model performance. Specifically, on USPTO-Condition, the utilization of reaction information and 3D structure brings average performance improvements of 3.9% and 2.5%, respectively; while in Pistachio, the improvements are 1.9% and 1.0%. Moreover, the reaction information and 3D information are complementary, and their combination results in improvements of 6.4% and 2.9% on USPTO-Condition and Pistachio-Condition, respectively.

Table 4: Influence of reaction information (Reaction Edge) and 3D structure (3D Stru) on the prediction of chemical reaction conditions.

| Dataset | Reaction Edge | 3D Stru | ACC↑ |
|---|---|---|---|
| USPTO-Condition | ✗ | ✗ | 0.3050 |
| | ✗ | ✓ | 0.3090 |
| | ✓ | ✗ | 0.3133 |
| | ✓ | ✓ | **0.3246** |
| Pistachio-Condition | ✗ | ✗ | 0.3806 |
| | ✗ | ✓ | 0.3819 |
| | ✓ | ✗ | 0.3852 |
| | ✓ | ✓ | **0.3915** |

### 4.4 REACTION YIELD PREDICTION

**Dataset.** Buchwald-Hartwig (B-H) (Ahneman et al., 2018) involves six molecules as reactants, with products comprised of a single molecule. B-H is used to create B-H-1 to B-H-4 through different train-test splits, with increasing challenges due to distribution differences. The molecule number involved in each reaction varies in Suzuki-Miyaura (S-M) (Perera et al., 2018). USPTO-Yield (Schwaller et al., 2021b) is divided into Gram and Subgram. We also notice that in the small-scale B-H and S-M datasets, there are only dozens of different molecular types, some of which are reagents; meanwhile, the USPTO-Yield dataset contains a significant amount of noise. This makes it difficult for the model to capture the relatively complex and variable 3D information, preventing it from learning the correct 3D priors. Therefore, we only test the role of reaction information in the yield prediction task.

**Evaluation Metrics.** The proposed method simultaneously outputs the mean $y$ and variance $\sigma^2$ of the predicted yield. Following Schwaller et al. (2021b); Kwon et al. (2022b), we use the $R^2$ score to evaluate the accuracy of the output mean. We additionally introduce likelihood $(y-y')^2/\sigma^2$ and log variance $\log \sigma^2$ from negative log-likelihood (Lakshminarayanan et al., 2017) to evaluate the output variance, where $y'$ is the ground truth.

Table 5: Regression accuracy ($R^2 \uparrow$) for reaction yield prediction on the Buchwald-Hartwig (B-H), Suzuki-Miyaura (S-M), Gram, and Subgram datasets. B-H-1, B-H-2, B-H-3 and B-H-4 are more challenging splits of the B-H dataset. (*) indicates the results are reported from the original paper.

| Method | Representation | B-H | B-H-1 | B-H-2 | B-H-3 | B-H-4 | S-M | Gram | Subgram |
|---|---|---|---|---|---|---|---|---|---|
| DRFP* (Probst et al., 2022) | Fingerprint | 0.95 | 0.81 | 0.83 | 0.71 | 0.49 | 0.85 | **0.130** | 0.197 |
| Yield-Bert* (Schwaller et al., 2021b) | SMILES | 0.95 | **0.84** | 0.84 | 0.75 | 0.49 | 0.82 | 0.117 | 0.195 |
| Egret* (Yin et al., 2024) | SMILES | 0.94 | **0.84** | **0.88** | 0.65 | 0.54 | 0.85 | 0.128 | 0.206 |
| UGNN (Kwon et al., 2022b) | Molecular Graph | **0.97*** | 0.74* | **0.88*** | 0.72* | 0.50* | **0.89*** | 0.117 | 0.190 |
| D-MPNN (Heid & Green, 2021) | CGR | 0.94 | 0.80 | 0.82 | 0.73 | 0.55 | 0.85 | 0.125 | 0.202 |
| Rxn Hypergraph (Tavakoli et al., 2022) | Rxn Hypergraph | 0.96 | 0.81 | 0.83 | 0.71 | 0.56 | 0.85 | 0.118 | 0.196 |
| Reaction Graph (ours) | Reaction Graph | **0.97** | 0.80 | **0.88** | **0.76** | **0.68** | **0.89** | 0.129 | **0.216** |

Table 6: Likelihood $(y - y')^2 / \sigma^2$ and log variance $\log \sigma^2$ metrics on the Gram and Subgram datasets, where likelihood reflects the consistency between predicted variance and regression error, and log variance reflects the size of variance. Within these methods, only UGNN has variance output.

| Methods | Representation | Likelihood↓ | | Log Variance↓ | |
|---|---|---|---|---|---|
| | | Gram | Subgram | Gram | Subgram |
| UGNN (Kwon et al., 2022b) | Molecular Graph | **1.02** | 1.20 | 6.94 | 7.36 |
| Reaction Graph (ours) | Reaction Graph | 1.06 | **1.18** | **5.86** | **6.14** |

**Comparison Methods.** DRFP (Probst et al., 2022) utilizes reaction fingerprint. Yield-Bert (Schwaller et al., 2021b) and Egret (Yin et al., 2024) are based on SMILES. UGNN (Kwon et al., 2022b) employs MG and simultaneously predicts yield and uncertainty. D-MPNN (Heid & Green, 2021) adopts CGR, while Rxn Hypergraph (Tavakoli et al., 2022) uses its uniquely designed graph representation.

**Results.** As shown in Tab. 5, RG achieves the highest accuracy on six out of eight yield prediction datasets. Especially on the more challenging B-H-4 and Subgram datasets, RG achieves improvements of $21.4\%$ and $4.9\%$, respectively. Besides, according to Tab. 6, both methods provide uncertainty that accurately reflects the actual error levels, while RG further reduces prediction uncertainty by $12.3\%$ on Gram and $13.3\%$ on Subgram. However, the quality and complexity of the Gram and Subgram datasets restrict further performance improvement in existing yield prediction methods.

### 4.5 REACTION CLASSIFICATION

**Dataset.** The USPTO-TPL is from Schwaller et al. (2021a), with labels generated by 1000 reaction templates, making it relatively simple. We construct the more challenging Pistachio-Type dataset from Pistachio, with labels generated by NameRXN[2] based on rules.

**Evaluation Metrics.** Similar to Schwaller et al. (2021a); Lu & Zhang (2022), we use accuracy (ACC), confusion entropy (CEN), Matthews Correlation Coefficient (MCC) and Macro F1 (F1) to evaluate the performance. CEN assesses misclassifications to quantify the uncertainty of predictions, while MCC and F1 provide a more comprehensive measure of classification accuracy.

**Comparison Methods.** DRFP (Probst et al., 2022) uses reaction fingerprint. RXNFP (Schwaller et al., 2021a) and T5Chem (Lu & Zhang, 2022) are based on SMILES. D-MPNN (Heid & Green, 2021) adopts CGR, while Rxn Hypergraph (Tavakoli et al., 2022) relies on its own graph representations.

**Results.** According to Tab. 7, RG surpasses advanced models on both USPTO-TPL and Pistachio-Type, demonstrating the effectiveness of the proposed designs. Compared to the state-of-the-art T5Chem, RG reduces the classification error by $66.6\%$ and achieves nearly $100\%$ accuracy on USPTO-TPL. The superior performance on USPTO-TPL is due to the limited number of reaction templates, which simplifies the classification task. RG's precise identification of the reaction center

---

[2]https://www.nextmovesoftware.com/namerxn.html

Table 7: Reaction classification results on the USPTO-TPL and Pistachio-Condition datasets. Evaluation metrics include accuracy (ACC), confusion entropy (CEN), Matthews Correlation Coefficient (MCC) and Macro F1 (F1). (*) indicates that the result is sourced from the original paper.

| Method | USPTO-TPL | | | | Pistachio-Type | | | |
|---|---|---|---|---|---|---|---|---|
| | ACC↑ | CEN↓ | MCC↑ | F1↑ | ACC↑ | CEN↓ | MCC↑ | F1↑ |
| DRFP | 0.977* | 0.011* | 0.977* | 0.972 | 0.899 | 0.149 | 0.890 | 0.898 |
| RXNFP | 0.989* | 0.006* | 0.989* | 0.986 | 0.948 | 0.078 | 0.944 | 0.946 |
| T5Chem | 0.995* | 0.003* | 0.995* | - | 0.976 | 0.041 | 0.974 | 0.976 |
| D-MPNN | 0.997 | 0.001 | 0.997 | 0.996 | 0.982 | 0.033 | 0.980 | 0.982 |
| Rxn Hypergraph | 0.954 | 0.024 | 0.953 | 0.935 | 0.911 | 0.129 | 0.903 | 0.910 |
| Reaction Graph (ours) | **0.999** | **0.001** | **0.999** | **0.998** | **0.987** | **0.024** | **0.986** | **0.987** |

(detailed in Sec. 4.1) enhances template discrimination capability, bringing further performance improvements. On the complex Pistachio-Type dataset, RG exceeds the best performance by 1.2% on MCC and 1.1% on F1, highlighting its superiority in modeling reactions.

**Ablation Study.** We investigate the influence of reaction information and 3D structure in RG on the reaction classification task. As shown in Tab. 8, integrating reaction information reduces classification error by an average of 77% on USPTO-TPL and 54.1% on Pistachio-Type. On the other hand, 3D structure can also enhance the accuracy across both datasets. The results suggest that reaction information and 3D structures mutually enhance each other, improving the understanding of reaction mechanisms.

Table 8: Influence of the proposed reaction information (Reaction Info) and 3D structure (3D Stru) modeling methods on the USPTO-Condition and Pistachio-Condition datasets, using ACC, CEN and MCC as classification metrics.

| Reaction Info | 3D Stru | USPTO-TPL | | | Pistachio-Type | | |
|---|---|---|---|---|---|---|---|
| | | ACC↑ | CEN↓ | MCC↑ | ACC↑ | CEN↓ | MCC↑ |
| ✗ | ✗ | 0.9921 | 0.0037 | 0.9921 | 0.9658 | 0.0559 | 0.9627 |
| ✗ | ✓ | 0.9955 | 0.0021 | 0.9955 | 0.9669 | 0.0538 | 0.9640 |
| ✓ | ✗ | 0.9978 | 0.0010 | 0.9977 | 0.9862 | 0.0262 | 0.9850 |
| ✓ | ✓ | **0.9991** | **0.0004** | **0.9991** | **0.9873** | **0.0242** | **0.9862** |

## 5 CONCLUSION

In this paper, we propose a unified 3D Reaction Graph (RG) for chemical reaction modeling. Unlike existing methods, the RG is equipped with enhanced capabilities for modeling reaction changes and 3D structures. We conduct extensive experiments across various tasks and datasets, demonstrating RG's effectiveness in understanding chemical reactions. Furthermore, since it is independent of any specific GNN architecture, the RG representation may show increased potential as the underlying network backbone is improved.

**Limitations and Future Work.** Firstly, like most data-driven methods, the quality of data significantly impacts the performance of our method. Specifically, inaccuracies in the 3D coordinates of atoms can lead to inferior results. Thus, developing accurate and efficient methods for 3D prediction could further enhance our approach. Secondly, integrating scientific knowledge and rules into deep learning is crucial for AI applications in science. In RG, we incorporate atom-level reaction relationships into the graph, thereby enhancing our understanding of reactions. However, this is not sufficient. Further exploration of methods to inject chemical laws into deep neural networks could significantly enhance the effectiveness of the proposed method. These potentials can be further investigated in the future.

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

# Reaction Graph: Toward Modeling Chemical Reactions with 3D Molecular Structures

# Appendix

CONTENTS

# A    SUPPLEMENTARY RELATED WORKS

## A.1    MOLECULE AND REACTION REPRESENTATION

The representation of molecules and reactions plays a crucial role in chemical property prediction models. The representation of molecules and reactions shares similarities. By appropriately combining molecular representations, one can derive a representation of a reaction. Various methods have been proposed for representing molecules, encompassing both explicit and implicit expressions.

Explicit expressions provide interpretative molecular representations and include 1D strings (Weininger, 1988; Krenn et al., 2020; O'Boyle & Dalke, 2018; Heller et al., 2015), molecular fingerprints (Morgan, 1965; Glen et al., 2006; Rogers & Hahn, 2010; Probst et al., 2022), 2D molecular graphs (Gilmer et al., 2017; Kong et al., 2022; Li et al., 2024; Zhang et al., 2021), and 3D molecular point clouds (Schütt et al., 2017b;a; Thomas et al., 2018; Zhou et al., 2023).   1D strings, such as the commonly used Simplified molecular-input line-entry system (SMILES) (Weininger, 1988) and the more robust SELFIE (Krenn et al., 2020), are suitable inputs for natural language processing models like BERT (Devlin et al., 2018) and LLaMA(Touvron et al., 2023). Molecular fingerprints, typically fixed-length binary sequences, are ideal inputs for machine learning models.  However, fingerprints have limited expressive capacity.  2D molecular graphs come in various forms: some treat atoms as nodes and bonds as edges (Gilmer et al., 2017), while other works use a hierarchical structure that incorporates both fine-grained atomic-level information and coarse-grained molecular fragment information (Kong et al., 2022; Luong & Singh, 2023; Zhang et al., 2021).  These graphs are usually processed by GNNs.  3D molecular point clouds are generally used for tasks like molecular dynamics calculations (Batzner et al., 2022; Anderson et al., 2019) and quantum chemical property predictions(Wang et al., 2024; Du et al., 2024), and are typically processed by equivalent neural networks.

Implicit molecular representations, on the other hand, are vectorized representations obtained through feature extraction by neural networks from explicit molecular representations(Gómez-Bombarelli et al., 2018; Jaeger et al., 2018; Jin et al., 2020; Zhu et al., 2023).  These vectorized representations contain the inherent properties of the molecules as well as richer predictive information derived from the model's training priors, making them suitable for downstream tasks.

The representation of chemical reactions corresponds to the aforementioned molecular representations. For reaction SMILES strings, they essentially combine the SMILES of reactant and product molecules using pre-defined symbols. Some reaction fingerprints result from bitwise operations on the molecular fingerprints of reactants and products (Gao et al., 2018; Chen & Li, 2024).  In the case of molecular graphs, previous works on reaction-related tasks often simply combine molecular graphs (Ryou et al., 2020; Maser et al., 2021; Kwon et al., 2022b) or use a hierarchical structure (Tavakoli et al., 2022) to obtain graph representations for reactions. However, these methods have limitations in expressive capacity, due to the lack of reaction change information. There are also representations like Condensed Graph of Reaction (CGR) to depict changes from reactants to products, which also accurately characterize the features of the chemical reaction (Varnek et al., 2005). Compared to CGR, RG explicitly distinguishes and makes connections between reactant and product molecules, allowing Reaction Graph (RG) to model both 3D molecular structure and 3D reaction transformations. A detailed discussion on the difference between RG and CGR is in Sec. B.

## A.2    EQUIVALENT AND INVARIANT NEURAL NETWORK

To enhance our model's understanding of reaction properties related to 3D molecular structures, we integrate 3D information into RG in an invariant manner. This method ensures that the prediction results remain unaffected by the rotation and translation of molecules in 3D space. Invariant neural networks (INNs) are a special type of equivalent neural networks (ENNs). They achieve invariance through the use of transformation-invariant features, such as distances and angles (Han et al., 2024). Initially, DTNN (Schütt et al., 2017b) utilizes distances between atoms with Gaussian basis embedding. Later, SchNet (Schütt et al., 2017a) improves performance by using learnable radial basis function (RBF) kernel embeddings. DimeNet (Gasteiger et al., 2020) is the first to introduce angle features in quantum chemistry property prediction tasks, and GemNet (Gasteiger et al., 2021) further enhances this by incorporating dihedral angles. Subsequently, ComENet (Wang et al., 2022) and SphereNet (Liu et al., 2022) introduce new dihedral angle representations to reduce computational

costs. Additionally, models like ClofNet (Du et al., 2022) and LEFTNet (Du et al., 2024) use a local coordinate system to guarantee equivalence and propose the concept of completeness. Some works also explore the use of Transformer architectures (Vaswani et al., 2017), such as Graphormer (Ying et al., 2021) and UniMol (Zhou et al., 2023).

While considering the expressive ability and completeness of the invariant neural network, we also take into account the specific task scenario. For reaction-related tasks, it is crucial for RG to accurately express molecular geometric features like bond lengths and bond angles while respecting the flexibility and rotational degrees of freedom of molecules. This suggests that we may not need to input too many multi-body features into the model, as that could lead to redundant information.

To be detailed, a molecule has different conformations due to the rotation of single bonds. In different conformations of the same molecule, bond lengths and bond angles generally change only slightly, while the torsion angles can vary significantly. Reaction Graph employs bond length and bond angle features but does not utilize torsion angles. Therefore, the 3D features of Reaction Graph are nearly invariant to different conformations (i.e. theoretically the selection of conformer will not affect the performance of Reaction Graph).

Reaction Graph is nearly invariant to different conformations. Similarly, the properties of chemical reactions are also invariant to different conformations. For example, the fact that $D$ is the catalyst for reaction $AB >> C$ does not change with the specific conformations of $A$, $B$, and $C$. Thus, Reaction Graph is consistent with the task objectives. Moreover, the invariance of the reaction graph to different conformations enhances the consistency of 3D features across different training samples, making it easier for the model to learn.

In contrast, torsion angles and pairwise distances vary with specific conformations. However, the properties of chemical reactions remain invariant to different conformations. Therefore, torsion angles and pairwise distances provide redundant information. Redundant information makes it difficult for the model to interpret the 3D features. Therefore, Reaction Graph do not include torsion angles and pairwise distances.

We also demonstrate the effectiveness of our design in Sec. G.5.1.

### A.3 REACTION CONDITION PREDICTION

Reaction conditions are pivotal for chemical synthesis and drug design. Suitable reaction conditions can significantly enhance the progress of reactions and increase yields, whereas unsuitable conditions can impede reactions and result in the wastage of raw materials. In computer-aided synthesis planning (CASP), designing an efficient route to synthesize the target product requires not only predicting the starting materials, such as reactants, but also determining the optimal conditions for each step of the reaction.

For the task of reaction condition prediction, given the representation of a reaction, the model is designed to predict the required catalysts, solvents, as well as the temperature, pressure, and other reaction conditions needed for the reaction to occur. In the early stages, machine learning models used for predicting reaction conditions were primarily based on knowledge graph reasoning (Segler & Waller, 2017), database similarity searches (Lin et al., 2016), and expert systems (Marcou et al., 2015).

With (Gao et al., 2018) and others pioneering the use of neural networks trained on large datasets for the prediction of reaction conditions, increasing attention has been given to the potential of deep learning in this task. Researchers attempt to improve upon the inputs to the network. (Afonina et al., 2021) uses ISIDA fingerprints as descriptors for reactions, (Walker et al., 2019) uses MACCS keys as inputs, and (Chen & Li, 2024) uses the difference between the product and reactant Morgan fingerprints as input. Other researchers focus on enhancements in network architecture. (Ryou et al., 2020), (Maser et al., 2021), and (Kwon et al., 2022a) use GNNs for feature extraction, while (Andronov et al., 2023) and (Wang et al., 2023b) attempt to use Transformers to extract features of reactions. Additionally, (Kwon et al., 2022a) and (Karpovich et al., 2023), aiming to address the one-to-many relationship between reactions and reaction conditions, opt to use Variational Autoencoders (VAEs) (Kingma & Welling, 2014) for generating reaction conditions, allowing for multiple sets of reaction condition labels to be produced for a single input reaction.

## A.4   REACTION YIELD PREDICTION

In the context of CASP, the yield of the synthetic route is also an important evaluation factor. For a high-quality synthetic route, it is essential to take into account both the cost of materials and the overall yield. However, for some single-step reactions generated based on retrosynthesis models, we may not find corresponding yield data in the database. This necessitates the use of data-driven yield prediction models to predict the yields of these reactions, thereby providing a more reliable yield measure for multi-step retrosynthetic route planning programs.

The objective of a reaction yield prediction model is to estimate the yield of a chemical reaction, ranging from 0-100%, based on the provided reaction representation. Reaction yield typically indicates the percentage of reactant molecules converted into the desired product. Similar to reaction condition prediction tasks, early approaches utilize fingerprints for reaction yield prediction (Ahneman et al., 2018), but these methods are generally applied to specific types of chemical reactions. With the advent of Transformers, efforts to achieve yield prediction using language models like BERT emerge, exemplified by models such as YieldBERT (Schwaller et al., 2021b) and Egret (Yin et al., 2024). These approaches benefit from low data acquisition costs, making them suitable for large-scale datasets. However, their performance often hinges on carefully designed pre-training tasks and the scale of the dataset. Additionally, there are methods employing GNNs (Kwon et al., 2022b), but many of these face challenges as mentioned earlier. Recently, researchers have begun exploring Bayesian Neural Networks (BNNs) (Gal & Ghahramani, 2016; Kendall & Gal, 2017), which output both the yield and the associated confidence, thereby enhancing training stability and result interpretability (Kwon et al., 2022b; Chen et al., 2024). Some studies also attempt to integrate reaction conditions or multimodal information to improve practical applicability and model accuracy (Yin et al., 2024; Chen et al., 2024), while the potential of pre-training remains to be explored (Shi et al., 2024).

## A.5   REACTION CLASSIFICATION

The classification of chemical reaction types is based on the changes between reactants and products, as well as the underlying reaction principles. Identifying the type of an unknown reaction helps us compare it with those already in the database, thereby understanding the potential characteristics of the reaction and aiding in decision-making for other reaction-related tasks. Additionally, classifying reaction types tests a model's understanding of reaction mechanisms, contributing to performance analysis and interpretability of the model.

Earlier, people use rule-based methods to classify chemical reactions (Ghiandoni et al., 2019). These methods are accurate and have strong interpretability, but they require a huge amount of labor. In recent years, with the rise of machine learning, data-driven methods begin to emerge (Schwaller et al., 2019). RXNFP (Schwaller et al., 2021a) employs Bert to extract implicit vector representations of reactions from SMILES strings, achieving excellent clustering results for reaction types. DRFP (Probst et al., 2022) combines circular fingerprints and hash-based fingerprint generation methods to efficiently represent a chemical reaction in an interpretable manner. T5Chem (Lu & Zhang, 2022) utilizes a specially designed multi-task decoder for reaction type classification, enhancing the model's understanding of reaction mechanisms. Furthermore, works like Egret (Yin et al., 2024) use reaction type classifiers trained on the Pistachio dataset for data analysis, aiming to evaluate the model's yield prediction capabilities.

## B   MOLECULE AND REACTION REPRESENTATIONS

In deep learning tasks, the representation of chemical molecules and reactions is used to describe the chemical features of molecules and reactions. Modalities can be divided into 1D strings and fingerprints, 2D molecular or reaction graphs, and 3D molecular or reaction point clouds. In terms of interpretability, features can be classified as explicit and implicit. Explicit features can be understood by humans or explained using simple rules, while implicit features are usually representation vectors obtained through data-driven learning from black-box neural networks. They contain rich information but are difficult to interpret. Tab. 9 lists examples of various representation forms for aspirin, and Tab. 10 lists the representations of chemical reaction for synthesizing aspirin.

Table 9: Example of different types of molecular representations of aspirin, including string-based representations such as SMILES, SELFIES, DeepSMILES, fingerprint-based representations like ECFP and MACCS, molecular graph and improvements, as well as 3D molecular point cloud (graph).

| Type | Representation |
|---|---|
| SMILES | $CC(=O)Oc1ccccc1C(=O)O$ |
| SELFIES | $[C][C][Branch1\_2][C][=O][O][C][=C][C][=C][C][=C]$ $[Ring1][Branch1\_2][C][Branch1\_2][C][=O][O]$ |
| DeepSMILES | $CC=O)Occcccc6C=O)O$ |
| ECFP | [ 1 1 1 0 1 1 0 1 1 0 1 1 0 1 0 1 1 1 1 0 0 0 1 1 0 1 0 0 1 1 0 0 ] |
| MACCS | [ 0 0 0 0 0 0 0 0 0 0 0 0 0 0 0 0 0 0 0 0 0 0 0 0 0 0 0 0 0 0 0 0 0 0 0 0 0 0 0 0 0 0 0 0 0 0 0 0 0 0 0 0 0 0 0 0 0 0 0 0 0 0 0 0 0 0 0 0 0 0 0 0 0 0 0 0 0 0 0 0 0 0 0 0 0 0 1 0 0 0 0 0 0 0 0 0 0 0 0 0 0 0 0 0 1 0 0 0 0 0 0 0 0 0 1 0 0 1 1 0 0 0 0 0 0 0 0 1 0 0 1 1 0 0 1 1 0 1 0 0 0 1 0 1 0 1 0 0 1 0 1 1 0 1 1 1 1 0 ] |
| Molecular Graph |  |
| PS Graph |  |
| Point Cloud |  |

## B.1 ONE-DIMENSIONAL

### B.1.1 STRING

In deep learning chemistry tasks, a common representation form is the string representation. String representations are typically composed of a sequence of characters that express atoms, bonds, topological structures, and even stereochemical information, following a specific grammar that can be parsed by computer programs and is generally human-readable. This representation form is widely used in chemical databases to describe molecules and reactions. Due to its similarity to natural language, networks such as Transformers and RNNs, which can handle variable-length data, are often employed to extract features from these string representations. String representation data is readily available and generally possesses strong expressive capabilities, as it reflects not only the molecular composition but also topological and 3D information. However, there exists a many-to-one relationship between strings and molecules. Although existing datasets provide canonization algorithms, most of them may have flaws, making it difficult for models to learn the correct relationships be-

Table 10: Example of different types of reaction representations, including string-based SMILES and SMARTS, fingerprint-based DRFP and ISIDA, graph representations Molecular Graph, CGR, Rxn Hypergraph and Reaction Graph, as well as 3D reaction point cloud (graph).

| Type | Representation |
|---|---|
| SMILES | $c1ccc(c(c1)C(=O)O)O.CC(=O)OC(=O)C$ 
 $>> CC(=O)Oc1ccccc1C(=O)O$ |
| SMARTS | $CC(=O)O[C:2]([CH3:1]) = [O:3].[OH:4][c:5]1[cH:6]$ 
 $[cH:7][cH:8][cH:9][c:10]1[C:11](=[O:12])[OH:13]$ 
 $>> [CH3:1][C:2](=[O:3])[O:4][c:5]1[cH:6][cH:7]$ 
 $[cH:8][cH:9][c:10]1[C:11](=[O:12])[OH:13]$ |
| DRFP | [ 0 1 1 1 1 0 1 0 0 0 0 0 1 1 1 1 1 0 1 1 0 1 1 0 1 1 0 1 0 0 0 0 ] |
| ISIDA | [ 0 0 0 1 1 0 1 0 1 0 0 0 0 0 0 1 1 0 1 1 1 0 0 0 1 0 0 0 0 0 1 1 ] |
| Molecular Graph |  |
| CGR |  |
| Rxn Hypergraph |  |
| Reaction Graph (Omit Angular Edges and 3D) |  |
| Point Cloud |  |

tween molecules. This leads to a reliance on optimization methods such as pre-training, which often do not yield satisfactory results.

**SMILES.** (Weininger, 1988) expression is the most commonly used string representation in deep learning chemistry tasks. It is composed of symbols representing atoms or groups, symbols for

bonds, labels for rings, case rules for indicating aromaticity, and symbols for chirality, all working together to represent the atomic composition and topological structure of a molecule.

The SMILES representation of aspirin, as shown in Tab. 9, uses atomic symbols to represent the atoms in the molecule. For example, the symbol for carbon is $C$, for oxygen is $O$, and bromine, which does not appear in the example, has the symbol $Br$. Hydrogen atoms are typically omitted. For chemical bonds, a single bond is represented by a dash (-), which is usually omitted, a double bond is represented by an equals sign (=), and a triple bond is represented by a hash sign (#).

It's important to note the presence of parentheses in the structure, which indicate a branch; for instance, the expression $C(=O)$ signifies a branch with an oxygen atom connected to the carbon by a double bond. Additionally, we observe the structure $c1ccccc1$, where the numbers 1 at both ends indicate that the corresponding carbon atoms are connected by a bond to form a ring. The lowercase letter indicates that this is an aromatic ring.

Furthermore, SMILES may include functional groups or atoms with valence states enclosed in brackets, such as $[O-]$ and $[C@H]$. Here, $[O-]$ denotes an oxygen atom with a negative charge, and the (@) symbol typically indicates a chiral center, with different chiralities distinguished by $[C@H]$ and $[C@@H]$. For mixtures, we generally use the (.) symbol to separate the SMILES expressions of different molecules, while for ionic compounds, ionic bonds can be represented as ( ), and various ions can also be written in the form of a mixture.

For chemical reactions, the SMILES notation generally follows the format of $reactants >$ $reagents > products$, as shown in Tab. 10, which does not include reagents. Currently, mainstream Python libraries such as RDKit[3] and OpenBabel[4] can parse or generate SMILES expressions for molecules or chemical reactions. They can convert between SMILES and structured molecular data, and efficiently calculate various chemical properties.

The SMILES expression is simple and easy to understand, but its drawback is that it may not always be valid. For example, a SMILES expression like $C(=O)(=O)(=O)$ has more bonds on the carbon than its outermost electron count allows, making such a structure unreasonable. Additionally, in cases where branches or ring structures are complex, the SMILES can become very long. This leads to language models encoding long token sequences for SMILES or having difficulty generating SMILES sequences, prompting subsequent improvements to the SMILES representation.

**SMARTS.** (Daylight Chemical Information Systems, 2007) is an extension of SMILES used to describe patterns or substructures within molecular structures. It allows the use of wildcards and logical operators to represent more complex chemical queries. For chemical reactions, SMARTS strings can include atomic mapping information. For example, in the aspirin synthesis reaction shown in Tab. 10, we can find the symbol $[C:2]$ in both the reactants and products, indicating that this carbon atom is the same in both. Similarly, the oxygen atoms in $[OH:4]$ and $[O:4]$ refer to the same atom. With this atomic mapping information, we can better understand the reaction mechanisms and the changes that occur before and after the reaction, as well as construct CGRs and Reaction Graphs (RGs) for reaction characterization. The atomic mapping relationships are determined based on the reaction mechanism and the bond-breaking positions, and traditional algorithms often struggle to achieve reliable matching. Therefore, the conversion from SMILES to SMARTS strings with mapping information is typically performed using neural networks. Tools like RXNMapper[5] provide convenient mapping functionalities to accomplish this task.

**SELFIES.** (Krenn et al., 2020) is a fully robust grammar designed to ensure the validity of molecular strings. As shown in Tab. 9, the smallest unit of a SELFIES string is a symbol enclosed in brackets []. The SELFIES grammar consists of five types of symbols: atom symbols, branch symbols, ring symbols, index symbols, and special symbols. Atom symbols, such as $[=O]$, include bond information and atom types, similar to SMILES, where single bonds can be omitted. Branch symbols, like $[Branch1\_2]$, indicate that the subsequent symbol $[C]$ will be interpreted as an index symbol rather than representing a carbon atom. All symbols in SELFIES can be escaped as index symbols and correspond to a number. Here, $[C]$ is designated to correspond to the number 0, indicating that the length of the branch is $0 + 1 = 1$ (since a branch must have at least one atom, we add 1). This

---

[3]https://www.rdkit.org/

[4]https://openbabel.org/index.html

[5]https://github.com/rxn4chemistry/rxnmapper

means that the first valid atom $[=O]$ following the $[C]$ character belongs to the branch. The number 2 indicates that this branch will be connected to the main chain in a double bond format.

The same applies to rings; for example, the [Ring1] followed by $[Branch1\_2]$ is also an index symbol corresponding to the number 4, indicating that the previous sequence of $[C][=C][C][=C][C][=C]$ consists of a total of $4 + 2 = 6$ atoms forming a ring (since a ring must have at least 2 atoms, we add 2). Special symbols, such as (.), are used to separate the SELFIES expressions of multiple molecules in a mixture. The SELFIES parsing program scans from left to right, keeping track of the remaining number of bonds that can be connected to the current atom. If the number of bonds in the following symbol exceeds this value, that symbol is discarded. Thus, we can consider that SELFIES ensures its legality through the design of the parsing program rules. Any sequence of symbols generated by the model can be interpreted as a valid molecule.

Currently, converting SMILES to SELFIES is typically accomplished using the Python SELFIES toolkit[6]. The conversion from SMILES strings to SELFIES is efficient and does not impose significant preprocessing overhead on the dataset; since each symbol can be treated as a token during tokenization, the token sequence of SELFIES is also not excessively long. We consider this a clever design, but it can lead to a single symbol having multiple meanings (for example, it could represent either an atom or an index). Additionally, for specific tasks, the validity of the generated output does not equate to its rationality or correctness. Therefore, even with the support of SELFIES, the design of the model and the data are crucial.

**DeepSMILES.** (O'Boyle & Dalke, 2018) appears very similar to that of the original SMILES expression, consisting of symbols representing atoms, bond symbols, single right parentheses, and numeric indicators for rings. The key difference is that only right parentheses are present in the expression, and each ring is represented by a single number. For example, in the case of the number 6 indicating a ring, it represents that the preceding six atoms $ccccc$ form a ring, making it more concise compared to SMILES and akin to the SELFIES representation.

For branches, DeepSMILES employs a clever method of expression. Since both SMILES branches and standard mathematical expressions use nested parentheses, one can draw a parallel to how computers process simple arithmetic expressions. Typically, an arithmetic expression is first converted into an expression tree, which is then arranged into a postfix expression through postorder traversal, also known as Reverse Polish Notation. For instance, the expression $2 \times (4 + 6) + 3 \times (7 - 5)$ can be converted to $2\ 4\ 6\ +\ \times\ 3\ 7\ 5\ -\ \times\ +$. This representation allows for writing a mathematical expression without needing parentheses.

Similarly, by using a comparable approach, SMILES expressions can be transformed into a format that only contains right parentheses, where each parenthesis corresponds to a branch and an atom. For example, as in Tab. 9, in the expression $=O$), the $=O$ symbol indicates that it is part of a branch. This method helps ensure that the matching of parentheses and ring numbers does not obstruct the generative model. However, compared to the complete robustness of SELFIES, DeepSMILES strings may still represent invalid molecules. Additionally, since the length of the branch corresponds to the number of right parentheses, DeepSMILES strings can become quite long (for example, $B(c1ccccc1)(O)O$ converts to DeepSMILES as $Bccccc6))))))O)O$, which is longer than the original SMILES string).

Currently, the conversion from SMILES to DeepSMILES can be accomplished using the Python DeepSMILES toolkit[7].

### B.1.2 FINGERPRINT

Another 1D representation form is fingerprint representation. Explicit fingerprint representations typically manifest as binary sequences, where each bit indicates the presence or absence of a substructure or chemical property. In contrast, implicit fingerprint representations are usually derived from other explicit representation forms through neural network processing, resulting in latent vectors. These vectors are generally of fixed length, making them easily manageable by most neural network architectures, such as MLPs.

---

[6]https://github.com/aspuru-guzik-group/selfies
[7]https://github.com/baoilleach/deepsmiles

Explicit fingerprints can significantly enhance the performance of specific tasks by selecting the most relevant features and discarding irrelevant ones; however, this relies on careful manual design and has limited generalization ability. Additionally, explicit fingerprints have a many-to-one relationship with molecules or reactions, which restricts their expressive power and limits performance in large-scale complex data tasks. On the other hand, implicit representations obtained through data-driven learning possess strong expressive capabilities, but extracting these representations depends on the original explicit representations, network architecture, training objectives, and data. These aspects are all popular research directions in the field of deep learning chemistry.

**ECFP.** (Rogers & Hahn, 2010) generates fingerprints by traversing the atoms of a molecule and their surrounding environments, capturing the topological structure information of the molecule. The ECFP generation algorithm starts by assigning an initial identifier to each atom based on its features. Then, it undergoes several iterations where, in each iteration, the identifier of each atom is combined with the identifiers of its neighbors, and a hash function generates new identifiers. The identifiers from the nth iteration actually contain information about the topology within a radius of n+1 around the atom. Each iteration's identifiers are recorded. Ultimately, a fixed-length bit string is used to represent all these identifiers, as shown in Tab. 9. The simplest way to construct this bit string is by taking each identifier's value modulo the bit string length and setting the corresponding position to 1.

ECFP is efficient to construct and easy to input into various machine learning models. It can be generated using popular chemical toolkits like RDKit, where it is referred to as Morgan Fingerprint. ECFP is generally designed for molecules; if it is to be used for tasks related to chemical reactions, a common approach is to compute an ECFP for both the reactants and products, then concatenate or differentiate the fingerprints of the reactants and products to form the fingerprint for the chemical reaction. However, large molecules and extensive datasets can lead to different molecules having the same ECFP. Additionally, the expressiveness of the ECFP based on a bit string is limited. This necessitates longer ECFP lengths for challenging tasks, such as data-driven prediction of reaction conditions, where the ECFP can reach lengths of up to $2^{14}$ bits.

**MACCS.** (Durant et al., 2002) fingerprints are 166-bit long fingerprints, where each bit represents the presence or absence of a specific substructure or chemical property of a molecule. The meanings of these bits are predefined, unlike Morgan circular fingerprints, which derive bit indices through operations like taking modulo of identifiers. For example, the MACCS fingerprint shown in Tab. 9 has 167 bits, as computer indexing typically starts from 0, and the MACCS fingerprint generated by toolkits usually has a 0 filled in the first bit.

In the observed example fingerprint, the 234-th bit from the end is set to 1, indicating the presence of a ring, an oxygen atom, and a six-membered ring within the molecule. MACCS fingerprints can be generated using toolkits such as RDKit or OpenBabel. The predefined molecular fragment information gives MACCS fingerprints strong interpretability, and this information is relatively well correlated with various chemical properties of the molecule. For small datasets, MACCS fingerprints can provide strong priors and help avoid overfitting.

However, for larger datasets, the expressive capacity of the 166-bit MACCS bit string gradually reveals its limitations. Moreover, this fingerprint design is entirely reliant on human input; while it may perform well on certain tasks, it has poor generalization ability and high design costs.

**ISIDA.** (Varnek et al., 2005) is a method for describing molecular and supramolecular structures as well as reactions based on sequence molecular fragments, with its reaction descriptors constructed using CGR graphs, as shown in Tab. 10. The fragment descriptors constructed using the ISIDA method are referred to as ISIDA fingerprints.

First, the algorithm extracts sequence fragments from each molecule in the dataset. For each atom in the molecule, the algorithm performs a random walk around the atom based on specified minimum and maximum path lengths, recording the atomic bonds along the walk. These path fragments serve as substructure templates. The occurrence of each substructure template in each molecule is recorded, and this information is used to construct the fingerprint for each molecule. The simplest approach is to count all templates appearing in the entire dataset and represent the occurrence of a specific template in a molecule as a bit string for that molecule. Of course, there are various construction methods available.

This construction process is automated, and the sequence fragments are essentially subgraphs, sharing similarities with the previously mentioned fingerprints. Additionally, since ISIDA fingerprints are based on CGR graphs, they can better reflect the changes occurring during chemical reactions compared to reaction fingerprints formed by direct differentiation or concatenation of molecular fingerprints.

To generate ISIDA fingerprints for molecules in a specific dataset, we can first use the Python CGR-Tools[8] to create CGR graphs and export them as SDF files, and then use the Python CIMTools[9] package or ISIDA/Fragmentor 2017[10] to extract and construct the ISIDA fingerprints.

**DRFP.** (Probst et al., 2022) shares similarities with Morgan Circular Fingerprints. DRFP first extracts a list of circular molecular n-grams from the reactants and products, then calculates the symmetric difference of the n-gram lists. For each n-gram in the resulting symmetric difference list, a descriptor is computed. Finally, similar to the method used in Morgan Circular Fingerprints, the set of descriptors is converted into a fixed-length descriptor vector. This approach is akin to the differentiation method in Morgan Circular Fingerprints, highlighting the changes that occur before and after the reaction. DRFP can be generated using the Python DRFP[11] library, allowing for the adjustment of various related parameters.

## B.2 Two-Dimensional

In 2D graph representations, each node typically corresponds to an atom in a molecule or reaction, while edges correspond to bonds. Each node and edge has associated attributes to express its chemical features, such as atom type and bond type. Graph representations are usually processed by GNNs or Graph Transformers to extract features. Specially designed molecular graphs or reaction graphs incorporate additional vertices and edges to enable neural networks to extract features more efficiently.

Due to the similarity between graphs and the real-world existence of chemical molecules, graph representations possess strong expressive capabilities. The specially designed features of nodes and edges can create a one-to-one correspondence between the 2D graph and the molecules or reactions, aiding the network in correctly modeling the relationships between them. However, most feature extraction methods based on graph representations either struggle to model long-range relationships or rely on dense graphs, which can lead to overfitting. Improving graph representations for more efficient and accurate feature extraction is currently a key area of research. Below are some related graph representations.

**Molecular Graph (MG).** (Duvenaud et al., 2015) uses graphs to represent molecules, where nodes correspond to atoms and edges correspond to bonds, with both nodes and edges carrying chemical information about the atoms and bonds, respectively. Such graphs are typically undirected and are often stored in a computer as an edge matrix of size $E \times 2$, instead of adjacency matrix, where the first column represents the starting point and the second column represents the endpoint. Additionally, there are attribute matrices of size $V \times D_V$ and $E \times D_E$. Sometimes, the edge matrix is also stored in forms such as an adjacency matrix.

As shown in Tab. 9 and 10, the visualization of the aspirin MG reveals that the molecule has 13 nodes and 13 edges, thus $V = 13$ and $E = 13$. In molecular property prediction tasks, hydrogen atoms are typically omitted, significantly reducing the size of MG and minimizing redundant information. However, in molecular dynamics calculations, hydrogen atoms are generally not omitted. Currently, most chemical toolkits support the conversion from SMILES strings to MGs.

**PS Graph.** (Kong et al., 2022) is a hierarchical molecular graph constructed based on subgraphs, containing both the original atomic-level molecular graph and fragment-level molecular graph. As shown in Tab. 9, each fragment graph's nodes correspond to a set of adjacent atoms in the atomic-level molecular graph; for example, the second node of the fragment graph corresponds to the benzene ring substructure. These subgraphs are extracted from the dataset using the Principle Subgraph Mining method.

---

[8]https://cgrtools.readthedocs.io/index.html

[9]https://github.com/cimm-kzn/CIMtools

[10]https://infochim.u-strasbg.fr/downloads/manuals/Fragmentor2017/Fragmentor2017_Manual_nov2017.pdf

[11]https://github.com/reymond-group/drfp

Specifically, the algorithm first extracts each type of atom present in the dataset to create an initial subgraph library, where these subgraphs consist of only one node. Then, the subgraph library is iteratively expanded. During each iteration, each molecule can be represented as a fragment-level molecular graph using the current subgraphs in the library. For each subgraph in a molecule, the algorithm attempts to merge it with each adjacent subgraph to form new subgraphs. Finally, the frequency of all new subgraphs is counted, and the most frequently occurring ones are added to the subgraph library. This process is repeated until the number of subgraphs reaches the desired threshold.

This work is initially used for hierarchical generation of molecules but later found to also benefit molecular property prediction. In feature extraction, works like GraphFP choose to process both the atomic-level molecular graph and the fragment-level molecular graph through two different GNNs with distinct parameters, ultimately merging the feature vectors to obtain a representation vector for the entire molecule. This method combines the coarse-grained features and fine-grained features of the molecule.

However, it is noted that the aforementioned subgraph extraction method still incurs significant resource overhead, as extracting each subgraph requires traversing the entire dataset, making it challenging to run on large molecular datasets with millions of entries.

**Condense Graph of Reaction (CGR).** (Varnek et al., 2005) is a graph specifically designed for chemical reactions, as shown in Tab. 10. Previously, we mentioned that ECFP-based reaction fingerprints are typically constructed using concatenation or differential methods. CGR can be seen as an analogous approach applied to graphs, modeling the reaction as the superposition of molecules participating in a chemical reaction, where each node represents the mixed features of the same atom in the reactants and products, while each edge represents the mixed characteristics of the same chemical bond in the reactants and products. The mixing of characteristics is typically done through operations such as subtraction or concatenation. If a certain atom or edge is absent in either the reactants or products, the feature for the absent side is set to zero.

CGRs have a smaller size and are computationally efficient, but they may sacrifice the relative independence of the molecules. Meanwhile, the introduction of zero features can also lead to redundancy. Algorithms for building CGR are provided in toolkits like CGRTools and Chemprop[12].

**Rxn Hypergraph.** (Tavakoli et al., 2022), like the CGR graph, is specifically designed for chemical reactions. As in Tab. 10, all atoms in the molecule are connected to a Mol Hypernode that represents the molecule, while all Mol Hypernodes in the reactants or products are connected to a Rxn Hypernode that represents the entirety of the reactants or products. Furthermore, the Rxn Hypernodes in the reactants are interconnected, and the Rxn Hypernodes in the products are also interconnected, with no connections between the reactants and products.

Unlike the CGR graph, the Rxn Hypergraph is designed to adapt the RGAT architecture. The hierarchical design allows it to focus on features with different granularities within the molecule, such as atomic-level information and molecular-level information during feature extraction. However, the reactant molecules remain isolated from the product molecules, with no direct connections between them. Only after features have been extracted from both the reactant and product graphs are they concatenated for downstream tasks.

The Rxn Hypergraph facilitates information exchange between reactant molecules and between product molecules, combining fine-grained and coarse-grained features, and it can be constructed without the need for atomic mapping information. However, the Rxn Hypergraph does not leverage the prior knowledge of changes inherent in chemical reactions, which may affect its expressive power. We couldn't find any existing toolkits that can generate Rxn Hypergraphs. The Rxn Hypergraph used in our tests is implemented based on the formulation in Tavakoli et al. (2022).

### B.3 THREE-DIMENSIONAL

3D molecules or chemical reactions are typically represented using point cloud or 3D molecular graph (Thomas et al., 2018; Fuchs et al., 2020), as shown in Tab. 9 and 10. In 3D point cloud representations, each node corresponds to an atom and includes not only chemical attributes but

---

[12]https://github.com/chemprop/chemprop

Table 11: Ability comparison of graph representations of reaction. **Itn** represents internal message transmission within a molecule, **R to R** represents message transmission between reactants, **P to P** represents message transmission between products, **R to P** represents message transmission between reactants and products, **Mol** and **Rxn** indicate the ability to express an independent molecule and an entire chemical reaction during the message passing stage, and **2D** and **3D** represent the inclusion of 2D and 3D topological structures.

| Representation | Message Passing | | | | Expression | | | |
|---|---|---|---|---|---|---|---|---|
| | Itn | R to R | P to P | R to P | Mol | Rxn | 2D | 3D |
| MG | ✓ | | | | ✓ | | ✓ | |
| Rxn Hypergraph | ✓ | ✓ | ✓ | | ✓ | | ✓ | |
| CGR | ✓ | ✓ | ✓ | ✓ | | ✓ | ✓ | |
| RG (ours) | ✓ | ✓ | ✓ | ✓ | ✓ | ✓ | ✓ | ✓ |

also information about 3D spatial positioning, such as coordinates. 3D point clouds are usually processed using point cloud convolution or point cloud Transformers to extract features. To facilitate computation, some methods set a cutoff distance for convolution (Schütt et al., 2017a), creating edges between two atoms within this cutoff distance, thereby forming a 3D graph representation. This approach is similar to molecular graph representations, but it appears to be denser and requires additional handling of 3D information.

Since molecules can undergo transformations such as translation and rotation in space, there is a many-to-one relationship between 3D point clouds or graphs and the molecules or reactions. Therefore, a key focus in 3D chemical tasks is called equivariance, which ensures that the neural network can understand transformations of molecules or the molecules that compose a chemical reaction in 3D space from the design perspective. In terms of 3D point clouds or graph representations, we also aim to design invariant or equivariant features to facilitate subsequent processing by equivariant neural networks. Such 3D point clouds or graphs are generally used in the molecular field for molecular dynamics or quantum chemical property calculations (Batzner et al., 2022), and 3D reaction point cloud can be used for predicting molecular conformations of transition states in chemical reactions (Jackson et al., 2021). However, they are relatively rare in predicting yields, reaction conditions, and reaction types.

### B.4    COMPARISON OF GRAPH REPRESENTATIONS OF REACTION

As shown in Tab. 11, we compare the message passing and expressive capabilities of various reaction representation forms. All representations allow for message passing between atoms within a molecule. However, MG lacks edges connecting different molecules, thereby preventing message passing across atoms in separate molecules. In contrast, Rxn Hypergraph connects all atoms within a molecule through a Mol Hypernode, with Mol Hypernodes in both reactants and products forming a dense graph. This structure enables message passing among atoms within the reactants and products; however, it still does not facilitate message passing between reactants and products.

On the other hand, CGR models the reaction as a superposition of molecules. During the message passing stage in GNNs, the atomic features of the same atom in reactants and products are stored in a single node and processed through fully connected layers simultaneously, which results in a loss of relative independence between the molecules.

Similarly, RG connects nodes of the same atom in reactants and products by reaction edges. This improvement also enables message passing between reactants or products through reaction edges → bond edges/angular edges → reaction edges. It achieves global message passing while preserving locality for feature extraction, which helps mitigate the risk of overfitting.

In comparison to the Rxn Hypergraph, our method ensures the independence of molecules, facilitating feature extraction, graph modification, and the incorporation of 3D information. Additionally, to the best of our knowledge, we are the first to model bond angle features through adding edges to a unified graph representation, rather than through model architecture improvements or by introducing an additional line graph. Moreover, our approach is orthogonal to the Rxn Hypergraph and

other enhancements based on MG, allowing for straightforward integration. The effectiveness of combining these methods warrants further investigation.

# C  IMPLEMENTATION DETAILS

## C.1  GRAPH REPRESENTATIONS

### C.1.1  REACTION GRAPH

In the implementation, each node and edge of Reaction Graph (RG) contains more information. The attributes of node $i$ can be represented as vector $\boldsymbol{n}_i \in \mathbb{R}^{D_n}$ (distinguished from $\boldsymbol{v}_i$ in the Sec. 3), and the attributes of edge $ij$ can be represented as vector $\boldsymbol{e}_{ij} \in \mathbb{R}^{D_e}$, where $\boldsymbol{v}_i$ and $\boldsymbol{e}_{ij}$ are the concatenations of all attribute values of the node and edge, respectively. Tab. 12 provides an example of attribute values. Tab. 13 provides detailed dimensions of node attributes and edge attributes for each dataset. Note that the edge attributes here do not include $l_{ij}$, which is stored in a separate matrix $\boldsymbol{L}$ according to our previous definition in Sec. 3.

**Torsion Angle Extension.** Reaction Graph can be easily extended to model torsion angle. We only need to add a new type of edge called torsion angular edges based on the original Reaction Graph. Specifically, for each edge $BC$ in the molecular graph, we search for each edge $AB$ connected to node $B$ and each edge $CD$ connected to node $C$. If $A$ is different from $C$, and $D$ is different from $B$, and $A$ is different from $D$, we add a torsion angular edge $AD$. With the already constructed bond angular edges, the length of any edge in the tetrahedron $ABCD$ is determined, thereby uniquely defining the tetrahedron. Once the tetrahedron $ABCD$ is defined, the dihedral angle between the planes $ABC$ and $BCD$ is determined, which corresponds to the torsion angle of $BC$.

**One-hot Length Embedding Baseline.** In our experiments, we also use one-hot length embeddings, and divide the lengths into 16 bins. The distribution of bond lengths in the dataset is not uniform. If we use the previous method of uniform division, most of the one-hot embeddings for the bond lengths will be the same. This results in a loss of 3D information. Therefore, we adopt a frequency-based division method. First, we count all the bond lengths in the dataset and evenly divide these lengths into 16 bins. This approach provides a higher degree of differentiation for the key lengths and has stronger expressive capability. The specific bond length distribution and the division method are shown in Fig. 5.

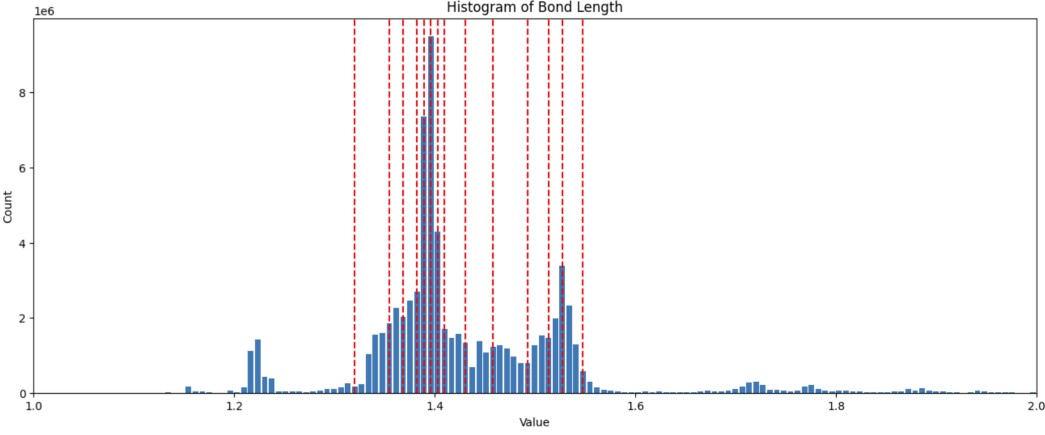

Figure 5: Histogram of the bond length distribution of molecular graphs on the USPTO-Condition dataset, along with the division method of the bins.

### C.1.2  RXN HYPERGRAPH WITH REACTION HYPERNODE

We can model the interactions between reactants and products by adding another hypernode in the Rxn Hypergraph. Specifically, we can connect the hypernodes of reactants and products with an

Table 12: Example of node and edge attributes in Reaction Graph. Each attribute is represented by a one-hot vector, and the final node attribute or edge attribute is the concatenation of all relevant attributes.

| | | |
|---|---|---|
| **Node** | **Atom Type** | C, O, N, F, P, S, Cl, Br, I |
| | **Charge Type** | -4, -3, -2, -1, 0, 1, 2, 3, 4, 5, 6, 7 |
| | **Degree Type** | 0, 1, 2, 3, 4, 5, 6, 7, 8 |
| | **Hybridization Type** | SP, SP2, SP3, SP3D, SP3D2, S |
| | **Num of Hydrogens** | 1, 2, 3, 4, 5, 0 |
| | **Valence Type** | 0, 1, 2, 3, 4, 5, 6, 7, 8, 9, 10, 11, 12, 13, 14 |
| | **Ring Size Type** | 3, 4, 5, 6, 7, 8 |
| **Edge** | **Edge Type** | Bond Edge, Reaction Edge, Angular Edge |
| | **Bond Type** | Single, Double, Triple, Aromatic |
| | **Chirality Type** | Clockwise, Counterclockwise |
| | **Stereochemistry Type** | BondCis, BondTrans |

Table 13: The dimensions of node and edge attribute vector for each dataset, where $D_n$ denotes node attribute vector dimensions, and $D_e$ denotes edge attribute vector dimensions.

| Dataset | $D_n$ | $D_e$ |
|---|---|---|
| USPTO-Condition | 110 | 13 |
| Pistachio-Condition | 117 | 13 |
| Buchwald-Hartwig | 43 | 10 |
| Suzuki-Miyaura | 49 | 10 |
| USPTO-Yield | 128 | 14 |
| USPTO-TPL | 135 | 14 |
| Pistachio-Type | 127 | 17 |

additional hypernode. However, this method still exhibits limitations compared to our Reaction Graph.

**Lack of Atomic Mapping.** To effectively track the transformation of substructures in a reaction, GNNs need to establish a mapping that identifies the correspondence between the substructures in reactants and products. However, the hypernode method falls short in directly representing this mapping, posing challenges in accurately expressing the interactions between reactants and products.

**Limited Capability for Reaction Modeling.** When a node has too many neighbors, it may lead to an information loss during message aggregation in GNNs (referred to as the Over-squashing Issue). This additional hypernode also faces the similar Over-squashing problem. Because the additional Hypernodes serve as a bridge for interactions between reactants and products, it usually has massive numbers of neighbors. Since it is difficult for a single node to carry so much information, the additional hypernode method may have a limited capability for reaction modeling, especially when many large reactants and products involved.

### C.1.3 CGR WITH 3D

In our supplementary experiments, we added 3D bond length information to the CGR in D-MPNN. Specifically, since the Chemprop library does not provide implementations for conformation calculations, we implemented a 3D Featurizer based on the Chemprop library code. The conformation information is stored as coordinates on the nodes. Note that in the CGR, one node corresponds to two atoms, so each node stores the coordinate information of the two atoms from the reactants and products. During inference, we calculate the bond lengths using the atomic coordinates, embedding them with a 16-dimensional RBF kernel, and then concatenate the bond length embeddings to the edge feature vector. Additionally, each edge in the CGR corresponds to two chemical bonds from the reactants and products, so each edge contains two length embeddings. Furthermore, all model architectures and hyperparameter settings (except for the input layer dimension of the edge features) remain consistent with the original model.

### C.1.4 RXN HYPERGRAPH WITH 3D

In our supplementary experiments, we add 3D bond length information to the Rxn Hypergraph. Specifically, for each edge in the Rxn Hypergraph, we calculate the length using the coordinates of the two atoms it connects, just like in a molecular graph. Since the Rxn Hypergraph contains hypernodes, which do not have actual physical coordinates, we set the lengths of all edges connected to hypernodes (including molecular hypernodes, reactant hypernodes, and product hypernodes) to 0. This method ensures the invariance of the 3D information. Before the message passing begins, we use a 4-dimensional RBF kernel to embed the edges in the Rxn Hypergraph and concatenate the edge length embeddings to the edge features. Other hyperparameters remain the same as in the original model.

## C.2 MODEL DETAILS

### C.2.1 OURS MODEL

**Node Embedding.** We simply use a fully connected (FC) layer to embed the node vector $\boldsymbol{n}_i$, resulting in $\boldsymbol{v}_i \in \mathbb{R}^{D_v}$.

**Edge Embedding.** We first use the RBF kernel to embed the edge length $l_{ij}$ as $\boldsymbol{l}_{ij} \in \mathbb{R}^{D_l}$, and concatenate the edge length embedding to the edge attribute vector to obtain $\boldsymbol{e}'_{ij} = [\boldsymbol{e}_{ij}; \boldsymbol{l}_{ij}]$. Then, $\boldsymbol{e}'_{ij}$ is directly inputted into a FC layer and the output vector is reshaped into matrix $\mathcal{M}_{ij} \in \mathbb{R}^{D_v \times D_v}$. This approach is similar to the method of generating weight matrices in hypernetworks.

**Vertex-Edge Integration.** We refer to the method in Kwon et al. (2022b), using residual connections to enhance training stability and employing GRU to facilitate node attribute updates. Specifically, the node attributes $\boldsymbol{v}_i$ will serve as the memory state of the GRU, while the messages $\boldsymbol{v}_i^t + \sum_{j \in \mathbb{N}_i} \mathcal{M}_{ij} \cdot \boldsymbol{v}_j^t$ aggregated during each MPNN iteration will be treated as the input to the GRU, updating the node attributes instead of directly assignment. The final feature vector of node $i$ is calculated by $\boldsymbol{v}'_i = [\boldsymbol{v}_i^0; \boldsymbol{v}_i^{T_1}]$.

**Attention-based Aggregation.** We use $[\boldsymbol{q}^{t+1}; \boldsymbol{h}^t]$ as the input to the LSTM to obtain $\boldsymbol{h}^{t+1}$. The final reaction representation vector $\boldsymbol{r}$ is obtained by mapping $[\boldsymbol{q}^{T_2}; \boldsymbol{h}^{T_2-1}]$ through a FC layer with PReLU activation.

**Output.** This module is specifically designed for each downstream task. For the condition prediction task, the output module is consistent with CRM.

$$\boldsymbol{z}_1 = f_{c_1}(\boldsymbol{r}; \theta_{c_1}), \tag{5}$$

$$\boldsymbol{z}_i = f_{c_i}(\boldsymbol{r}, \boldsymbol{z}_1, \ldots, \boldsymbol{z}_{i-1}; \theta_{c_i}), \tag{6}$$

where $f_{c_i}$ linearly map each previously predicted condition $\boldsymbol{z}_j$ to embedding vector $\boldsymbol{h}_{z_j} \in \mathbb{R}^{D_z}$, concatenate all $\boldsymbol{z}_j$ with reaction feature $\boldsymbol{r}$ and feed them into a 2-layer FC classifier with ReLU activation to predict the next probability vector $\boldsymbol{z}_i \in \mathbb{R}^{D_{c_i}}$ of the $i$-th condition.

The yield prediction module uses a 3-layer MLP. It incorporates PReLU activation and Dropout layers, and simultaneously outputs the predicted mean $\mu_y$ and logarithmic variance $\log \sigma_y^2$ of yield.

$$\mu_y, \log \sigma_y^2 = f_r(\boldsymbol{h}; \theta_r). \tag{7}$$

The reaction classification head also uses a 3-layer MLP with PReLU activation to output the predicted probability vector of reaction type.

**Hyperparameter Settings.** For USPTO-Condition, USPTO-TPL, and Pistachio-Type, we set the edge length embedding dimension $D_l$ to 16, with a total edge attribute dimension of $D_l + D_e = 29$. For Pistachio-Condition, $D_l = 1$, with $D_l + D_e = 14$. For the yield prediction task, since 3D information is not used, $D_l = 0$.

For reaction condition prediction and reaction classification tasks, we set the embedding dimension $D_v$ of nodes to 200, the number of MPNN iterations $T_1$ to 3, the number of Set2Set aggregation iterations $T_2$ to 2, the dimension of the GRU memory state to be the same as $D_v$, while the LSTM memory state dimension is set to $2 \times D_v$. The final output reaction feature vector $r$ has a dimension $D_r$ of 4096. For yield prediction task, $D_v$ is set to 64, $D_r$ is set to 1024, and $T_1 = T_2 = 3$.

The hidden layer dimension of the reaction condition output head $f_c$ is 512, and the dimension of the condition embedding $D_z$ is 256. The hidden layer dimension of the reaction yield output head $f_r$ is 512, with a dropout rate of 0.1. The hidden layer dimension of the reaction classification output head is 4096.

For specific hyperparameter selection methods, please refer to Sec. G.

### C.2.2 OTHER MODELS

**Bond Angle Model.** We adopt the approach of DimeNet (Gasteiger et al., 2020) that integrates bond angles, explicitly providing angular information. In terms of implementation, we refer to the

official DimeNet code[13] and first construct a graph of edges, where nodes represent edges and edges represent angles in the molecular graph (MG). The node features in the edge graph contain messages and lengths of the edges from the MG, while the edge features in the edge graph represent the angles. During each iteration, we first perform message passing in the edge graph to obtain the features of the vertices (which correspond to the original messages in MG), and then conduct message passing in MG. Unlike the official implementation, the original DimeNet employs targeted embedding methods for bond angles and bond lengths to compute quantum chemistry-related features. To make the bond angle model more suitable for reaction property prediction tasks, we adopt the method used in the advanced molecular property prediction model GEM (Fang et al., 2022), which employs RBF kernels for 3D feature embedding[14], with an embedding dimension of 16. We concatenate the edge attributes with the 3D feature embeddings to create new edge attributes. Other settings, such as hidden layer dimensions, iteration counts, and modules, remain consistent with our model.

**CRM.** We use the open-source code[15] provided by CRM (Gao et al., 2018) to test the results on the Pistachio-Condition dataset. Since Pistachio-Condition is similar in scale to USPTO-Condition, we refer to the hyperparameter settings in the reproduction code of Parrot (Wang et al., 2023b).

**Parrot.** We use the open-source code[16] provided by Parrot (Wang et al., 2023b). Specifically, since Parrot employs a SMILES tokenizer, and the Pistachio-Condition dataset contains vocabulary that is different from the USPTO-Condition dataset, we couldn't use the officially provided pre-trained parameters. Therefore, we first extract the vocabulary from Pistachio-Condition, then apply the RCM pre-training proposed by Parrot, and finally perform supervised training for fine-tuning on the Pistachio-Condition dataset.

**AR-GCN & CIMG.** Both methods provide corresponding open-source code[17] [18], but the network structure design of AR-GCN (Maser et al., 2021) only accommodates a specified number of reactants and products, while CIMG (Zhang et al., 2022) does not provide the corresponding training code. Therefore, we only refer to the reproduction results in Parrot (Wang et al., 2023b) for reference and do not train on Pistachio-Condition.

**D-MPNN.** We use the D-MPNN (Heid & Green, 2021) and CGR graph construction code provided in the Chemprop library for model training and refer to the official training notebook. We maintain the default settings for hyperparameters, training strategies, loss functions, and optimizers in all task training.

**Rxn Hypergraph.** The Rxn Hypergraph (Tavakoli et al., 2022) does not provide related open-source code, but the structure of the Rxn Hypergraph is described in detail in the paper. Therefore, we reproduce the construction of the Rxn Hypergraph using the DGL library[19]. Additionally, the Rxn Hypergraph paper conducts experiments using RGAT, but does not specify the exact type of RGAT used. Thus, based on the descriptions in the paper, we employ EGAT provided in DGL, which is a commonly used RGAT, for result reproduction.

**UGNN.** We use the open-source code[20] provided by UGNN (Kwon et al., 2022b) to train on the USPTO-Yield dataset. Since the molecules in the USPTO-Yield dataset are different from those in the HTE dataset, we adapt the dataset preprocessing script in the open-source code to handle the USPTO-Yield dataset and adjust the input dimension of the fully connected layer in the input layer accordingly to accommodate the reaction data from USPTO-Yield, while keeping other hyperparameters unchanged.

**DRFP.** We use the DRFP library[21] to generate DRFP fingerprints. According to the description in the original paper, we select 2048-dimensional DRFP fingerprints and train an MLP with a hidden layer dimension of 1664 and a tanh activation function.

---

[13]https://github.com/gasteigerjo/dimenet

[14]https://github.com/PaddlePaddle/PaddleHelix/tree/dev/apps/pretrained_compound/ChemRL/GEM

[15]https://github.com/Coughy1991/Reaction_condition_recommendation

[16]https://github.com/wangxr0526/Parrot

[17]https://github.com/slryou41/reaction-gcnn

[18]https://github.com/zbc0315/synprepy

[19]https://www.dgl.ai/

[20]https://github.com/seokhokang/reaction_yield_nn/

[21]https://github.com/reymond-group/drfp

**RXNFP.** We use the RXNFP[22] library to test the performance of Pistachio-Type. Similar to Parrot, we are unable to train using the original token settings of RXNFP. Therefore, we construct a vocabulary for the Pistachio-Type dataset and perform MLM pre-training on RXNFP, followed by fine-tuning for the reaction classification task. The hyperparameter settings for the model architecture remain at their default values.

**T5Chem.** We use the official open-source code[23] of T5Chem (Lu & Zhang, 2022) to train on the USPTO-yield, Pistachio-Type, and Suzuki-Miyaura datasets. The official pre-trained parameters provided by T5Chem are based on character tokenization. After checking, we find that T5Chem's vocabulary includes all characters from our datasets. Therefore, we adopt the pre-trained parameters provided by T5Chem and train on the downstream tasks for each dataset. The hyperparameters remain at default settings.

**UniMol.** We use UniMol as a comparison method in our supplementary experiments. We utilize the official code[24] and released pre-trained parameters of UniMol. Specifically, to enable UniMol to handle reaction task, we use the Transformer-based feature extraction module from UniMol, and maintain consistency with our model in subsequent modules. The hyperparameter settings for the feature extraction module align with the default values provided in the UniMol official code, while the hyperparameters for other modules are consistent with our model.

**Vector Scalarization Model.** In the supplementary experiments, in addition to testing the performance of explicitly incorporating bond angles, we also test the implementation of ENN using vector scalarization methods, specifically referring to PaiNN. Since PaiNN has not released an official implementation, we refer to a third-party implementation[25], and develop a DGL version. The original PaiNN does not support embedding of scalar edge attributes. However, due to its reliance on scalarization to achieve equivariance, we concatenate the edge attributes with the embedding of edge lengths for the message computation in PaiNN. Additionally, since the lengths of reaction edges are set to 0, they may lead to errors in embedding and vector message calculations involving division by edge lengths. Therefore, we uniformly increase all edge lengths by 1 before embedding. As edge lengths are scalar features, performing addition and subtraction operations does not affect the model's equivariance or the effectiveness of the embeddings. Furthermore, we only compute vector messages for edges other than the reaction edge, setting the vector message for the reaction edge to 0 to avoid division by zero errors. Similarly, we set the hidden layer dimensions of vector scalarization model to be consistent with our model, while setting the edge length embedding dimension to 20.

**GAT & GIN.** In the appendix, we test the performance of the GAT and GIN models. We implement GAT and GIN by utilizing the EGAT and GINE modules from DGL, where EGAT employs 8-head attention, and the GINE module adds an additional BatchNorm1D layer after each iteration to maintain numerical stability. Other hyperparameters, such as iterations, hidden layer dimensions, and subsequent modules, are consistent with our model.

**ReaMVP.** In the appendix, we test the performance of ReaMVP without large-scale yield dataset pre-training. In this way, the yield data used for training is the same as our model and other methods (4k), rather than utilizing a much larger dataset (600k) for training. This better reflects the model's performance on the benchmark, allowing for a fair comparison. Specifically, we obtained the model code and pre-trained parameters from the official repository[26], and keep the default hyperparameter setting. But since the database used by ReaMVP has become proprietary data, we are unable to acquire the dataset and reproduce their pre-training process on our model.

**Set Transformer.** We attempt to use the Set Transformer as our aggregation module. Specifically, we use the Set Transformer Encoder and Decoder implemented by DGL toolkit for feature aggregation. We set the hidden dimension to 200, attention head number to 8, number of layer to 2, for both encoder and decoder.

---

[22]https://github.com/rxn4chemistry/rxnfp
[23]https://github.com/HelloJocelynLu/t5chem
[24]https://github.com/deepmodeling/Uni-Mol
[25]https://github.com/nityasagarjena/PaiNN-model
[26]https://github.com/Meteor-han/ReaMVP

## C.3   TRAINING DETAILS

All the model training is completed on RTX 4090 using CUDA version 11.3, with PyTorch 1.12.1 and DGL 0.9.1.post1 to build the model and training framework. For the models using PyTorch Geometry, we use PyTorch Geometry 2.5.2[27] for training and reproduction. We use Scikit-Learn[28] and PyCM[29] to compute various evaluation metrics. For the replication of other works, we set up the corresponding environments using the official requirements files or guidelines provided. For the experimental models designed by us, we use the same optimizer, learning rate scheduler and training strategies.

**Leaving Group Identification.** We use the average of the cross-entropy loss for all atoms and the cross-entropy loss for leaving group atoms as the loss function for model training. The training, testing, and validation sets are distinguished according to the USPTO-Condition in an 8:1:1 ratio. A learning rate of 5e-4 is used, along with a ReduceLROnPlateau training schedule with mode set to $min$, a factor of 0.1, patience of 5, and a minimum learning rate of 1e-8. Training is conducted with a learning rate of 5e-4, a weight decay of 1e-10, and the Adam optimizer with beta values of [0.9, 0.999]. We use a batch size of 32 with 4 accumulation steps (equivalent to a batch size of 128). The training lasts for 50 epochs with early stopping applied. We choose 666 as the random seed and take the best evaluation epoch as the result, with a total training duration of approximately 6 hours.

**Reaction Condition Prediction.** We use an improved cross entropy loss to optimize the condition prediction model:

$$\mathcal{L}_{\theta,\mathcal{G}} = -\sum_{i=1}^{N_1}\sum_{j=1}^{N_{c_i}} w_{ij} \cdot c_{ij}' \log \frac{exp(z_{ij})}{\sum_{k=1}^{N_{c_i}} exp(z_{ik})}, \tag{8}$$

$$c_{ij}' = \lambda_i c_{ij} + (1-\lambda_i)/D_{c_i}, \tag{9}$$

where $w_{ij}$ is the weight of class $j$ of $i$-th type of reaction condition, $\lambda_i$ is the label smoothing factor (Szegedy et al., 2016), $z_i$ is the predicted result vector for the $i$-th reaction condition output by the model, while $c_i$ is the one-hot encoding of the ground truth for the reaction condition. We use a two-stage training strategy to train our reaction condition model, with a learning rate of 5e-4, weight decay of 1e-10, and the Adam optimizer with beta values of [0.9, 0.999] for each stage. We employ a ReduceLROnPlateau training schedule with mode set to $min$, a factor of 0.1, patience of 5, and a minimum learning rate of 1e-8. For Pistachio-Condition, we split the dataset into training, validation, and test sets in an 8:1:1 ratio. In the first stage, we train for 50 epochs, using $\lambda$ of 0.9, 0.8, 0.8, 0.7, and 0.7 for catalyst1, solvent1, solvent2, reagent1, and reagent2, respectively. For the $None$ category of solvent2 and reagent2, we set the learning rate weight $w$ to 0.1. And for USPTO-Condition dataset, we set all $\lambda$ and $w$ to 1. After completing the first stage of training, we reset and initialize the weights of the output module using a normal distribution with a mean of 0 and a standard deviation of 0.1, and freeze the feature extraction and aggregation modules for the second stage of training, which lasted for 50 epochs. We use a batch size of 32 with 4 accumulation steps (equivalent to a batch size of 128). We choose 666 as the random seed and take the best evaluation epoch as the result, with a total training duration of approximately 2 days. The training times of other models and representations on USPTO-Condition are roughly the same as the ratio of the training times we presented in Fig. 7.

**Reaction Yield Prediction.** We use the training method of BNNs (Gal & Ghahramani, 2016; Kendall & Gal, 2017) combined with L2 regularization to train the yield model:

$$\mathcal{L}_{\theta,\mathcal{G}} = (1-\gamma) * (y - \mu_y)^2 + \gamma * (\frac{(y-\mu_y)^2}{\sigma_y^2} + \log \sigma_y^2) + \lambda \sum_{M}^{M \in \theta} \|M\|_2^2, \tag{10}$$

where $y \in [0,1]$ is the ground truth yield corresponding to $\mathcal{G}$ in the dataset, $\gamma$ and $\lambda$ are two regulatory factor, $\mu_y$ and $\sigma_y$ are the model output mean and variance, and $\theta$ is the set of model parameters.

---

[27]https://pytorch-geometric.readthedocs.io/en/latest/

[28]https://github.com/scikit-learn/scikit-learn

[29]https://github.com/sepandhaghighi/pycm

The first term makes $\mu_y$ close to the ground truth yield, and the second loss is used to train $\sigma_y$ to reflect prediction uncertainty. and the last term is L2 regularization.

For the HTE dataset, our training framework remains consistent with UGNN, using a learning rate of 1e-3, weight decay of 1e-5, and the Adam optimizer with beta values of [0.9, 0.999]. For the hyperparameters of the model, we set $\gamma$ to 1e-1 and $\lambda$ to 1e-5. We train for a total of 500 epochs, shuffle the dataset at the beginning of each epoch, and employ the MultiStepLR training strategy with milestones set to [400, 450] and gamma set to 0.1. The training batch size is 32, with an accumulation step of 4 (equivalent to a batch size of 128). Training on each dataset takes approximately 20 minutes.

For the USPTO-Yield dataset, we train for 30 epochs (starts to overfit around the 20th epoch each time), modifying the MultiStepLR milestones to [20, 25]. We set $\gamma$ to 1e-1 and $\lambda$ to 0. We randomly split off 10% of the training set as a validation set and again choose 10 random seeds to take the best results. Training for Gram takes about 45 minute, while Subgram takes approximately 1.5 hours.

**Reaction Classification.** For the USPTO-TPL and Pistachio-Type datasets, we use the same training framework, employing cross entropy loss function and the Adam optimizer with learning rate of 5e-5, weight decay of 1e-10, and beta values of [0.9, 0.999]. We utilize ReduceLROnPlateau training scheduler with mode set to $min$, a factor of 0.1, patience of 5, and a minimum learning rate of 1e-8. We use a batch size of 32 with 4 accumulation steps (equivalent to a batch size of 128) and train for 100 epochs. We randomly split off 10% of the training set as a validation set and find that the dataset quality is relatively high, allowing us to achieve good performance without careful hyperparameter tuning and early stopping. Therefore, we take the final epoch as the result. We choose 123 as the random seed. The training for Pistachio-Type takes 60 hours, while USPTO-TPL takes 40 hours.

## D   ALGORITHM AND PIPELINE DETAILS

### D.1   REACTION GRAPH CONSTRUCTION

**SMILES Formatting.** To elaborate on our process of constructing RGs, we start with the most common SMILES expressions from the database. Given a SMILES expression, the reaction expression takes the form: $A>B>C\ D$, where $A$ and $C$ represent the reactants and products, respectively, and $D$ contains additional information about the reaction. Our construction of RG does not require this extra information, so we can remove part $D$. Part $B$ typically represents the reagents. In the task of reaction condition prediction, $B$ is one of the prediction targets, so we convert part $B$ into our reagent classification labels. While in the tasks of predicting reaction types and yields, such reagents will be treated as additional prior. Therefore, in these tasks, we concatenate part $B$ with part $A$ using a (.) symbol, treating it as $A$. At this point, the SMILES of the chemical reaction simplifies to $A>>C$.

**Reaction Validation.** Second, we need to verify the validity of the molecules in the reaction and the reaction formula itself. We use the RDKit to read in reactants $A$ and products $C$. If RDKit can successfully parse parts $A$ and $C$, we consider the SMILES of the reactants and products to be valid. Next, we predict the atomic mapping of the reaction by RXNMapper and validate reaction formula by checking if the atoms with mapping in the reactants and products correspond one-to-one. Although this cannot guarantee that all reactions passing the tests are valid, it significantly ensures the quality of our reaction data.

**Molecular Graph Construction.** Thirdly, we will move on to the molecular graph (MG) construction. Specifically, for each molecule in the reactants and products, we compute the attributes of each atom and bond by RDKit, as the example in Tab. 12, and use DGL to construct graph-format data. Each atom corresponds to a node, while each bond corresponds to an edge in the graph. Note that this is slightly differs from the RG we defined earlier in Sec. 3. This simplification of RG is made for the sake of ease of writing and understanding, and the actual nodes and edges in RG contain more information. In addition to the aforementioned attributes of the bond, the edge also has a flag to indicate the type of edge (as there will also be reaction edges and angular edges). After generating MGs of all molecules, we will integrate them into a single graph for easier manipulation.

**Atomic Coordinate Calculation.** We calculate the 3D coordinates of atoms in each molecule. We first attempt to optimize the conformation using MMFF94; if it fails, we switch to UFF (Casewit et al., 1992). If it still fails, we use the 2D coordinates of MG as a substitute. This ensures that

most molecules contain 3D information, while the proportion of structurally complex molecules is small, so using 2D coordinates as an approximation does not significantly impact the model's understanding of 3D information. We use RDKit and OpenBabel in this process to calculate conformers, aiming to cover as many molecules as possible.

**Angular Edge Construction.** We add angular edges to the graph to construct triangles to convey bond angle informations. Specifically, for each MG, we will find all edge pairs of the form $a, b$ and $b, c$ by traversing the adjacency list, and check if there is already an edge connecting $a, c$. If not, we will construct an angular edge connecting $a, c$ to implicitly convey the angle information of the angle $\angle abc$.

**Reaction Edges Construction.** Finally, we will construct the reaction edges. For all atoms with indices $m$ in the reactants and $n$ in products with the same mapping label, we will construct a reaction edge to connect them. This represents that the same atom before and after the reaction will have a reaction edge connecting them, facilitating the model's extraction of changes occurring to these atoms during the reaction while maintaining the model's perception of the independence of these molecules.

In the feature extraction stage, the model will only use edge length information and not coordinate information, which ensures that RG is invariant to translation and rotation of each molecule, as can be easily proven through simple derivation. Intuitively, the length information is inherently invariant to 3D rotation and translation, and our reaction edges are also independent of the rotational and translational properties between the molecules.

**Torsion Angular Edge Construction.** Torsion angular edge is an optional extension of Reaction Graph. In the molecular graph, we traverse all triplets $(ab, bc, cd)$ formed by the edges $ab$, $bc$, and $cd$, where $a$, $b$, $c$, and $d$ are distinct nodes. We then check if there is already a bond edge or angular edge $ad$ present. If not, we add a torsional angular edge $ad$ to implicitly convey the torsional angle of bond $bc$. Additionally, leveraging the advantages of the reaction graph, we can also conduct targeted designs, such as adding the torsional angular edge $ad$ only for edges $bc$ that are single bonds, to simplify calculations. This will be explored in future work.

## D.2   REACTION CONDITION PREDICTION

Due to the use of a special multi-label classification head akin to CRM for reaction condition prediction, the training and inference processes differ somewhat from traditional classification tasks. Specifically, our model needs to predict five types of reaction conditions, which are catalyst, solvent1, solvent2, reagent1 and reagent2, with the input being the representation vector of the chemical reaction along with the previous reaction conditions. As shown in Alg. 1, during training, we first use a GNN to extract features from RG, which are then aggregated into a reaction representation vector. When the model predicts each reaction condition, we provide the reaction representation vector along with the ground truth labels of the previous reaction conditions, rather than the previously predicted labels, to enhance the stability of model training.

As shown in Alg. 2, before inference, we define how many candidate labels to generate for each reaction condition $(n_1, n_2, n_3, n_4, n_5)$. For example, if we allow the model to generate 3 types of solvent1 and 5 types of reagent1, the total number of reaction conditions generated by the model would be $1 \times 3 \times 1 \times 5 \times 1$. We use the inference method of beam search. When inferring the $i$-th reaction condition, we input the reaction representation vector extracted by the model along with all combinations of the previously predicted $n_1 \times n_2 \times \cdots \times n_{i-1}$ reaction conditions into the current classification head to obtain the classification labels for the new reaction condition. The resulting $n_1 \times n_2 \times \cdots \times n_{i-1}$ labels are not one-hot vectors but rather a vector composed of floating-point numbers. We then select the top $n_i$ labels for each label based on the predefined number of candidate labels, forming $n1 \times n_2 \times \cdots \times n_{i-1} \times n_i$ combinations of reaction conditions, and continue the generation process. The specific inference algorithm is as follows:

## D.3   ATTENTION WEIGHTS VISUALIZATION SAMPLE SELECTION

As mentioned in Sec. 4, we use the attention map of the model trained on USPTO-Condition to observe the model's understanding of the reaction. We select two chemical reactions related to 3-

---

**Algorithm 1** Train Reaction Condition Prediction Model

---

1: **Input:** Reaction Graph $\mathcal{G}$, ground truth labels $\boldsymbol{c}_{gt} = [\boldsymbol{c}_1, \boldsymbol{c}_2, \ldots, \boldsymbol{c}_5]$, label smoothing coefficients $\lambda_1, \ldots, \lambda_5$, learning rate $\alpha$, model parameters $\theta$, feature extraction and aggregation module $f$, output modules $g_1, g_2, \ldots, g_5$
2: **Output:** trained model parameters $\theta'$
3: **Constant:** dimension of reaction condition labels $D_1, \ldots, D_5$
4:
5: **procedure** TrainConditionModelOneIter($\mathcal{G}, \boldsymbol{c}_{gt}, \alpha, \theta, \lambda_1, \ldots, \lambda_5$)
6:     **let** $\boldsymbol{r} \leftarrow f(\mathcal{G}; \theta)$
7:     **let** $\hat{\boldsymbol{c}} = [\hat{\boldsymbol{c}}_1, \hat{\boldsymbol{c}}_2, \ldots, \hat{\boldsymbol{c}}_5]$ be zero vector like $\boldsymbol{c}$
8:     **let** $loss \leftarrow 0$
9:     **for** $i$ **from** 1 **to** 5 **do**
10:        **let** $\boldsymbol{s}_i \leftarrow \lambda_i \boldsymbol{c}_i + (1 - \lambda_i)/D_i$
11:        **let** $\hat{\boldsymbol{c}}_i \leftarrow g_i(\boldsymbol{r}, \boldsymbol{c}_1, \ldots, \boldsymbol{c}_{i-1}; \theta)$
12:        $loss \leftarrow loss+$ CrossEntropy($\boldsymbol{s}_i, \hat{\boldsymbol{c}}_i$)
13:     **end for**
14:     **let** $\theta' \leftarrow \theta - \alpha \cdot \nabla_\theta loss$
15:     **return** $\theta'$
16: **end procedure**

---

---

**Algorithm 2** Reaction Condition Beam Search Inference

---

1: **Input:** Reaction Graph $\mathcal{G}$, candidate labels $(n_1, n_2, n_3, n_4, n_5)$, model parameters $\theta$, feature extraction and aggregation module $f$, output modules $g_1, g_2, \ldots, g_5$
2: **Output:** Predicted reaction condition combinations $\boldsymbol{C} \in \mathbb{R}^{(n_1 n_2 \ldots n_{i-1}) \times 5}$
3:
4: **procedure** ConditionBeamSearch($\mathcal{G}, \boldsymbol{r}, (n_1, n_2, n_3, n_4, n_5), \theta$)
5:     **let** $\boldsymbol{r} \leftarrow f(\mathcal{G}; \theta)$
6:     **let** $\boldsymbol{c}_1 \leftarrow g_i(\boldsymbol{r}; \theta)$
7:     **let** $c_{11}, \ldots, c_{1n_1}$ be top $n_1$ of $\boldsymbol{c}_1$
8:     **let** $\mathcal{C}$ be $[[c_{11}], \ldots, [c_{1n_1}]]$
9:     **for** $i$ **from** 2 **to** 5 **do**
10:        **let** $\mathcal{C}_{new}$ be a empty list
11:        **for** $j$ **from** 0 **to** $\mathcal{C}.length - 1$ **do**
12:           **let** $\boldsymbol{c}_i \leftarrow g_i(\boldsymbol{r}, \mathcal{C}[j]; \theta)$
13:           **let** $c_{i1}, \ldots, c_{in_i}$ be top $n_i$ of $\boldsymbol{c}_i$
14:           **for** $k$ **from** 1 **to** $n_i$ **do**
15:              **let** $\boldsymbol{c}_{new} \leftarrow Concatenate([\mathcal{C}[j], [c_{ik}]])$
16:              **append** $\boldsymbol{c}_{new}$ **to** $\mathcal{C}_{new}$
17:           **end for**
18:        **end for**
19:        $\mathcal{C} \leftarrow \mathcal{C}_{new}$
20:     **end for**
21:     **let** $C \leftarrow$ Stack($\mathcal{C}$)
22:     **return** $C$
23: **end procedure**

---

Amino-5-bromobenzoic acid as example, and we obtained these example molecules and reactions through an automated script.

Specifically, we first set up a series of frequently occurring functional groups based on our observation criteria, such as chlorine atoms, bromine atoms, hydroxyl groups, carboxyl groups, and amino groups. For all reactant molecules that appear in the reaction dataset, we filter out those that possess two or more different functional groups from the aforementioned list. We then find all reactions involving these molecules as reactants in the validation and test set.

Secondly, based on the predicted mapping, we identify the reaction centers of these reactions (i.e., the atoms where nearby bonds undergo breaking and forming changes) and locate the functional

groups associated with the reaction centers. If there are molecule with two different reaction centers in two reactions, we select that molecule as observation subject. Ultimately, we filter out 3-Amino-5-bromobenzoic acid and the related two reactions.

### D.4 LEAVING GROUP EXTRACTION

We use a dataset labeled with Leaving Groups (LvGs) for supervised training in the LvG identification task. We obtain the LvG dataset based on the processed RG dataset. First, for each RG in the dataset, we remove all reaction edges and the nodes at both ends of these edges. This step helps us eliminate all non-LvG atoms that appear in both the reactants and products, leaving the remaining connected subgraphs as the LvGs. We assign temporary labels to these connected subgraphs and mark this label on the original RG nodes for later replacement with the LvG labels.

Next, we collect all the LvGs extracted from the entire dataset to form a LvG list and perform graph hashing on these LvG subgraphs. Since our graphs are stored in DGL format, we use a GNN with a randomization parameter to perform the graph hashing operation, ultimately generating a 128-dimensional hash value for each graph. We count all the hash values that occur and identify LvGs with frequencies above threshold, resulting in 204 types of LvG labels, while other LvGs are uniformly assigned a $Other$ label.

Finally, we build a mapping between temporary labels and LvG labels through LvG list, and update the labels in the original RG to the LvG labels, with assigning a label of $None$ to atoms that do not belong to LvGs. Thus, we complete the creation of the LvG dataset.

### D.5 REACTION BIN-PACKING

We use an algorithm similar to bin-packing to construct reaction bins in the experiment of testing the model's runtime on the USPTO-Condition. We first specify a series of atom quantities and place reactions into $N$ bins of the same capacity, aiming to make the sum of their atom quantities as close as possible to the bin's capacity. A greedy algorithm is designed to solve this problem, as shown in Alg. 3.

## E COMPLEXITY ANALYSIS AND TEST

### E.1 THEORETICAL ANALYSIS OF COMPUTATIONAL EFFICIENCY

**Number of Edges.** The factors affecting computational complexity of Reaction Graph (RG) include the specific implementation of the graph, as well as the number of atoms $N$, the number of bonds $E$, and the maximum degree $D$ of the nodes. The number of reaction edges cannot exceed half the number of graph nodes $\frac{V}{2}$. This is because a reactant atom can match at most one product atom, meaning each node in the graph can be connected to at most one reaction edge. And the number of angular edges is at most $ND^2$, which is comparable to the theoretical computational complexity of DimeNet. However, the message passing process of angular edges is conducted simultaneously with other edges, which enhances parallelism, while in DimeNet, the computation of angular information and the message passing are conducted in two separate steps. Therefore, if $K$ message passing steps involving angluar information are performed, the time complexity for DimeNet is approximately $2K$, while angular edges only require $K + 1$. Additionally, since RG connects multiple molecules with different roles in to a unified graph, employing ENNs like PaiNN in molecular tasks may struggle to maintain equivariance in RG. In contrast, invariance is easier to ensure. When using invariant features, it is also important to convey multi-body characteristics, and angular edges meet these requirements while being relatively computationally efficient. Last but not least, we believe that explicitly adding edges to convey multi-body features is a worthwhile area of research. Although, in theory, our simple attempt is still not as computationally efficient as methods like ComENet, it requires minimal modifications to the GNN architecture. Furthermore, it remains to be explored whether we can adding angular edges only at critical positions to convey key information, which is not feasible for methods like DimeNet.

If we consider only the reaction edges, the computation of RGs is efficient compared to other graph representation improvements, given the same level of representational capability. First, the number

**Algorithm 3** Reaction Bin-Packing

1: **Input:** Number of bins $N$ and capacity of each bin $C$, number of atoms of reactions $\mathcal{A} = [a_1, a_2, \ldots, a_M]$
2: **Output:** Indices of reactions in each bin $\mathcal{B} = [\mathcal{L}_1, \mathcal{L}_2, \ldots, \mathcal{L}_N]$
3:
4: **procedure** ReactionBinPacking($N, C, \mathcal{A}$)
5:    **shuffle** $\mathcal{A}$
6:    Initialize $\mathcal{B}$ as a list of lists of length $N$
7:    Initialize $\mathcal{W}$ as a zero list of length $N$
8:    **for** each reaction $a_i$ **in** $\mathcal{A}$ **do**
9:       **let** $d_{min} \leftarrow 0$
10:      **let** $k \leftarrow 0$
11:      **for** $j \leftarrow 1$ **to** $N$ **do**
12:         $d \leftarrow C - (\mathcal{W}[j] + a_i)$
13:         **if** $d < d_{min}$ and $d > 0$ **do**
14:           $d_{min} \leftarrow d$
15:           $k \leftarrow j$
16:         **end if**
17:      **end for**
18:      **if** $k > 0$ **do**
19:         **push** $i$ into $\mathcal{B}[k]$
20:         $\mathcal{W}[k] \leftarrow \mathcal{W}[k] + a_i$
21:      **end if**
22:    **end for**
23:    **return** $\mathcal{B}$
24: **end procedure**

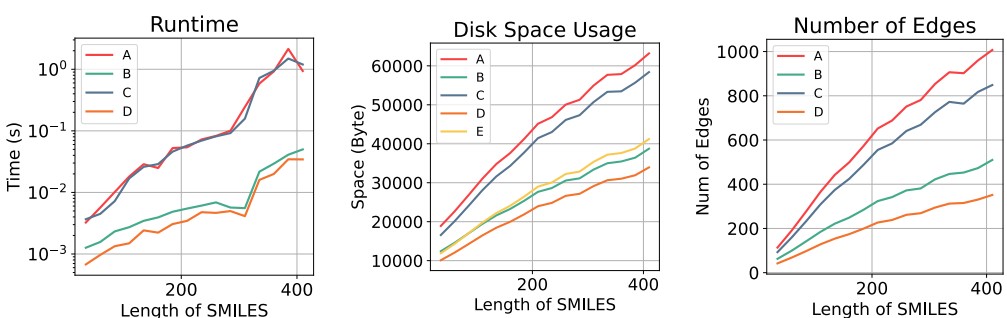

Figure 6: Graph construction algorithm metrics of different graph representations, including (A) RG, (B) RG with only reaction edges, (C) RG with only 3D information and angular edges , (D) molecular graph (MG), and (E) MG with 3D information. Due to the overlap between groups (C) and (E) of computation time, and since the number of edges in group (D) is equal to that in group (E), group (E) is omitted in both figures.

of edges in a RG is significantly lower than that in Rxn Hypergraph, with Rxn Hypergraph having at least $E + V$ edges, while RG has at most $E + \frac{V}{2}$. Compared to Condense Graph of Reaction (CGR), although it has fewer nodes and edges (approximately half the size when there are not many leaving group atoms), each node and edge in CGR actually needs to contain information from both the reactants and products, effectively doubling the feature dimensions. Additionally, for leaving group atoms, the edge and node features may also contain slight redundancies.

**Computational Complexity.** We first define:

- $V$: Number of atoms
- $M$: Number of molecules
- $E$: Number of chemical bonds

Table 14: Data pre-processing time for different representations. The molecular graph can be viewed as a reaction graph without reaction information and 3D structure information. Time units are milliseconds (ms).

| Representation | Information | | Total Length of Reaction SMILES | | | | | | | | | | | | |
|---|---|---|---|---|---|---|---|---|---|---|---|---|---|---|---|
| | Reac. | 3D | 35 | 60 | 85 | 110 | 135 | 160 | 185 | 210 | 235 | 260 | 285 | 310 | 335 | 360 |
| Reaction Graph | ✔ | ✔ | 3.2 | 5.6 | 10.1 | 18.1 | 28.7 | 24.9 | 52.5 | 54.0 | 72.4 | 82.8 | 100.3 | 244.4 | 585.6 | 923.3 |
| w/o Reaction Edge | ✘ | ✔ | 3.6 | 4.4 | 7.2 | 16.8 | 26.0 | 28.8 | 45.9 | 57.5 | 69.3 | 81.3 | 91.9 | 157.8 | 722.1 | 945.9 |
| w/o Angular Edge | ✔ | ✘ | 1.2 | 1.5 | 2.3 | 2.7 | 3.5 | 3.9 | 4.8 | 5.4 | 6.1 | 6.8 | 5.6 | 5.5 | 21.7 | 29.3 |
| Molecular Graph (w/o 3D) | ✘ | ✘ | 0.7 | 1.0 | 1.3 | 1.5 | 2.4 | 2.2 | 3.0 | 3.4 | 4.7 | 4.6 | 4.9 | 4.1 | 15.9 | 19.9 |
| Molecular Graph (w/ 3D) | ✘ | ✔ | 2.9 | 4.1 | 7.3 | 16.5 | 23.5 | 26.8 | 49.5 | 54.7 | 69.7 | 82.4 | 90.5 | 156.4 | 718.7 | 932.1 |

- $D$: Maximum degree of atomic nodes
- $L$: Length of the molecular SMILES string
- $H$: Hidden feature dimension of atoms
- $A$: Number of atoms in a graph node
- $N$: Number of iterations for the GNN/Transformer
- $P$: Number of message passing in a GNN iteration

We simply assume that the messages are calculated by passing the hidden features of atoms through a fully connected layer.

- **Molecular Grpah.** Since each edge needs to calculate the message once in each GNN interaction, and the complexity of fully connected layer is $O(H^2)$, each node in molecular graph represents one atom, the complexity of the original Molecular Graph is $O(ENH^2)$.

- **Reaction Graph (ours).** Reaction Graph introduces two types of edges: reaction edges and angular edges.
  - Reaction Edge: The number of reaction edges $E_{reaction}$ cannot exceed the number of product atoms $V_{product}$, and the number of product atoms cannot exceed half of the total atoms, so $E_{reaction} = V_{product} \leq V/2$. Therefore, the computational cost introduced by the reaction edges is $O(V/2 \cdot NH^2) = O(VNH^2)$.
  - Angular Edge: Since a bond angle is formed between every two neighboring edges of each atom node, and the number of bond angles is equal to the number of angle edges. Therefore, the number of angle edges $E_{angular} = VD^2$. Therefore, the computational cost introduced by the angular edges is $O(VD^2 \cdot NH^2)$.

  The overall complexity of Reaction Graph is $O[(E + V + VD^2) \cdot NH^2] = O[(E + VD^2) \cdot NH^2]$. When 3D information is not used, the complexity of the Reaction Graph is $O[(E + V) \cdot NH^2]$.

- **Bond Angle Model.** Our proposed angular edges are more efficient than explicitly using bond angles. With the angular edge method, there is only one message passing per round of GNN iteration. In contrast, when angle features are explicitly integrated, each GNN iteration requires $P$ message passing operations. The complexity is $O[(E+VD^2) \cdot PNH^2]$.

- **Rxn Hypergrpah.** Rxn Hypergraph adds edges connecting each atom to the corresponding molecule hypernode, introducing a computational cost of $O(VNH^2)$. The molecule nodes between reactants are connected pairwise, all linking to the reactant hypernode. And it is similar for the products. The cost for these new edges are $O[(M^2 + M) \cdot NH^2] = O(M^2 \cdot NH^2)$. Finally, the cost for Rxn Hypergrpah is $O[(E + V + M^2) \cdot NH^2]$

- **Condense Graph of Reaction.** In Condense Graph of Reaction (CGR), each node represents $A$ atoms from the original graph, and each edge represents $A$ edges from the original graph. Therefore, the number of edges in the CGR becomes $E/A$, and the dimension of the hidden layer for the nodes becomes $AH$. Thus, the time complexity of the CGR is $O[E/A \cdot N \cdot (AH)^2] = O(E \cdot NAH^2)$.

- **SMILES-based Methods.** SMILES-based methods typically use Transformers as the backbone. In Transformers, the time complexity for each component is:
  1. Tokenization: The cost of regex tokenizer is $O(L)$

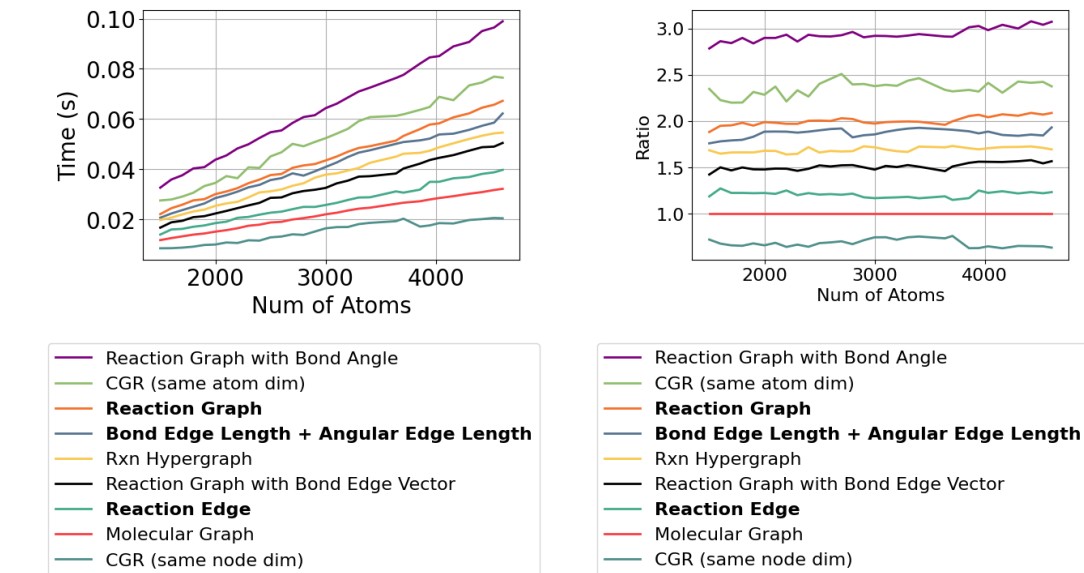

Figure 7: Computational time of models and graph representations at different atom counts. Legend order matches line order.

Figure 8: Computational time ratios of models and graph representations in relation to molecular graph at different atom counts.

2. Embedding: The cost of embedding layer is $O(LH)$

3. Attention: The size of the attention matrix is $L^2$, so the complexity for attention matrix calculation is $O(L^2H)$

4. Feed Forward: Each token's embedding is processed through a fully connected layer, so the time complexity is $O(LH^2)$

And there's $N$ layers of Attention and Feed Forward. The overall time complexity of Transformer is $O[L + LH + N \cdot (LH^2 + L^2H)] = O[N \cdot (LH^2 + L^2H)]$

### E.2 EXPERIMENTS OF CONSTRUCTION AND COMPUTATION

**Construction.** As shown in Fig. 6, the construction time for all graphs exhibits a more than linear increase with the length of SMILES strings, which is consistent with our theoretical derivations. The construction time for RG with angular edges is almost the same as that for 3D MG, so the curve is omitted. In RG, the additional overhead of computing 3D structures is the most significant, approximately 10-100 times that of other parts. Regarding hard disk space usage, the additional space occupied by angular edges is the most noticeable, while the extra space overhead caused by reaction edges and conformation storage is relatively small. This corresponds to the situation with the number of edges, where the number of angular edges is significantly more than other types of edges, about 2-3 times the number of edges in the original MG.

Tab. 14 shows the influence of reaction edges, angular edges, and bond edge length on data pre-processing time in RG, as the total length of reaction SMILES increases. During data pre-processing, the primary time cost comes from 3D conformer calculation. But it is an unavoidable cost for any method that utilizes 3D structure information.

**Computation.** We also test the computational efficiency of different graph representations and model architectures. To ensure a fair comparison, we use the same GNN architecture (except for the experimental group that uses bond angles and bond vectors). Our experimental group includes those mentioned in the Sec. 4, as well as the newly introduced groups, including: (1) Molecular Graph, (2) Reaction Edge, (3) Bond Edge Length + Angular Edge Length, (4) Reaction Graph (with Angular Edge Length), (5) Reaction Graph with Bond Angle, (6) Reaction Graph with Bond Edge Vector, (7) Rxn Hypergraph, (8) CGR with same atomic feature dimension with MG, (9) CGR with same

Table 15: Model inference time for different representations on the same backbone. Time units are milliseconds (ms).

| Representation | Information | | Number of atoms $|V|$ in Reaction Graph | | | | |
|---|---|---|---|---|---|---|---|
| | Reaction Infomation | 3D Structure | 2500 | 3000 | 3500 | 4000 | 4500 |
| Reaction Graph | ✓ | ✓ | 37.28 | 43.77 | 50.57 | 58.03 | 65.45 |
| w/o Reaction Edge | ✗ | ✓ | 35.36 | 40.92 | 48.68 | 53.30 | 58.79 |
| w/o Angular Edge | ✓ | ✗ | 22.51 | 25.96 | 29.82 | 34.21 | 38.77 |
| Molecular Graph | ✗ | ✗ | 18.59 | 21.93 | 25.39 | 28.30 | 31.48 |
| Rxn Hypergraph | ✗ | ✗ | 31.18 | 37.28 | 43.31 | 48.53 | 54.06 |
| Condensed Graph of Reactions | ✓ | ✗ | 44.55 | 52.48 | 60.75 | 66.32 | 75.34 |

Table 16: Inference time comparison for Reaction Graph and SMILES-based methods. According to the result, Reaction Graph requires smaller computation cost.

| Methods | Representation | Time |
|---|---|---|
| T5Chem | SMILES | 3min 19s |
| RXNFP | SMILES | 23min 43s |
| Reaction Graph (ours) | Reaction Graph | **2min 36s** |

node feature dimension with MG. Note that in CGR, one node is the superposition of two atoms, and one edge in the superposition of two bonds.

Results are shown in Fig. 7 and 8. Based on observations, the improvement of reaction edges in RG is efficient compared to other two-dimensional representations. It connects chemical reactions as a whole without compressing the atomic property dimensions, ensuring the relative independence of molecules, while only introducing about $20\%$ additional computational overhead. In contrast, Rxn Hypergraph introduces more edges, increases the computational time by $60\%$. CGR, while having the same atomic property dimensions, shows a significant increase in overhead by $130\%$. However, we also find that CGR, despite may lose some expressive ability with the same node dimensions, only requires $70\%$ of the computational time of MG, which is suitable for real-time applications and large-scale training.

Compared to other 3D representations, angular edge are more efficient than explicitly introducing bond angle, as concluded in Sec. 4. Although the computational efficiency slightly lags behind bond vector, which only introduces 50% computational overhead, the equivariance of bond vector and the efficiency of ENN architectures have not yet been proven in reaction systems, resulting in suboptimal performance in Sec. G. In contrast, angular edge significantly enhance performance while maintaining moderate computational overhead.

**Compared with Baselines.** We compare the inference time of Reaction Graph and other graph-based representations on the same backbone. The results are shown in Tab. 15. Due to incorporating additional reaction information and 3D structures, our method requires slightly more time than molecular graphs and Rxn Hypergraph representations. However, it delivers significant performance improvement, justifying the additional time cost. The CGR introduces larger computation overhead because each of its nodes is a combination of two atoms, requiring larger hidden layer dimensions in the neural network.

We compare the inference time of Reaction Graph and smiles-based methods. The results are shown in the table below, where Reaction Graph, as a graph-based model, exhibits the shortest inference time.

## F    DATASET PREPROCESSING AND ANALYSIS

For each dataset, we test various metrics related to dataset distribution and complexity, including the frequency of each label, the frequency of each atom, the total number of atoms and molecules, the total number of atoms per molecule and per reaction and so on. For clarity, we sort all data categories by frequency in ascending order except for the yield dataset, and use a logarithmic scale. Due to the large number of data points, we use line charts to display the data distribution. In each chart, the

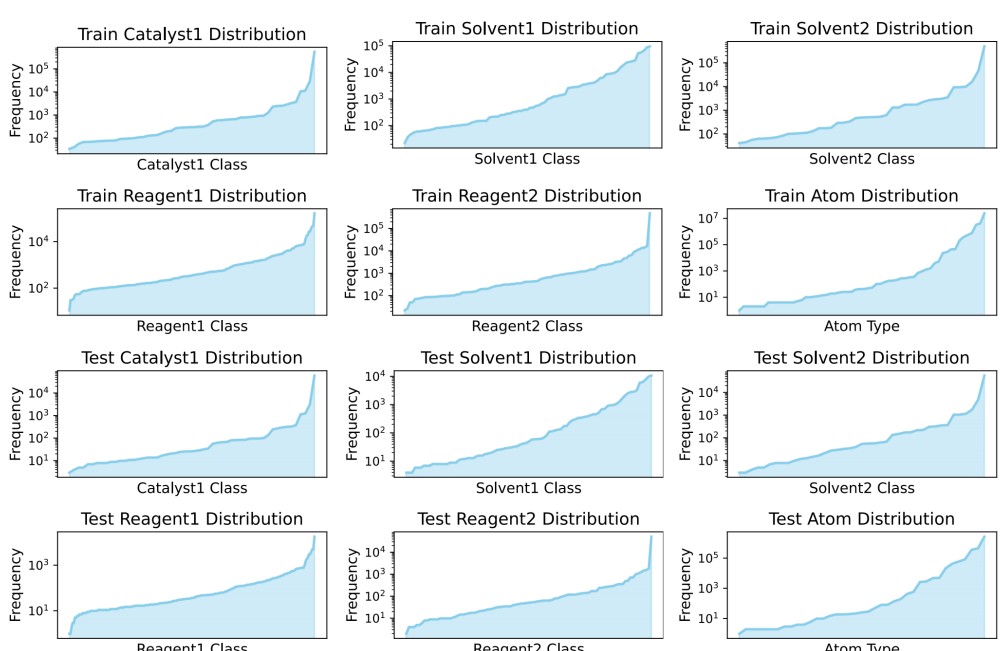

Figure 9: Data distributions of USPTO-Condition, including distribution of the number of data points for each category of reaction conditions, as well as the distribution of the occurrence of atom types.

Table 17: Summary of USPTO-Condition, including metrics for measuring the complexity of data points, the overall complexity of the dataset, and the distribution differences between the train/val set and the test set.

| Metric | Train/Val | Test |
|---|---|---|
| Size | 612,666 | 68,075 |
| Mean Length | 96.55 | 98.44 |
| Mean Atoms | 18.20 | 18.53 |
| Mean Molecules | 2.88 | 2.89 |
| Max Length | 868 | 825 |
| Max Atoms | 347 | 280 |
| Atoms | 60 | 39 |
| Mean Atoms per Molecule | 24.22 | 22.92 |
| Max Atoms per Molecule | 176 | 165 |
| Max Molecules | 55 | 24 |
| Molecules | 849,149 | 135,066 |
| Different Molecules (Train/Val and Test) | 787,898 | |
| Num Catalyst 1 | 54 | |
| Num Solvent 1 | 85 | |
| Num Solvent 2 | 41 | |
| Num Reagent 1 | 223 | |
| Num Reagent 2 | 95 | |

x-axis represents the data categories, while the y-axis represents the frequency of that category in the dataset.

Table 18: Changes in the dataset size of USPTO-Condition during the preprocessing process.

| Stage | Num of Data Rows |
|---|---|
| Reaction Data Extraction | 3,130,812 |
| Mapping and duplication Dropping | 1,117,867 |
| Filter Conditions | 680,741 |
| Split Train/Val/Test | 544,591 : 68,075 : 68,075 |
| Graph Dataset Generation | 544,125 : 68,018 : 68,002 |

## F.1 USPTO-CONDITION

The USPTO-Condition dataset is a collection of reaction condition data extracted from the entire USPTO database. We use scripts provided in Parrot (Wang et al., 2023b) to process the raw dataset. First, for each reaction data in USPTO with SMILES, we extract all corresponding reaction condition annotations. Then, since the SMILES in USPTO do not contain mapping, the RXNMapper tool is used to predict atomic mapping for the reactions. Molecules in reactants that do not contain any mapping markers are considered reagents. During this process, all SMILES that cannot be parsed or whose mappings do not correspond are discarded, and duplicated data rows are dropped. Afterward, we count the frequency of each catalyst, solvent, and reagent, and remove data with condition frequencies below a certain threshold 100. We check reagents with ions and removed non-electrically neutral combinations. Then, to standardize the output format, we delete all data with catalyst $> 1$, solvent $> 2$, reagent $> 2$, and those without reaction conditions. Finally, the dataset is split into training, validation, and test sets. The various metrics of the dataset are shown in Tab. 17, while the distributions of data are shown in Fig. 9.

Based on the divided USPTO-Condition, we use the Reaction Graph (RG) construction algorithm in Sec. D to build a graph dataset. For molecules that could not generate conformations, we try two strategies: using 2D graphs as substitutes and directly discarding them. Considering the size of the dataset and the proportion of molecules that could not generate conformations, we believe that directly discarding them is simple and reasonable. The number of remaining data samples at each stage is in Tab. 18.

## F.2 PISTACHIO-CONDITION

For the Pistachio database, we adopt the same approach as with the USPTO-Condition dataset. The difference is that most reactions in Pistachio come with mapping information and are also annotated with atmosphere labels. Therefore, during the mapping step, we only predict the missing mapping information. In the filter step, since atmosphere labels are too sparse to be suitable for classification, and to maintain consistency with the USPTO-Condition dataset, we discard all reaction entries with atmosphere labels. Due to the differences in label distribution between the Pistachio and USPTO, we filter condition label earlier and do not use a set frequency threshold to filter reaction conditions simultaneously. Instead, we prioritize selecting catalysts, then solvents from the remaining data, and finally reagents. This approach maximizes the retention of the relatively sparse catalyst data. The thresholds we set are as follows. Finally, we also split the dataset into 8:1:1 for training, testing, and validation. The various metrics of the dataset are shown in Tab. 19, while the distributions of data are shown in Fig. 10.

Similarly, based on the split dataset, we further generate the RG dataset. And the number of remaining data samples at each stage is as shown in Tab. 20.

Based on the data, we find that the original dataset of Pistachio is quite large, but it also includes some duplicate reactions and reactions without condition labels. The final dataset size is similar to that of USPTO-Condition. However, when observing the complexity of the dataset, we notice that the reaction conditions in Pistachio-Condition are sparse, with more extreme data points; for instance, the largest molecules have over 400 atoms, and the average data length and number of atoms are also greater. Additionally, the distribution of the dataset is relatively uneven. The reason the model achieves higher classification accuracy on Pistachio-Condition is also due to the uneven

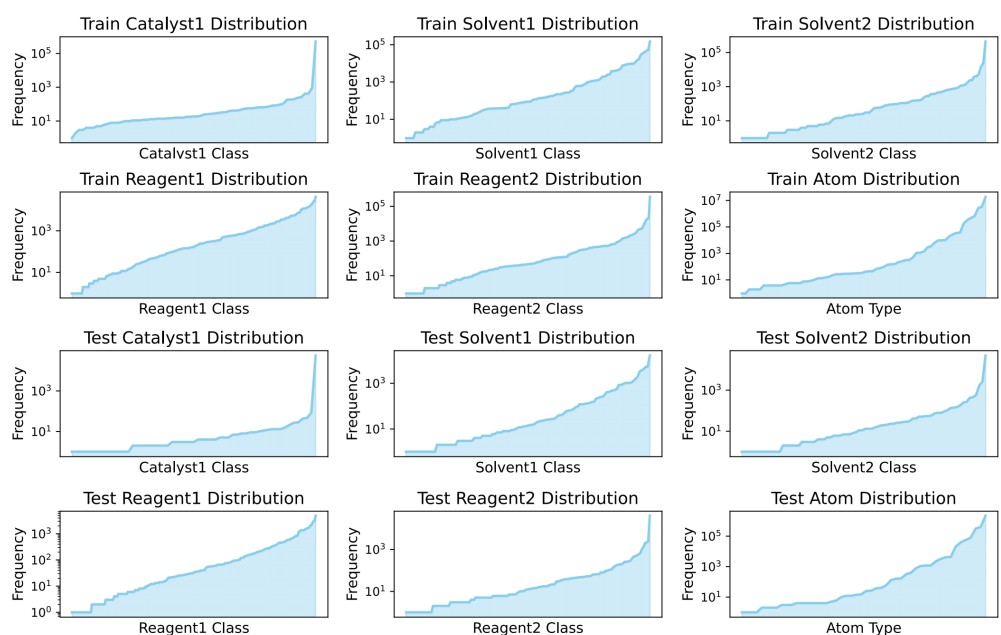

Figure 10: Data distributions of Pistachio-Condition, including distribution of the number of data points for each category of reaction conditions, as well as the distribution of atom types.

Table 19: Summary of Pistachio-Condition dataset, including metrics for measuring the complexity of data points, the overall complexity of the dataset, and the distribution differences between the train/val set and the test set.

| Metric | Train/Val | Test |
|---|---|---|
| Size of Dataset | 506,224 | 56,247 |
| Mean Length per Reaction | 99.66 | 99.93 |
| Mean Atoms per Reaction | 17.44 | 17.48 |
| Atom Types | 68 | 50 |
| Mean Atoms per Molecule | 24.27 | 22.73 |
| Max Atoms of Molecule | 218 | 209 |
| Mean Molecules per Reaction | 3.04 | 3.04 |
| Max Length of Reaction | 988 | 878 |
| Max Atoms of Reaction | 440 | 408 |
| Max Molecules of Reaction | 54 | 18 |
| Molecule Types | 761,357 | 110,771 |
| Different Molecules (Train/Val and Test) | 718,036 | |
| Catalyst1 Types | 69 | |
| Solvent1 Types | 134 | |
| Solvent2 Types | 108 | |
| Reagent1 Types | 267 | |
| Reagent2 Types | 198 | |

distribution of data labels, where the $None$ class accounts for the majority, which can be observed in the confusion matrix from Sec. G.

Table 20: Changes in the dataset size of Pistachio-Condition during the preprocessing process.

| Stage | Num of Data Rows |
|---|---|
| Reaction Data Extraction | 145,035,928 |
| Filter Conditions | 1,981,125 |
| Mapping and duplication Dropping | 562,471 |
| Split Train/Val/Test | 449,977 : 56,247 : 56,247 |
| Graph Dataset Generation | 449,902 : 56,240 : 56,234 |

Table 21: Summary of Buchwald-Hartwig on metrics reflecting dataset complexity.

| Metric | Value |
|---|---|
| Size of Dataset | 3,955 |
| Mean Length pre Reaction | 216.65 |
| Mean Atoms pre Reaction | 116.15 |
| Max Length pre Reaction | 300 |
| Max Atoms of Reaction | 146 |
| Molecules pre Reaction | 7 |
| Molecule Types | 51 |
| Atom Types | 10 |
| Mean Atoms per Molecule | 13.20 |
| Max Atoms of Molecule | 46 |

Table 22: Summary of Suzuki-Miyaura on various metrics reflecting dataset complexity.

| Metric | Value |
|---|---|
| Size of Dataset | 5,760 |
| Mean Length pre Reaction | 196.925 |
| Mean Atoms pre Reaction | 110 |
| Max Length pre Reaction | 270 |
| Max Atoms of Reaction | 144 |
| Mean Molecules pre Reaction | 12.14 |
| Max Molecules pre Reaction | 15 |
| Molecule Types | 43 |
| Atom Types | 16 |
| Mean Atoms per Molecule | 12.14 |
| Max Atoms of Molecule | 42 |

## F.3 HTE

We use the dataset provided by UGNN (Kwon et al., 2022b), which contains the SMILES expressions of reactions and yield data. Due to the small number of molecules included, it is difficult for models to learn from the limited conformational data. Therefore, we only generate RGs with reaction edges and did not discard any data. Consequently, the final dataset size and metrics are as shown in Tab. 21 and 22.

From the data, we can observe that the number of data points in the HTE dataset is nearly a thousand times smaller than that in our previous datasets. The variety of molecules is relatively limited, and the complexity of the data points is not high, with all data points belonging to the same reaction. The distribution of the training and testing sets in the original dataset is basically consistent, so we do not list them separately. In the above data, all data points in the Buchwald-Hartwig dataset involve six molecules as reactants and reagents, while the products consist of a single molecule. In contrast, the Suzuki-Miyaura dataset may involve a different number of molecules for each reaction, and its overall complexity is higher than that of the Buchwald-Hartwig dataset. Previous work also proposes using Test datasets with greater distribution differences between the training and testing sets to evaluate the model's generalization performance. Tab. 23 lists some metrics for the Test datasets.

For Test datasets, we can clearly see significant differences between the training and testing sets, which also include different types of molecules. This requires the model to accurately model the relationship between molecules and reaction structures in order to obtain reasonable extrapolation results.

## F.4 USPTO-YIELD

We use the USPTO-Yield dataset provided in Yield-Bert (Schwaller et al., 2021b), which is extracted from the USPTO database and is relatively noisy with a complex distribution. The dataset is processed similarly as described above, with some specific metrics shown in Tab. 24 and 25, and the distribution of data shown in Fig. 11.

Table 23: Summary of Test datasets of Buchwald-Hartwig, including metrics for measuring the complexity of data points, the overall complexity of the dataset, and the distribution differences between the train/val set and the test set.

| Metric | Test1 | | Test2 | | Test3 | | Test4 | |
|---|---|---|---|---|---|---|---|---|
| | train | test | train | test | train | test | train | test |
| Mean Length per Reaction | 213.23 | 224.62 | 214.61 | 221.41 | 214.53 | 221.60 | 214.55 | 221.54 |
| Mean Atoms per Reaction | 16.40 | 17.05 | 16.56 | 16.67 | 16.53 | 16.74 | 16.48 | 16.85 |
| Max Length of Reaction | 289 | 300 | 300 | 300 | 300 | 300 | 300 | 300 |
| Max Atoms of Reaction | 138 | 146 | 146 | 146 | 146 | 146 | 146 | 146 |
| Molecule Types | 45 | 38 | 46 | 38 | 46 | 38 | 46 | 37 |
| Mean Atoms per Molecule | 13.18 | 14.50 | 13.61 | 13.95 | 13.54 | 14.03 | 13.30 | 14.16 |
| Max Atoms per Molecule | 46 | 46 | 46 | 46 | 46 | 46 | 46 | 46 |
| Different Molecules (Train/Val and Test) | 13 | | 13 | | 13 | | 14 | |
| Molecule per Reaction | 7 | | | | | | | |
| Atom Types | 10 | | | | | | | |
| Size of Dataset | 3955 | | | | | | | |

Table 24: Summary of Gram dataset, including metrics for measuring the complexity of data points, the overall complexity of the dataset, and the distribution differences between the train/val set and the test set.

| Metric | Train/Val | Test |
|---|---|---|
| Size of Dataset | 156,565 | 39,137 |
| Mean Length per Reaction | 115.97 | 116.03 |
| Mean Atoms per Reaction | 60.44 | 60.46 |
| Mean Molecules per Reaction | 6.63 | 6.63 |
| Max Length of Reaction | 560 | 493 |
| Max Atoms of Reaction | 257 | 244 |
| Mean Atoms per Molecule | 20.27 | 19.33 |
| Max Atoms of Molecule | 116 | 103 |
| Atom Types | 67 | 57 |
| Max Molecules of Reaction | 59 | 32 |
| Molecule Types | 230,450 | 74,116 |
| Different Molecules (Train/Val and Test) | 197,580 | |

Table 25: Summary of Subgram dataset, including metrics for measuring the complexity of data points, the overall complexity of the dataset, and the distribution differences between the train/val set and the test set.

| Metric | Train/Val | Test |
|---|---|---|
| Size of Dataset | 240,326 | 60,075 |
| Mean Length per Reaction | 150.24 | 150.67 |
| Mean Atoms per Reaction | 79.13 | 79.35 |
| Mean Molecules per Reaction | 6.88 | 6.89 |
| Max Length of Reaction | 696 | 641 |
| Max Atoms of Reaction | 352 | 260 |
| Mean Atoms per Molecule | 25.88 | 24.91 |
| Max Atoms of Molecule | 166 | 143 |
| Atom Types | 67 | 57 |
| Max Molecules of Reaction | 38 | 29 |
| Molecule Types | 400,811 | 123,077 |
| Different Molecules (Train/Val and Test) | 353,890 | |

According to the data, we find that the yield distribution in USPTO-Yield is quite uneven, and there is a significant difference in the yield distributions between Gram and Subgram, as noted in the original Yield-Bert paper (Schwaller et al., 2021b). Additionally, although both the Gram and Subgram datasets contain a considerable number of entries and the length and complexity of the reaction data points appear to be low, as mentioned in Yield-Bert, the quality of the yield data cannot be guaranteed, and there is considerable complexity in the yields. For reactions of the same type, the variance in yields is relatively large, which further increases the task difficulty.

## F.5  USPTO-TPL

In the USPTO-TPL dataset from RXNFP (Schwaller et al., 2021a), we find from Fig. 12 and Tab. 26 that the data distribution is also complex, with uneven labels and a certain scale. Compared to other datasets extracted from USPTO, their data point and label distribution are quite similar. However, due to the labels being extracted based on 1,000 templates, the model can easily infer the relationship between the reactions and the labels, resulting in lower difficulty.

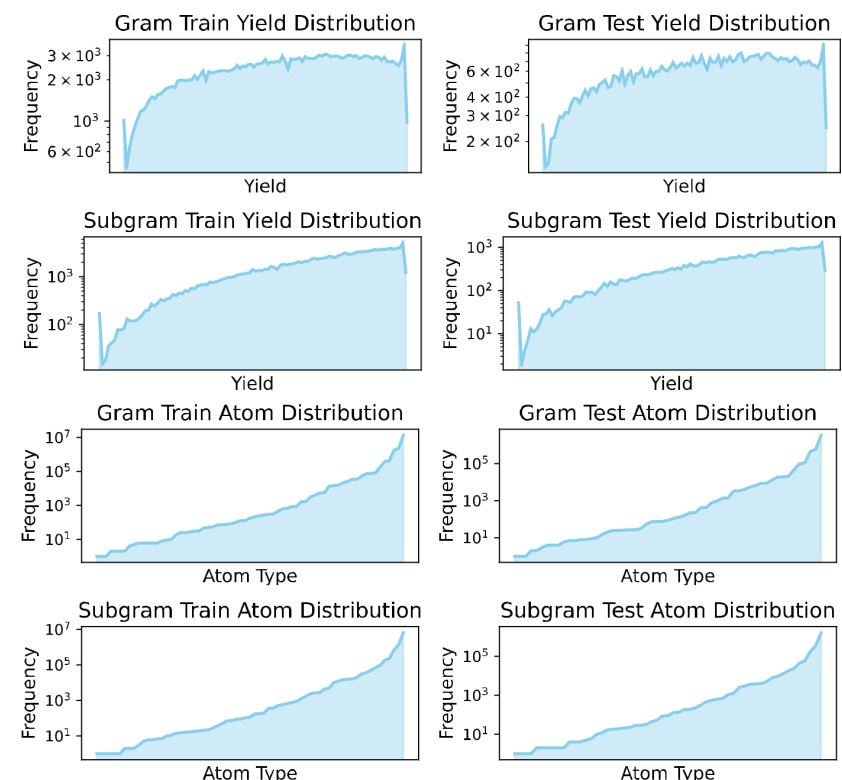

Figure 11: Data distributions of USPTO-Yield, including distribution of the number of data points in each bin of yield, and the distribution of atom types.

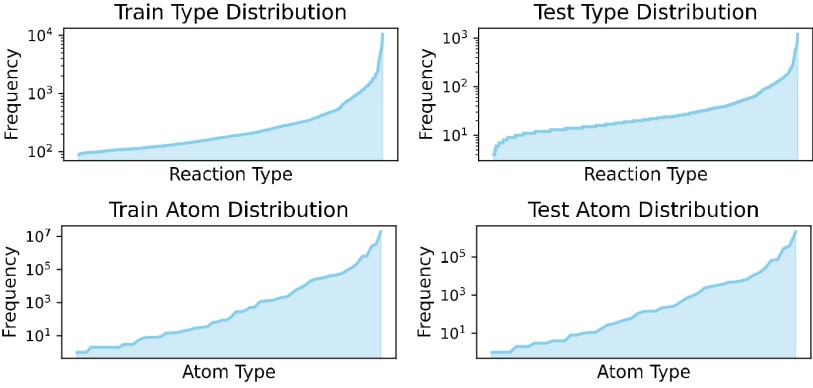

Figure 12: Data distributions of USPTO-TPL, including the distribution of the number of data points for various reaction types, as well as the distribution of atom types.

## F.6 PISTACHIO-TYPE

Considering the scale of the dataset and the difficulty of preprocessing, we extract a portion of the large reaction classification dataset provided by Pistachio to form the Pistachio-Type dataset. First, for the reaction dataset provided by Pistachio, we simplify its reaction type labels into 13 major categories, with each category containing multiple reaction templates, making it more complex compared to the USPTO-TPL, where one template corresponds to one category. Then, we discard one category with too little data. From the remaining 12 categories, we select an equal amount of data from each category to form the Pistachio-Type dataset, and then divide it into training and test sets. This approach ensures a more uniform data distribution.

Table 26: Summary of USPTO-TPL, including metrics for measuring the complexity of data points, the overall complexity of the dataset, and the distribution differences between the train/val set and the test set.

| Metric | Train/Val | Test |
|---|---|---|
| Size | 400,604 | 44,511 |
| Mean Length | 124.92 | 125.15 |
| Mean Atoms | 11.29 | 11.32 |
| Mean Molecules | 5.91 | 5.91 |
| Max Length | 599 | 493 |
| Max Atoms | 332 | 243 |
| Atoms | 66 | 51 |
| Mean Atoms per Molecule | 24.99 | 23.31 |
| Max Atoms per Molecule | 164 | 99 |
| Max Molecules | 59 | 31 |
| Molecules | 581,458 | 88,219 |
| Different Molecules (Train/Val and Test) | 543,559 | |
| Num Types | 1000 | |

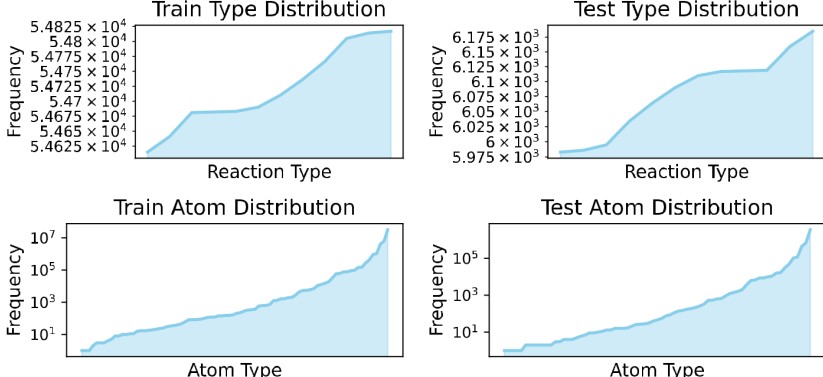

Figure 13: Data distributions of Pistachio-Type, including the distribution of the number of data points for various reaction types, as well as the distribution of atom types.

Table 27: Summary of Pistachio-Type, including metrics for measuring the complexity of data points, the overall complexity of the dataset, and the distribution differences between the train/val set and the test set.

| Metric | Train/Val | Test |
|---|---|---|
| Size | 656,640 | 72,960 |
| Mean Length | 133.91 | 133.98 |
| Mean Atoms | 13.04 | 13.04 |
| Mean Molecules | 5.36 | 5.35 |
| Max Length | 2706 | 1118 |
| Max Atoms | 926 | 480 |
| Mean Atoms per Molecule | 25.30 | 23.61 |
| Max Atoms per Molecule | 420 | 289 |
| Max Molecules | 64 | 48 |
| Molecules | 1,007,041 | 148,239 |
| Different Molecules (Train/Val and Test) | 946,727 | |
| Num Types | 12 | |

The final Pistachio-Type dataset still has a large scale, with data distribution shown in Fig. 13 and dataset metrics shown in Tab. 27. Although its label distribution is relatively even, each label contains multiple subcategories, involving far more reaction templates than 1000 categories. Additionally, by observing various metrics of the data points, we find that the complexity of chemical reactions in the Pistachio-Type dataset is significantly higher compared to the USPTO-TPL dataset. Therefore, the Pistachio-Type dataset is closer to real data distributions and application scenarios, and it presents a higher level of difficulty.

## G    SUPPLEMENTARY EXPERIMENTS

### G.1    ATTENTION WEIGHTS ANALYSIS

#### G.1.1    ATTENTION WEIGHTS VISUALIZATION

As mentioned in Sec. 4, we use the attention map of 3-Amino-5-bromobenzoic acid and two related reactions as example. We provide the original images of the attention weights drawn by RDKit, and additionally selected 4 groups of observation subjects as supplements. Each group contains a molecule with multiple active functional groups and two associated reactions, with each reaction occurring on different functional groups of the molecule. The results are shown in Fig. 14

The results show that for these five molecules and their respective two reactions, the Reaction Graphs help the model accurately identify the reaction centers while reducing attention to irrelevant atoms. In contrast, the model using molecular graphs exhibits a similar distribution of attention weights for the target molecules in both reactions, making it difficult to identify the correct reaction center in either reaction. It frequently focuses on functional groups that are less relevant to the reaction, leading to errors in prediction.

#### G.1.2    FAILURE MODE ANALYSIS

We analyzed the edge cases and failure modes of the reaction center identification results for molecular graphs and Reaction Graph. As shown in Fig. 14, molecular graphs often mislocate the reaction center when multiple functional groups are present, as the model cannot obtain information related to the reaction changes from the molecular graph, making it difficult to determine which functional group is involved in the reaction. In contrast, incorrect localization of the reaction center in Reaction Graph is relatively rare, but still exist. As shown in Fig. 15, the typically reasons are:

1. Errors in the reaction itself;

2. Distraction of attention due to overly complex molecular structures;

3. Encountering rare chemical reactions that do not occur on functional groups.

Although the model using Reaction Graph still encounters failures, we observe that even in the failure cases of Reaction Graph, the model still shows a higher level of attention to the reaction center compared to molecular graphs. This validates the strong adaptability of Reaction Graph to edge cases.

#### G.1.3    ATTENTION WEIGHTS STATISTICAL VALIDATION

We conducted additional experiments on dataset scale to quantitatively validate that the Reaction Graph aids in identifying reaction centers.

We use USPTO dataset and use the algorithm described in Sec. D.3 to extract a test set of 1000 molecules, as well as 2000 corresponding reactions. We use the condition prediction model trained on USPTO-Condition to process these reactions.

Table 28: Influence of reaction information on attention weight results on USPTO.

| Method | Proportion | Ratio $a/b$ |
|---|---|---|
| Molecular Graph | 0.4326 | 1.4542 |
| Reaction Graph | **0.5079** | **2.0390** |

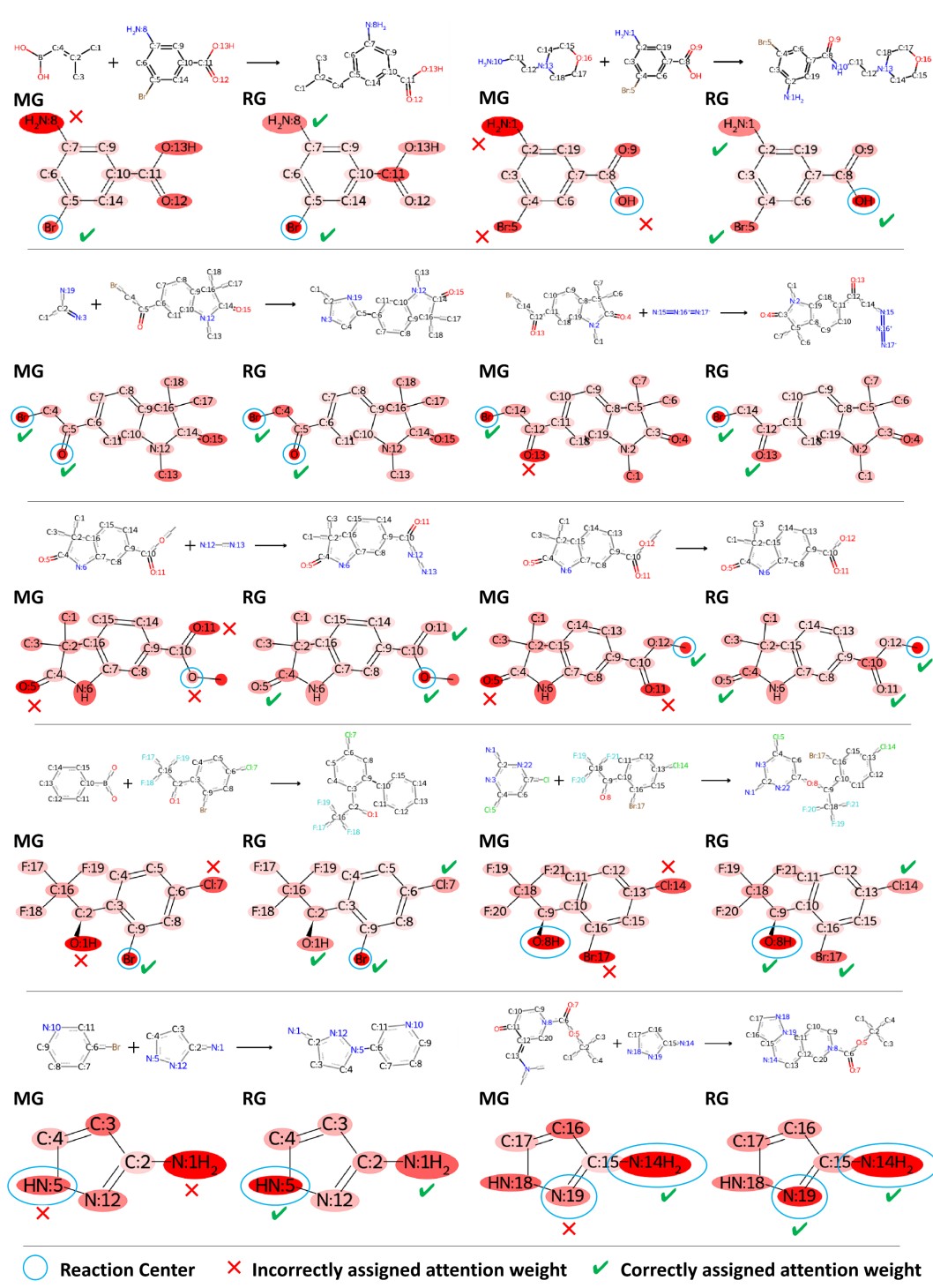

Figure 14: Visualization results of attention weights of the reaction condition prediction model on molecular graph and Reaction Graph. The depth of red represents the magnitude of attention weights, with deeper shades indicating larger attention weights.

Attention weights are extracted for reacting atoms and non-reacting atoms from the Set2Set layer for each target molecule. We calculate two metrics to evaluate the model's attention on the reacting

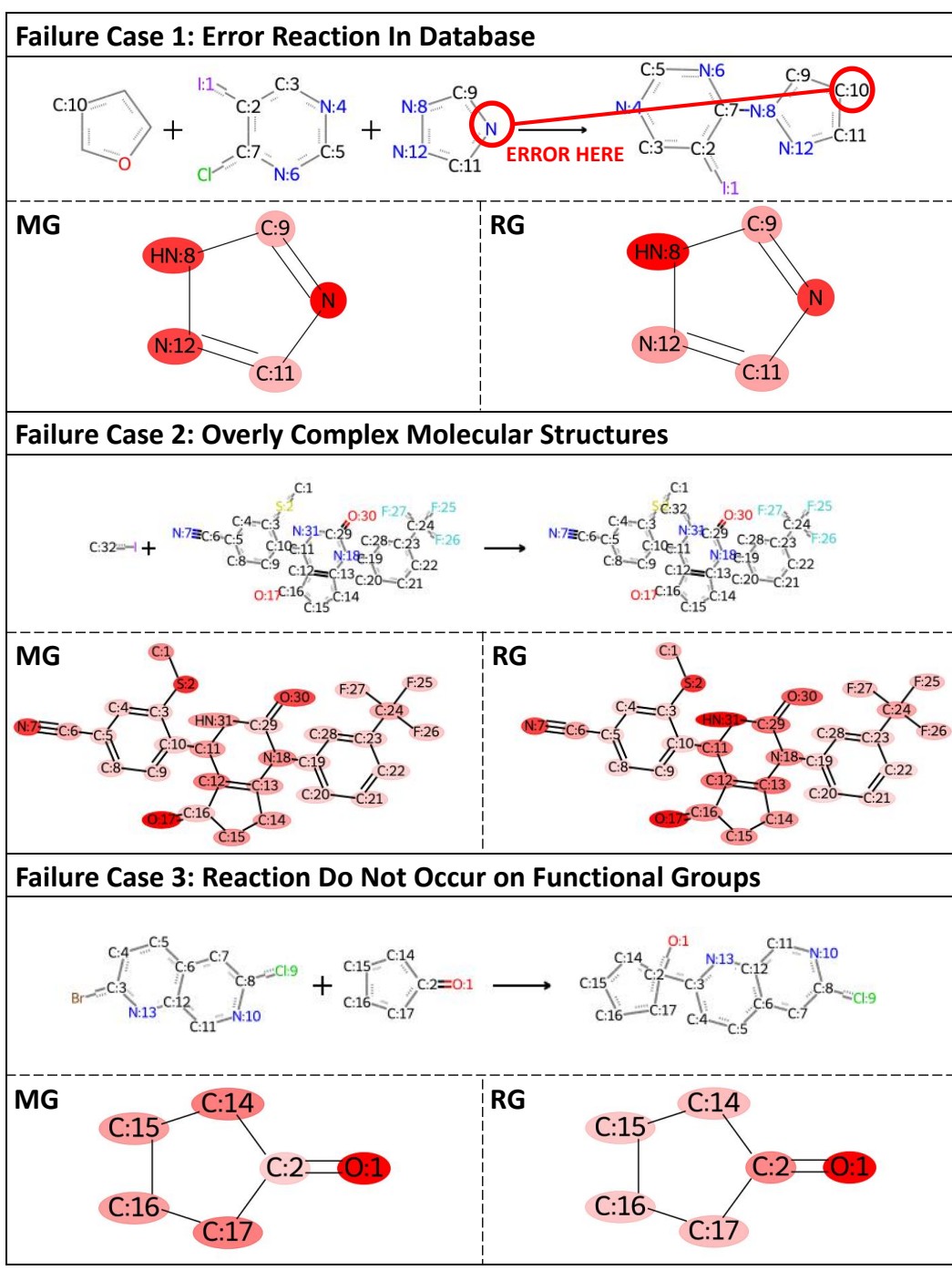

Figure 15: Visualization of attention weights for the failure cases. For the Reaction Graph, these cases can be categorized into three types, each being relatively rare. However, even in failure cases, the Reaction Graph (RG) still outperforms the Molecular Graph (MG).

atoms. First, we consider the proportion of attention weights for reacting atoms relative to the total. Second, we calculate the average attention weights for reacting atoms and non-reacting atoms, denoted as $a$ and $b$, and computed their ratio $a/b$. The results are shown in Tab. 28

Table 29: The proportion of attention weight for the reacting atoms, across various reaction types.

| Method | 0 | 1 | 2 | 3 | 4 | 5 | 6 | 7 | 8 | 9 | 10 | 11 |
|---|---|---|---|---|---|---|---|---|---|---|---|---|
| Molecular Graph | 0.65 | 0.49 | 0.50 | 0.50 | 0.61 | 0.35 | 0.58 | 0.50 | 0.38 | 0.46 | 0.28 | 0.04 |
| **Reaction Graph** | **0.70** | **0.53** | **0.54** | **0.54** | **0.66** | **0.38** | **0.65** | **0.57** | **0.40** | **0.52** | **0.32** | **0.06** |

Table 30: The ratio $a/b$ of attention weight between reacting and non-reacting atoms, across various reaction types.

| Method | 0 | 1 | 2 | 3 | 4 | 5 | 6 | 7 | 8 | 9 | 10 | 11 |
|---|---|---|---|---|---|---|---|---|---|---|---|---|
| Molecular Graph | 4.65 | 1.92 | 2.87 | 3.07 | 2.25 | 0.67 | 4.56 | 2.56 | 0.84 | 2.84 | 0.55 | 0.05 |
| **Reaction Graph** | **6.07** | **2.35** | **3.21** | **3.79** | **2.86** | **0.84** | **6.73** | **3.39** | **0.99** | **3.66** | **0.64** | **0.09** |

We further test the relationship between the attention weight results and reaction types, calculating the aforementioned metrics for each reaction type in the dataset. The results are shown in the Tab. 29 and 30. The correspondence between reaction type numbers and specific reaction types can be referenced in the Tab. 39.

### G.1.4 RELATION WITH DFT

We attempt to compare the attention map results from Molecular Graph and Reaction Graph with the theoretical calculation results from DFT. We aim to find the relationship between the attention map and chemical theoretical calculations to provide deeper insights. We use the def2-svp basis set and exploring two approaches: calculating a single molecule solely and jointly calculating the reaction system. We mainly observe the connection between electron density and attention weights. The results are shown in Fig. 16.

By observing the relationship between attention weights and electron density, we find that the attention weights of the Molecular Graph have a certain correlation with the electron density calculated for the molecule. Similarly, the attention weights of the Reaction Graph also show a correlation with the reaction system. However, this correlation does not seem to be very pronounced. In our ongoing work, we are also employing other analytical methods or higher-precision calculations to further analyze the connections.

## G.2 LEAVING GROUP IDENTIFICATION ANALYSIS

### G.2.1 LEAVING GROUP ANALYSIS DETAILS

We use a leaving group (LvG) identification task to further demonstrate that RG have stronger expressive ability than MG. To obtain the LvG identification dataset, we randomly selected a subset of 120,000 chemical reactions from USPTO database and then converted them into a RG dataset by using the algorithm in Sec. D.

During training, we attach a FC layer to the last layer of the feature extraction module to map it into node-level multi-class classification labels, supervised by the LvG labels. In MG, where message cannot be passed between molecules, the model can only select the most active fragment of the molecule as the LvG. This approach relies on the characteristics of the molecule rather than the reaction, significantly limiting classification accuracy. However, in RG, where message can be passed between reactants and products, the model can identify the atomic groups that leave during the reaction. As a result, it can achieve higher accuracy.

### G.2.2 ERROR ANALYSIS OF MISCLASSIFIED CASES

We conduct a detailed analysis of the failure cases in the leaving group recognition task for molecular graphs and Reaction Graph, and we attempt to identify the failure modes.

We visualize the leaving group identification results of molecular graph and Reaction Graph. As shown in Fig. 17, molecular graph often misidentify leaving group positions, hindering accurate

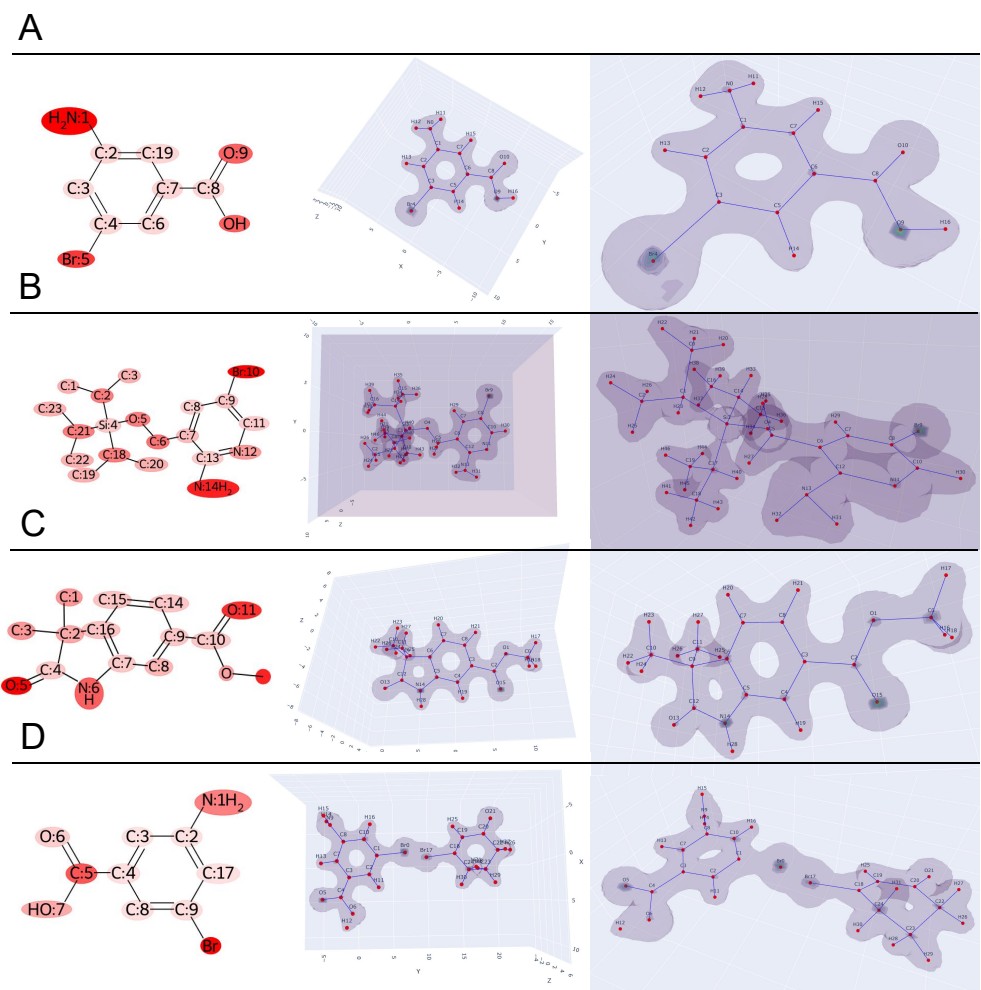

Figure 16: Visualization of the attention weights and the DFT calculation result. We observe the connection between electron density and attention weights. A, B and C are calculated for a single molecule, and use Molecular Graph. D is calculated for a reaction system, and use Reaction Graph.

classification. In contrast, Reaction Graph position leaving groups correctly at most time, with minor errors mainly in specific type classification. This shows that Reaction Graph effectively helps the model to focus on features related to the reaction.

### G.2.3 ADDITIONAL BASELINE METHODS

We conduct extra experiment with D-MPNN and Rxn Hypergraph, and the results are shown in Tab. 31.

According to the results, Reaction Graph achieved the best performance in leaving group identification compared to other baselines.

### G.3 TASKS

### G.3.1 REACTION CONDITION PREDICTION

**Metric Calculation.** In the reaction condition prediction task, the most commonly used metric is the top-$k$ accuracy. Formally speaking, given a ground truth condition $c =$

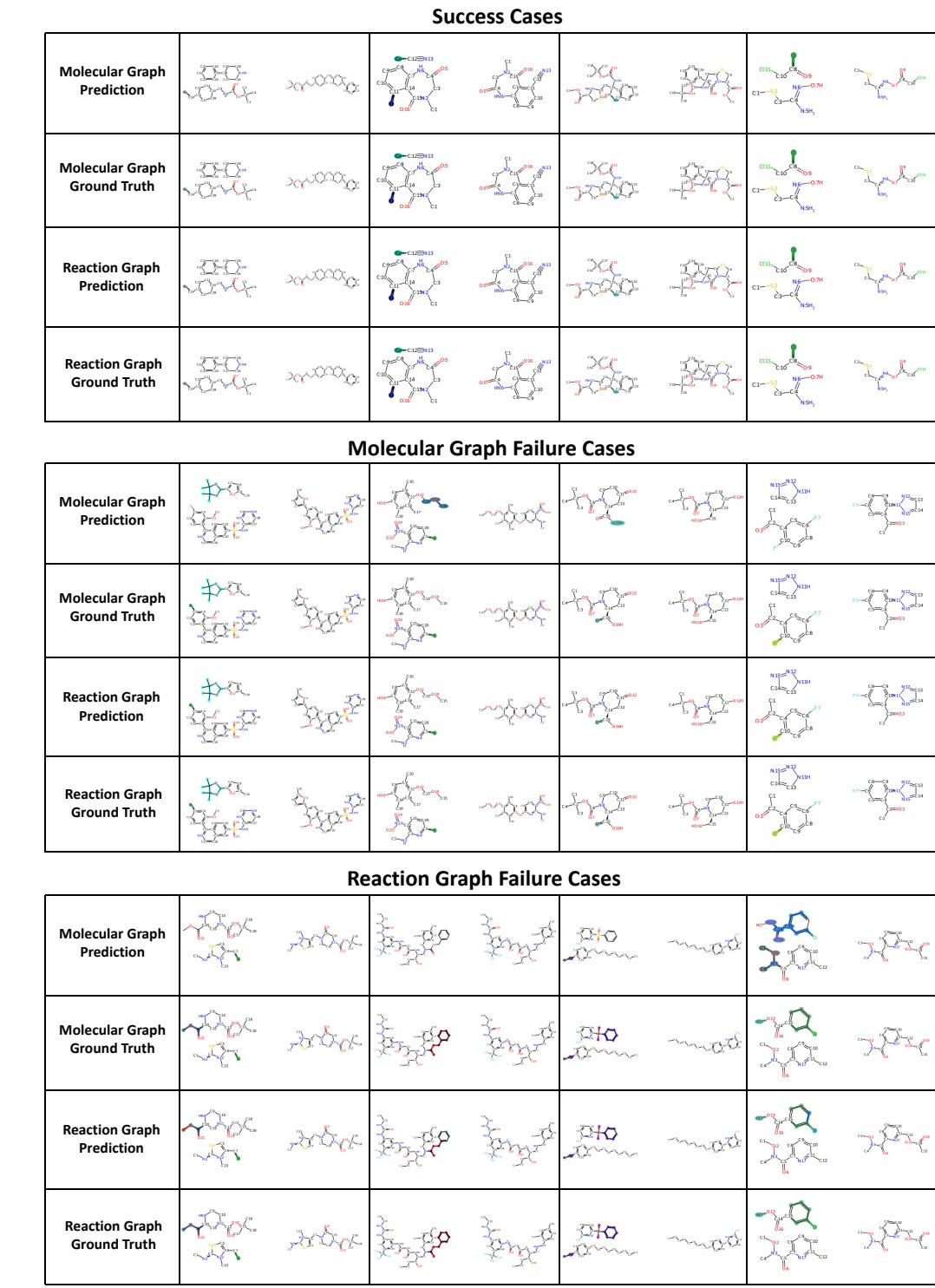

Figure 17: Visualization of the success and failure cases on leaving group identification. Molecular graphs often exhibit errors in positioning the leaving group when the molecule has multiple functional groups, while errors in Reaction Graph are relatively rare, typically occurring due to incorrect classification of certain atoms.

Table 31: Performance of different baseline methods on the Leaving Group Identification task.

| Method | Overall | | | LvG | | |
|---|---|---|---|---|---|---|
| | ACC | CEN | MCC | ACC | CEN | MCC |
| Molecular Graph | 0.950 | 0.036 | 0.549 | 0.448 | 0.201 | 0.519 |
| Rxn Hypergraph | 0.969 | 0.026 | 0.743 | 0.679 | 0.150 | 0.699 |
| D-MPNN | 0.993 | 0.003 | 0.949 | 0.902 | 0.051 | 0.899 |
| Reaction Graph | **0.997** | **0.002** | **0.973** | **0.947** | **0.031** | **0.945** |

$[\boldsymbol{c}_{catalyst}, \boldsymbol{c}_{solvent1}, \boldsymbol{c}_{solvent2}, \boldsymbol{c}_{reagent1}, \boldsymbol{c}_{reagent2}]$, to calculate top-$k$ accuracy, we allow the model to generate $k$ sets of labels $\mathbb{C} = \{\hat{\boldsymbol{c}}^1, \hat{\boldsymbol{c}}^2, \ldots, \hat{\boldsymbol{c}}^k\}$. Let $a$ represent the correctness of model's prediction on this data sample, and $a_{category}$ represent the correctness of model's prediction on this one label category, where category can be one of catalyst, solvent1, solvent2, reagent1 and reagent2. We have:

$$a = \begin{cases} 1, & \hat{\boldsymbol{c}}^i = \boldsymbol{c}, \exists\, i \in \{0, 1, \ldots, k\}, \\ 0, & otherwise, \end{cases} \tag{11}$$

$$a_{category} = \begin{cases} 1, & \hat{\boldsymbol{c}}^i_{category} = \boldsymbol{c}_{category}, \exists\, i \in \{0, 1, \ldots, k\}, \\ 0, & otherwise, \end{cases} \tag{12}$$

And assume the correctness of model on data sample $i$ is $a^i$, then the overall top-$k$ accuracy on the whole dataset is $\overline{a} = \frac{\sum_i^{N_d} a^i}{N_d}$, and the top-$k$ accuracy of a label category is $\overline{a}_{category} = \frac{\sum_i^{N_d} a^i_{category}}{N_d}$, where $N_d$ is the number of data samples in the test set.

It is important to note that the overall accuracy calculation aligns with our intuitive understanding, while the accuracy calculation for each label category is not. This is because it is possible that $\hat{\boldsymbol{c}}^i_{category} = \hat{\boldsymbol{c}}^j_{category}$, where $i, j \in \{0, 1, \ldots, k\}$, meaning the number of different classification results output by the model is actually less than $k$. Therefore, this evaluation criterion is stricter than our intuitive understanding. In other words, we do not generate $k$ labels for each category separately to calculate the top-$k$ accuracy of it.

**Additional Experimental Setting.** To fully validate the performance of the designs we propose, we conduct the following additional experiments:

- First, following Gao et al. (2018); Wang et al. (2023b), we test the top-$k$ accuracy for the comparison methods in Sec. 4.

- Second, we test the influence of different hyperparameter combinations on the model's performance using USPTO-Condition dataset.

- Thirdly, we conduct ablation study on the model and training methods, and use the top-$k$ curves from the two-stage training to explain the effectiveness of the improvements..

- Fourth, due to the more active field of molecular research, we attempt to introduce advanced molecular representation models for reaction tasks and compare them with our models specifically designed for reaction design.

- Fifth, we explore applying reaction information from the Reaction Graph to other model architectures to verify the generality and foundational nature of the proposed design.

- Sixth, we also conduct an in-depth analysis of the classification results of RG on two reaction classification datasets.

- Seventh, we attempted to merge the USPTO and Pistachio datasets to observe whether the model could achieve better generalization performance.

Table 32: Detailed results of top-$k$ accuracies of comparison methods on USPTO-Condition, including performance for each reaction condition category.

| Method | Conditions | Top-1↑ | Top-3↑ | Top-5↑ | Top-10↑ | Top-15↑ |
|---|---|---|---|---|---|---|
| CRM | catalyst | 0.9219 | 0.9219 | 0.9219 | 0.9219 | 0.9219 |
| | solvent-1 | 0.5015 | 0.6640 | 0.7055 | 0.7340 | 0.7346 |
| | solvent-2 | **0.8130** | 0.8369 | 0.8461 | 0.8525 | 0.8527 |
| | reagent-1 | 0.4972 | 0.6597 | 0.7402 | 0.8184 | 0.8516 |
| | reagent-2 | 0.7622 | 0.8408 | 0.8664 | 0.8876 | 0.8986 |
| | overall | 0.2596 | 0.3771 | 0.4206 | 0.4612 | 0.4717 |
| AR-GCN | catalyst | 0.9024 | 0.9024 | 0.9024 | 0.9024 | 0.9024 |
| | solvent-1 | 0.4114 | 0.5787 | 0.6295 | 0.6635 | 0.6650 |
| | solvent-2 | 0.8093 | 0.8093 | 0.8093 | 0.8093 | 0.8093 |
| | reagent-1 | 0.4200 | 0.5740 | 0.6667 | 0.7515 | 0.7622 |
| | reagent-2 | 0.7486 | 0.7486 | 0.7486 | 0.7486 | 0.7486 |
| | overall | 0.1460 | 0.2374 | 0.2733 | 0.3121 | 0.3261 |
| CIMG-Condition | catalyst | 0.9146 | 0.9146 | 0.9146 | 0.9146 | 0.9146 |
| | solvent-1 | 0.4218 | 0.6139 | 0.6542 | 0.6780 | 0.6789 |
| | solvent-2 | 0.8110 | 0.8110 | 0.8110 | 0.8110 | 0.8110 |
| | reagent-1 | 0.4351 | 0.5685 | 0.6665 | 0.7462 | 0.7598 |
| | reagent-2 | 0.7574 | 0.7574 | 0.7574 | 0.7574 | 0.7574 |
| | overall | 0.1839 | 0.2714 | 0.3026 | 0.3391 | 0.3525 |
| Parrot | catalyst | 0.9250 | 0.9250 | 0.9250 | 0.9250 | 0.9250 |
| | solvent-1 | 0.5018 | 0.6858 | **0.7311** | **0.7536** | **0.7543** |
| | solvent-2 | 0.8096 | 0.8426 | 0.8521 | 0.8582 | 0.8585 |
| | reagent-1 | 0.5039 | 0.6820 | 0.7629 | **0.8436** | **0.8776** |
| | reagent-2 | 0.7648 | 0.8486 | 0.8774 | 0.8998 | 0.9110 |
| | overall | 0.2691 | 0.4035 | 0.4510 | 0.4914 | 0.5031 |
| D-MPNN | catalyst | 0.9198 | 0.9198 | 0.9198 | 0.9198 | 0.9198 |
| | solvent-1 | 0.4621 | 0.6295 | 0.6583 | 0.7177 | 0.7192 |
| | solvent-2 | 0.8120 | 0.8120 | 0.8120 | 0.8120 | 0.8120 |
| | reagent-1 | 0.4777 | 0.6272 | 0.7449 | 0.8067 | 0.8089 |
| | reagent-2 | **0.7702** | 0.7702 | 0.7702 | 0.7702 | 0.7702 |
| | overall | 0.1977 | 0.3000 | 0.3341 | 0.3780 | 0.3924 |
| Rxn Hypergraph | catalyst | 0.9160 | 0.9160 | 0.9160 | 0.9160 | 0.9160 |
| | solvent-1 | 0.4676 | 0.6309 | 0.6767 | 0.7077 | 0.7095 |
| | solvent-2 | 0.8089 | 0.8089 | 0.8089 | 0.8089 | 0.8089 |
| | reagent-1 | 0.4761 | 0.6246 | 0.7105 | 0.7844 | 0.7937 |
| | reagent-2 | 0.7642 | 0.7642 | 0.7642 | 0.7642 | 0.7642 |
| | overall | 0.2127 | 0.3084 | 0.3447 | 0.3808 | 0.3927 |
| Reaction Graph | catalyst | **0.9316** | **0.9316** | **0.9316** | **0.9316** | **0.9316** |
| | solvent-1 | **0.5429** | **0.6925** | 0.7265 | 0.7475 | 0.7481 |
| | solvent-2 | 0.8075 | **0.8564** | **0.8654** | **0.8723** | **0.8725** |
| | reagent-1 | **0.5343** | **0.6982** | **0.7713** | 0.8420 | 0.8729 |
| | reagent-2 | 0.7630 | **0.8663** | **0.8928** | **0.9119** | **0.9193** |
| | overall | **0.3246** | **0.4343** | **0.4715** | **0.5061** | **0.5181** |

- Finally, we also tried to divide the Pistachio dataset into different scaffold sizes to see at what data volume the 3D information would start to take effect.

**Top-$k$ Accuracy of Comparison Methods.**

According to Tab. 32 and 33, the model using RG surpasses existing methods on the majority of top-$k$ accuracy, further demonstrating the effectiveness of our proposed approach. We also note

Table 33: Detailed results of top-$k$ accuracies on Pistachio-Condition, including performance for each reaction condition category.

| Method | Conditions | Top-1↑ | Top-3↑ | Top-5↑ | Top-10↑ | Top-15↑ |
|---|---|---|---|---|---|---|
| CRM | catalyst | 0.9943 | 0.9943 | 0.9943 | 0.9943 | 0.9943 |
| | solvent-1 | 0.5188 | 0.6954 | 0.7281 | 0.7580 | 0.7584 |
| | solvent-2 | 0.8406 | 0.9004 | 0.9077 | 0.9134 | 0.9135 |
| | reagent-1 | 0.4287 | 0.5990 | 0.6640 | 0.7350 | 0.7643 |
| | reagent-2 | 0.7120 | 0.8326 | 0.8603 | 0.8924 | 0.9055 |
| | overall | 0.3300 | 0.4692 | 0.5098 | 0.5476 | 0.5538 |
| Parrot | catalyst | 0.9951 | 0.9951 | 0.9951 | 0.9951 | 0.9951 |
| | solvent-1 | 0.5084 | 0.7667 | 0.7981 | 0.8064 | 0.8065 |
| | solvent-2 | 0.8417 | 0.9124 | 0.9160 | 0.9167 | 0.9168 |
| | reagent-1 | 0.4315 | 0.6438 | 0.7236 | 0.8057 | 0.8314 |
| | reagent-2 | 0.7341 | 0.8642 | 0.8944 | **0.9225** | **0.9348** |
| | overall | 0.3500 | 0.5323 | 0.5883 | 0.6263 | 0.6301 |
| D-MPNN | catalyst | 0.9940 | 0.9940 | 0.9940 | 0.9940 | 0.9940 |
| | solvent-1 | 0.5015 | 0.6485 | 0.6871 | 0.7830 | 0.7872 |
| | solvent-2 | 0.8614 | 0.8614 | 0.8614 | 0.8614 | 0.8614 |
| | reagent-1 | 0.4061 | 0.5727 | 0.6844 | 0.7461 | 0.7480 |
| | reagent-2 | 0.7491 | 0.7491 | 0.7491 | 0.7491 | 0.7491 |
| | overall | 0.2586 | 0.3422 | 0.3775 | 0.4415 | 0.4693 |
| Rxn Hypergraph | catalyst | 0.9945 | 0.9945 | 0.9945 | 0.9945 | 0.9945 |
| | solvent-1 | 0.5173 | 0.6841 | 0.7466 | 0.7895 | 0.7925 |
| | solvent-2 | **0.8619** | 0.8619 | 0.8619 | 0.8619 | 0.8619 |
| | reagent-1 | 0.4246 | 0.5793 | 0.6608 | 0.7373 | 0.7470 |
| | reagent-2 | **0.7520** | 0.7520 | 0.7520 | 0.7520 | 0.7520 |
| | overall | 0.2881 | 0.3671 | 0.4117 | 0.4636 | 0.4851 |
| Reaction Graph | catalyst | **0.9952** | **0.9952** | **0.9952** | **0.9952** | **0.9952** |
| | solvent-1 | **0.5579** | **0.7791** | **0.8032** | **0.8130** | **0.8130** |
| | solvent-2 | 0.8539 | **0.9214** | **0.9248** | **0.9261** | **0.9261** |
| | reagent-1 | **0.4884** | **0.6767** | **0.7456** | **0.8123** | **0.8384** |
| | reagent-2 | 0.7416 | **0.8733** | **0.8964** | 0.9206 | 0.9319 |
| | overall | **0.3915** | **0.5566** | **0.6039** | **0.6384** | **0.6432** |

Table 34: Top-$k$ accuracies on USPTO-Condition under different hyperparameter combinations, including hidden layer dimensions and iterations of GNN and pooling. Each has 3 candidate values.

| Hidden Dim $D_v$ | MPNN Iters $T_1$ | Pooling Iters $T_2$ | Top-1↑ | Top-3↑ | Top-5↑ | Top-10↑ | Top-15↑ |
|---|---|---|---|---|---|---|---|
| 200 | 3 | 2 | **0.325** | **0.434** | **0.472** | 0.506 | 0.518 |
| 50 | 3 | 2 | 0.307 | 0.419 | 0.456 | 0.492 | 0.505 |
| 100 | 3 | 2 | 0.315 | 0.427 | 0.464 | 0.500 | 0.512 |
| 200 | 2 | 2 | 0.320 | 0.433 | 0.471 | **0.508** | **0.520** |
| 200 | 4 | 2 | 0.317 | 0.428 | 0.466 | 0.502 | 0.514 |
| 200 | 3 | 1 | 0.319 | 0.430 | 0.467 | 0.502 | 0.514 |
| 200 | 3 | 3 | 0.318 | 0.429 | 0.465 | 0.501 | 0.514 |

that Parrot shows performance advantages in certain specific condition categories, indicating the effectiveness and potential of large-scale pre-training.

**Hyperparameter Selection.** Due to training cost issues, we choose to test several hyperparameters that have the greatest impact on performance, specifically the hidden layer dimension $D_v$ of MPNN and the number of iterations $T_1$, as well as the number of iterations $T_2$ for Set2Set. According to Tab. 34, the hyperparameter combination we selected achieves optimal overall performance. Further

Table 35: Influence of the output head on model performance. We test the prediction accuracy using different output heads based on Molecular Graph on the USPTO-Condition dataset. The results show that the CRM output head we adopt is more effective.

| Method | Cond | Top-1↑ | Top-3↑ | Top-5↑ | Top-10↑ | Top-15↑ |
|---|---|---|---|---|---|---|
| MLP Output Head | c1 | 0.922 | 0.922 | 0.922 | 0.922 | 0.922 |
| | s1 | 0.501 | 0.653 | **0.712** | **0.734** | **0.7347** |
| | s2 | 0.799 | 0.799 | 0.799 | 0.799 | 0.799 |
| | r1 | **0.503** | 0.638 | 0.727 | 0.797 | 0.811 |
| | r2 | **0.763** | 0.763 | 0.763 | 0.763 | 0.763 |
| | all | 0.249 | 0.305 | 0.318 | 0.387 | 0.422 |
| CRM Output Head | c1 | **0.926** | **0.926** | **0.926** | **0.926** | **0.926** |
| | s1 | **0.511** | **0.656** | 0.693 | 0.715 | 0.716 |
| | s2 | **0.803** | **0.853** | **0.863** | **0.869** | **0.869** |
| | r1 | 0.500 | **0.671** | **0.743** | **0.821** | **0.856** |
| | r2 | 0.753 | **0.861** | **0.887** | **0.908** | **0.917** |
| | all | **0.298** | **0.400** | **0.437** | **0.472** | **0.484** |

Table 36: Influence of the two-stage training on model performance, using USPTO-Condition. The model is based on Reaction Graph. The results indicate that the proposed two-stage training strategy significantly improves the top-$k$ accuracy.

| Method | Cond | Top-1↑ | Top-3↑ | Top-5↑ | Top-10↑ | Top-15↑ |
|---|---|---|---|---|---|---|
| One-Stage | c1 | 0.928 | 0.928 | 0.928 | 0.928 | 0.928 |
| | s1 | 0.517 | 0.664 | 0.702 | 0.725 | 0.725 |
| | s2 | 0.799 | 0.855 | **0.866** | **0.873** | **0.874** |
| | r1 | 0.508 | 0.685 | 0.758 | 0.833 | 0.869 |
| | r2 | 0.754 | 0.866 | 0.891 | 0.911 | **0.920** |
| | all | 0.304 | 0.413 | 0.449 | 0.484 | 0.495 |
| Two-Stage | c1 | **0.932** | **0.932** | 932 | **0.932** | **0.932** |
| | s1 | **0.543** | **0.693** | **0.727** | **0.748** | **0.748** |
| | s2 | **0.808** | **0.856** | 0.865 | 0.872 | 0.873 |
| | r1 | **0.534** | **0.698** | **0.771** | **0.842** | **0.873** |
| | r2 | **0.763** | 0.866 | **0.893** | **0.912** | 0.919 |
| | all | **0.325** | **0.434** | **0.472** | **0.506** | **0.518** |

reducing the number of iterations for MPNN may enhance the model's top-15 performance, but it significantly affects the Top-1 performance. We also observe that as the hidden layer dimension increases, the model's performance gradually improves, indicating that our model has scalability. Due to GPU memory limitations, we do not continue to try higher hidden layer dimensions to ensure high training efficiency.

**Model Architecture and Training Strategy Ablation.** We primarily improve the output module in the reaction condition task. Following Gao et al. (2018), the CRM output head we adopt supports Beam Search, which significantly enhances the performance of multi-label multi-class classification tasks. In contrast, the MLP outputs all reaction conditions simultaneously, which results in a loss of overall accuracy. To demonstrate the effectiveness of our model architecture design, we eliminate additional influencing factors and test the model performance of the MLP output head and the CRM output head we adopted under the Molecular Graph.

Based on the results shown in Tab. 35, the CRM output head we adopt effectively improves the model's overall top-$k$ accuracy. However, upon examining the data, we find that the model using the MLP output head achieves accuracy that is nearly comparable to that of the CRM model under each type of reaction condition, indicating that the difference in overall accuracy arises from the mismatched combinations of generated reaction conditions. Compared to MLP output module, CRM, which supports beam search, can fully consider the previous prediction results when predicting the next reaction condition, allowing it to generate more suitable combinations of reaction conditions.

For the training strategy, we primarily propose a two-stage training approach, while also incorporating the technique of label smoothing. We test the effects of these two factors on model performance in one experiment. Specifically, we use a loss function that includes label smoothing and class weight to train the RG model on the USPTO-Condition dataset and observe the changes in validation top-$k$ accuracy.

According to the results in Tab. 18, the two-stage training strategy we propose effectively enhances the model's performance. Observing the top-$k$ accuracy curves from the two-stage training, we can see that even without using the two-stage method, the model still achieves accuracy comparable to that of the two-stage training at certain moments for each top-$k$ metric. However, the timing of the best performance differs. Especially for the top-15 accuracy, it shows a sign of overfitting after reaching its peak. After using two-stage training, the timing for achieving the best metrics from top-1 to top-15 accuracy is basically consistent.

Although there are certain differences between the validation and test sets, comparing the results in Fig. 18 and Tab. 36 reveals that the inclusion of label smoothing results in a more severe imbalance in model top-k performance. While label smoothing effectively improves the top-1 accuracy, it leads to a significant decrease in top-15 accuracy in the later stages of training. Therefore, it is important to select an appropriate training strategy based on actual usage conditions.

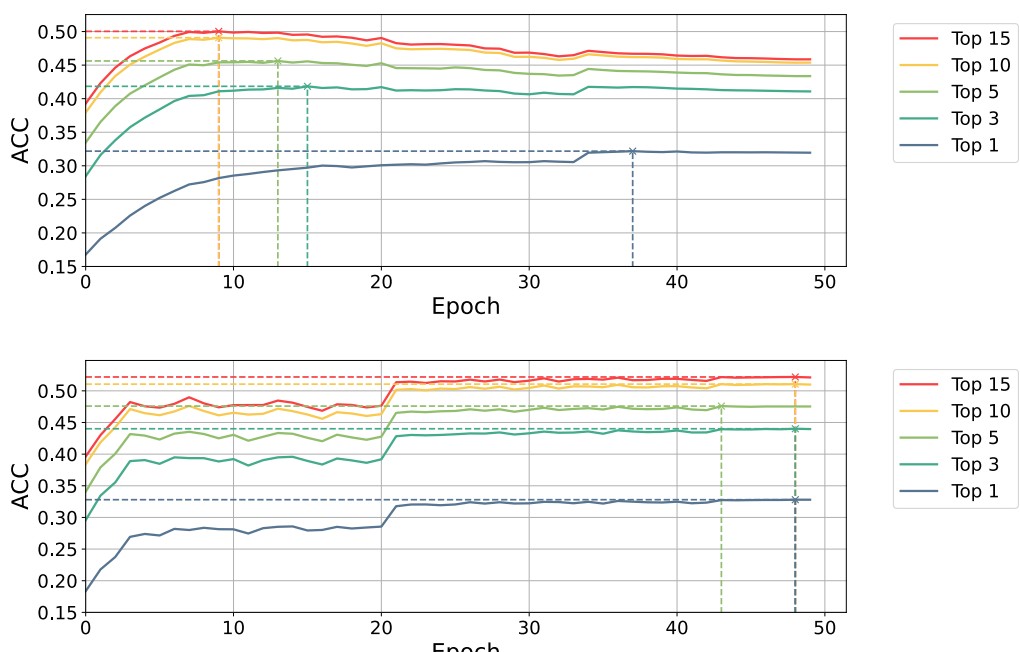

Figure 18: Top-$k$ validation accuracies of two-stage training, where the positions of the optimal results are marked with crosses. The upper figure is from the first stage, and the lower figure is from the second stage. The positions where the accuracies change sharply correspond to the points where the scheduler adjusts the learning rate.

Table 37: Comparison of top-$k$ accuracy between the state-of-the-art molecular representation model and Reaction Graph, using the USPTO-Condition dataset.

| Method | Condition | Top-1↑ | Top-3↑ | Top-5↑ | Top-10↑ | Top-15↑ |
|---|---|---|---|---|---|---|
| UniMol | catalyst | 0.9271 | 0.9271 | 0.9271 | 0.9271 | 0.9271 |
| | solvent-1 | 0.5170 | 0.6602 | 0.6931 | 0.7159 | 0.7165 |
| | solvent-2 | 0.7963 | 0.8462 | 0.8594 | 0.8715 | 0.8718 |
| | reagent-1 | 0.5107 | 0.6814 | 0.7497 | 0.8219 | 0.8540 |
| | reagent-2 | 0.7611 | 0.8645 | 0.8882 | 0.9091 | **0.9203** |
| | overall | 0.2955 | 0.4054 | 0.4397 | 0.4689 | 0.4766 |
| Reaction Graph | catalyst | **0.9316** | **0.9316** | **0.9316** | **0.9316** | **0.9316** |
| | solvent-1 | **0.5429** | **0.6925** | **0.7265** | **0.7475** | **0.7481** |
| | solvent-2 | **0.8075** | **0.8564** | **0.8654** | **0.8723** | **0.8725** |
| | reagent-1 | **0.5343** | **0.6982** | **0.7713** | **0.8420** | **0.8729** |
| | reagent-2 | **0.7630** | **0.8663** | **0.8928** | **0.9119** | 0.9193 |
| | overall | **0.3246** | **0.4343** | **0.4715** | **0.5061** | **0.5181** |

**Molecular Representation Method in Reaction Field.** We attempt to use UniMol for reaction condition prediction tasks on USPTO. To adapt it for reaction condition prediction, we change its output head to CRM and adopt a two-stage training strategy. The final results are as follows:

Based on the experimental results in Tab. 37, we find that UniMol, which utilizes 3D information and is based on Graph Transformer, not only achieves state-of-the-art performance in molecular representation but also excels in reaction condition prediction tasks. However, Reaction Graph, which is specifically designed for chemical reactions, demonstrates a greater advantage in comparison, underscoring the importance of modeling based on the intrinsic characteristics of chemical reactions.

Table 38: Influence of the proposed reaction information on the accuracies of different methods, on USPTO-Condition dataset.

| Model Architecture | Reaction Information | Top-1↑ | Top-3↑ | Top-5↑ | Top-10↑ | Top-15↑ |
|---|---|---|---|---|---|---|
| Bond Vector Model | ✗ | 0.282 | 0.393 | 0.432 | 0.468 | 0.481 |
| | ✓ | **0.290** | **0.403** | **0.439** | **0.475** | **0.488** |
| Bond Angle Model | ✗ | 0.307 | 0.420 | 0.456 | 0.493 | 0.505 |
| | ✓ | **0.318** | **0.429** | **0.466** | **0.502** | **0.514** |
| EGAT (Monninger et al., 2023) | ✗ | 0.297 | 0.406 | 0.442 | 0.478 | 0.489 |
| | ✓ | **0.304** | **0.417** | **0.453** | **0.490** | **0.502** |
| GINE (Hu et al., 2019) | ✗ | 0.289 | 0.396 | 0.432 | 0.468 | 0.481 |
| | ✓ | **0.299** | **0.406** | **0.441** | **0.475** | **0.487** |

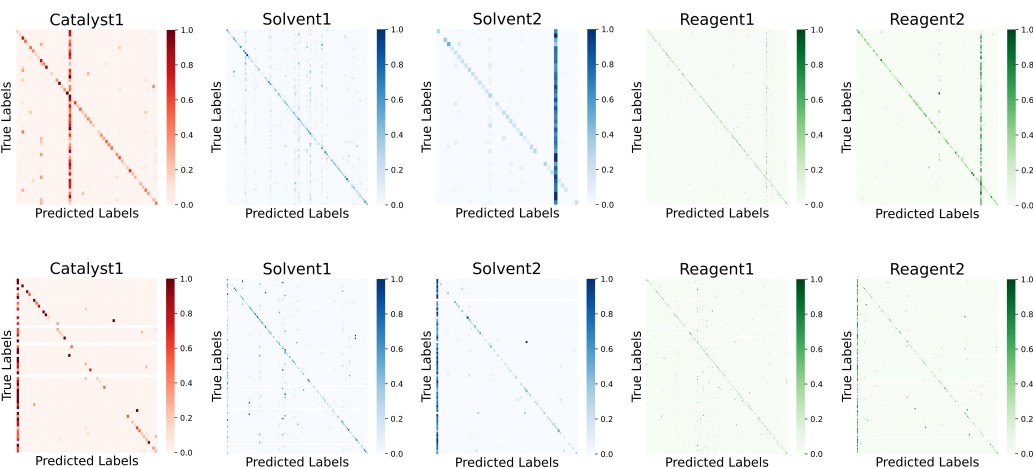

Figure 19: Normalized confusion matrices of condition prediction results on USPTO Condition dataset, using the proposed model architecture with Reaction Graph.

**Applicability of Reaction Information.** We attempt to incorporate the proposed designs into other model architectures to explore whether RG can be used to enhance various GNN-based methods. Specifically, we test the Bond Vector Model and Bond Angle Model designed by us, as well as the EGAT and GINE models from DGL library.

The results in Tab. 46 show that reaction information improves the top-$k$ accuracy of all methods, including INNs, ENNs and vanilla GNNs. This demonstrates the broad effectiveness of integrating reaction information.

**Classification Result Analysis.**

We perform a confusion matrix analysis of the model's classification results in Fig. 19. Due to the extremely uneven data distribution in the dataset, we apply row normalization to the confusion matrix values, dividing each row's values by the sum of that row. Based on the results, we observe that the diagonal of the confusion matrix is darker, indicating that the model can achieve correct classification results for most categories. However, we also notice that the columns corresponding to the $None$ category and some frequently occurring categories are similarly dark. This is due to the data distribution of the model. From the label distribution of the dataset shown in Sec. F, we can see that the number of certain high-frequency categories is significantly higher than that of other categories, leading the model to favor predicting the labels of these categories. Although the category weights and label smoothing methods we employ can alleviate this issue, improving performance on these sparse samples still relies on the inclusion of high-quality data.

Table 39: Correspondence between reaction type labels and reaction types. The class here correspond to the label of Pistachio-Type.

| Label | Name |
|---|---|
| 0 | Unrecognized |
| 1 | Heteroatom alkylation and arylation |
| 2 | Acylation and related processes |
| 3 | C-C bond formation |
| 4 | Heterocycle formation |
| 5 | Protections |
| 6 | Deprotections |
| 7 | Reductions |
| 8 | Oxidations |
| 9 | Functional group interconversion (FGI) |
| 10 | Functional group addition (FGA) |
| 11 | Resolutions |

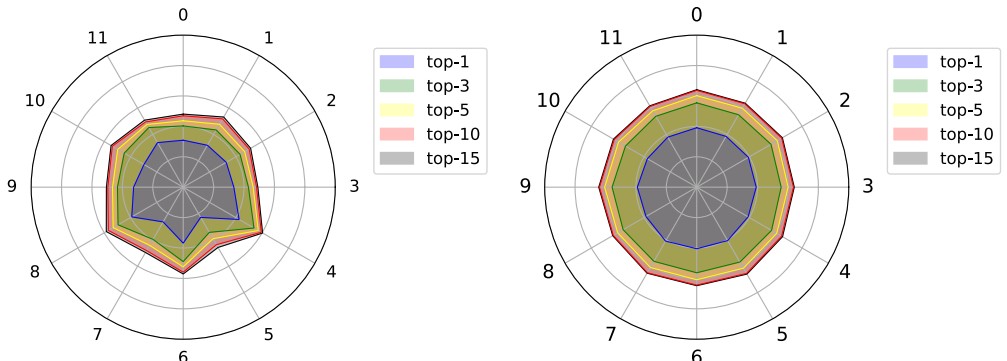

Figure 20: Radar chart of condition prediction accuracy under various reaction types. The results show that the USPTO-Condition prediction results have stronger correlation with reaction categories than that of Pistachio-Condition.

In addition, we analyze the classification accuracy of different categories across the two datasets. The correspondence between the reaction type indices and the names of reaction types is shown in Tab. 39, while the results are illustrated in Fig. 20. The categories in the USPTO-Condition dataset are classified using the NameRXN tool from Wang et al. (2023b), while the category labels in the Pistachio-Condition dataset are derived from the labels in the Pistachio database. Based on our observations, we find that the accuracy of reaction condition predictions in the USPTO-Condition dataset is somewhat correlated with the reaction categories, while this correlation is less evident in the Pistachio dataset. This also reflects the impact of data distribution on model performance, leading to higher classification accuracy on Pistachio compared to USPTO-Condition. It indicates that if we want to enhance the model's performance further, we can consider using external datasets to augment the data for reaction categories with fewer samples. Similarly, we observe that as $k$ increases, the growth of top-$k$ accuracy gradually slows down, suggesting that the choice of top-15 is reasonable. Further increasing the value of $k$ does not effectively improve the model's performance and may lead to higher costs for actual experimental validation.

**Merging USPTO and Pistachio.** In this paper, we focus on proposing a novel chemical reaction representation, which can enhance model performance within a fixed data volume. Exploring the effect of training with large-scale data is another interesting research topic, which is not the focus of this paper. Nevertheless, we conduct additional experiments to explore whether combining the USPTO and Pistachio datasets can provide valuable insights.

Table 40: Reaction classification results before and after mixing USPTO with Pistachio, on the test set of original USPTO-Condition and the OOD test set of Pistachio.

| Method | USPTO-Condition Test Set | | | | | Pistachio-OOD Test Set | | | | |
|---|---|---|---|---|---|---|---|---|---|---|
| | Top-1↑ | Top-3↑ | Top-5↑ | Top-10↑ | Top-15↑ | Top-1↑ | Top-3↑ | Top-5↑ | Top-10↑ | Top-15↑ |
| w/o Pistachio | **0.325** | 0.434 | **0.472** | **0.506** | **0.518** | 0.177 | 0.235 | 0.260 | 0.284 | 0.292 |
| w Pistachio | 0.323 | **0.436** | 0.470 | 0.500 | 0.511 | **0.275** | **0.455** | **0.517** | **0.567** | **0.578** |

Table 41: The influence of adding 3D information at different scaffold sizes on Pistachio-Condition.

| Scaffold Ratio | 1 | 1/2 | 1/4 | 1/8 | 1/16 | 1/32 |
|---|---|---|---|---|---|---|
| **w 3D** | **0.3915** | **0.3550** | **0.3146** | **0.2768** | **0.2244** | 0.1857 |
| **w/o 3D** | 0.3852 | 0.3503 | 0.3116 | 0.2716 | 0.2210 | **0.1907** |

In the implementation, since the reaction category labels between the two datasets are not completely consistent, we select reactions from Pistachio that match the categories in USPTO. We perform joint training using the merged dataset.

The results on the USPTO test set are as the left side of Tab. 40. We find that mixing the two datasets for training does not improve the model's performance on a single dataset. This may result from the significant differences in data distribution between USPTO and Pistachio. Although they are large in scale, their sparse annotations still cannot cover the diverse chemical space.

We further evaluate the effect of mixed data training on model generalization. Specifically, we extract a subset of data from Pistachio to serve as the test set. This subset is out-of-distribution (OOD), which can reflect the model's generalization performance. The results are as the right side of Tab. 40. Training with mixed data significantly improved performance on the Pistachio-OOD test set, while keeping the performance of model on the original USPTO-Condition dataset, indicating a benefit for generalization.

**Relation between Scaffold Size and 3D Information Utilization.** Based on different scaffold sizes, we design 6 progressively challenging experiments on Pistachio. The test set is consistent across all groups. From the first to the sixth group, the training set size halves each time, reducing from full size to 1/32 of the original. For each group, we compare the results with and without 3D information. As shown in Tab. 41, adding 3D information is helpful in the first five groups. The performance in the sixth group decreased. That is because it is difficult to effectively learn 3D information with limited data.

### G.3.2 REACTION YIELD PREDICTION

**Metric Calculation.** In yield prediction task, in addition to the commonly used $R^2$ metric, Mean Absolute Error (MAE) and Root Mean Squared Error (RMSE) are also frequently utilized. Assuming the ground truth yield of $i$-th data sample is $y_i$, the model's output mean is $\hat{\mu}_{y_i}$, then the MAE of the prediction result is $\frac{\sum_i^{N_d} |y_i - \hat{\mu}_{y_i}|}{N_d}$, RMSE is $\sqrt{\frac{\sum_i^{N_d} (y_i - \hat{\mu}_{y_i})^2}{N_d}}$, and $R^2$ is $1 - \frac{\sum_i^{N_d} (y_i - \hat{\mu}_{y_i})^2}{\sum_i^{N_d} (y_i - \overline{y})^2}$.

We also used the Negative Log-Likelihood (NLL) to evaluate the fitting performance of the model that incorporates uncertainty and to assess the reasonableness of its output variance. The specific calculation method for NLL is as follows:

$$NLL = \sum_i^{N_D} [\frac{(y_i - \mu_{y_i})^2}{2\sigma_{y_i}^2} + \frac{1}{2} \log(2\pi\sigma_{y_i}^2)], \tag{13}$$

where we call the first term $\frac{(y_i - \mu_{y_i})^2}{2\sigma_{y_i}^2}$ as Calibration, and the second term $\frac{1}{2}\log(2\pi\sigma_{y_i}^2)$ as Tolerance. Additionally, follow (Schwaller et al., 2021b; Kwon et al., 2022b), we also evaluate the standard deviation of the above metrics under ten repetitions of the experiment.

**Additional Experimental Setting.** For the USPTO-Yield and HTE datasets, we refer to relevant literature and introduced more baseline data for comparison. Additionally, for the larger USPTO-Yield dataset, we conducted a reaction type analysis to study the yield fitting performance across different types of reactions.

Specifically, on HTE datasets, we introduce traditional methods such as DFT and MFF as comparisons. Due to the limited number of molecules in the HTE dataset, we also compare the performance of the model using one-hot labels of molecules (Onehot). Additionally, we include the performance of T5Chem, as well as the SOTA model RMVP based on large-scale data pretraining. On USPTO-Yield, we introduce HRP as our extra baselines. Since T5Chem does not provide details of the experiments, we cannot determine whether it uses average results or optimal values. As a result, we exclude it from the comparison of results.

Based on the results shown in Tab. 42, we find that our model remains highly competitive against the baseline. Compared with non-pretrained models and methods, whether based on neural networks or theoretical calculations, most of our performance is at the forefront. When compared to the RMVP model pre-trained on large-scale datasets, we achieve similar $R^2$ scores using only the original training samples of over two thousand. However, we also observe that for the more challenging Test4 dataset, our model exhibits significant variance. In ten repeated experiments, our model achieves a maximum $R^2$ score of 0.77, while the minimum $R^2$ score is only 0.58. This is due to the number of samples in the dataset and is also related to the structure of GNNs, as we find that simpler fingerprint-based models have relatively smaller training variance compared to GNN-based or Transformer-based methods. This is an issue we need to address in the future.

Similarly, as the results shown in Tab. 43, in the USPTO-Yield dataset, the performance of our model remains at a leading level.

Additionally, we analyze the regression performance for different reaction types in the USPTO-Yield dataset, as shown in Fig. 21. We classify the data samples in USPTO-Yield using a classifier trained on the Pistachio-Type dataset and obtain category labels for analysis. The results show a strong correlation between regression performance and reaction categories, with noticeable differences in the Gram and Subgram datasets.

We also identify some commonalities in the two radar charts, such as the generally poor yield prediction $R^2$ metrics for (5) Protections and (10) FGA reactions. However, when comparing $R^2$ metrics with RMSE and MAE, we find that the poor $R^2$ scores for (5) and (10) are due to the complexity of the data distribution, as the variance of the data samples in these two categories is significantly greater than that of other categories. Additionally, we ob-

Table 43: Regression accuracy ($R^2$) on USPTO-Yield dataset, with more comparison methods.

| Model | Gram | Subgram |
|---|---|---|
| DRFP | **0.130** | 0.197 |
| Yield-Bert | 0.117 | 0.195 |
| T5Chem | 0.116 | 0.202 |
| Egret | 0.128 | 0.206 |
| UGNN | 0.117 | 0.190 |
| HRP | 0.129 | 0.200 |
| D-MPNN | 0.125 | 0.202 |
| Rxn Hypergraph | 0.118 | 0.196 |
| RG | 0.129 | **0.216** |

serve similar situations for (3) C-C bond formation and (6) Deprotections in the Gram dataset. This indicates that the yield distribution in large-scale datasets is relatively complex, making the task more challenging, and there is still significant room for exploration.

### G.3.3 REACTION CLASSIFICATION

**Metric Calculation.** We primarily used ACC, CEN, MCC and F1 Score as evaluation metrics for reaction classification. In implementation, we used the library functions from PyCM to compute these metrics.

Specifically, we refer to the calculation method in Schwaller et al. (2021a). Let $C$ be the number of classes, and class labels are from 1 to $C$. $M \in [0,1]^{C \times C}$ is the confusion matrix where the element $M_{i,j}$ represents the number of instances that belong to the true class $i$ but are predicted to be in class $j$. $M_{i,j}$ can be calculated using the following formula:

Table 42: Detailed HTE yield prediction results with more comparison methods, using MAE, RMSE and $R^2$ metrics.

| Models | B-H | S-M | Test1 | Test2 | Test3 | Test4 |
|---|---|---|---|---|---|---|
| DFT | - | - | - | - | - | - |
| | - | - | - | - | - | - |
| | 0.92 | - | 0.8 | 0.77 | 0.64 | 0.54 |
| Onehot | - | - | - | - | - | - |
| | - | - | - | - | - | - |
| | 0.89 | - | 0.69 | 0.67 | 0.49 | 0.49 |
| MFF | - | - | - | - | - | - |
| | - | - | - | - | - | - |
| | 0.93 | - | 0.85 | 0.71 | 0.64 | 0.18 |
| Y-B | 3.99±0.15 | 8.13±0.34 | 7.35±0.10 | 7.27±0.72 | 9.13±0.75 | 13.67±1.07 |
| | 6.01±0.27 | 12.07±0.46 | 11.44±0.34 | 11.14±1.27 | 14.28±0.82 | 19.68±1.40 |
| | 0.95±0.01 | 0.82±0.01 | **0.84±0.01** | 0.84±0.03 | 0.75±0.04 | 0.49±0.05 |
| Y-B-A | 3.09±0.12 | 6.60±0.27 | 7.02±0.76 | 6.59±0.33 | 11.05±0.95 | 18.42±0.62 |
| | 4.80±0.26 | 10.52±0.48 | 11.76±1.40 | 9.89±0.74 | 18.04±1.40 | 24.28±0.49 |
| | **0.97±0.01** | 0.86±0.01 | 0.81±0.05 | 0.87±0.02 | 0.59±0.07 | 0.16±0.03 |
| DRFP | 4.03±0.13 | 7.00±0.20 | 8.16±0.07 | 7.69±0.10 | 8.92±0.06 | 12.42±0.07 |
| | 6.08±0.28 | 11.00±0.40 | 11.99±0.10 | 11.26±0.16 | 15.04±0.06 | 18.76±0.11 |
| | 0.95±0.01 | 0.85±0.01 | 0.81±0.01 | 0.83±0.00 | 0.71±0.01 | 0.49±0.00 |
| T5 | - | - | - | - | - | - |
| | - | - | - | - | - | - |
| | 0.97 | 0.86 | 0.81 | 0.91 | 0.79 | 0.63 |
| Egret | 4.47±0.23 | 7.00±0.20 | **6.97±0.47** | 6.31±0.37 | 10.40±0.71 | 12.37±0.83 |
| | 6.61±0.30 | 11.00±0.40 | 11.03±0.64 | 9.41±0.98 | 16.58±1.33 | 17.88±0.99 |
| | 0.94±0.01 | 0.85±0.01 | **0.84±0.01** | 0.88±0.03 | 0.65±0.06 | 0.54±0.06 |
| UGNN | **2.92±0.06** | 6.12±0.22 | 8.08±0.83 | 6.30±0.65 | 8.99±0.31 | 13.19±0.75 |
| | **4.43±0.09** | 9.47±0.46 | 13.75±1.18 | 9.48±1.03 | 14.94±0.62 | 18.77±0.57 |
| | **0.97±0.01** | **0.89±0.01** | 0.74±0.04 | 0.88±0.03 | 0.72±0.02 | 0.50±0.03 |
| RMVP | 5.13±0.23 | 7.37±0.21 | 9.60±0.69 | 8.02±1.23 | 10.74±0.72 | 13.59±1.28 |
| | 7.52±0.49 | 10.79±0.25 | 13.33±0.63 | 11.18±1.54 | 15.38±1.20 | 18.16±1.40 |
| | 0.92±0.01 | 0.85±0.01 | 0.76±0.02 | 0.83±0.05 | 0.70±0.05 | 0.53±0.08 |
| RMVP (Pretrained) | 3.11±0.07 | 6.59±0.20 | 7.28±0.12 | **6.08±0.15** | 8.97±0.49 | **10.61±0.66** |
| | 4.63±0.14 | 10.37±0.42 | **10.77±0.14** | **8.72±0.18** | **12.79±0.77** | **14.62±0.93** |
| | **0.97±0.01** | 0.86±0.01 | **0.84±0.01** | **0.90±0.01** | **0.79±0.03** | **0.69±0.04** |
| D-MPNN | 4.72±0.08 | 7.96±0.21 | 8.20±0.28 | 7.81±0.81 | 9.04±0.26 | 12.25±0.23 |
| | 6.41±0.15 | 10.84±0.44 | 12.09±1.00 | 11.21±1.10 | 14.46±0.60 | 17.76±0.63 |
| | 0.94±0.01 | 0.85±0.01 | 0.80±0.03 | 0.82±0.03 | 0.73±0.02 | 0.55±0.03 |
| Rxn Hypergraph | 3.44±0.04 | 7.83±0.31 | 8.44±0.30 | 7.20±0.52 | 9.01±0.21 | 11.62±0.85 |
| | 5.45±0.08 | 11.06±0.46 | 11.65±0.49 | 11.18±0.60 | 14.98±0.48 | 17.65±0.42 |
| | 0.96±0.01 | 0.85±0.01 | 0.81±0.01 | 0.83±0.02 | 0.71±0.02 | 0.56±0.02 |
| RG | 3.07±0.06 | **6.08±0.26** | 7.84±0.20 | 6.23±0.45 | **8.64±0.35** | 10.81±1.37 |
| | 4.64±0.09 | **9.32±0.47** | 12.05±0.30 | 9.20±0.71 | 13.50±0.61 | 15.05±1.40 |
| | **0.97±0.01** | **0.89±0.01** | 0.80±0.01 | 0.88±0.02 | 0.76±0.02 | 0.68±0.06 |

$$M_{i,j} = \sum_{k=1}^{N} \delta(y_k, i) \cdot \delta(\hat{y}_k, j),$$

where $N$ is the total number of samples, $y_k$ is the true label of the $k$-th sample, $\hat{y}_k$ is the predicted label of the $k$-th sample, and $\delta(a, b)$ is the Kronecker delta function defined as:

$$\delta(a, b) = \begin{cases} 1 & \text{if } a = b \\ 0 & \text{if } a \neq b \end{cases}$$

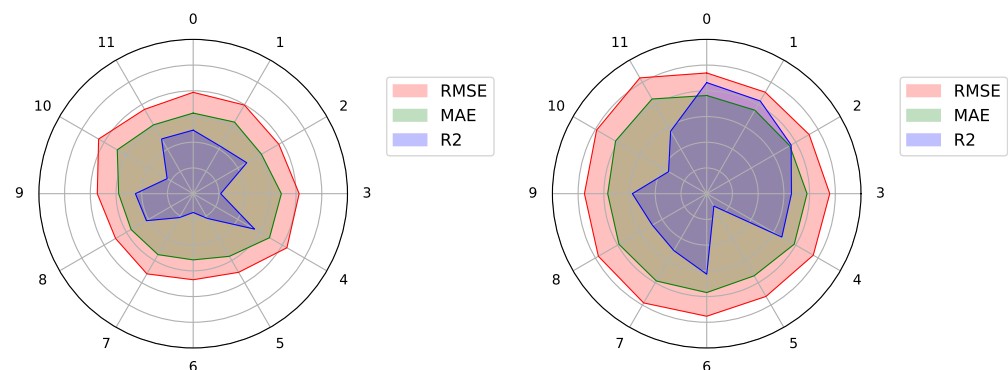

Figure 21: Radar chart of yield prediction metrics under various reaction types. The results show a strong correlation between the accuracy of yield prediction and reaction types in the USPTO-Yield.

From the confusion matrix, the following metrics can be calculated:

- **Accuracy (ACC/Micro F1).** The accuracy is the ratio of the number of correct predictions to the total number of predictions. It is calculated using the following formula:

$$\text{Accuracy} = \frac{\sum_{i=1}^{C} M_{i,i}}{N}$$

- **Macro F1.** The macro F1 score is the unweighted average of the F1 scores of each class. It is calculated using the following formula:

$$\text{Precision}_j = \frac{M_{j,j}}{\sum_{i=1}^{C} M_{i,j}}$$

$$\text{Recall}_j = \frac{M_{j,j}}{\sum_{i=1}^{C} M_{j,i}}$$

$$F1_j = 2 \cdot \frac{\text{Precision}_j \cdot \text{Recall}_j}{\text{Precision}_j + \text{Recall}_j}$$

- **Confusion Entropy (CEN).** The confusion entropy is a measure of the uncertainty in the confusion matrix. It is calculated using the following formula:

$$P_{i,j}^{j} = \frac{M_{i,j}}{\sum_{k=1}^{C} (M_{j,k} + M_{k,j})}, \quad P_{i,j}^{i} = \frac{M_{i,j}}{\sum_{k=1}^{C} (M_{i,k} + M_{k,i})}$$

$$\text{CEN}_j = - \sum_{k=1, k \neq j}^{C} \left( P_{j,k}^{j} \log_{2(C-1)} \left( P_{j,k}^{j} \right) + P_{k,j}^{j} \log_{2(C-1)} \left( P_{k,j}^{j} \right) \right)$$

$$P_j = \frac{\sum_{k=1}^{C} (M_{j,k} + M_{k,j})}{2 \sum_{k,l=1}^{C} M_{k,l}}$$

$$\text{CEN} = \sum_{j=1}^{C} P_j \text{CEN}_j$$

- **Matthews Correlation Coefficient (MCC).** The Matthews correlation coefficient is usually used in binary classification problems. However, it can be extended to multi-class classification problems using the following formula:

$$\text{cov}(X, Y) = \sum_{i,j,k=1}^{C} (M_{i,i} M_{k,j} - M_{j,i} M_{i,k})$$

Table 44: USPTO-TPL results.

| Models | ACC | CEN | MCC |
|--------|-----|-----|-----|
| DRFP | 0.977 | 0.011 | 0.977 |
| RXNFP | 0.989 | 0.006 | 0.989 |
| T5Chem | 0.995 | 0.003 | 0.995 |
| HRP | 0.991 | 0.005 | 0.990 |
| Rxn Hypergraph | 0.990 | 0.005 | 0.990 |
| D-MPNN | 0.997 | 0.001 | 0.997 |
| Reaction Graph | **0.999** | **0.001** | **0.999** |

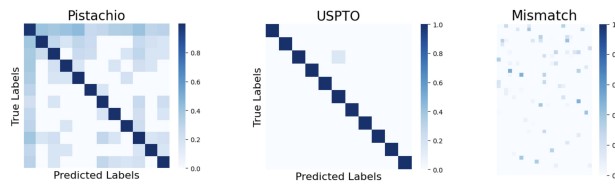

Figure 22: Scaled confusion matrices of reaction classification results.

$$\mathrm{cov}(X, X) = \sum_{i=1}^{C} \left[ \left( \sum_{j=1}^{C} \mathrm{M}_{j,i} \right) \left( \sum_{k,l=1,k\neq i}^{C} \mathrm{M}_{l,k} \right) \right]$$

$$\mathrm{cov}(Y, Y) = \sum_{i=1}^{C} \left[ \left( \sum_{j=1}^{C} \mathrm{M}_{i,j} \right) \left( \sum_{k,l=1,k\neq i}^{C} \mathrm{M}_{k,l} \right) \right]$$

$$\mathrm{MCC} = \frac{\mathrm{cov}(X, Y)}{\sqrt{\mathrm{cov}(X, X) \times \mathrm{cov}(Y, Y)}}$$

**Additional Experimental Setting.**

- **Additional Baselines.** we additionally introduce HRP, which also tests the performance on USPTO-TPL, for comparison. The result is shown in Tab. 44. Reaction Graph demonstrates superior performance on all metrics.

- **Confusion Matrix Analysis.** We analyze the model's performance by testing the confusion matrix of the classification results. For USPTO-TPL, we introduce category mapping to group the original 1,000 categories into 12 reaction categories from Pistachio, making it easier to assess the model's classification performance. We first use the reaction classification model trained on Pistachio-Type to predict the types of chemical reactions in USPTO-TPL. Then, we count the frequency of each Pistachio classification result corresponding to each USPTO-TPL category, selecting the most frequent as the mapped category.

The results are shown in Fig. 22. The reaction type mapping results are displayed in the bitmap on the right of the figure, which contains 1,000 points, each representing a USPTO-TPL category, with values indicating the proportion of results other than the most frequent classification. We observe that the misclassification rate for all classification results does not exceed 0.5, and most data points have misclassification rates close to 0, demonstrating the validity of our mapping and the generalization performance of the model trained on Pistachio-Type. Since both datasets exhibit high classification accuracy, we perform row normalization and amplification for each point in the confusion matrix, with the amplification factor set to $V' = V^{0.2}$. The classification results for reactions have high credibility. However, we also note that the misclassification rate for the $Unknown$ category is significantly higher than for other categories. This aligns with our expectations, as the boundaries for the $Unknown$ category are the most ambiguous and its distribution is the most complex among the 12 categories. The purpose of training the reaction classification model is to address the classification issues of the $Unknown$ category. In the testing of Pistachio-Type, we find that the classification accuracy for the $Unknown$ category is relatively high. While this reflects the advanced performance of our model, it also indicates that our model's extrapolation capability is limited, which is an issue that we need to address in the future.

**Dimensionality Reduction Visualization.** We perform dimensionality reduction visualization on the reaction representation vector $r$. This experiment aim to observe the distribution of chemical reactions, as well as exploring the relationship between $r$ and chemical properties. We use three methods for dimensionality reduction visualization: TMAP,

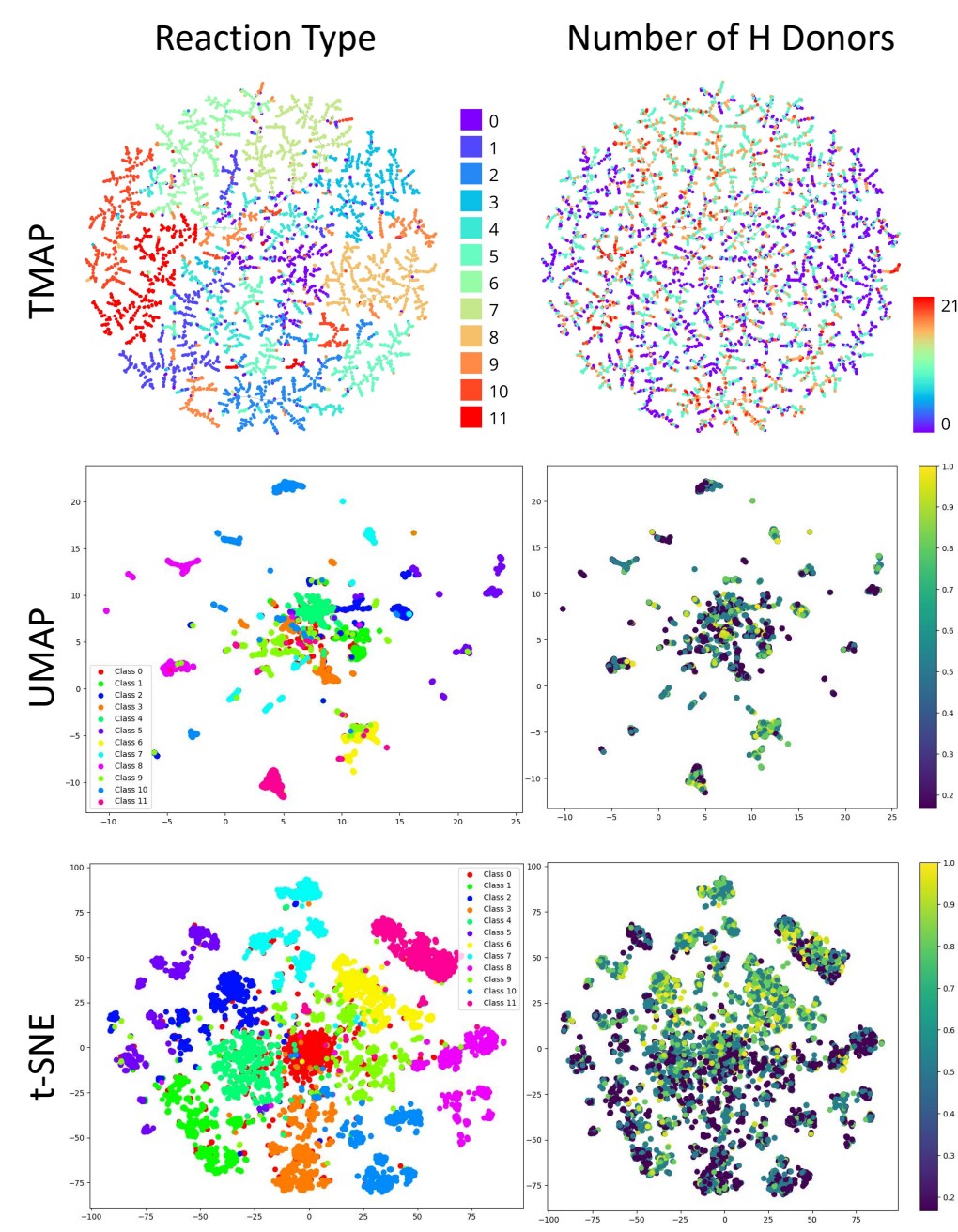

Figure 23: The dimensionality reduction visualization results of the reaction representation vectors $r$ extracted from the Reaction Graph. We utilize three unsupervised dimensionality reduction methods: TMAP, UMAP, and t-SNE. The figure shows the relationship between the dimensionality reduction results and reaction types, as well as the number of H donors.

UMAP, and t-SNE. Specifically, TMAP and UMAP are implemented using the TMAP[30] and UMAP[31] libraries in Python, while t-SNE is implemented using Scikit-Learn[32].

---

[30]https://tmap.gdb.tools/

[31]https://umap-learn.readthedocs.io/en/latest/

[32]https://github.com/scikit-learn/scikit-learn

Table 45: Influence of different edge length embedding methods on model's performance, using USPTO-Condition dataset.

| Method | Top-1↑ | Top-3↑ | Top-5↑ | Top-10↑ | Top-15↑ |
|---|---|---|---|---|---|
| Linear Projection Embedding | 0.3173 | 0.4303 | 0.4684 | 0.5048 | 0.5164 |
| Discretization Embedding | 0.3101 | 0.4201 | 0.4569 | 0.4926 | 0.5046 |
| RBF kernel Embedding (ours) | **0.3246** | **0.4343** | **0.4715** | **0.5061** | **0.5181** |

Table 46: Influence of different vertex-edge integration methods on condition prediction task, using USPTO-Condition dataset.

| Method | Top-1↑ | Top-3↑ | Top-5↑ | Top-10↑ | Top-15↑ |
|---|---|---|---|---|---|
| Bond Vector Model | 0.290 | 0.403 | 0.439 | 0.475 | 0.488 |
| Bond Angle Model | 0.318 | 0.429 | 0.466 | 0.502 | 0.514 |
| EGAT | 0.304 | 0.417 | 0.453 | 0.490 | 0.502 |
| GINE | 0.299 | 0.406 | 0.441 | 0.475 | 0.487 |
| Ours | **0.325** | **0.434** | **0.472** | **0.506** | **0.518** |

The result is shown in Fig. 23. Based on the results, we can see a clear correlation between the dimensionality reduction results and reaction types. Additionally, the results indicate a connection between the reaction representation vectors and the number of H donors. This indicates that the reaction representation vectors extracted by the Reaction Graph implicitly contain high-dimensional features of chemical properties.

## G.4 ARCHITECTURE MODULES

### G.4.1 EDGE EMBEDDING

We adopt RBF kernel to calculate edge length embedding. RBF can effectively capture the non-linear relationships between distance and molecular property. This method lifts the scalar edge lengths into a high-dimensional vector that can be more easily utilized by machine learning models, focus more on local structural pattern, and produces smooth mappings which help models to capture variations in continuous data. These advantages make RBF kernel suitable for tasks involving local continuous spatial relationships, such as in molecular structures where edge lengths indicate bond distances.

To demonstrate the efficiency of RBF kernel embedding, we conduct experiments to compare it with different embedding methods, using USPTO-Condition dataset. We use linear projection and discretization embedding as baselines. Specifically, linear projection directly inputs the edge lengths into a fully connected layer to obtain embeddings. Discretization embedding divides the edge lengths into multiple bins to obtain one-hot embeddings. The embeddings are then concatenated with the edge feature vectors. Detailed bin division method is discussed in Sec. C.1.1.

As shown in the Tab. 45 , RBF kernel outperforms other embedding methods, demonstrating its efficiency. Directly using linear mapping may cause the distance feature to lose its non-linearity. Meanwhile, using discrete features can easily lead to information loss.

### G.4.2 VERTEX-EDGE INTEGRATION

We conduct experiment to tested different vertex-edge integration methods to demonstrate the efficiency of our chosen approach. The baselines include Bond Vector Message Passing Model following PaiNN (Schütt et al., 2021), the Bond Angle Message Passing Model following DimeNet (Gasteiger et al., 2020), EGAT (Monninger et al., 2023) and GINE Hu et al. (2019). Our method follows UGNN (Kwon et al., 2022b) and MPNN (Gilmer et al., 2017)

The results show that our method is superior to other methods, demonstrating its efficiency.

### G.4.3 AGGREGATION

**Influence of Attention Mechanism and LSTM.** Effectively identifying and leveraging the chemical reaction mechanism to understand and reason about reactions is challenging. In our work, we use an attention mechanism to adaptively capture the most important cues for reaction modeling. However, since these cues are not always easy to identify in one time, we employ an LSTM to progressively and interactively discover them. As shown in Fig. 3, the attention-based aggregation module with LSTM accurately locates the reaction center on Reaction Graph.

We conduct experiments to demonstrate the roles of attention and LSTM. Specifically, we compare the performance of our method with the aggregation module without using attention and LSTM. We use USPTO-Condition dataset to evaluate the model's performance. For aggregation module without both attention and LSTM, we use SumPooling. For the attention aggregation module without LSTM, we set the number of iterations for Set2Set to 1, which is equivalent to not using LSTM.

Table 47: Influence of attention mechanism and LSTM on model performance, using USPTO-Condition dataset.

| Method | Top-1↑ | Top-3↑ | Top-5↑ | Top-10↑ | Top-15↑ |
|---|---|---|---|---|---|
| w/o Attention & w/o LSTM | 0.3159 | 0.4276 | 0.4642 | 0.4983 | 0.5110 |
| w/ Attention & w/o LSTM | 0.3187 | 0.4303 | 0.4670 | 0.5018 | 0.5136 |
| w/ Attention & w/ LSTM | **0.3246** | **0.4343** | **0.4715** | **0.5061** | **0.5181** |

The result in the table above shows that the attention mechanism and LSTM contributes to the performance.

**Set2Set vs. Set Transformer.** We use Set2Set to capture the global representation of a Reaction Graph by aggregating node features. We compare the ability of Set2Set and Set Transformer (Lee et al., 2019) for capturing the global representation of a Reaction Graph. The detailed implementation of Set Transformer can be found in Sec. C.

The results indicate that, when used as an aggregation module, Set2Set surpasses the Set Transformer by 5% to 10% in performance and is also more computationally efficient.

We further analyze the reasons for the unsatisfactory performance of the Set Transformer in the reaction property prediction task. We summarize two points:

1. The set message passing in the Set Transformer disrupts the topological constraints of the graph, leading to the loss of structural information.
2. The seed vector in the Set Transformer is fixed, whereas the adaptive seed vector derived from the graph in Set2Set is more advantageous.

We demonstrate these points through experiments. The result is shown in Tab. 49. Specifically, we gradually reduce the number of layers in the Set Transformer Encoder to observe whether the set message passing has a negative effect on graph information extraction. Additionally, We compare the performance of the Set Transformer Decoder (without adaptive seed vector) and the Set2Set(with adaptive seed cector) to observe whether the adaptive seed vector contributes to the model's performance. We use a subset from USPTO-Condition for training efficiency.

According to the results, with the set message passing reduces, the performance gradually improves. This result indicates that set message passing exhibits a negative effect when extracting features from the Reaction Graph. Meanwhile, adaptive seed vector achieves better performance than directly using a fixed seed vector, validating its effectiveness.

### G.5 GRAPH REPRESENTATION

### G.5.1 3D REPRESENTATION IN REACTION GRAPH

**Different 3D Representations.** We demonstrate the efficiency of our 3D representation design through experiments. Reaction Graph includes bond length and bond angle information, where

Table 48: Influence of Set Transformer and Set2Set on condition prediction result and inference time, using USPTO-Condition dataset.

| Method | Top-1↑ | Top-3↑ | Top-5↑ | Top-10↑ | Top-15↑ | Inference Time↓ |
|---|---|---|---|---|---|---|
| Set Transformer | 0.2940 | 0.4079 | 0.4471 | 0.4847 | 0.4968 | 6min 1s |
| Set2Set | **0.3246** | **0.4343** | **0.4715** | **0.5061** | **0.5181** | **2min 36s** |

Table 49: The influence of set message passing and adaptive seed vector on model's performance. Using a 1/8 subset of USPTO-Condition for training efficiency.

| Method | Set Message Passing | Adaptive Seed Vector | ACC↑ |
|---|---|---|---|
| Set Transformer | 2 | ✗ | 0.1597 |
| Set Transformer | 1 | ✗ | 0.1634 |
| Set Transformer | 0 | ✗ | 0.1700 |
| Set2Set (ours) | 0 | ✓ | **0.1773** |

Table 50: Influence of different 3D representation methods on Reaction Graph in the task of predicting reaction conditions, using USPTO-Condition dataset.

| Method | Top-1↑ | Top-3↑ | Top-5↑ | Top-10↑ | Top-15↑ |
|---|---|---|---|---|---|
| Without 3D Information | 0.3133 | 0.4248 | 0.4613 | 0.4961 | 0.5094 |
| Bond Length | 0.3165 | 0.4251 | 0.4616 | 0.4971 | 0.5090 |
| Bond Length+Explicit Bond Angle | 0.3179 | 0.4290 | 0.4656 | 0.5018 | 0.5146 |
| Atom Coordinate | 0.3123 | 0.4243 | 0.4628 | 0.4987 | 0.5111 |
| Equivariant Neural Networks | 0.2899 | 0.4026 | 0.4390 | 0.4749 | 0.4879 |
| Bond Length+Torsion Angular Edge | 0.3022 | 0.4087 | 0.4467 | 0.4821 | 0.4935 |
| Bond Length + Angular Edge (ours) | **0.3246** | **0.4343** | **0.4715** | **0.5061** | **0.5181** |

bond angle is implicitly conveyed by angular edge. We compare Reaction Graph with methods that use explicit bond angle, atomic coordinates, equivariant neural networks, and torsion angles.

The method using explicit bond angle is implemented by directional message passing module from DimeNet (Gasteiger et al., 2020). The atomic coordinate method is implemented by concatenating the atomic XYZ coordinate with the atomic attributes. The equivariant neural network is implemented by replacing the length information in Reaction Graph with vector, and adopt PaiNN (Schütt et al., 2021) as our vertex-edge integration module. The torsion angle is implemented by extending the angular edge in Reaction Graph to torsion angular edge, and details can be found in Sec. C and Sec. D.

As shown in the Tab. 50, Bond Length+Angular Edge achieves the best performance. This demonstrates the effectiveness of our approach.

**3D Information on Angular Edge.** We recognize that the role of the angular edge may not only be to provide angular information but also to serve as a shortcut for message passing. Therefore, we conduct an additional ablation experiments.

To analyze the role of the angular edge as a 3D prior, we set it's length to 0. This approach retains the connection of the angular edge to two-hop neighbors, thus also serving as a shortcut. To analyze the effect of offering a shortcut by angular edge, we remove angular edge from Reaction Graph.

The experimental results are shown in the Tab. 51. Based on the results, for angular edge, the role of being a shortcut is far less significant than its provision of 3D priors. This also demonstrates that angular edge effectively provides 3D bond angle information.

**Impact of 3D Information Accuracy and Label Quality on Model Performance.** The accuracy of 3D information does affect model performance, but the main factors that cause performance bottlenecks are the quality of labels in the dataset. We illustrate this through theoretical analysis and experiments.

Table 51: Impact of connection and length (for 3D modeling) on Angular Edge. Angular Edge is mainly used for providing 3D information, instead of providing shortcut for message passing.

| Method | Top-1↑ | Top-3↑ | Top-5↑ | Top-10↑ | Top-15↑ |
|---|---|---|---|---|---|
| w/o connection & w/o length for 3D modeling | 0.3165 | 0.4251 | 0.4616 | 0.4971 | 0.5090 |
| w/ connection & w/o length for 3D modeling | 0.3150 | 0.4259 | 0.4637 | 0.4991 | 0.5107 |
| w/ connection & w/ length for 3D modeling (ours) | **0.3246** | **0.4343** | **0.4715** | **0.5061** | **0.5181** |

1. **Theoretical Analysis**

   We demonstrate that 3D information and labels jointly influence model performance. Assume that the neural network is sufficiently trained and perfectly fits the data.

   - **Noisy Labels.** For a target function $f : \mathbb{X} \to \mathbb{Y}$ of neural network training, the quality of the label $Y \in \mathbb{Y}$ has a significant impact on the neural network performance. If there is noise $\Delta Y_{\text{error}}$ in $Y$, then the fitted network $f_1 : \mathbb{X} \to \mathbb{Y}$ will have output $f_1(X) = Y + \Delta Y_{\text{error}}$ with input $X$. This results in $f_1$ being unable to correctly model the relationship between $X$ and $Y$.

   - **Label Sparsity.** Assume the annotations $(X, Y)$ are sparse in training set, $f_2 : \mathbb{X} \to \mathbb{Y}$ is the fitted neural network on the sparse training set. Then for outliers in the test data $(X + \Delta X, Y + \Delta Y)$, the theoretical upper bound $|\Delta X| \cdot (L_f + L_{f_2})$ of the prediction error $\Delta Y_{outlier}$ will increase, where $X + \Delta X, Y + \Delta Y$ are the input and ground truth label of the outlier, $\Delta Y_{outlier} = f_2(X + \Delta X) - (Y + \Delta Y) = [f_2(X + \Delta X) - f_2(X)] - \Delta Y$, and $L_f$, $L_{f_2}$ are the Lipschitz constant[33] of $f$ and $f_2$, respectively.

   - **Noisy 3D Information.** The accuracy of 3D information also affects the neural network performance. Assume that the error of 3D information is $\Delta X_{error}$, and the network trained with error samples $(X + \Delta X_{error}, Y)$ is $f_3 : \mathbb{X} \to \mathbb{Y}$. We have $f_3(X + \Delta X_{error}) = Y$. Then $f_3$ will produce bias $\Delta Y_{errorX}$ during inference like $f_3(X) = Y + \Delta Y_{errorX}$ , where $|\Delta Y_{errorX}| = |f_3(X) - f_3(X + \Delta X_{error})| \leq |\Delta X_{error}| \cdot L_{f_3}$, and $L_{f_3}$ is the Lipschitz constant of $f_3$.

   The 3D structure is only part of the structural information, and $|\Delta X_{error}|$ is controllable when MMFF can converge. However, the distinction $|\Delta Y_{\text{error}}|$ between labels and the distance $|\Delta X|$ between train-test samples can be very large. Therefore, we can conclude that $|\Delta Y_{errorX}|_{max} \ll |\Delta Y_{\text{error}}|_{max}$ and $|\Delta Y_{errorX}|_{max} \ll |\Delta Y_{outlier}|_{max}$, where $| \cdot |_{max}$ denotes the upper bound. This conclusion implies that the quality and sparsity of labels have a greater impact on model performance, while the 3D accuracy has a relatively small influence.

2. **Experiment**

   We design experiments to explore the impact of 3D accuracy and label quality on model performance.

   - **Accuracy of 3D Information.** The computation time of DFT on dataset containing millions of samples is far beyond our processing capacity. We are currently unable to further enhance 3D accuracy. In this case, to evaluate the impact of 3D accuracy on model performance, we adopt a reasonable alternative approach. Specifically, we reduce the 3D accuracy by adding normal noise to original MMFF calculation results. The larger the noise, the lower the accuracy. The results are shown in Tab. 52. The results show that the accuracy of 3D information does affect the performance of the model. As the noise level increases, the model's performance gradually declines.

   - **Label Quality.** Label sparsity, including insufficient quantity or uneven distribution, is an important aspect of label quality. To evaluate the effect of label sparsity on model performance, we conduct experiments on the condition prediction task. Specifically,

---

[33]The Lipschitz continuity of a function $f$ is defined as the existence of a Lipschitz constant $L$ such that for all $X_1$ and $X_2$ in the domain of $f$, the inequality $|f(X_1) - f(X_2)| \leq L|X_1 - X_2|$ holds. This means that the rate of change of the function $f$ is bounded by $L$, ensuring that small changes in the input lead to controlled changes in the output. Neural networks can be considered Lipschitz continuous under certain conditions, particularly when they are composed of layers with Lipschitz continuous activation functions and bounded weights.

Table 52: The influence of 3D accuracy on model performance in condition prediction task. The noise level reflects the accuracy of 3D information. The smaller the noise, the higher the 3D accuracy. We use 1/8 of the USPTO-Condition dataset.

| Noise Level | Top-1 | Top-3 | Top-5 | Top-10 | Top-15 |
|---|---|---|---|---|---|
| 0.0 | **0.1773** | **0.2749** | **0.3146** | **0.3534** | **0.3670** |
| 0.05 | 0.1738 | 0.2650 | 0.3044 | 0.3443 | 0.3580 |
| 0.1 | 0.1695 | 0.2633 | 0.2990 | 0.3420 | 0.3570 |
| 0.2 | 0.1635 | 0.2643 | 0.2986 | 0.3397 | 0.3556 |
| 0.4 | 0.1518 | 0.2510 | 0.2940 | 0.3346 | 0.3486 |

Table 53: The influence of label sparsity on model performance in condition prediction task. The degree of label sparsity can be reflected by the size of the scaffold.

| Scaffold Ratio | 1 | 1/2 | 1/4 | 1/8 | 1/16 | 1/32 |
|---|---|---|---|---|---|---|
| **w/o 3D** | 0.3852 | 0.3503 | 0.3116 | 0.2716 | 0.2210 | **0.1907** |
| **w 3D** | **0.3915** | **0.3550** | **0.3146** | **0.2768** | **0.2244** | 0.1857 |

Table 54: Influence of an additional Reaction Hypernode in Rxn Hypergraph across various tasks.

| Method | Cond (T1↑) | | Yield (R2↑) | | | | | | | | Type (ACC↑) | |
|---|---|---|---|---|---|---|---|---|---|---|---|---|
| | U-C | P-C | BH | BH1 | BH2 | BH3 | BH4 | SM | Gram | Subgram | U-T | P-T |
| w/o Hypernode | 0.213 | 0.288 | 0.96 | 0.81 | 0.83 | 0.71 | 0.56 | 0.85 | 0.118 | 0.196 | 0.954 | 0.911 |
| with Hypernode | 0.211 | 0.289 | 0.96 | **0.82** | 0.81 | 0.75 | 0.57 | 0.86 | 0.112 | 0.187 | 0.984 | 0.936 |
| Reaction Graph (ours) | **0.324** | **0.392** | **0.97** | 0.80 | **0.88** | 0.76 | 0.68 | **0.89** | **0.129** | **0.216** | **0.999** | **0.987** |

we separate Pistachio-Condition into multiple scaffolds of different sizes. The size of scaffold can reflect the degree of label sparsity. Specifically, the test set is consistent across all scaffolds. From the first to the sixth group, the training set size halves each time, reducing from full size to $1/32$ of the original. The result is shown in Tab. 53. It is observed that the model's performance decreases with the increase of label sparsity. An interesting point is that, in the smallest scaffold, the effect of 3D information is affected by label sparsity. These results demonstrate that label sparsity is the source of bottleneck for the model.

Based on the results in Tab. 52 and 53, we find that the accuracy of 3D information has a significant impact on model performance. However, the quality of the labels is even more critical. The quality of the labels acts as a bottleneck for model performance, limiting the 3D information to further enhance model performance.

### G.5.2 RXN HYPERGRAPH WITH REACTION HYPERNODE

We can model the interactions between reactants and products by adding another hypernode in the Rxn Hypergraph. Specifically, we can connect the hypernodes of reactants and products with an additional hypernode. The detailed implementation can be found in Sec. C.

To test the performance of this idea, we conduct experiments on various tasks using the hypernode method. The results are presented in Tab. 54.

According to the result, the hypernode method shows improvement in reaction classification by increasing the accuracy on USPTO-TPL by 3% and that of Pistachio-Type by 2.5%, while it has a slight impact on condition and yield prediction. This may be because reaction classification is relatively intuitive, while condition and yield prediction are complex. To improve the performance of condition and yield prediction, more accurate interaction modeling (e.g., Reaction Graph) is needed. Reaction Graph surpasses the performance of the hypernode method, demonstrating its efficiency in interaction modeling.

Table 55: Influence of 3D bond length information on D-MPNN's performance in reaction condition prediction task, using USPTO-Condition dataset.

| Method | 3D Info. | Top-1↑ | Top-3↑ | Top-5↑ | Top-10↑ | Top-15↑ |
|---|---|---|---|---|---|---|
| D-MPNN | w/o 3D | 0.1977 | 0.3000 | 0.3341 | 0.3780 | 0.3924 |
| D-MPNN | w 3D | **0.2030** | **0.3059** | **0.3410** | **0.3830** | **0.3971** |
| Rxn Hypergraph | w/o 3D | 0.2127 | 0.3084 | 0.3447 | 0.3808 | 0.3927 |
| Rxn Hypergraph | w/ 3D | **0.2149** | **0.3113** | **0.3464** | **0.3825** | **0.3949** |

Table 56: Results for leaving group identification of Molecular Graph and Reaction Graph on USPTO, including mean and standard deviation from multiple trials with different random seeds.

| Method | Overall | | | LvG-Specified | | |
|---|---|---|---|---|---|---|
| | ACC | CEN | MCC | ACC | CEN | MCC |
| Molecular Graph | 0.950±0.001 | 0.037±0.001 | 0.538±0.006 | 0.423±0.004 | 0.209±0.002 | 0.497±0.005 |
| Reaction Graph | **0.996±0.001** | **0.002±0.001** | **0.971±0.001** | **0.944±0.001** | **0.035±0.001** | **0.942±0.001** |

### G.5.3 3D INFORMATION IN OTHER GRAPHS

To explore whether 3D information can be effective in other models performing reaction-related tasks, we attempt to incorporate bond length information into the D-MPNN and Rxn Hypergraph. The detailed implementation can be found in Sec. C.1.3 and Sec. C.1.4.

We test the reaction condition prediction performance of D-MPNN and Rxn Hypergraph on USPTO-Condition before and after adding 3D information, and the results are shown in Tab. 55.

According to the result, the incorporation of 3D information effectively improved the top-$k$ performance of D-MPNN and Rxn Hypergraph. The average top-$k$ performance is increased by 1.8% and 0.7% for D-MPNN and Rxn Hypergraph relatively, and the Top-1 accuracy is increased by 2.7% for D-MPNN, demonstrating the efficiency of 3D structural priors in reaction property prediction tasks.

## G.6 STATISTICAL ANALYSIS

### G.6.1 STANDARD DEVIATIONS

To better reflect the model's performance, we not only test the model's best performance on each dataset but also evaluate the standard deviations of the model's performance under multiple random seeds. Specifically, for the results of the Buchwald-Hartwig and Suzuki-Miyaura datasets, we adopt the method of averaging the results from 10 tests as the final outcome. Therefore, we mainly focus on the additional testing of the datasets extracted from USPTO and Pistachio, including datasets on leaving group identification, reaction condition predition, reaction yield prediction, and reaction classification.

In particular, due to the large size of the datasets extracted from USPTO and Pistachio, and for the convenience of t-test statistical significance analysis calculations, we choose the integer four as the number of random tests, which is closest to the square of the t-value (1.96) for a 95% confidence interval. We calculate the results of training under four different random seeds, and then compute the mean and standard deviation of the results.

The mean and standard deviation results of leaving group identification are in Tab. 56, reaction conditions prediction is in Tab. 57, yield is in Tab. 58, and reaction classification is in Tab. 59.

Based on the results, we observe that the standard deviations of the training results are smaller and more stable ($\leq 0.01$) on larger datasets, such as USPTO and Pistachio. In contrast, some test splits of the smaller Buchwald-Hartwig and Suzuki-Miyaura datasets exhibit larger training standard deviations ($> 0.01$).

Table 57: Results for reaction condition prediction of Reaction Graph on USPTO-Condition and Pistachio-Condition, including mean and standard deviation from multiple trials with different random seeds.

| Dataset | Top-1 | Top-3 | Top-5 | Top-10 | Top-15 |
|---|---|---|---|---|---|
| USPTO-Condition | 0.322±0.002 | 0.432±0.003 | 0.469±0.003 | 0.504±0.003 | 0.516±0.003 |
| Pistachio-Condition | 0.391±0.001 | 0.556±0.001 | 0.602±0.001 | 0.636±0.001 | 0.641±0.001 |

Table 58: Results for reaction yield prediction of Reaction Graph on yield datasets, including mean and standard deviation from multiple trials with different random seeds.

| Metrics | BH | BH1 | BH2 | BH3 | BH4 | SM | Gram | Subgram |
|---|---|---|---|---|---|---|---|---|
| Mean±Std of $R^2$ | 0.97±0.01 | 0.80±0.01 | 0.88±0.02 | 0.76±0.02 | 0.68±0.06 | 0.89±0.01 | 0.125±0.006 | 0.211±0.003 |

Table 59: Results for reaction classification of Reaction Graph on USPTO-TPL and Pistachio-Type dataset, including mean and standard deviation from multiple trials with different random seeds.

| Metrics | USPTO-TPL | | | Pistachio-Type | | |
|---|---|---|---|---|---|---|
| | ACC | CEN | MCC | ACC | CEN | MCC |
| Mean±Std | 0.999±0.001 | 0.001±0.001 | 0.999±0.001 | 0.986±0.002 | 0.026±0.004 | 0.985±0.003 |

### G.6.2 STATISTICAL SIGNIFICANCE TESTS

Statistical significance testing is essential for determining whether the Reaction Graph demonstrates an improvement over the baseline method. For each task, we selected the optimal baseline and employed the $t$-test to assess statistical significance. The formula used is $t = \sqrt{n} \cdot \frac{x-b}{s}$, where $t$ represents the t-statistic, indicating the size of the difference relative to the variation in the sample data; $n$ is the sample size, which in this case is 4; $x$ is the mean performance of the Reaction Graph; $b$ is the baseline performance; and $s$ is the standard deviation of the performance measurements. By calculating the $t$-statistic, we can evaluate whether the observed improvement in the Reaction Graph is statistically significant compared to the baseline method.

The significance level P corresponding to the $t$-value can be found in the lookup table[34], and we use $-\log P$ to make the results easier to observe; the larger the value, the higher the probability that the Reaction Graph shows an improvement over the original methods.

We choose the existing optimal model as the baseline method to calculate the $t$-values, which indicates the Molecular Graph in leaving group identification, Parrot (Wang et al., 2023b) in condition prediction, UGNN (Kwon et al., 2022b) in yield prediction, and D-MPNN (Heid & Green, 2021) in reaction classification.

The final calculated significance levels of Reaction Graph on leaving group identification is in Tab. 60, reaction conditions prediction is in Tab. 62, yield is in Tab. 63, and reaction classification is in Tab. 61.

Based on the results of the $t$-test statistical significance analysis, the advantages achieved by our model on the majority of datasets and metrics have a high level of confidence ($-logP > 3$ indicates confidence $> 95\%$). This indicates that the improvement of the Reaction Graph over existing methods is significant, demonstrating the effectiveness of our approach.

### G.6.3 CROSS-VALIDATION RESULTS

Cross-validation can reduce the bias introduced by the unevenly distributed chemical reaction dataset, ensuring the model's generalization. We evenly split the dataset used in each task and tested the model's performance using K-fold cross-validation.

---

[34]https://en.wikipedia.org/wiki/Student%27s_t-distribution

Table 60: Results for statistical significance analysis of Reaction Graph on leaving group identification in USPTO, using $t$-test method from the mean and standard deviations of multiple trials with different random seeds. The baseline method is Molecular Graph.

| Metrics | Overall | | LvG-Specified | |
|---------|---------|------|---------------|------|
| | ACC | MCC | ACC | MCC |
| $-logP$ | 13.47 | 20.19 | 20.75 | 20.28 |

Table 61: Results for statistical significance analysis of Reaction Graph on reaction classification on USPTO-TPL and Pistachio-Type, using $t$-test method from the mean and standard deviations of multiple trials with different random seeds. The baseline method is D-MPNN.

| Metrics | USPTO-TPL | | Pistachio-Type | |
|---------|-----------|------|----------------|------|
| | ACC | MCC | ACC | MCC |
| $-logP$ | 4.27 | 4.27 | 4.27 | 3.80 |

Table 62: Results for statistical significance analysis of Reaction Graph on reaction condition prediction in USPTO-Condition and Pistachio-Condition, using $t$-test method from the mean and standard deviations of multiple trials with different random seeds. The baseline method is Parrot. The value in the table is $-logP$.

| Dataset | Top-1 | Top-3 | Top-5 | Top-10 | Top-15 |
|---------|-------|-------|-------|--------|--------|
| USPTO-Condition | 11.81 | 8.69 | 7.38 | 6.43 | 6.43 |
| Pistachio-Condition | 13.12 | 11.52 | 9.90 | 8.90 | 9.18 |

Table 63: Results for statistical significance analysis of Reaction Graph on reaction yield prediction in Buchwald-Hartwig, Suzuki-Miyaura and USPTO-Yield dataset, using $t$-test method from the mean and standard deviations of multiple trials with different random seeds. The baseline method is UGNN.

| Metrics | BH | BH1 | BH2 | BH3 | BH4 | SM | Gram | Subgram |
|---------|------|------|------|------|------|------|------|---------|
| $-logP$ | 0.69 | 8.74 | 0.69 | 5.52 | 6.69 | 0.69 | 3.27 | 7.84 |

Table 64: Cross-validation results for leaving group identification of Molecular Graph and Reaction Graph on USPTO, including mean and standard deviation from multiple trials with different train/test splits.

| Method | Overall | | | LvG-Specified | | |
|--------|---------|-----|-----|---------------|-----|-----|
| | ACC | CEN | MCC | ACC | CEN | MCC |
| Molecular Graph | 0.948±0.004 | 0.038±0.003 | 0.535±0.015 | 0.418±0.010 | 0.203±0.001 | 0.495±0.008 |
| Reaction Graph | **0.995±0.002** | **0.004±0.001** | **0.955±0.012** | **0.915±0.022** | **0.050±0.011** | **0.912±0.022** |

Specifically, for the yield dataset, the two small HTE datasets, Buchwald-Hartwig and Suzuki-Miyaura, already have 10 predefined splits. Therefore, we directly use the previously tested results. For the larger condition and type datasets extracted from USPTO and Pistachio, we perform random shuffling before splitting. Due to their large scale, the data distribution differences within each split are minimal. Considering the training cost and the reuse of experiment results, we choose to divide them into 9 folds (excluding the original test set) and conduct three random tests. Including the validation results from the original splits, there are a total of four tests. We take the average as the result of K-fold cross-validation to demonstrate the differences in dataset distribution.

The final K-fold cross validation results of Reaction Graph on leaving group identification is in Tab. 64, reaction conditions prediction is in Tab. 65, yield is in Tab. 66, and reaction classification is in Tab. 67.

Based on the results of cross-validation, for large reaction datasets (USPTO, Pistachio), the data distribution differences between different splits are minimal due to the sufficient amount of data, leading to small standard deviations ($\leq 0.01$) in the training results. For the high-quality

Table 65: Cross-validation results for reaction condition prediction of Reaction Graph on USPTO-Condition and Pistachio-Condition, including mean and standard deviation from multiple trials with different train/test splits.

| Dataset | Top-1 | Top-3 | Top-5 | Top-10 | Top-15 |
|---|---|---|---|---|---|
| USPTO-Condition | 0.320±0.003 | 0.430±0.002 | 0.467±0.002 | 0.503±0.002 | 0.516±0.002 |
| Pistachio-Condition | 0.391±0.002 | 0.556±0.002 | 0.603±0.001 | 0.636±0.001 | 0.641±0.001 |

Table 66: Cross-validation results for yield prediction of Reaction Graph from multiple trials with different train/test splits.

| Metrics | Buchwald-Hartwig | | Suzuki-Miyaura | |
|---|---|---|---|---|
| | Mean | Std | Mean | Std |
| $R^2$ | 0.97 | 0.01 | 0.89 | 0.01 |

Table 67: Cross-validation results for reaction classification of Reaction Graph on USPTO-TPL and Pistachio-Type dataset, including mean and standard deviation of ACC, MCC and CEN metrics from multiple trials with different train/test splits.

| Metrics | USPTO-TPL | | | Pistachio-Type | | |
|---|---|---|---|---|---|---|
| | ACC | CEN | MCC | ACC | CEN | MCC |
| Mean±Std | 0.999±0.001 | 0.001±0.001 | 0.999±0.001 | 0.987±0.002 | 0.025±0.003 | 0.986±0.002 |

Table 68: Confidence interval results ([Min,Max]) for leaving group identification of Molecular Graph and Reaction Graph on USPTO, calculated using normal distribution method from mean and standard deviations of multiple trials with different random seeds.

| Method | Type | Overall | | | LvG-Specified | | |
|---|---|---|---|---|---|---|---|
| | | ACC | CEN | MCC | ACC | CEN | MCC |
| Molecular Graph | Min | 0.949 | 0.036 | 0.532 | 0.419 | 0.207 | 0.492 |
| | Max | 0.951 | 0.038 | 0.544 | 0.427 | 0.211 | 0.502 |
| Reaction Graph | Min | **0.995** | **0.001** | **0.970** | **0.943** | **0.034** | **0.941** |
| | Max | **0.997** | **0.003** | **0.972** | **0.945** | **0.036** | **0.943** |

Table 69: Confidence interval results ([Min,Max]) for reaction condition prediction of Reaction Graph on USPTO-Condition and Pistachio-Condition dataset, calculated using normal distribution method from mean and standard deviations of multiple trials with different random seeds.

| Dataset | Type | Top-1 | Top-3 | Top-5 | Top-10 | Top-15 |
|---|---|---|---|---|---|---|
| USPTO-Condition | Min | 0.320 | 0.429 | 0.466 | 0.501 | 0.513 |
| | Max | 0.324 | 0.435 | 0.472 | 0.507 | 0.519 |
| Pistachio-Condition | Min | 0.390 | 0.555 | 0.601 | 0.635 | 0.640 |
| | Max | 0.392 | 0.557 | 0.603 | 0.637 | 0.642 |

Table 70: Confidence interval results ([Min,Max]) for reaction yield prediction of Reaction Graph on Buchwald-Hartwig, Suzuki-Miyaura and USPTO-Yield dataset, calculated using normal distribution method from mean and standard deviations of multiple trials with different random seeds.

| Metrics | Bound | BH | BH1 | BH2 | BH3 | BH4 | SM | Gram | Subgram |
|---|---|---|---|---|---|---|---|---|---|
| $R^2$ | Min | 0.96 | 0.79 | 0.87 | 0.75 | 0.64 | 0.88 | 0.119 | 0.208 |
| | Max | 0.98 | 0.81 | 0.89 | 0.77 | 0.72 | 0.90 | 0.131 | 0.214 |

Buchwald-Hartwig and Suzuki-Miyaura datasets, the standard deviations introduced by different training/testing splits is also small ($\leq 0.01$).

Table 71: Confidence interval results ([Min,Max]) for reaction classification of Reaction Graph on USPTO-TPL and Pistachio-Type dataset, calculated using normal distribution method from mean and standard deviations of multiple trials with different random seeds.

| Metrics | Type | USPTO-TPL | | | Pistachio-Type | | |
|---|---|---|---|---|---|---|---|
| | | ACC | CEN | MCC | ACC | CEN | MCC |
| Mean±Std | Min | 0.998 | 0.000 | 0.998 | 0.984 | 0.022 | 0.982 |
| | Max | 1.000 | 0.002 | 1.000 | 0.988 | 0.030 | 0.988 |

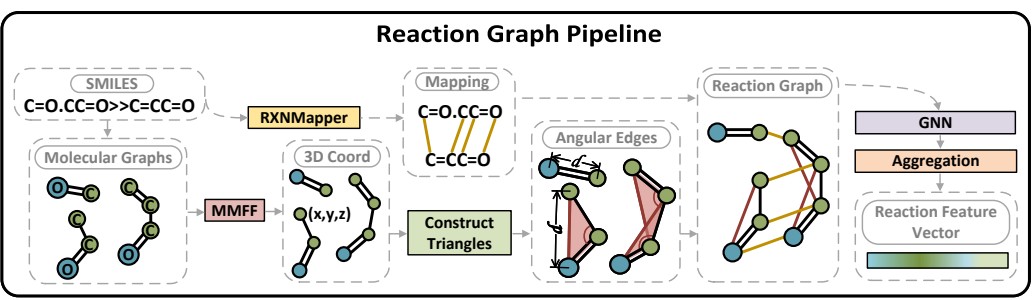

Figure 24: Illustration of Reaction Graph (RG). Our method first constructs molecular graph based on SMILES and predict atomic mapping for creating reaction edges using RXNMapper (Schwaller et al., 2020). Then, 3D atom coordinates are calculated using MMFF94 (Halgren, 1996) and angular edges are constructed for each bond angle. Finally, a GNN is used to extract the unified reaction feature vector based on RG.

### G.6.4 CONFIDENCE INTERVALS

We utilize the normal distribution method to calculate the confidence interval (CI) for our model using the formula $CI = \bar{x} \pm z \cdot \frac{s}{\sqrt{n}}$, where $\bar{x}$ represents the sample mean, $z$ is the z-score corresponding to the desired confidence level, $s$ is the standard deviation of the sample, and $n$ is the sample size. For a 95% confidence level, the corresponding z-score is $z = 1.96$.

The confidence interval results of leaving group identification are in Tab. 68, reaction condition prediction is in Tab. 69, yield is in Tab. 70, and reaction classification is in Tab. 71.

The results of our confidence interval analysis align with our results in the mean and standard deviation section (Sec. G.6.1). In larger datasets, such as USPTO and Pistachio, the 95% confidence interval for model performance is narrow ($\leq 0.01$), indicating that the model demonstrates stable performance. However, in smaller datasets like Buchwald-Hartwig and Suzuki-Miyaura, some test sets exhibit wider confidence intervals ($> 0.01$), suggesting that the model experiences a degree of training variance that can not be ignored.

## H SUPPLEMENTARY FIGURES

### H.1 REACTION GRAPH PIPELINE

To clearly illustrate the process of constructing the Reaction Graph (RG), we outline the pipeline for both the RG construction and feature extraction. As depicted in Fig. 24, we start with the SMILES representation of a chemical reaction. We first convert it into a molecular graph. In molecular graph, atoms are represented as nodes and chemical bonds as edges.

Simultaneously, we employ RXNMapper (Schwaller et al., 2020) to predict the atomic mapping relationships within the chemical reaction. These mappings are essential for constructing reaction edges that connect corresponding atoms in the reactants and products. Next, we calculate the 3D atomic coordinates using the MMFF94 method (Halgren, 1996), and we create angular edges for each bond angle.

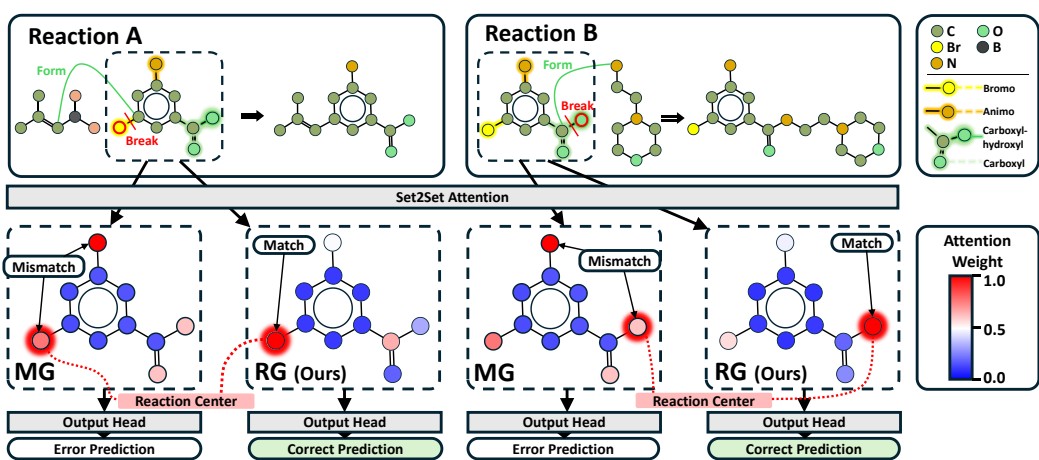

Figure 25: Visualization of attention weights and prediction results for two reactions involving the bromo and carboxyl-hydroxyl groups in $C_7H_6BrNO_2$. The model using the Molecular Graph (MG) focuses on atoms that are less relevant to the reaction, thus leading to prediction errors. In contrast, the model equipped with the Reaction Graph (RG) accurately concentrates on the reaction center.

Table 72: Comparison between Data Preprocessing Tools, including functionality, precision, time efficiency, application scope, and limitations.

| Tools | Function | Precision | Efficiency | Application Scope | Limitation |
|---|---|---|---|---|---|
| ETKDG | Calculate/Optimize 3D Conformation | Low | High | Small Organic Molecules | Fail on Some Metal-Complex |
| UFF | Calculate/Optimize 3D Conformation | Relatively Low | High | Universal | Low Accuracy |
| MMFF | Calculate/Optimize 3D Conformation | Medium | High | Small Organic Molecules | Fail on Some Metal-Complex |
| DFT | Calculate/Optimize 3D Conformation | High | Low | Depend on Basis Set | Slow |
| RXNMapper | Predict Atom Mapping | Relatively High | High | Small Molecules | Lack of Rule Constraints |
| NameRXN | Predict Atom Mapping + Classify Reaction | High | High | Limited by Manual Rule | Manual Rule Based |

At this stage, the construction of the RG is complete. Finally, we utilize a GNN to extract the features of the constructed RG, and aggregate the node features into a unified reaction feature vector. This feature vector is then used for various reaction property prediction tasks.

## H.2 ATTENTION WEIGHT VISUALIZATION

In the reaction task, the attention weight for each atom reflect the model's understanding of the reaction mechanism. The more the model focuses on the reaction center, the more accurate its understanding of the reaction mechanism, leading to better prediction results. Conversely, if the model pays more attention to atoms unrelated to the reaction, it will struggle to understand the reaction correctly, resulting in prediction errors.

Fig. 25 illustrates the attention weights and condition prediction results for two reactions. Both reactions utilize $C_7H_6BrNO_2$ as the reactant, with the reaction center lying in the bromo group and the carboxyl-hydroxyl group, respectively.

The result shows that the model using the Molecular Graph focuses on atoms that are less relevant to the reaction, thus leading to prediction errors. In contrast, the model equipped with the Reaction Graph accurately concentrates on the reaction center, thereby making the correct prediction.

## I TOOLKITS

We use a series of tools for chemical property calculations, data analysis, and Reaction Rraph construction. We have conducted a brief comparison and summary of these tools, and the results are in Tab. 72.

## I.1 CALCULATE/OPTIMIZE 3D CONFORMATION

ETKDG, UFF, MMFF, and DFT are algorithms for calculating or optimizing 3D conformations. ETKDG and MMFF are accurate for small organic molecules but less effective for larger ones and metal complexes. UFF is more suitable for handling metal complexes. DFT, based on quantum chemistry, provides high precision for various compounds but is inefficient. In this paper, we use large real-world chemical databases, USPTO and Pistachio, with total millions of entries and complex molecular distributions. We find that even with the simplest basis sets, DFT is too time-consuming. Hence, we initialize conformations using ETKDG and then optimize them with MMFF94. For molecules that MMFF94 cannot handle (e.g., involving heavy metals), we use UFF for conformation optimization.

Chemical toolkits such as RDKit and OpenBabel provide implementations of the MMFF and UFF algorithms, with RDKit offering an easy way to initialize conformations using ETKDG. As for calculations like DFT, chemical toolkits such as Psi4 and PySCF provide relevant functionalities. These tools all offer interfaces in Python.

## I.2 PREDICT ATOM MAPPING

RXNMapper and NameRXN are tools for atomic mapping. RXNMapper is open-source and based on an unsupervised language model, capable of providing efficient and accurate predictions. NameRXN is a commercial rule-based tool, offering highly reliable results, though some reactions fall outside its rule coverage. After evaluating the mapping performance on the USPTO dataset, we choose RXNMapper because it demonstrates high stability and most of its predictions are accurate.

## I.3 CLASSIFY REACTION

NameRXN is also a tool for reaction classification. In this paper, the labels of the Pistachio dataset are annotated using NameRXN.

## J TERMINOLOGY LIST

We explain the methods mentioned in the paper in the Sec. A and Sec. B, and here we provide additional explanations for some remaining concepts.

1. **Graph Neural Network (GNN).** A type of neural network designed to process data structured as graphs, capturing relationships between nodes.

2. **Message Passing (MP).** A mechanism in graph neural networks where nodes exchange information to update their representations based on neighboring nodes.

3. **Molecule.** A group of atoms bonded together, representing the smallest fundamental unit of a chemical compound.

4. **Edge.** A connection between two nodes in a graph, representing a relationship or interaction.

5. **Node/Vertex.** A fundamental unit in a graph representing an entity, such as an atom in a molecular graph.

6. **Atom.** The basic unit of a chemical element, consisting of protons, neutrons, and electrons.

7. **Bond.** A lasting attraction between atoms that enables the formation of chemical compounds.

8. **Bond Length.** The distance between the nuclei of two bonded atoms in a molecule.

9. **Bond Angle.** The angle formed between three atoms, where the central atom is bonded to the other two.

10. **Reaction.** A process in which one or more substances are transformed into different substances.

11. **Retrosynthesis.** The process of deconstructing a complex molecule into simpler precursor structures to design synthetic pathways.

12. **Catalyst.** A substance that accelerates a chemical reaction without being consumed in the process.

13. **Solvent.** A substance, typically a liquid, in which solutes are dissolved to form a solution.

14. **Reagent.** A substance used in a chemical reaction to detect, measure, or produce other substances.

15. **Computational Chemistry.** The use of computer simulations to solve chemical problems and predict molecular behavior.

16. **Atom Mapping.** The process of identifying corresponding atoms in different molecular structures, often used in reaction analysis.

17. **High-throughput Experiment (HTE).** A method that allows rapid testing of multiple conditions or compounds simultaneously to accelerate research.

18. **Conformer.** A specific spatial arrangement of atoms in a molecule that can be interconverted by rotation around single bonds.

19. **Conformation.** The 3D shape of a molecule resulting from the rotation around its bonds.

20. **Bond Edge.** Represents an edge in the reaction graph that corresponds to a chemical bond, distinguished from Reaction Edge and Angular Edge.

