# OpenReview forum: "Reaction Graph: Toward Modeling Chemical Reactions  with 3D Molecular Structures"
_ICLR.cc/2025/Conference — Submitted to ICLR 2025_

### Official Review · Reviewer_L6NH · 2024-10-22

**Soundness:** 2
**Presentation:** 2
**Contribution:** 2
**Rating:** 3
**Confidence:** 4

**Summary:**

This paper is focused on the problem of modelling chemical reactions using machine learning techniques. The potential application of this work lies within the material science domain, primarily in Chemistry and drug design.
The authors introduce the concept of Reaction Graph (RG), which is a graph representation enhanced by 3D structural information.

One of the key objections against previous methods in the field of study is that so-called "reaction information" that is relevant to the understanding of chemical reactions is usually overlooked and not used. In previous reaction modeling works, the main focus is still molecular representation learning. With that in mind, the authors propose a novel unified graph representation that includes a reaction edge. This additional edge links the same atoms on either side of the chemical reaction - from reagents to products respectively, hence, the inductive bias about the reaction itself is preserved during the message passing.

Experiments were conducted for a wide range of tasks, and the method shows promising results.

**Strengths:**

The formal requirements are met, and the overall structure of the paper is cohesive.

 - Fig. 2 is a good example of an illustration of the overall pipeline and model architecture.
 - Novelty in terms of expansion of data representation in chemical reaction modeling.
 - Multiple experiments are performed (reaction condition prediction, reaction yield prediction, etc.), as well as the ablation study.

**Weaknesses:**

Figures 1 and 3 are overloaded and therefore hard to interpret.

 - Fig. 1 loses its informativity due to the number of subgraphs included. Most of it showcases the schematics of other existing methods, while only the far right is important for the reader (Reaction Graph Pipeline). Since Fig.1 is referenced many times throughout the paper, those references point to a small fraction of the data. I'd suggest thinking about another layout, either several illustrations for several disjoint figures or even removing some unrelated to the contribution of this work illustrations (e.g. Molecule Graph, Rnx Hypergraph). You can still reference these works in the text.

 - In Fig. 3 it is not clear how to look for the "Attention weight" since the color bar has all the colors for a) different atom types; b) bromo-/amino-/other group highlights. Another question that arises, is what does "Mismatch" imply?


 I would strongly recommend proofreading the article again to correct some sentences and formulations. Examples may include:
 - poor sentence formulation: "If the edge is empty or a reaction edge, the length is set to 0."
 - "...**a** attention-based aggregation..."

**Questions:**

1) For additional 3D information one can utilize xyz coordinates of atoms, and therefore capture spatial information. Have you considered using coordinates?

2) During a couple of iterations of message passing, node embedding will get a cumulative update from their second neighbor in the molecule chain. From this perspective, I don't see how angular edges are conceptually any different from the concept of additional message passing in approaches with functional groups [1] or pseudoatoms/neural atoms [2]. Can you elaborate more on this point?

3) In the section "Leaving Group Identification Analysis", could you explain what does "Overall" column have to do with the LvG identification task?


4) In the subsection "Dataset" of section "4.3 Reaction Condition Prediction" - it isn't clear what was meant by "... construct Pistachio-Condition from the Pistachio database by thorough cleaning and filtering". What was the procedure and what was the purpose of this "filtering" of data?
_____
**References:**

[1] Hajiabolhassan, Hossein, et al. "FunQG: Molecular representation learning via quotient graphs." Journal of chemical information and modeling 63.11 (2023): 3275-3287.

[2] Li, Xuan, et al. "Neural Atoms: Propagating Long-range Interaction in Molecular Graphs through Efficient Communication Channel." The Twelfth International Conference on Learning Representations. 2024.

---

> ### Author Response · Authors · 2024-11-24
> **Part Ⅰ: Figure Revision, XYZ Coordinates Experiment and Difference from FunQG/Neural Atoms**
>
> Thank you for acknowledging our **novelty** and the structure of the paper **cohesive**. We also appreciate your constructive suggestions.
>
> ## Why did we include both the schematics of other existing methods and the schematics of our method in Fig.1?
>
> The main purpose of Fig. 1 is to guide readers through the transition from existing approaches to our novel contribution. This helps readers efficiently compare methods and grasp the unique aspects of our approach.
>
> We also include a revised schematic of our Reaction Graph Pipeline, which can be found in Sec. H.1.
>
> ## How to look for the "Attention weight" and what does "Mismatch" imply in Fig.3?
>
> In Figure 3, the node colors in the schematics below "Set2Set Attention" represent Attention Weights, corresponding to the colors in the Colorbar.
>
> "Mismatch" indicates that the atom with the highest Attention Weight is not the same as the reaction center atom (i.e. the atom where the reaction takes place). This situation occurs in the molecular graph, indicating that the model using the molecular graph has incorrectly understood the mechanism of the chemical reaction.
>
> We also added the revised figure to avoid the color conflict between atomic types and the Attention Weights in Sec.4.1 and Sec.H.1.
>
> ## Sentences and Formulations
>
> We have addressed these issues in the revision version.
>
> ## Why not use XYZ Coordinate
>
> Directly using XYZ coordinates is not rotational and transcational invariant. The property of the chemical reaction is independent of the pose of the molecule. If we directly use the XYZ coordinates, the chemical reaction representation will change with their pose, increasing the chanllenge for neural networks to recognizing reactions.
>
> To verify the above statements, we conduct the following experiment: we directly use XYZ coordinates as an additional node feature, and compare its performance with that of our Reaction Graph on USPTO-Condition dataset. The result is as shown below:
>
> **Table 1:** Influence of XYZ coordinates on Reaction Graph.
> | **Method**                  | **Top-1** ↑ | **Top-3** ↑ |
> |-----------------------------|--------------|--------------|
> | with XYZ Coordinate         | 0.3123      | 0.4243       |
> | without 3D Information      | 0.3133      | 0.4248       |
> | Reaction Graph (ours)      | **0.3246**  | **0.4343**   |
>
>
> The results show that our method effectively provides 3D prior, while the XYZ coordinates is difficult for the model to understand and utilize, resulting in drop in performance. We provide specific implementations and result analyses in the Sec.G.5.1.
>
> ## Difference between our Reaction Graph’s Angular Edge and the supernode in Functional Groups or Pseudoatoms/Neural Atoms.
>
> ### Motivation
>
> Functional Groups[1] or Pseudoatoms/Neural Atoms[2] aims to facilitate long-range message passing. Our Angular Edge is designed to incorporate 3D information for structure modeling purposes.
>
> ### Methodology
>
> Functional Groups or Pseudoatoms/Neural Atoms shorten message passing path by reducing the size of the molecular graph. Our Angular Edge constructs triangles in molecular graph to convey bond angle feature.
>
> ### Experiment
>
> To verify these distinctions, we set the lengths of the Angular Edges to zero to eliminate the 3D information while still connecting related atoms. The results are as follows:
>
> **Table 2:** Impact of connection and length (for 3D modeling) on Angular Edge. Angular Edge is mainly used for providing 3D infromation, instead of long-range message passing.
> | **Method**                                           | **Top-1** ↑ | **Top-3** ↑ |
> |-----------------------------------------------------|--------------|--------------|
> | w/o connection & w/o length for 3D modeling         | 0.3165      | 0.4251       |
> | w/ connection & w/o length for 3D modeling          | 0.3150      | 0.4259       |
> | w/ connection & w/ length for 3D modeling           | **0.3246**  | **0.4343**   |
>
> The result shows that the Angular Edge without 3D prior did not have a significant impact on performance, while Angular Edge with edge length significantly improved performance. This indicates that the role of the Angular Edge in providing messages is greater than its role in passing messages.
>
> [1] Hajiabolhassan, Hossein, et al. "FunQG: Molecular representation learning via quotient graphs." Journal of chemical information and modeling 63.11 (2023): 3275-3287.
>
> [2] Li, Xuan, et al. "Neural Atoms: Propagating Long-range Interaction in Molecular Graphs through Efficient Communication Channel." The Twelfth International Conference on Learning Representations. 2024.

---

> ### Author Response · Authors · 2024-11-24
> **Part Ⅱ: Overall Column in LvG Identification and Purpose/Procedure of Filtering Data**
>
> ## What Does Overall Column Have to Do with the LvG Identification Task?
>
> In a chemical reaction, a leaving group (LvG) is an atomic group. It is present in the reactants but absent in the products. Leaving groups are classified into different types based on their structures.
>
> Leaving group identification is an atom-level multi-classification task. It determines if an atom belongs to a leaving group. It also identifies the type of leaving group.
>
> The Overall column records the classification accuracy for all atoms, while the LvG column excludes non-leaving group atoms when calculating accuracy. The Overall column indicates the model's ability to locate leaving groups, whereas the LvG column assesses its ability to distinguish between different types of leaving groups.
>
> For example:
>
> **Table 3:** Leaving group (LvG) identification results of Molecular Graph and Reaction Graph representations, with Overall and LvG atom-specific evaluation.
> | **Representation**         | **Overall**             |                     |                     | **LvG Atom-Specific** |                     |                     |
> |----------------------------|-------------------------|---------------------|---------------------|-----------------------|---------------------|---------------------|
> |                            | **ACC** ↑               | **CEN** ↓           | **MCC** ↑           | **ACC** ↑             | **CEN** ↓           | **MCC** ↑           |
> | Molecular Graph        | 0.950                   | 0.036               | 0.549               | 0.448                 | 0.201               | 0.519               |
> | Reaction Graph (ours)  | **0.997**               | **0.002**           | **0.973**           | **0.947**             | **0.031**           | **0.945**           |
>
>
> As shown in the table above, in the molecular graph, the Overall column shows relatively low accuracy, suggesting that the model mislocates leaving groups. In contrast, the Reaction Graph not only locates leaving groups correctly but also classifies them accurately.
>
> ## What Was The Purpose and Procedure of Filtering of Data
> ### Purpose
> The purpose of filtering is to enhance the quality of the dataset by eliminating erroneous or incomplete records from the reaction database. We extract reaction task datasets from real-world reaction databases. These databases often contain errors and missing labels. Without filtering, these issues can negatively impact model performance.
> ### Procedure
> The cleaning steps starts with parsing each reaction with chemical toolkit (e.g. RDKit) and discarding unrecognized reactions. Then, reactions which lack task-related labels (e.g. reaction conditions) are removed. More details on the data processing workflow and results can be found in Sec.F.

---

> ### Author Response · Authors · 2024-11-27
>
> Dear Reviewer L6NH,
>
> Thank you very much for your time and effort in reviewing our submission.
> We have tried our best to address the concerns and problems mentioned.
> We look forward to your feedback and further discussion.
> Feel free to let us know if there is anything unclear.
> We are happy to provide additional information.
> Thank you once again.
>
> Best wishes,
>
> Paper 4277 Authors

---

> ### Author Response · Authors · 2024-12-03
>
> Dear Reviewer L6NH,
>
> Thank you very much for your efforts in reviewing our paper.
> The discussion phase is nearing its end, and we are looking forward to receiving your feedback on our responses.
> Should you have any further concerns, we are glad to provide additional information.
> Thank you once again.
>
> Best regards,
>
> Paper 4277 Authors

---

### Official Review · Reviewer_xQ99 · 2024-10-28

**Soundness:** 3
**Presentation:** 3
**Contribution:** 2
**Rating:** 8
**Confidence:** 3

**Summary:**

This paper presents a new method, Reaction Graph, for modeling chemical reactions. The main contributions include:

1. Constructing a reaction graph that captures interatomic relationships in both reactants and products.
2. Incorporating 3D structural information into reaction modeling.

Extensive experiments validate the effectiveness of the proposed method across various tasks related to chemical reactions.

**Strengths:**

1. The paper is well-written and easy to understand.
2. The motivation for introducing a reaction graph to unify reactants and products in reaction modeling is clear and reasonable.
3. The authors conducted comprehensive experiments across three different chemical reaction tasks, demonstrating the effectiveness of the proposed method.

**Weaknesses:**

1. The concept of unifying reactants and products seems incremental (modeling individual molecules -> unifying reactants/products (Rxn Hypergraph) -> unifying both reactants and products (Reaction Graph)).
2. The idea of utilizing 3D structures has been extensively explored in molecular modeling, and thus the novelty of extending its application to reaction modeling is limited.
3. The efficiency analysis is insufficient, both experimentally and theoretically. Experimentally, the paper compares the efficiency of the proposed method with and without 3D structure information, rather than against other baseline methods. Theoretically, the complexity analysis in Appendix E.1 focuses solely on graph size without providing a formal complexity calculation of the entire computation process. Additionally, the authors should account for the additional computational burden in preprocessing steps (such as atomic mapping prediction).

**Questions:**

1. Is it possible to model the interactions between reactants and products by simply adding another hypernode in the Rxn Hypergraph method (connecting to the existing hypernodes of reactants and products)?
2. In the attention-based aggregation process (Line 230), why the authors choose the LSTM-based method which is relatively old (in 2016)? Have the authors explored more recent techniques such as SetTransformer[1]?

[1] Lee et al. Set transformer: A framework for attention-based permutation-invariant neural networks. In ICML, 2019.

---

> ### Author Response · Authors · 2024-11-23
> **Part Ⅰ: The Novelty of Modeling the 3D Interaction between Reactant and Product**
>
> Thank you for acknowledging our paper **well-written** and our approach **clear and reasonable**. We also appreciate your constructive suggestions.
>
> ## Innovative Modeling of the Interaction between Reactant and Product
>
> When employing AI techniques to solve chemistry problems, it involves two key aspects: molecular structure modeling and molecule interaction modeling. Existing methods typically focus on individual molecule modeling, which can address a few challenges such as molecule property, function, and attribute prediction. However, when it comes to chemical reactions, molecule interaction modeling at the atom level is required. Due to the lack of interaction modeling, existing methods have to first employ individual molecule modeling for reactants and products and then concatenate the global features for reaction reasoning, where the relation between reactants and products are not well processed. In contrast, our Reaction Graph fills the gap and can capture the interactions between molecules at the atomic level.
>
>
> ## Incorporating 3D Information in Reactions vs. Incorporating 3D Information Molecules
>
> 3D information is crucial in modeling molecular structures, yet its application in molecular interactions (reactions) remains underexplored. Our Reaction Graph focuses on modeling the interactions between reactants and products. It innovatively integrates 3D information with reaction edges to model structural transformations at the atomic level. Additionally, it incorporates angular edges to represent bond angle features, which address the rotational degrees of freedom. These aspects are treated distinctly in structural and interaction modeling. This innovative approach positions Reaction Graph as a pioneering 3D method specifically tailored for the molecular interaction modeling of chemical reactions.

---

> ### Author Response · Authors · 2024-11-23
> **Part Ⅱ: Analyzing the Additional Computational Cost in Preprocessing Reaction Graph**
>
> ## Analyzing the Additional Computational Cost in Preprocessing
>
> The proposed Reaction Graph includes angular edges, bond edge lengths, and reaction edges.
>
> To obtain angular edges and bond edge lengths, it is necessary to calculate the 3D conformers of reactants and products based on their SMILES representations. To acquire reaction edges, the RXNMapper tool is used to calculate the mapping of atoms involved in reactions. Therefore, the preprocessing time primarily involves the calculation of reaction edges and the generation of 3D conformers.
>
> The table below details the time required to acquire this information during data preprocessing, in relation to the SMILES length (number of atoms). During data preprocessing, the primary time expenditure is attributed to the calculation of 3D conformers. However, this is an unavoidable and inevitable cost for any method that utilizes 3D structural information.
>
> **Table 1:** Data pre-processing time for different representations under varying numbers of graph nodes. Time units are milliseconds (ms).
> | **Representation**    | **Reaction Edge Calculation** | **3D Conformer Generation** | **35** | **60** | **85** | **110** | **135** | **160** | **185** | **210** | **235** | **260** | **285** | **310** | **335** | **360** |
> |-|-|-|-|-|-|-|-|-|-|-|-|-|-|-|-|-|
> | Reaction Graph        | √                | √              | 3.2    | 5.6    | 10.1   | 18.1    | 28.7    | 24.9    | 52.5    | 54.0    | 72.4    | 82.8    | 100.3   | 244.4   | 585.6   | 923.3   |
> | w/o Reaction Edge     | ×                | √              | 3.6    | 4.4    | 7.2    | 16.8    | 26.0    | 28.8    | 45.9    | 57.5    | 69.3    | 81.3    | 91.9    | 157.8   | 722.1   | 945.9   |
> | w/o Angular Edge      | √                | ×               | 1.2    | 1.5    | 2.3    | 2.7     | 3.5     | 3.9     | 4.8     | 5.4     | 6.1     | 6.8     | 5.6     | 5.5     | 21.7    | 29.3    |
> | Molecular Graph       | ×                 | ×               | 0.7    | 1.0    | 1.3    | 1.5     | 2.4     | 2.2     | 3.0     | 3.4     | 4.7     | 4.6     | 4.9     | 4.1     | 15.9    | 19.9    |
>
> More details of the preprocessing cost analysis can be found in Sec.E.

---

> ### Author Response · Authors · 2024-11-23
> **Part Ⅲ: Computational Time Analysis for Reaction Graph and Other Models**
>
> ## Computational Time Analysis for Reaction Graph and Other Models
>
> ### Our method
> 1. **Theoretical Analysis**
>
>     We define all graph-based representations of chemical reactions as $ G(V, E) $, where $ V $ represents the set of atoms and $ E $ represents the set of bonds. The computational complexity of the Reaction Graph depends on the number of atoms $ |V| $, the number of bonds $ |E| $, the maximum degree $ D $ of the atoms, and the specific implementation of the graph.
>
>     In Reaction Graph, a reactant atom can match with at most one product atom, so the number of reaction edges does not exceed half the total number of atoms, $ |V|/2 $. Furthermore, as defined for angular edges in Fig. 1, the maximum number of angular edges is $ |V| D^2 $.
>
>     Thus, the computational complexity contributed by bond edges is $ |E| $, by reaction edges is approximately $ |V|/2 $, and by angular edges is around $ |V| D^2 $. Consequently, the total computational complexity of Reaction Graph can be approximated as $ |E| + |V|/2 + |V| D^2 $.
>
>
>
>
> 2. **Experimental Results**
>
>     We assess the impact of reaction edges and angular edges in the Reaction Graph on model inference speed as the number of atoms $ |V| $ increases. The findings are presented in rows 1-3 of the table below. The time cost associated with reaction edges is lower than that caused by angular edges.
>
>
>
> ### Compared with Other Methods
> 1. **Theoretical Analysis**
>     + Graph-based Method
>
>         When the molecular graph is represented as $G(V,E)$, $\vert V \vert$ indicates the total number of atoms in reactants and products, and $\vert E \vert$ represents the total count of bonds. The computation cost for the molecular graph is $\vert E \vert$. The computation cost of Rxn Hypergraph is at least $\vert E \vert + \vert V \vert$. The edge count in Condensed Graph of Reactions (CGR) is roughly $\vert E \vert /2$. However, when handling the same amount of information, the dimensions of the fully connected layers in CGR far exceed those in the Reaction Graph and Rxn Hypergraph. This leads to greater computational overhead.
>
>     + SMILES-based Method
>
>         SMILES-based methods are generally based on Transformer. The complexity of Transformer is $ \vert L \vert ^2 $, where $\vert L \vert$ is the length of the input sequence. As a result, the computation cost of SMILES-based methods are roughly $\vert L \vert ^2$. Since there is no direct conversion between the length of SMILES strings $\vert L \vert$ and the number of nodes $\vert V \vert$ and edges $\vert E \vert$ in molecular graphs, we directly measured their computational efficiency through experiments.
>
> 2. **Experimental Results**
>     + Compared with graph-based methods
>
>         We compare the inference time of Reaction Graph and other graph-based methods. The results are shown in rows 4-6 of the table below. Due to incorporating additional reaction information and 3D structures, our method requires slightly more time than molecular graphs and Rxn Hypergraph representations. However, it delivers significant performance improvement, justifying the additional time cost.
>
>
>         **Table 2:** Model inference time for different graph representations under varying numbers of graph nodes. Time units are milliseconds (ms).
>         | **Representation** | **Reaction Information** | **3D Structure** | 2500  | 3000  | 3500  | 4000  | 4500  |
>         |-|-|-|-|-|-|-|-|
>         | Reaction Graph                           | √                        | √                | 37.28 | 43.77 | 50.57 | 58.03 | 65.45 |
>         | w/o Reaction Edge                        | ×                        | √                | 35.36 | 40.92 | 48.68 | 53.30 | 58.79 |
>         | w/o Angular Edge                         | √                        | ×                | 22.51 | 25.96 | 29.82 | 34.21 | 38.77 |
>         | Molecular Graph                          | ×                        | ×                | 18.59 | 21.93 | 25.39 | 28.30 | 31.48 |
>         | Rxn Hypergraph                           | ×                        | ×                | 31.18 | 37.28 | 43.31 | 48.53 | 54.06 |
>         | Condensed Graph of Reactions             | √                        | ×                | 44.55 | 52.48 | 60.75 | 66.32 | 75.34 |
>
>     + Compared with SMILES-based methods
>
>         We compare the inference time of Reaction Graph and SMILES-based methods. The results are shown in the table below, where Reaction Graph, as a graph-based model, exhibits the shortest inference time.
>
>
>         **Table 3:** Inference time comparison for Reaction Graph and other methods.
>         | **Methods**             | **Representation**    | **Time**      |
>         |-|-|-|
>         | T5Chem                  | SMILES                | 3min 19s     |
>         | RXNFP                   | SMILES                | 23min 43s    |
>         | Reaction Graph (ours)   | Reaction Graph        | **2min 36s**     |
>
>
> The details of the cost analysis can be found in Sec.E.

---

> ### Author Response · Authors · 2024-11-23
> **Part Ⅳ: Exploring Revised Rxn Hypergraph and Set Transformer**
>
> ## Is it possible to model the interactions between reactants and products by adding another hypernode in the Rxn Hypergraph
>
> We can model the interactions between reactants and products by adding another hypernode in the Rxn Hypergraph. Specifically, we can connect the hypernodes of reactants and products with an additional hypernode.
>
> However, this method still exhibits limitations compared to our Reaction Graph.
>
> 1. **Lack of Atomic Mapping.** To effectively track the transformation of substructures in a reaction, GNNs need to establish a mapping that identifies the correspondence between the substructures in reactants and products. However, the hypernode method falls short in directly representing this mapping, posing challenges in accurately expressing the interactions between reactants and products.
>
> 2. **Limited Capability for Reaction Modeling .** When a node has too many neighbors, it may lead to an information loss during message aggregation in GNNs (referred to as the Over-squashing Issue). This additional hypernode also faces the similar Over-squashing problem. Because the additional Hypernodes serve as a bridge for interactions between reactants and products, it usually has massive numbers of neighbors. Since it is difficult for a single node to carry so much information, the additional hypernode method may have a limited capability for reaction modeling, especially when many large reactants and products involved.
>
> We conduct experiments on various tasks using the hypernode method. The results are presented as below:
>
> + Reaction Condition Prediction:
>
>     **Table 4:** Influence of the additional hypernode on the performance of Rxn Hypergraph in condition prediction.
>     | Method              | USPTO-Condition   | Pistachio-Condition   |
>     |-|-|-|
>     | Rxn Hypergraph w/o additional Hypernode       | 0.213 | 0.288 |
>     | Rxn Hypergraph w/  additional Hypernode   | 0.211 | 0.289 |
>     | Reaction Graph (ours) | **0.324** | **0.392** |
> + Reaction Yield Prediction
>
>     **Table 5:** Influence of the additional hypernode on the performance of Rxn Hypergraph in yield prediction.
>     | Method              | BH   | BH1  | BH2  | BH3  | BH4  | SM   | Gram | Subgram |
>     |-|-|-|-|-|-|-|-|-|
>     | Rxn Hypergraph w/o additional Hypernode       | 0.96 | 0.81 | 0.83 | 0.71 | 0.56 | 0.85 | 0.118| 0.196   |
>     | Rxn Hypergraph w/  additional Hypernode      | 0.96 | **0.82** | 0.81 | 0.75 | 0.57 | 0.86 | 0.112| 0.187   |
>     | Reaction Graph (ours) | **0.97** | 0.80 | **0.88** | **0.76** | **0.68** | **0.89** | **0.129** | **0.216** |
> + Reaction Classification
>
>     **Table 6:** Influence of the additional hypernode on the performance of Rxn Hypergraph in reaction classification.
>     | Method              | USPTO-TPL   | Pistachio-Type   |
>     |-|-|-|
>     | Rxn Hypergraph w/o additional Hypernode      | 0.954 | 0.911 |
>     | Rxn Hypergraph w/  additional Hypernode      | 0.984 | 0.936 |
>     | Reaction Graph (ours) | **0.999** | **0.987** |
>
> The hypernode method shows improvement in reaction classification, while it has a slight impact on condition and yield prediction. This may be because reaction classification is relatively intuitive, while condition and yield prediction are complex. To improve the performance of condition and yield prediction, more accurate interaction modeling (e.g., Reaction Graph) is needed. Reaction Graph surpasses the performance of the hypernode method, demonstrating its efficiency in interaction modeling. Details can be found in Sec.C.1.2 and G.5.2.
>
> ##  Set2Set vs. SetTransformer
>
> We use Set2Set to capture the global representation of a Reaction Graph by aggregating   node features.
> We compare the ability of  Set2Set and Set Transformer for capturing the global representation of a Reaction Graph.
>
> **Table 7:** Influence of Set Transformer on the model's performance in USPTO-Condition dataset.
> | **Method**          | **Top-1** ↑ | **Top-3** ↑ | **Top-5** ↑ | **Top-10** ↑ | **Top-15** ↑ | **Inference Time** ↓ |
> |-|-|-|-|-|-|-|
> | **Set Transformer** | 0.2940       | 0.4079       | 0.4471       | 0.4847        | 0.4968        | 6 min 1 s               |
> | **Set2Set**        | **0.3246**       | **0.4343**       | **0.4715**       | **0.5061**        | **0.5181**        | **2 min 36 s**              |
>
> The results indicate that, when used as an aggregation module, Set2Set surpasses the Set Transformer by 5% to 10% in performance and is also more computationally efficient. Detailed implementation methods and analysis can be found in the Sec.C.2.2 and Sec.G.4.3.

---

> > ### Comment · Reviewer_xQ99 · 2024-11-26
> > **Thank you for the responses**
> >
> > I thank the authors for their responses and the additional experiments. Regarding the responses, I think the complexity analysis should be calculated more rigorously (using the "big $\mathcal{O}$ notation"). Additionally, providing an analysis of why Set2Set performs better would be helpful. Overall, I still remain positive about this work and will maintain my score.

---

> > > ### Author Response · Authors · 2024-11-27
> > > **Part Ⅴ-1: Complexity Analysis with Big O**
> > >
> > > Thank you for your prompt response and constructive feedback. We sincerely appreciate your suggestion, and we have included a complexity analysis with Big O notation in Sec. E.1. Below is a summary.
> > >
> > > ## Using Big O in The Complexity Analysis
> > >
> > > The big O is the upper bound of complexity. If there exist positive constants $ C $ and $ n\_0 $ such that for all $ n \\geq n\_0 $, the following holds:
> > >   $
> > >   T(n) \\leq C \\cdot f(n),
> > >   $
> > >   then we say that $ T(n) $ is $ O(f(n)) $,
> > >   where $ T: \\mathbb{R} \\to \\mathbb{R} $ represents the computational complexity of an algorithm for an input size $ n $, and $ f: \\mathbb{R} \\to \\mathbb{R} $ is a known function that describes the upper bound growth rate.
> > >
> > > An important property of Big O is that it is independent of lower-order terms and constants. For example, $O(2n) = O(n)$ and $O(n^2 + n) = O(n^2)$. However, as shown in Figure 7 and Figure 8 in Section E.2, the computational complexity of graph-based methods differs by constants. The absence of constant in Big O makes it difficult to reflect the distinctions between different methods, which is not conducive to analysis.
> > >
> > > To reflect the differences among various methods, we abstract the distinctions into explicit variables during analysis, such as the hidden feature dimension of atoms. Through these explicit variables, we can observe the differences between these methods intuitively.

---

> > > ### Author Response · Authors · 2024-11-27
> > > **Part Ⅴ-2: Complexity Analysis with Big O**
> > >
> > > ## Computational Complexity Analysis with Big O
> > >
> > > Since Big O analysis enhances the comprehensiveness of our analytical results, we supplement it in Sec.E.1.
> > >
> > > We first define:
> > > - $V$: Number of atoms
> > > - $M$: Number of molecules
> > > - $E$: Number of chemical bonds
> > > - $D$: Maximum degree of atomic nodes
> > > - $L$: Length of the molecular SMILES string
> > > - $H$: Hidden feature dimension of atoms
> > > - $A$: Number of atoms in a graph node
> > > - $N$: Number of iterations for the GNN/Transformer
> > > - $P$: Number of message passing in a GNN iteration
> > >
> > > We simply assume that the messages are calculated by passing the hidden features of atoms through a fully connected layer.
> > > - **Molecular Grpah.** Since each edge needs to calculate the message once in each GNN interaction, and the complexity of fully connected layer is $O(H^2)$, each node in molecular graph represents one atom, the complexity of the original Molecular Graph is $O(ENH^2)$.
> > > - **Reaction Graph (ours).**  Reaction Graph introduces two types of edges: reaction edges and angular edges.
> > >     - Reaction Edge: The number of reaction edges $E\_{reaction}$ cannot exceed the number of product atoms $V\_{product}$, and the number of product atoms cannot exceed half of the total atoms, so $E\_{reaction} = V\_{product}\\leq V/2$. Therefore, the computational cost introduced by the reaction edges is $O(V/2\\cdot NH^2) = O(VNH^2)$.
> > >     - Angular Edge: Since a bond angle is formed between every two neighboring edges of each atom node, and the number of bond angles is equal to the number of angle edges. Therefore, the number of angle edges $E\_{angular} = VD^2$. Therefore, the computational cost introduced by the angular edges is $O(VD^2 \\cdot NH^2)$.
> > >
> > >     The overall complexity of Reaction Graph is $O[(E+V+VD^2)\\cdot NH^2] = O[(E+VD^2)\\cdot NH^2]$. When 3D information is not used, the complexity of the Reaction Graph is $O[(E+V)\\cdot NH^2]$.
> > > - **Bond Angle Model.** Our proposed angular edges are more efficient than explicitly using bond angles. With the angular edge method, there is only one message passing per round of GNN iteration. In contrast, when angle features are explicitly integrated, each GNN iteration requires $ P $ message passing operations. The complexity is $ O[(E+VD^2)\\cdot PNH^2] $.
> > > - **Rxn Hypergrpah.** Rxn Hypergraph adds edges connecting each atom to the corresponding molecule hypernode, introducing a computational cost of $ O(VNH^2) $. The molecule nodes between reactants are connected pairwise, all linking to the reactant hypernode. And it is similar for the products. The cost for these new edges are $O[(M^2 + M)\\cdot NH^2] = O(M^2\\cdot NH^2)$. Finally, the cost for Rxn Hypergrpah is $O[(E + V + M^2)\\cdot NH^2]$
> > > - **Condense Graph of Reaction.** In Condense Graph of Reaction (CGR), each node represents $A$ atoms from the original graph, and each edge represents $A$ edges from the original graph. Therefore, the number of edges in the CGR becomes $ E/A $, and the dimension of the hidden layer for the nodes becomes $ AH $. Thus, the time complexity of the CGR is $ O[E/A\\cdot N\\cdot (AH)^2]$ = $ O(E\\cdot NAH^2)$.
> > > - **SMILES-based Methods**. SMILES-based methods typically use Transformers as the backbone. In Transformers, the time complexity for each component is:
> > >     - Tokenization: The cost of regex tokenizer is $O(L)$
> > >     - Embedding: The cost of embedding layer is $O(LH)$
> > >     - Attention: The size of the attention matrix is $L^2$, so the complexity for attention matrix calculation is $O(L^2H)$
> > >     - Feed Forward: Each token's embedding is processed through a fully connected layer, so the time complexity is $O(LH^2)$
> > >
> > >     And there's $N$ layers of Attention and Feed Forward. The overall time complexity of Transformer is $O[L + LH + N\\cdot (LH^2 + L^2H)] = O[N\\cdot (LH^2 + L^2H)]$

---

> > > ### Author Response · Authors · 2024-11-27
> > > **Part Ⅵ-1: The Analysis for Set2Set and Set Transformer (Set Transformer)**
> > >
> > > We analyze the reasons that the Set Transformer does not perform well in our task:
> > > + The Set Transformer performs set message passing operation, which is lack the explicit constraints of the edges in the graph. It causes the loss of structural information in molecules.
> > > + Seed vector is used for aggregating the graph features through attention. The seed vector of Set Transformer is fixed for all graphs, while the seed vector of Set2Set is derived from and adapted to a specific graph.
> > >
> > > To prove our point, we provide detailed formulations and experiments.
> > >
> > > ## Formulations and Analysis
> > > For simplicity, we will use single-head attention in the following formulations, and use single seed vector for aggregation. Some feed-forward operations that are unrelated to our analysis are omitted.
> > >
> > > ### Set Transformer
> > > Set Transformer is divided into two parts: the Encoder and the Decoder. The Encoder performs set message passing, while the Decoder performs aggregation.
> > > + **Encoder.**
> > >     + (1) Formulation:
> > >
> > >         Assume the hidden feature of node $i$ is $\\boldsymbol{n}\_i$. For each node $i$, the Encoder computes its query vector $\\boldsymbol{q}\_i = \\boldsymbol{M}\_q \\cdot \\boldsymbol{n}\_i$, key vector $\\boldsymbol{k}\_i = \\boldsymbol{M}\_k \\cdot \\boldsymbol{n}\_i$ and value vector $\\boldsymbol{v}\_i = \\boldsymbol{M}\_v \\cdot \\boldsymbol{n}\_i$, where $\\boldsymbol{M}\_q, \\boldsymbol{M}\_k, \\boldsymbol{M}\_v$ are linear projection matrics. Then it performs attention operation to update hidden features:
> > >
> > >         $$
> > >         \\boldsymbol{n}\_i' = \\boldsymbol{q}\_i + \\sum\_{j=1}^N w(\\boldsymbol{q}\_i \\cdot \\boldsymbol{k}\_j)\\boldsymbol{v}\_j,
> > >         $$
> > >
> > >         where $w$ is an activation function (e.g. Softmax). This process is equivalent to message passing in a fully connected graph:
> > >
> > >         $$
> > >         \\boldsymbol{n}\_i' = f(\\boldsymbol{n}\_i) + \\sum\_{j \\in \\mathbb{N\_i}} g(\\boldsymbol{n}\_i,\\boldsymbol{n}\_j,\\boldsymbol{e}\_{ij}),
> > >         $$
> > >
> > >         where $f$ is the self update function, $g$ is the message function, and $\\mathbb{N}\_i$ is the set of neighbor nodes of $i$.
> > >     + (2) Analysis:
> > >
> > >         In this encoding process, the edge information $\\boldsymbol{e}\_{ij}$ of Reaction Graph is missing. Additionally, $\\mathbb{N}\_i$ is the set of all nodes, resulting in the loss of neighbor topology information. This type of message passing has a negative impact on reaction feature extraction.
> > > + **Decoder.**
> > >     + (1) Formulation:
> > >
> > >         Assume the hidden feature of node $i$ after Encoder is $\\boldsymbol{n}'\_i$, and $\\boldsymbol{s}$ is the learnable seed vector for aggregation. The decoder first calculate the key vector $\\boldsymbol{k}'\_i = \\boldsymbol{M}'\_k \\cdot \\boldsymbol{n}'\_i$ and value vector $\\boldsymbol{k}'\_i = \\boldsymbol{M}'\_k \\cdot \\boldsymbol{n}'\_i$ for all nodes, and then perform attention aggregation on seed vector:
> > >
> > >         $$\\boldsymbol{s}' = \\boldsymbol{s} + \\sum\_{j=1}^N (\\boldsymbol{s} \\cdot \\boldsymbol{k}'\_j)\\boldsymbol{v}'\_j.$$
> > >
> > >         Finally, the decoder outputs the reaction representation vector $\\boldsymbol{r}$ by passing $ \\boldsymbol{s}' $ through a sequence of feed-forward and self attention layer, denoted as:
> > >         $$
> > >         \\boldsymbol{r} = \\text{OutputLayers}(\\boldsymbol{s}'),
> > >         $$
> > >         and it is important to notice that the node feature $\\boldsymbol{n}\_i$ do not involve anymore in these output layers.
> > >
> > >     + (2) Analysis
> > >
> > >         The decoder uses a fixed seed vector $\\boldsymbol{s}$ for all graphs. Thus, its aggregation process lacks focus on the uniqueness of reaction.

---

> ### Author Response · Authors · 2024-11-27
> **Part Ⅵ-2: The Analysis for Set2Set and Set Transformer (Set2Set and Experiment)**
>
> ## Formulations and Analysis
> ### Set2Set
>
> + **Formulation.**
>
>     As detailed in Sec.2 and Sec.C.2.1, the process of Set2Set can be formulize as:
>
>     $$
>     \\boldsymbol{q}^{t+1} = \\sum\_{i=1}^{N} w(\\boldsymbol{n}\_i \\cdot \\boldsymbol{h}^{t}) \\times \\boldsymbol{n}\_i,~~~~~~\\boldsymbol{h}^{t+1}, \\boldsymbol{c}^{t+1} = \\text{LSTM}([\\boldsymbol{q}^{t+1}; \\boldsymbol{h}^t];~\\boldsymbol{h}^t, \\boldsymbol{c}^t),
>     $$
>
>     $$
>     \\boldsymbol{r} = \\boldsymbol{W}\\cdot[\\boldsymbol{q}^{T}; \\boldsymbol{h}^{T-1}] + \\boldsymbol{b},
>     $$
>
>     where $w$ is an activation function, $\\boldsymbol{h}$ and $\\boldsymbol{c}$ are the hidden state and cell state of LSTM. $\\boldsymbol{W}$ and $\\boldsymbol{b}$ are learnable parameters of a fully connected layer. After $T$ iterations, we concatenate LSTM's hidden state $\\boldsymbol{h}^{T-1}$ with the aggregated node features $\\boldsymbol{q}^{T}$, and acquire the final reaction representation $\\boldsymbol{r}$ by linear projection.
>
> + **Analysis.**
>
>     1. *Without Unnecessary Message Passing.* Set2Set does not introduce unnecessary set message passing operations, preserving the topological features of the graph.
>
>     2. *Adaptive Aggregation.* The role of $ \\boldsymbol{h}\_t $ is similar to the seed vector $ s $ in Set Transformer. However, $ \\boldsymbol{h}\_t $ is specifically extracted for the current graph, whereas $ \\boldsymbol{s} $ remains fixed across all graphs. This adaptive seed vector $\\boldsymbol{h}\_t$ enhances the adaptability of the aggregation process to the graph.
>
> ## Experiment
>
> We design the following experiment to demonstrate our point:
>
> 1. We gradually reduce the number of layers in the Set Transformer Encoder to observe whether the set message passing has a negative effect on graph information extraction.
>
> 2. We compare the performance of the Set Transformer Decoder (without adaptive seed vector) and the Set2Set(with adaptive seed cector) to observe whether the adaptive seed vector contributes to the model's performance.
>
> The result is shown in Table 1. Details can be found in Set.G.4.3.
>
> **Table 1:** The influence of set message passing and adaptive seed vector on model's performance. Using a subset of USPTO-Condition for training efficiency.
>
> |**Method**|||**Set Message Passing**|**Adaptive Seed Vector**|**ACC**↑|
> |-|-|-|-|-|-|
> |Set Transformer|||2|×|0.1597|
> |Set Transformer|||1|×|0.1634|
> |Set Transformer|||0|×|0.1700|
> |Set2Set (ours)|||0|√|**0.1773**|
>
> According to the results, with the set message passing reduces, the performance gradually improves. The results indicate that set message passing exhibits a negative effect when extracting features from the Reaction Graph. Meanwhile, adaptive seed vector achieves better performance than directly using a fixed seed vector, validating its effectiveness.
>
>
> Thank you once again for your valuable contributions to improving this paper. If you feel that our efforts have addressed your concerns, we would greatly appreciate it if you could consider raising your rating. Your support is crucial to our work. Should you require any further information, please feel free to contact us. We look forward to your response.

---

> > ### Comment · Reviewer_xQ99 · 2024-11-28
> > **Thank you for your further responses**
> >
> > I thank the authors for their further responses and clarification. The further explanation about Set2Set sounds reasonable to me. I have updated my score to 8. Good luck to the authors.

---

> > > ### Author Response · Authors · 2024-11-28
> > >
> > > Dear Reviewer xQ99,
> > >
> > > We are happy to have addressed your concerns. Thank you for raising the score, and we greatly appreciate your constructive contribution to this work.
> > >
> > > Best regards,
> > >
> > > Paper 4277 Authors

---

### Official Review · Reviewer_uB2j · 2024-10-30

**Soundness:** 2
**Presentation:** 3
**Contribution:** 2
**Rating:** 6
**Confidence:** 4

**Summary:**

The paper presents Reaction Graph (RG), a novel approach for representing chemical reactions using 3D molecular structures. It introduces a unified graph representation integrating reactants and products, incorporating 3D structural information. The work demonstrates applications in areas like leaving group identification.

**Strengths:**

- Innovative unified graph representation combining reactants and products
- Integration of 3D structural information into reaction modeling
- Promising results in leaving group identification analysis
- Clear understanding of current limitations and proposed solutions

**Weaknesses:**

1. Reproducibility Issues:
Code not available during review process. The statement "code will be available publicly" is insufficient for scientific reproducibility. The review process itself should include access to the code and clear instructions for reproducing the results. This is particularly important in computational chemistry where reproducibility should be more straightforward than in experimental settings.

2. Statistical Analysis Deficiencies:
The paper lacks crucial statistical analysis across all results. No uncertainty measurements (standard deviations, confidence intervals) are provided for the accuracy metrics. Without these, it's impossible to determine if the reported improvements are statistically significant or merely random variations. The authors should implement:
     - Multiple trials with different random seeds
     - Standard deviations for all metrics
     - Appropriate statistical significance tests
     - Cross-validation results
     - Confidence intervals for performance metrics


3. Limited Validation:
The authors' claim on page 6, line 295 that "The experiment results validate our hypothesis" is based on just two or three examples. This is methodologically unsound. Such limited examples cannot provide sufficient statistical power or account for the full diversity of chemical reactions. The paper needs:
     - Comprehensive testing across many reaction types
     - Analysis of edge cases and failure modes
     - Statistical validation of the hypothesis
     - Discussion of potential selection bias in examples

4. Technical Clarity Issues:
- The methodology surrounding the Reaction edge introduction requires significant clarification on several fronts. First, there is a citation issue regarding RXNMapper - the authors cite a paper that merely uses RXNMapper rather than the original methodology paper (available at https://chemrxiv.org/engage/chemrxiv/article-details/60c74b2aee301c3c2cc79dac ). This should be corrected to properly credit the original work.
- More substantially, the use of force field calculations for atomic coordinates raises several crucial questions that need to be addressed. The authors mention using MMFF94 to calculate atom coordinates, but provide no details about the conformational analysis. It's essential to know whether they used the lowest energy conformation, how they handled multiple possible conformations, and what impact different conformational choices might have on their results. Given that molecular conformation can significantly affect interatomic distances and geometric parameters, this could have substantial implications for their method's reliability and reproducibility.
- The description of the edge length attribute needs elaboration. While it appears to be based on pairwise distances between selected vertices, the exact methodology isn't clear. If these are indeed simple pairwise distances, how do they account for conformational flexibility? If there's something more sophisticated at play, it needs to be explicitly described.
- A fundamental question arises about whether the reaction graph generation uses products or reactants as its basis. This has important implications for the method's practical applicability. If it uses only reactants, the authors need to explain their methodology for 3D structure generation, particularly regarding the ordering of molecules and the calculation of geometric parameters like bond angles and pairwise distances.
- The handling of molecular geometry deserves special attention. The authors should clarify their approach to calculating bond angles and whether they considered torsion angles in their analysis. The role of molecular conformation in these calculations is particularly important - did they investigate how different conformations affect their results? What criteria did they use to select specific conformations? Was any sensitivity analysis performed to assess the impact of conformational changes on their method's performance?
- Additionally, there's an inconsistency in terminology throughout the paper that needs to be addressed. The authors alternate between using "atoms" for nodes, and "bonds" and "edges" for connections. While these terms are often used interchangeably in chemical graph theory, for clarity and precision, they should establish their terminology early in the paper and maintain consistency throughout. This would help readers better understand the relationship between the chemical and mathematical aspects of their approach.
- These technical details are crucial for understanding the method's limitations and potential applications, as well as for ensuring reproducibility of the results. The authors should provide a comprehensive description of these aspects in their revision.

5. Leaving Group Analysis While the leaving group identification analysis shows promising results, it requires:
- Error analysis of misclassified cases
- Comparison with additional baseline methods
- Cross-validation results
- Discussion of systematic errors or biases
- Analysis of chemical pattern dependencies

6. Runtime Analysis The runtime discussion in section 4.2 lacks essential details
- System specifications
- Benchmark conditions
- Variability in performance
- Hardware dependencies

**Questions:**

- How will you ensure reproducibility by making code available and providing clear instructions?
- Can you provide comprehensive statistical analysis including standard deviations and confidence intervals?
- What was your methodology for handling multiple possible conformations in the MMFF94 calculations?
- Could you clarify whether reaction graph generation uses products or reactants as its basis?
- How do different conformations affect your results and what criteria were used for conformation selection?
- Can you expand the leaving group analysis with error analysis and comparison to additional baselines?
- Will you provide detailed runtime analysis including system specifications and benchmark conditions?

---

> ### Author Response · Authors · 2024-11-25
> **Part Ⅰ: Ensuring Reproducibility: Code, Guidelines, and Parameters**
>
> Thank you for acknowledging that we propose an "**innovative unified graph representation combining reactants and products**" and that our method demonstrates "**promising results**". We also appreciate your suggestions.
> **Part Ⅰ- Part Ⅵ** are responses to weaknesses, and **ReplyⅠ- Repy Ⅶ** are responses to questions.
>
> ## Reproducibility
>
> We provide reproducibility support in the following ways:
> + **Code**: We have submitted the code in initial supplementary materials. It includes dataset processing, Reaction Graph construction, network architecture and training framework.
> It also contains implementations of the compared baselines.
>
> + **Guidelines**: We provide detailed guidelines in Sec.C and Sec.F. The guidelines include the operating environment, data preprocessing, hyperparameter settings, training methods, and other related aspects.
>
>
> + **Parameters**: In this revision, we provide the training parameter checkpoints in the link below.
> https://huggingface.co/reactiongraph/ReactionGraph

---

> ### Author Response · Authors · 2024-11-25
> **Part Ⅱ-1: Statistical Analysis**
>
> ## 1. Multiple Trials with Different Random Seeds
> We train and test the model on all tasks using different random seeds. The results are summarized in the Standard Deviations section. More details on the standard deviations of baseline methods have been added in Sec. G.3.2 and Sec. G.6.1.
>
>
> ## 2. Standard Deviations
> We summarize the standard deviations of all metrics. More detailed information, including the baselines and implementation details, is provided in the Sec. G.3.2 and Sec. G.6.1.
>
> **Table 1:** Results for leaving group identification on USPTO. The metrics include mean and standard deviations from multiple trials with different random seeds.
> | Type               | Overall ACC         | Overall CEN         | Overall MCC         | LvG ACC             | LvG CEN             | LvG MCC             |
> |-|-|-|-|-|-|-|
> | Molecular Graph| 0.950±0.001        | 0.037±0.001        | 0.538±0.006        | 0.423±0.004        | 0.209±0.002        | 0.497±0.005        |
> | Reaction Graph | 0.996±0.001        | 0.002±0.001        | 0.971±0.001        | 0.944±0.001        | 0.035±0.001        | 0.942±0.001        |
>
> **Table 2:** Results for reaction condition prediction on USPTO-Condition. The metrics includes mean and standard deviations from multiple trials with different random seeds.
> | Metrics | Top-1       | Top-3       | Top-5       | Top-10      | Top-15      |
> |-|-|-|-|-|-|
> | Mean±Std of Top-K Accuracy                         | 0.322±0.002 | 0.432±0.003 | 0.469±0.003 | 0.504±0.003 | 0.516±0.003 |
>
> **Table 3:** Results for reaction condition prediction on Pistachio-Condition. The metrics includes mean and standard deviations from multiple trials with different random seeds.
> | Metrics | Top-1       | Top-3       | Top-5       | Top-10      | Top-15      |
> |-|-|-|-|-|-|
> | Mean±Std of Top-K Accuracy                    | 0.391±0.001 | 0.556±0.001 | 0.602±0.001 | 0.636±0.001 | 0.641±0.001 |
>
> **Table 4:**  Results for reaction yield prediction on Buchwald-Hartwig dataset. The metrics includes mean and standard deviations from multiple trials with different random seeds.
> |  Metrics  |  BH       | BH1      | BH2      | BH3      | BH4      |
> |-|-|-|-|-|-|
> | Mean±Std of $R^2$   | 0.97±0.01| 0.80±0.01| 0.88±0.02| 0.76±0.02| 0.68±0.06|
>
> **Table 5:** Results for reaction yield prediction on Suzuki-Miyaura and USPTO-Yield dataset. The metrics includes mean and standard deviations from multiple trials with different random seeds.
> |  Dataset  |  SM       | Gram    |Subgram    |
> |-|-|-|-|
> | Mean±Std of $R^2$   |  0.89±0.01| 0.125±0.006 |0.211±0.003 |
>
> **Table 6:** Results for reaction classification on USPTO-TPL dataset. The metrics includes mean and standard deviations from multiple trials with different random seeds.
> | Metrics             | ACC   | CEN   | MCC   |
> |-|-|-|-|
> | Mean±Std        | 0.999±0.001 | 0.001±0.001 | 0.999±0.001 |
>
> **Table 7:** Results for reaction classification on Pistachio-Type dataset. The metrics includes mean and standard deviations from multiple trials with different random seeds.
> | Metrics             | ACC   | CEN   | MCC   |
> |-|-|-|-|
> | Mean±Std        | 0.986±0.002 | 0.026±0.004 | 0.985±0.003 |
>
> Based on the results, we observe that the standard deviations of the training results is smaller ($≤0.01$) on larger datasets (USPTO, Pistachio). In contrast, we observe a larger training standard deviations ($>0.01$) on some smaller test set.

---

> ### Author Response · Authors · 2024-11-25
> **Part Ⅱ-2: Statistical Analysis**
>
> ## 3. Statistical Significance Tests
>
> Statistical significance testing helps us determine whether Reaction Graph have improvement compared to the baseline method. We select the optimal baseline for each task for comparison. Specifically,  $t$-test method is adopted to calculate the statistical significance. $t$ is an indicator of the difference between two experimental groups. The formula is $t = \sqrt{n} \cdot \frac{x - b}{s}$ , where $b$ is the baseline performance and  $n = 4$ .
>
> The significance level $P$ corresponding to the $t$-value can be found in the lookup table (https://en.wikipedia.org/wiki/Student%27s_t-distribution). We use $-\log P$ to make the results easier to observe. The larger the value, the higher the probability that Reaction Graph is better. The final calculated significance levels are as follows. Details can be found in Sec.G.6.2.
>
> **Table 8:** Results for statistical significance analysis on leaving group identification in USPTO. The significance level is calculated by $t$-test method. The data is from the mean and standard deviations of multiple trials with different random seeds.
> | Metrics | Overall ACC   | Overall MCC   | LvG ACC   | LvG MCC   |
> |-|-|-|-|-|
> | -logP   | 13.47 | 20.19 | 20.75 | 20.28 |
>
>
> **Table 9:** Results for statistical significance analysis on reaction condition prediction in USPTO-Condition. The significance level is calculated by $t$-test method. The data is from the mean and standard deviations of multiple trials with different random seeds.
> | Metrics | Top-1  | Top-3  | Top-5  | Top-10 | Top-15 |
> |-|-|-|-|-|-|
> | -logP   | 11.81 | 8.69  | 7.38  | 6.43  | 6.43  |
>
> **Table 10:** Results for statistical significance analysis on reaction condition prediction in Pistachio-Condition. The significance level is calculated by $t$-test method. The data is from the mean and standard deviations of multiple trials with different random seeds.
> | Metrics | Top-1  | Top-3  | Top-5  | Top-10 | Top-15 |
> |-|-|-|-|-|-|
> |  -logP  | 13.12 | 11.52 | 9.90  | 8.90  | 9.18  |
>
> **Table 11:** Results for statistical significance analysis on reaction yield prediction in Buchwald-Hartwig. The significance level is calculated by $t$-test method. The data is from the mean and standard deviations of multiple trials with different random seeds.
> | $R^2$ Metrics | BH    | BH1   | BH2   | BH3   | BH4   |
> |-|-|-|-|-|-|
> | -logP   | 0.69  | 8.74  | 0.69  | 5.52  | 6.69  |
>
> **Table 12:** Results for statistical significance analysis on reaction yield prediction in Suzuki-Miyaura and USPTO-Yield. The significance level is calculated by $t$-test method. The data is from the mean and standard deviations of multiple trials with different random seeds.
> | $R^2$ Metrics| SM    | Gram  | Subgram |
> |-|-|-|-|
> | -logP   | 0.69  | 3.27  | 7.84    |
>
> **Table 13:** Results for statistical significance analysis on reaction classification in USPTO-TPL and Pistachio Type. The significance level is calculated by $t$-test method. The data is from the mean and standard deviations of multiple trials with different random seeds.
> | Metrics | U-T ACC   | U-T MCC   | P-T ACC   | P-T MCC   |
> |-|-|-|-|-|
> | -logP   | 4.27  | 4.27  | 4.27  | 3.80  |
>
> Based on the results of the $t$-test statistical significance analysis, the advantages achieved by our model have a high level of confidence ($-logP > 3$ indicates $confidence > 95\%$), across most of the datasets and metrics.

---

> ### Author Response · Authors · 2024-11-25
> **Part Ⅱ-3: Statistical Analysis**
>
> ## 4. Cross-validation Results
> Cross-validation can reduce the bias introduced by the unevenly distributed reaction dataset. This ensures the model's generalization. We evenly split the dataset used in each task. And we test the model's performance using K-fold cross-validation. The results are as follows. Details of can be found in Sec.G.6.3.
>
> **Table 14:** Cross-validation results for leaving group identification on USPTO. The metrics includes mean and standard deviations from multiple trials with different train/test splits.
> | Type               | Overall ACC         | Overall CEN         | Overall MCC         | LvG ACC             | LvG CEN             | LvG MCC             |
> |-|-|-|-|-|-|-|
> | Molecular Graph     | 0.948±0.004        | 0.038±0.003        | 0.535±0.015        | 0.418±0.010         | 0.203±0.001         | 0.495±0.008         |
> | Reaction Graph      | 0.995±0.002        | 0.004±0.001        | 0.955±0.012        | 0.915±0.022         | 0.050±0.011         | 0.912±0.022         |
>
> **Table 15:** Cross-validation results for reaction condition prediction on USPTO-Condition dataset. The metrics includes mean and standard deviations from multiple trials with different train/test splits.
> | Metrics | Top-1       | Top-3       | Top-5       | Top-10      | Top-15      |
> |-|-|-|-|-|-|
> | Mean±Std of Top-K Accuracy   | 0.320±0.003 | 0.430±0.002 | 0.467±0.002 | 0.503±0.002 | 0.516±0.002 |
>
> **Table 16:** Cross-validation results for reaction condition prediction on Pistachio-Condition dataset. The metrics includes mean and standard deviations from multiple trials with different train/test splits.
> | Metrics | Top-1       | Top-3       | Top-5       | Top-10      | Top-15      |
> |-|-|-|-|-|-|
> | Mean±Std of Top-K Accuracy | 0.391±0.002 | 0.556±0.002 | 0.603±0.001 | 0.636±0.001 | 0.641±0.001 |
>
> **Table 17:** Cross-validation results for reaction yield prediction on Buchwald-Hartwig and Suzuki-Miyaura dataset. The metrics includes mean and standard deviations from multiple trials with different train/test splits.
> | Metrics  | BH       | SM       |
> |-|-|-|
> | Mean±Std of $R^2$ | 0.97±0.01 | 0.89±0.01 |
>
> **Table 18:** Cross-validation results for reaction classification on USPTO-TPL dataset. The metrics includes mean and standard deviations from multiple trials with different train/test splits.
> | Metrics             | ACC   | CEN   | MCC   |
> |-|-|-|-|
> | Mean±Std           | 0.999±0.001 | 0.001±0.001 | 0.999±0.001 |
>
> **Table 19:** Cross-validation results for reaction classification on Pistachio-Type dataset. The metrics includes mean and standard deviations from multiple trials with different train/test splits.
> | Metrics             | ACC   | CEN   | MCC   |
> |-|-|-|-|
> | Mean±Std      | 0.987±0.002 | 0.025±0.003 | 0.986±0.002 |
>
> Based on the results of cross-validation, for large reaction datasets (USPTO, Pistachio), the data distribution differences between different splits are minimal due to the sufficient amount of data, leading to small standard deviations ($≤0.01$) in the training results. For the high-quality Buchwald-Hartwig and Suzuki-Miyaura datasets, the standard deviations introduced by different training/testing splits is also small ($≤0.01$).

---

> ### Author Response · Authors · 2024-11-25
> **Part Ⅱ-4: Statistical Analysis**
>
> ## 5. Confidence Intervals
> We use the normal distribution method to calculate the model's confidence interval. Specifically, $CI = x±z · s / \sqrt n$. We take a 95\% confidence level, where the corresponding $z = 1.96$. The result is as shown below, with more details in Sec.G.6.4.
>
> **Table 19:** Confidence interval results ([Min,Max]) for leaving group identification on USPTO. The interval is calculated using normal distribution method. The data is from mean and standard deviations of multiple trials with different random seeds.
> | Type  |    Metrics         | Overall ACC         | Overall CEN         | Overall MCC         | LvG ACC             | LvG CEN             | LvG MCC             |
> |-|-|-|-|-|-|-|-|
> |Molecular Graph| Min         | 0.949 | 0.036 | 0.532 | 0.419 | 0.207 | 0.492 |
> || Max         | 0.951 | 0.038 | 0.544 | 0.427 | 0.211 | 0.502 |
> |Reaction Graph| Min         | 0.995 | 0.001 | 0.970 | 0.943 | 0.034 | 0.941 |
> || Max         | 0.997 | 0.003 | 0.972 | 0.945 | 0.036 | 0.943 |
>
> **Table 20:** Confidence interval results ([Min,Max]) for reaction condition prediction on USPTO-Condition dataset. The interval is calculated using normal distribution method. The data is from mean and standard deviations of multiple trials with different random seeds.
> | Metrics | Top-1       | Top-3       | Top-5       | Top-10      | Top-15      |
> |-|-|-|-|-|-|
> | Min         | 0.320 | 0.429 | 0.466 | 0.501 | 0.513 |
> | Max         | 0.324 | 0.435 | 0.472 | 0.507 | 0.519 |
>
> **Table 21:** Confidence interval results ([Min,Max]) for reaction condition prediction on Pistachio-Condition dataset. The interval is calculated using normal distribution method. The data is from mean and standard deviations of multiple trials with different random seeds.
> | Metrics | Top-1       | Top-3       | Top-5       | Top-10      | Top-15      |
> |-|-|-|-|-|-|
> | Min         | 0.390 | 0.555 | 0.601 | 0.635 | 0.640 |
> | Max         | 0.392 | 0.557 | 0.603 | 0.637 | 0.642 |
>
> **Table 22:** Confidence interval results ([Min,Max]) for reaction yield prediction on Buchwald-Hartwig dataset. The interval is calculated using normal distribution method. The data is from mean and standard deviations of multiple trials with different random seeds.
> | $R^2$ Metrics | BH    | BH1   | BH2   | BH3   | BH4   |
> |-|-|-|-|-|-|
> | Min         | 0.96  | 0.79  | 0.87  | 0.75  | 0.64  |
> | Max         | 0.98  | 0.81  | 0.89  | 0.77  | 0.72  |
>
> **Table 23:** Confidence interval results ([Min,Max]) for reaction yield prediction on Suzuki-Miyaura and USPTO-Yield dataset. The interval is calculated using normal distribution method. The data is from mean and standard deviations of multiple trials with different random seeds.
> | $R^2$ Metrics| SM    | Gram  | Subgram |
> |-|-|-|-|
> | Min         | 0.88  | 0.119 | 0.208 |
> | Max         | 0.90  | 0.131 | 0.214 |
>
> **Table 24:** Confidence interval results ([Min,Max]) for reaction classification on USPTO-TPL dataset. The interval is calculated using normal distribution method. The data is from mean and standard deviations of multiple trials with different random seeds.
> | Metrics             | ACC   | CEN   | MCC   |
> |-|-|-|-|
> | Min         | 0.998 | 0.000 | 0.998 |
> | Max         | 1.000 | 0.002 | 1.000 |
>
> **Table 25:** Confidence interval results ([Min,Max]) for reaction classification on Pistachio-Type dataset. The interval is calculated using normal distribution method. The data is from mean and standard deviations of multiple trials with different random seeds.
> | Metrics             | ACC   | CEN   | MCC   |
> |-|-|-|-|
> | Min         | 0.984 | 0.022 | 0.982 |
> | Max         | 0.988 | 0.030 | 0.988 |
>
> According to the results, in larger datasets (USPTO, Pistachio), the 95% confidence interval for model performance is small ($≤0.01$). This result indicates that the model can achieve stable performance. However, in smaller datasets (Buchwald-Hartwig and Suzuki-Miyaura), some test sets have wider confidence intervals ($>0.01$). It suggests that the model's training variance cannot be ignored.

---

> ### Author Response · Authors · 2024-11-25
> **Part Ⅲ: Attention Weight Result Validation**
>
> ## 1. Comprehensive Testing Across Many Reaction Types
>
> In Sec.G.1.1, we provide examples of Attention Weights for various reaction types. The types include amidation, cyclization, and nucleophilic substitution reactions. We also include experiments on dataset scale as in Sec.G.1.2, and analyze the reaction type distribution of the selected dataset in Sec.F.1. It validates that the Reaction Graph effectively helps the model identify reaction centers across different reaction types.
>
> In addition, we conduct a detailed analysis of the results from the dataset scale tests. We calculate two metrics based on the type of response. The first is the proportion of attention weight for the reacting atoms, and the second is the ratio of attention weight between reacting and non-reacting atoms. The results are as follows:
>
> **Table 26:** The proportion of attention weight for the reacting atoms, across various reaction types.
> | Method | 0    | 1    | 2    | 3    | 4    | 5    | 6    | 7    | 8    | 9    | 10   | 11   |
> |----------|------|------|------|------|------|------|------|------|------|------|------|------|
> | Molecular Graph | 0.65 | 0.49 | 0.50 | 0.50 | 0.61 | 0.35 | 0.58 | 0.50 | 0.38 | 0.46 | 0.28 | 0.04 |
> | **Reaction Graph**  | **0.70** | **0.53** | **0.54** | **0.54** | **0.66** | **0.38** | **0.65** | **0.57** | **0.40** | **0.52** | **0.32** | **0.06** |
>
> **Table 27:** The ratio of attention weight between reacting and non-reacting atoms, across various reaction types.
> | Method | 0    | 1    | 2    | 3    | 4    | 5    | 6    | 7    | 8    | 9    | 10   | 11   |
> |----------|------|------|------|------|------|------|------|------|------|------|------|------|
> | Molecular Graph   | 4.65 | 1.92 | 2.87 | 3.07 | 2.25 | 0.67 | 4.56 | 2.56 | 0.84 | 2.84 | 0.55 | 0.05 |
> | **Reaction Graph**   | **6.07** | **2.35** | **3.21** | **3.79** | **2.86** | **0.84** | **6.73** | **3.39** | **0.99** | **3.66** | **0.64** | **0.09** |
>
> According to the results, the attention weight shows significant variation across different reaction types. Reaction Graph consistently maintains a clear advantage in all types. For details of these reaction types, please refer to Table 37 in the appendix in Sec.G.3.1.
>
> ## 2. Analysis of Edge Cases and Failure Modes
>
> We add visualization and analysis of boundary cases in Sec.G.2. The Reaction Graph may occasionally misidentify reaction centers, typically due to:
> + errors in the reaction itself
> + overly complex molecular structures
> + rare reactions that do not occur at functional groups
>
> ## 3. Statistical Validation of the Hypothesis
> We conduct additional experiments on dataset scale, and quantitatively validate that the Reaction Graph aids in identifying reaction centers. We use USPTO dataset and Condition Prediction Model as in the main text. Attention weights are extracted for Reacting Atoms and non-Reacting Atoms from the Set2Set layer.
>
> We calculate the proportion of attention weights for Reacting Atoms relative to the total, and the results are as follows:
>
> **Table 28:** The proportion of attention weights for Reacting Atoms to the total attention weights of all atoms. Compare between Molecular Graph and Reaction Graph.
> | Method                   | Proportion |
> |-|-|
> | Molecular Graph   | 0.4326     |
> | Reaction Graph (ours)    | **0.5079**     |
>
>
> We also calculate the average attention weights for Reacting Atoms and non-Reacting Atoms, denoted as $a$ and $b$. And we compute their ratio $a/b$. The results are as follows:
>
> **Table 29:** The ratio $a/b$ of the attention weights $a$ of Reacting Atoms to the attention weights $b$ of non-Reacting Atoms. Compare between Molecular Graph and Reaction Graph.
> | Method                   | Ratio $a/b$ |
> |-|-|
> | Molecular Graph   | 1.4542     |
> | Reaction Graph (ours)    | **2.0390**     |
>
> According to the results, the model using the Reaction Graph shows significantly stronger attention to reaction centers. Reaction Graph achieves a performance improvement of 40\%. Considering the sparsity of leaving groups, this is a substantial enhancement.
>
> ## 4. Discussion of Potential Selection Bias in Examples
>
> We use the following methods to reduce selection bias. These methods enhance the reliability of the conclusions from attention weight visualization.
>
> + **Automated Algorithms.** We use the algorithms described in Sec. D.3 to automatically extract visual samples from the dataset. This automated procedure reduces the selection bias introduced by subjective factors.
>
> + **Dataset Size Testing.** In addition to selecting visual samples, we also conducted attention weight testing on the dataset size. This method can avoid the selection bias introduced by sample selection. The results can be found in Section 3. Statistical Validation of the Hypothesis.

---

> ### Author Response · Authors · 2024-11-25
> **Part Ⅳ: Technical Clarity**
>
> Thank you for your comments. We provide further explanations step-by-step in the hope of addressing your concerns.
>
> + **Citation.** Thank you for pointing out this issue. We correct the citation of RXNMapper in this revision.
>
> + **Multiple Possible Conformations Handling.** Theoretically, for multiple possible conformations converged from MMFF, regardless of which one is chosen, the 3D features of Reaction Graph unchanged. Details will later be provided in Reply Ⅲ: Handling Multiple Conformation by Invariant 3D Feature Modeling.
>
> + **Elaboration of Edge Length.** We do not use pairwise distances. Instead, we construct 3D information based on molecular graphs as in Sec.3.1. We only calculate the distances for nodes $i$ and $j$ that are connected by edges in the graph. This approach can address the issue of flexibility. For details, please refer to Reply Ⅲ: Handling Multiple Conformation by Invariant 3D Feature Modeling.
>
> + **Basis of Reaction Graph** Details will later be provided in Reply Ⅳ: Reaction Graph uses Reaction Itself as Basis.
> + **Molecular Geometry Handling.**
>   + We do not explicitly calculate bond angle. Instead, we propose angular edge to implicitly represent bond angle. Details can be found in Sec.3.2 and Sec.C.1.
>   + Torsion angle is considered in our analysis in Sec.A.2. Detailed discussion can later be found in Reply Ⅲ: Handling Multiple Conformation by Invariant 3D Feature Modeling.
>   + Details of conformations selection can be found in Reply Ⅴ: Criteria for Conformer Selection
>   + The statistical analysis and cross validation provides assessment of the impact of conformational change. Please refer to Sec.G.6 and Part Ⅱ: Statistical Analysis. And Reply Ⅲ also provides methods to deal with conformational change.
> + **Terminology.** A terminology list will be provided to maintain consistency at Sec.J.
> + **Technical Detail.** Sec.C, Sec.D, and Sec.F provides comprehensive details of our method.

---

> ### Author Response · Authors · 2024-11-25
> **Part Ⅴ: Leaving Group Identification Analysis**
>
> ## 1. Error Analysis of Misclassified Cases
> Additional error analysis is provided in Sec.G.2.2. In summary, molecular graph often misidentify leaving group positions, thereby hindering accurate classification. In contrast, Reaction Graph position leaving groups correctly at most time. But it still have minor errors in specific type classification. This shows that Reaction Graph effectively helps the model to focus on reaction related features.
>
> ## 2. Comparison with Additional Baseline Methods
> We conduct extra experiments with D-MPNN and Rxn Hypergraph. The results are shown in the table below (Table 30). According to the results, Reaction Graph achieves the best performance in leaving group identification compared to other baselines.
>
> **Table 30:** Performance of different methods on the Leaving Group Identification task.
> | Method          | Overall   |    |    | LvG   |    |    |
> |-|-|-|-|-|-|-|
> |           | **ACC**   | **CEN**   | **MCC**   | **ACC**   | **CEN**   | **MCC**   |
> | Molecular Graph   | 0.950 | 0.036 | 0.549 | 0.448 | 0.201 | 0.519 |
> | Rxn Hypergraph    | 0.969 | 0.026 | 0.743 | 0.679 | 0.150 | 0.699 |
> | D-MPNN            | 0.993 | 0.003 | 0.949 | 0.902 | 0.051 | 0.899 |
> | Reaction Graph     | **0.997** | **0.002** | **0.973** | **0.947** | **0.031** | **0.945** |
>
>
>
> ## 3. Cross-validation Results
>  We provide cross-validation results of Leaving Group Identification on USPTO (Table 31).
> According to the results of cross-validation, Reaction Graph's performance in leaving group identification still far exceeds that of the molecular graph. This demonstrates that its design for reaction modeling effectively addresses the shortcomings of the molecular graph.
>
> **Table 31:** Cross-validation results for leaving group identification on USPTO. The metrics includes mean and standard deviations from multiple trials with different train/test splits.
> | Method          | Overall               |                |                | LvG               |                |                |
> |-|-|-|-|-|-|-|
> |           | **ACC**   | **CEN**   | **MCC**   | **ACC**   | **CEN**   | **MCC**   |
> | Molecular Graph   | $0.948 \pm 0.004$ | $0.038 \pm 0.003$ | $0.535 \pm 0.015$ | $0.418 \pm 0.010$ | $0.203 \pm 0.001$ | $0.495 \pm 0.008$ |
> | Reaction Graph     | $0.995 \pm 0.002$ | $0.004 \pm 0.001$ | $0.955 \pm 0.012$ | $0.915 \pm 0.022$ | $0.050 \pm 0.011$ | $0.912 \pm 0.022$ |
>
>
> ## 4. Discussion of Systematic Errors or Biases
> In leaving group identification, systematic errors may be introduced through:
>
> (1) Dataset biases
>
> (2) Randomness during the training process
>
> (3) Differences in the computing environment
>
> To address these issues, we follow your advise:
>
> (1) We use cross-validation across various dataset split to mitigate dataset biases
>
> (2) We adopt multiple random seeds to reduce systematic errors caused by randomness. According to the results in Table 32, the standard deviation for the leaving group identification across different trials is small. The small standard deviation indicates that the results reliably reflect the model's performance.
>
> (3) We provide detailed descriptions of the machines and environment in Sec. C.3.
>
> **Table 32:** Results for leaving group identification on USPTO. The metrics include mean and standard deviations from multiple trials with different random seeds.
> | Method          | ACC               | CEN               | MCC               | ACC               | CEN               | MCC               |
> |-|-|-|-|-|-|-|
> | Molecular Graph   | $0.950 \pm 0.001$ | $0.037 \pm 0.001$ | $0.538 \pm 0.006$ | $0.423 \pm 0.004$ | $0.209 \pm 0.002$ | $0.497 \pm 0.005$ |
> | Reaction Graph     | $0.996 \pm 0.001$ | $0.002 \pm 0.001$ | $0.971 \pm 0.001$ | $0.944 \pm 0.001$ | $0.035 \pm 0.001$ | $0.942 \pm 0.001$ |
>
>
>
> ## 5. Analysis of Chemical Pattern Dependencies
> In chemical reactions, the most important chemical pattern is the structure of the molecules, as well as the changes during the reaction. Through comparisons with more baselines in Table 30, we found that the patterns of reaction changes play a crucial role in the positioning of leaving groups. At the same time, the molecular topology patterns also support the classification of leaving groups.
>
> Reaction Graph conveys 3D structure while integrating patterns of reaction changes, thereby achieving the best performance in the leaving group identification task.

---

> ### Author Response · Authors · 2024-11-25
> **Part Ⅵ: Runtime Analysis Details**
>
> ## 1. System Specifications
> + CPU: Intel(R) Xeon(R) Gold 6226R CPU @ 2.90GHz, 64 Cores, 1200-3900 MHz
>
> + RAM: 472GB (32GB * 16), DDR4, 2666 MT/s
>
> + Disk: 72.4TB, HDD
>
> + GPU: NVIDIA GeForce RTX 4090, 24GB * 8
>
> ## 2. Benchmark Conditions
> All the test is conducted under Ubuntu 22.04.5 LTS, with cuda 11.0, pytorch 1.12.1+cu113, rdkit 2023.9.4, dgl 0.9.1.post1. Other toolkit versions are provided in the attachments.
>
> We use the time module to test the runtime.
>
> ## 3. Variability in Performance
>
> We use the number of nodes as an indicator of data complexity, and measure the variability in performance at different data scales.
>
> Specifically, we test the pre-processing efficiency (Table 33) and computational efficiency (Table 34) of different baselines and our method.
> According to the results in Table 33, the pre-processing time increases non-linearly with the number of nodes in the molecular graph. This is due to the complexity of the MMFF algorithm. We also observe certain fluctuations in computation time. This phenomenon is caused by the uncertainty in the iterative convergence of the MMFF algorithm.
> As shown in Table 34, the computation time for each type of graph increases linearly with the number of atoms. This trend is due to the relative sparsity of the molecular graph.
>
>
>
> **Table 33:** Data pre-processing time for different representations under varying numbers of graph nodes. Time units are milliseconds (ms).
> | **Representation**    | **Reaction Edge Calculation** | **3D Conformer Generation** | **35** | **60** | **85** | **110** | **135** | **160** | **185** | **210** | **235** | **260** | **285** | **310** | **335** | **360** |
> |-|-|-|-|-|-|-|-|-|-|-|-|-|-|-|-|-|
> | Reaction Graph        | √                | √              | 3.2    | 5.6    | 10.1   | 18.1    | 28.7    | 24.9    | 52.5    | 54.0    | 72.4    | 82.8    | 100.3   | 244.4   | 585.6   | 923.3   |
> | w/o Reaction Edge     | ×                | √              | 3.6    | 4.4    | 7.2    | 16.8    | 26.0    | 28.8    | 45.9    | 57.5    | 69.3    | 81.3    | 91.9    | 157.8   | 722.1   | 945.9   |
> | w/o Angular Edge      | √                | ×               | 1.2    | 1.5    | 2.3    | 2.7     | 3.5     | 3.9     | 4.8     | 5.4     | 6.1     | 6.8     | 5.6     | 5.5     | 21.7    | 29.3    |
> | Molecular Graph       | ×                 | ×               | 0.7    | 1.0    | 1.3    | 1.5     | 2.4     | 2.2     | 3.0     | 3.4     | 4.7     | 4.6     | 4.9     | 4.1     | 15.9    | 19.9    |
>
>
> **Table 34:** Model inference time for different graph representations under varying numbers of graph nodes. Time units are milliseconds (ms).
> | **Representation** | **Reaction Information** | **3D Structure** | 2500  | 3000  | 3500  | 4000  | 4500  |
> |-|-|-|-|-|-|-|-|
> | Reaction Graph                           | √                        | √                | 37.28 | 43.77 | 50.57 | 58.03 | 65.45 |
> | w/o Reaction Edge                        | ×                        | √                | 35.36 | 40.92 | 48.68 | 53.30 | 58.79 |
> | w/o Angular Edge                         | √                        | ×                | 22.51 | 25.96 | 29.82 | 34.21 | 38.77 |
> | Molecular Graph                          | ×                        | ×                | 18.59 | 21.93 | 25.39 | 28.30 | 31.48 |
> | Rxn Hypergraph                           | ×                        | ×                | 31.18 | 37.28 | 43.31 | 48.53 | 54.06 |
> | Condensed Graph of Reactions             | √                        | ×                | 44.55 | 52.48 | 60.75 | 66.32 | 75.34 |
>
>
> ## 4. Hardware Dependencies
> + Training (same batch size and code as ours)
>   + GPU: At least 24GB VRAM
>   + RAM: At least 16GB
>
> + Inference
>   + GPU: 2GB VRAM

---

> ### Author Response · Authors · 2024-11-25
> **Reply Ⅰ- Ⅱ :  Reproducibility Ensurance and Statistical Analysis**
>
> Thank you for your questions. We respond to the first two questions in **Part Ⅰ: Ensuring Reproducibility: Code, Guidelines, and Parameters** and **Part Ⅱ-1 ~ Part Ⅱ- 4: Statistical Analysis**.

---

> ### Author Response · Authors · 2024-11-25
> **Reply Ⅲ: Handling Multiple Conformation by Invariant 3D Feature Modeling**
>
> **1. The 3D features of RG are invariant to different conformations.**
>
>    A molecule has different conformations due to the rotation of single bonds. Reaction Graph (RG) have already considered multiple conformations of a molecule. Specifically, the 3D features used by RG are bond lengths and bond angles. In different conformations of the same molecule, bond lengths and bond angles generally change only slightly. Therefore, the 3D features of RG are invariant to different conformations. The conformational invariance of the RG ensures consistency of 3D features across training samples, making it easier for the model to learn.
>
>
> **2. Properties of chemical reactions are invariant to different conformations.**
>
>    There is a chemical reaction $A + B \xrightarrow{\text{D}} C$. The catalyst $D$ won't change with the conformations of $A$, $B$, or $C$. Thus, RG is consistent with the task objectives.
>
>
> **3. Compared to other 3D features that vary with conformations.**
>
>    We also explore other 3D features that vary with conformations, including torsion angles, bond vectors, and atom coordinates. The specific implementation of those methods can be referenced in Sec.C.1.1 and Sec.C.2.2. The results are shown in Table 35. It can be seen that RG, which includes only bond lengths and bond angles as 3D information, achieves the best results.
>
> **Table 35:** Influence of methods to handling multiple conformation on models performance. Bond Vector, Atom Coordinate and Torsion Angle will change with conformation. In contrast, Bond Length and Bond Angle are invariant to specific conformation.
> | **Method**                                        | **Top-1** ↑ | **Top-3** ↑ | **Top-5** ↑ | **Top-10** ↑ | **Top-15** ↑ |
> |-|-|-|-|-|-|
> | Without 3D Information                                                | 0.3133       | 0.4248       | 0.4613       | 0.4961        | 0.5094        |
> | With Bond Vector        | 0.2899       | 0.4026       | 0.4390       | 0.4749        | 0.4879        |
> | With Atom Coordinate            | 0.3123       | 0.4243       | 0.4628       | 0.4987        | 0.5111        |
> | With Bond Length, Bond Angle and Torsion Angle   | 0.3022       | 0.4087       | 0.4467       | 0.4821        | 0.4935        |
> | With Bond Length and Bond Angle (ours)       | **0.3246**       | **0.4343**       | **0.4715**       | **0.5061**        | **0.5181**        |

---

> ### Author Response · Authors · 2024-11-25
> **Reply Ⅳ: Reaction Graph uses Reaction Itself as Basis**
>
> The algorithm for constructing the reaction graph is described in Sec.3.2, Sec.C.1.1, and Sec.D.1. Here, we provide a brief summary:
>
> Reaction Graph presents a unified modeling of the entire reaction, rather than being based solely on part of the reaction. During the construction of Reaction Graph, the reactants and products are built simultaneously (including their conformations) in the first step. In this phase, they have no interdependencies or specific order. After each molecule is generated, the construction algorithm connects the reactants and products into a unified Reaction Graph. This stage is still based on both reactants and products.

---

> ### Author Response · Authors · 2024-11-25
> **Reply Ⅴ: Criteria for Conformation Selection**
>
> In Sec. D.1 and Sec. F.1, we detail our handling and selection of conformations.
>
> The selection of conformations theoretically does not affect the 3D information of Reaction Graph, as metioned in Reply Ⅲ. However, we still generate multiple conformations for each molecule to account for errors from the MMFF algorithm. Unlike molecular datasets, we do not generate multiple conformations for each reaction in reaction dataset. This is because the same molecule can appear in different reactions. We only need to calculate one conformation for each reaction, then the molecules in the dataset can have multiple conformations.
>
> For conformation for each molecule, we try to select the conformation calculated by the most accurate algorithm. Specifically, we use RDKIT, starting with the ETKDG algorithm for initialization. The MMFF algorithm is then used for optimization. If it fails to converge, we switch to the UFF algorithm. If that also fails, we keep the ETKDG results. If ETKDG fails initially, we use a simpler 2D conformation.
>
> Our basis is:
> + Large dataset allows the model to learn from conformation calculation errors.
> + We retain molecules whose conformation cannot be computed, because their structure and labels still provide valuable prior information for the model.

---

> ### Author Response · Authors · 2024-11-25
> **Reply Ⅵ: Leaving Group Identification Analysis and Additional Baselines**
>
> Thank you for your comment. We respond to this question in **Part Ⅴ: Leaving Group Identification Analysis**.

---

> ### Author Response · Authors · 2024-11-25
> **Reply Ⅶ: Detailed Runtime Analysis including System Specifications and Benchmark Conditions**
>
> For System Specifications and Benchmark Conditions, please refer to **Part Ⅵ: Runtime Analysis Details**.  The runtime analysis is as follows.
>
> ## Our method
> 1. **Theoretical Analysis**
>
>     We define all graph-based representations of chemical reactions as $ G(V, E) $, where $ V $ represents the set of atoms and $ E $ represents the set of bonds. The computation complexity of the Reaction Graph depends on the number of atoms $ |V| $, the number of bonds $ |E| $, the maximum degree $ D $ of the atoms, and the specific implementation of the graph.
>
>     In Reaction Graph, a reactant atom can match with at most one product atom, so the number of reaction edges does not exceed half the total number of atoms, $ |V|/2 $. Furthermore, as defined for angular edges in Fig. 1, the maximum number of angular edges is $ |V| D^2 $.
>
>     Thus, the computational complexity contributed by bond edges is $ |E| $, by reaction edges is approximately $ |V|/2 $, and by angular edges is around $ |V| D^2 $. Consequently, the total computational complexity of Reaction Graph can be approximated as $ |E| + |V|/2 + |V| D^2 $.
>
> 2. **Experimental Results**
>
>     We assess the impact of reaction edges and angular edges in the Reaction Graph on model inference speed as the number of atoms $ |V| $ increases. The findings are presented in rows 1-3 of Table 36. The time cost associated with reaction edges is lower than that caused by angular edges.
>
> ## Compared with Other Methods
> 1. **Theoretical Analysis**
>     + Graph-based methods
>
>         When the molecular graph is represented as $G(V,E)$, $\vert V \vert$ indicates the total number of atoms in reactants and products, and $\vert E \vert$ represents the total count of bonds. The computation cost for the molecular graph is $\vert E \vert$. The computation cost of Rxn Hypergraph is at least $\vert E \vert + \vert V \vert$. The edge count in Condensed Graph of Reactions (CGR) is roughly $\vert E \vert /2$. However, when handling the same amount of information, the dimensions of the fully connected layers in CGR far exceed those in the Reaction Graph and Rxn Hypergraph. This leads to greater computational overhead.
>
>     + SMILES-based methods
>
>         SMILES-based methods are generally based on Transformer. When the length of the input SMILES is $\vert L \vert$, the computation complexity is roughly $ \vert L \vert ^2 $. Since there is no direct conversion between the length of SMILES strings $\vert L \vert$ and the number of nodes $\vert V \vert$ and edges $\vert E \vert$ in molecular graphs, we directly measured their computational efficiency through experiments.
>
> 2. **Experimental Results**
>     + Compared with graph-based methods
>
>         We compare the inference time of Reaction Graph and other graph-based methods. The results are shown in rows 4-6 of Table 36. Due to incorporating additional reaction information and 3D structures, our method requires slightly more time than molecular graphs and Rxn Hypergraph representations. However, it delivers significant performance improvement, justifying the additional time cost.
>
>         **Table36:** Model inference time for different graph representations under varying numbers of graph nodes. Time units are milliseconds (ms).
>         | **Representation** | **Reaction Information** | **3D Structure** | 2500  | 3000  | 3500  | 4000  | 4500  |
>         |-|-|-|-|-|-|-|-|
>         | Reaction Graph                           | √                        | √                | 37.28 | 43.77 | 50.57 | 58.03 | 65.45 |
>         | w/o Reaction Edge                        | ×                        | √                | 35.36 | 40.92 | 48.68 | 53.30 | 58.79 |
>         | w/o Angular Edge                         | √                        | ×                | 22.51 | 25.96 | 29.82 | 34.21 | 38.77 |
>         | Molecular Graph                          | ×                        | ×                | 18.59 | 21.93 | 25.39 | 28.30 | 31.48 |
>         | Rxn Hypergraph                           | ×                        | ×                | 31.18 | 37.28 | 43.31 | 48.53 | 54.06 |
>         | Condensed Graph of Reactions             | √                        | ×                | 44.55 | 52.48 | 60.75 | 66.32 | 75.34 |
>
>     + Compared with SMILES-based methods
>
>         We compare the inference time of Reaction Graph and SMILES-based methods. The results are shown in the table below, where Reaction Graph, as a graph-based model, exhibits the shortest inference time.
>
>         **Table37:** Inference time comparison for Reaction Graph and other methods.
>         | **Methods**             | **Representation**    | **Time**      |
>         |-|-|-|
>         | T5Chem                  | SMILES                | 3min 19s     |
>         | RXNFP                   | SMILES                | 23min 43s    |
>         | Reaction Graph (ours)   | Reaction Graph        | **2min 36s**     |
>
> The details of the cost analysis can be found in Sec.E.

---

> ### Author Response · Authors · 2024-11-27
> **Feed back to Reviewer uB2j**
>
> Dear Reviewer uB2j,
>
> We are glad to have addressed your concerns. Thank you for raising the score, and we especially appreciate your efforts in improving this work.
>
> Best regards,
>
> Paper 4277 Authors

---

### Official Review · Reviewer_N9iv · 2024-11-01

**Soundness:** 2
**Presentation:** 3
**Contribution:** 2
**Rating:** 5
**Confidence:** 2

**Summary:**

The paper proposes a type of graph, called Reaction Graph, to model chemical reactions that are overlooked in existing molecular graphs. Reaction graph allows GNNs to capture chemical reactions during the message passing stage and hence improves the accuracy of chemical reaction condition prediction tasks.

**Strengths:**

1. The paper includes chemical reactions in the graph representation.

2. The paper also includes 3D molecular information in the reaction modeling.

3. The proposed method achieves better prediction accuracy.

**Weaknesses:**

1. The novelty of the paper is unclear. It's not clear why Reaction Graph is a novel graph representation.

2. The proposed solution seems to be rather ad hoc, for example why RBF is used to embed the edge length, the way the vertex-edge integration is conducted, and why an attention-based aggregation method with an LSTM is employed. In addition, the solution also looks to be pretty straightforward and its novelty is unclear.

3. The significance of the work is unclear to a general reader whose background is perhaps more in CS rather than in chemistry, and therefore it's hard to assess the significance of the results presented in Section 4.

**Questions:**

It would help readers to better appreciate the contributes of this work if the novelty of the work can be clearly justified, e.g., why Reaction Graph is a novel concept, why edge embedding, vertex-edge integration, and attention-based aggregation in the solution are reasonable and how do they compare with their alternatives, what is the novelty in the solution?

The significance of the work might be easier for experts in chemistry to understand and hence this work might be better to be submitted to venues in chemistry, or otherwise, I suggest the authors to significantly improve the technical depth of the work.

---

> ### Author Response · Authors · 2024-11-24
> **Part Ⅰ: Novelty of Reaction Graph**
>
> Thank you for acknowledging our method "achieves better prediction accuracy". We also appreciate your constructive suggestions.
>
> ## Novelty of Reaction Graph
>
> + **First graph to understand chemical reactions at the atom level.** When using deep neural networks for chemical reaction modeling, existing methods lack a mechanism to account for the interactions between atoms during a reaction. Typically, these methods use molecule-level global representations of reactants and products for chemical reaction reasoning, which fails to effectively capture the underlying reaction mechanisms.
>     In contrast, our Reaction Graph is equipped with the ability to model relationships at the atom level, thus enabling effective reasoning of reactions.
>
> + **First graph with 3D chemical reaction modeling.** Existing graph-based methods for representing chemical reaction only consider 2D molecular structure, while overlooking 3D structural information. In constrast, our Reaction Graph is capable of modeling 3D structural transformation during chemical reaction, which better captures the dynamics and changes occurring throughout the reaction.
>
> + **Data-and-Knowledge-Driven Chemical Reaction Reasoning.**
>     Existing methods for understanding chemical reactions typically employ either rule-based or data-driven approaches. In our work, we successfully integrate chemical knowledge and rules into deep learning to enhance reaction reasoning.

---

> ### Author Response · Authors · 2024-11-25
> **Part Ⅱ: The Reason for Using RBF kernel as Edge Embedding Module**
>
> ## Why RBF is used to embed the edge length?
>
> RBF can effectively capture the non-linear relationships between distance and molecular property. This method lifts the raw edge lengths into a high-dimensional vector that can be more easily utilized by machine learning models, focus more on local structural pattern, and produces smooth mappings which help in capturing variations in continuous data. These advantages make RBF kernel suitable for tasks involving local continuous spatial relationships, such as in molecular structures where edge lengths indicate bond distances.
>
> To demonstrate the efficiency of RBF kernel embedding, we conducted experiments to compare it with different embedding methods, using USPTO-Condition dataset.
>
> **Table 1:** Impact of different edge embedding methods on model's performace.
> | **Edge Embedding Method**                       | **Top-1** ↑ | **Top-3** ↑ | **Top-5** ↑ | **Top-10** ↑ | **Top-15** ↑ |
> |----------------------------------|--------------|--------------|--------------|---------------|---------------|
> | Linear Projection Embedding  | 0.3173      | 0.4303      | 0.4684      | 0.5048       | 0.5164       |
> | Discretization Embedding     | 0.3101      | 0.4201      | 0.4569      | 0.4926       | 0.5046       |
> | RBF kernel Embedding (ours)  | **0.3246**      | **0.4343**      | **0.4715**      | **0.5061**       | **0.5181**       |
>
>
> As shown in the table above, RBF kernel outperforms other embedding methods, demonstrating its efficiency. Details could be found in Sec.C.2.2 and Sec.G.4.1.

---

> ### Author Response · Authors · 2024-11-25
> **Part Ⅲ-1: Explanation of Vertex-Edge Integration (Design)**
>
> ## Explanation of Vertex-Edge Integration
> ### Design
>
> Reaction Graph involves multiple types of edges. Effectively utilizing these edges to improve the modeling of chemical reaction structures and interactions with vertices (atoms) is crucial for accurate reaction reasoning.
> In our method, inspired by Dynamic Convolution[1] and Hypernetworks[2], we first use these edges to generate kernel weights
>
> $\mathcal{M_{ij}} = \text{Reshape}(\boldsymbol{W} \cdot [\boldsymbol{l_{ij}};\boldsymbol{e_{ij}}]),$
>
> which are then applied to vertex features.
>
> $\\boldsymbol{v}^{t+1}\_i = \\boldsymbol{v}^{t}\_i + \\sum\_{j \\in \\mathbb{N}\_i} \\mathcal{M}\_{ij} \\cdot \\boldsymbol{v}^{t}\_j, $
>
> In this way, our method can effectively capture the reaction  information.
>
> Our vertex-edge integration module handles different types of edges in the Reaction Graph and reflects the complementary role of 3D information to the 2D topological structure.
>
> Specificly, we start from the basic paradigm of MPNN:
>
> $
>     \\boldsymbol{v}^{t+1}\_i = \\boldsymbol{v}^{t}\_i + \\sum\_{j \\in \\mathbb{N}\_i} f(\\boldsymbol{v}\_i,\\boldsymbol{e}\_{ij},\\boldsymbol{v}\_j),~~~~~\\boldsymbol{v}'\_i = \\boldsymbol{v}^{T\_1}\_i,
> $
>
> where $f$ is the message function to integrate vertex and edge information, $\\boldsymbol{v}\_i$ is the node feature, and $\\boldsymbol{e}\_{ij}$ is the edge feature. Reaction Graph contains three distinct types of edges, including bond egde for modeling interatomic force ($e\_{ij} = 1$), reaction edge for modeling atomic mapping ($e\_{ij} = 2$), and angular edge for modeling 3D multi-body relation ($e\_{ij} = 3$). Based on the principle of MPNNs, we believe the message function $f(\\boldsymbol{v}\_i,\\boldsymbol{e}\_{ij},\\boldsymbol{v}\_j)$ for these edges should be fundamentally different:
>
> $
> f =
> \begin{cases}
> f_{bond} & \text{if } e_{ij} = 1, \\\\
> f_{reaction} & \text{if } e_{ij} = 2, \\\\
> f_{angular} & \text{if } e_{ij} = 3.
> \end{cases}
> $
>
> We use linear layers to implement these different message functions as $f\_t(\\boldsymbol{v}\_i,\\boldsymbol{v}\_j) = \\mathcal{M}\_t \\boldsymbol{v}\_j$, where $t \\in\\{bond,reaction,angular\\}$. Based on these, the hypernetwork-like design can generate different linear layer weight matrices based on the different edge types to process node features (assume $\\boldsymbol{e}\_{ij}$ is the one-hot representation of $e\_{ij}$):
>
> $
> \\mathcal{M}\_t = \\text{Reshape}(\\boldsymbol{W} \\cdot \\boldsymbol{e}\_{ij})
> $
>
> Meanwhile, the weight matrices generated from edge length embeddings serves as the adaptation of edge type weight matrices as (omit $ij$ for convenience):
>
> $
> \\mathcal{M} = \\mathcal{M}\_t + \\Delta\mathcal{M}\_{3D},~~~~~~
> \\Delta\\mathcal{M}\_{3D} = l\_1\\mathcal{M}\_1 + l\_2\\mathcal{M}\_2 + \\dots + l\_k\\mathcal{M}\_k,
> $
>
> where $\\boldsymbol{l}$ is the RBF embedding, $\\mathcal{M}\_t \\in \\{\\mathcal{M}\_{bond}, \\mathcal{M}\_{reaction}, \\mathcal{M}\_{angular}\\}$, and $\\mathcal{M}\_1$ to $\\mathcal{M}\_k$ is the learningable 3D bias matrices. This approach is liken to using ControlNet to add prior information such as wireframes based on the original parameters of Stable Diffusion, and is consistent with the complementary role of 3D information to 2D topology and atomic information. The final formulation is as:
>
> $
>     \\mathcal{M}\_{ij} = \\text{Reshape}(\\boldsymbol{W} \\cdot [\\boldsymbol{l}\_{ij};\\boldsymbol{e}\_{ij}]),~~~~~~\\boldsymbol{v}^{t+1}\_i = \\boldsymbol{v}^{t}\_i + \\sum\_{j\\in \\mathbb{N}\_i} \\mathcal{M}\_{ij} \\cdot \\boldsymbol{v}^{t}\_j,~~~~~\\boldsymbol{v}'\_i = \\boldsymbol{v}^{T\_1}\_i,
> $
>
> In the implementation, we also introduce GRU and residual connections. Since the MPNN increases the receptive field of node features in each iteration, the gating mechanism of GRU helps retain the sturctural information of appropriate neighbor size (some neighbor structure with certain size might act as a functional group in molecule), while residual connection combines neighbor information with atomic features.
>
> For more details, please refer to Sec.C.2.
>
> [1]Chen Y, Dai X, Liu M, et al. Dynamic convolution: Attention over convolution kernels[C]//Proceedings of the IEEE/CVF conference on computer vision and pattern recognition. 2020: 11030-11039.
>
> [2] Ha D, Dai A M, Le Q V. HyperNetworks[C]//International Conference on Learning Representations. 2017. https://openreview.net/forum?id=rkpACe1lx.

---

> ### Author Response · Authors · 2024-11-25
> **Part Ⅲ-2: Explanation of Vertex-Edge Integration (Experiment)**
>
> ## Explanation of Vertex-Edge Integration
> ### Experiment
>
> We also conduct experiment to tested different vertex-edge integration methods and demonstrated that our chosen approach has the best performance.
>
> **Table 2:** Impact of different vertex-edge integration methods on model's performace.
> | Vertex-edge Integration Method                                | **Top-1** ↑ | **Top-3** ↑ | **Top-5** ↑ | **Top-10** ↑ | **Top-15** ↑ |
> |-|-|-|-|-|-|
> | Bond Vector Model (Following PaiNN)   | 0.290        | 0.403        | 0.439        | 0.475         | 0.488         |
> | Bond Angle Model (Following DimeNet)  | 0.318        | 0.429        | 0.466        | 0.502         | 0.514         |
> | GAT (Following EGAT)                  | 0.304        | 0.417        | 0.453        | 0.490         | 0.502         |
> | GIN (Following GINE)                  | 0.299        | 0.406        | 0.441        | 0.475         | 0.487         |
> | Ours                 | **0.325**       | **0.434**        | **0.472**        | **0.506**         | **0.518**         |
>
>
> The result shows that our method is superior to other methods, demonstrating its efficiency. Details could be found in Sec.C.2.2 and Sec.G.4.2.

---

> ### Author Response · Authors · 2024-11-25
> **Part Ⅳ: The Reason for Using Attention-based Aggregation with an LSTM**
>
> ## Why an attention-based aggregation method with an LSTM is employed?
>
> Effectively identifying and leveraging the chemical reaction mechanism to understand and reason about reactions is challenging. In our work, we use an attention mechanism to adaptively capture the most important cues for reaction modeling. However, since these cues are not always easy to identify in one time, we employ an LSTM to progressively and interactively discover them. As shown in Fig.3 in the main paper, the attention-based aggregation module with LSTM accurately locates the reaction center on Reaction Graph.
>
> We conducted experiments to demonstrate the roles of Attention and LSTM. Specifically, we compared the performance of the aggregation module without using Attention and LSTM on USPTO-Condition dataset.
>
> **Table 3:** Impact of Attention and LSTM in aggregation module.
> | **Aggregation Method**                           | **Top-1** ↑ | **Top-3** ↑ | **Top-5** ↑ | **Top-10** ↑ | **Top-15** ↑ |
> |--------------------------------------|--------------|--------------|--------------|---------------|---------------|
> | w/o Attention & w/o LSTM         | 0.3159       | 0.4276       | 0.4642       | 0.4983        | 0.5110        |
> | w/ Attention & w/o LSTM          | 0.3187       | 0.4303       | 0.4670       | 0.5018        | 0.5136        |
> | w/ Attention & w/ LSTM           | **0.3246**       | **0.4343**       | **0.4715**       | **0.5061**        | **0.5181**        |
>
>
> The result in the table above shows that the Attention mechanism and LSTM contributes to the performance. Detail implementations and analysis can be found in Sec.C.2.2 and Sec.G.4.3.

---

> ### Author Response · Authors · 2024-11-27
> **Part Ⅴ: Contributions to the Fields of Computer Science and Chemistry**
>
> ### 1. Contribution to Computer Science
> + **Promoting interdisciplinary development.** Interdisciplinary research is an important direction in computer science. We are the first to integrate interaction modeling from computer science into reaction tasks, thereby promoting the development of computer science methods in the field of chemistry.
>
> + **Application of computer science concepts.** The design of 3D information in Reaction Graph incorporates design principles from computer science, including modal consistency and algorithm parallelism. Our work demonstrates that the integration of computer science concepts is effective, and thus encourage their application in interdisciplinary research.
>
> + **Provides insights for computer science tasks.** Our approach handles complex individual interactions and geometric information. This also provides insights for graph representation learning and 3D geometric data modeling tasks in computer science.
>
> ### 2. Contribution to Chemistry
> + **State-of-the-art performance.** Our proposed Reaction Graph achieves state-of-the-art performance in multiple chemical reaction tasks. This provides better support for chemical tasks such as synthetic design, thereby driving chemical discoveries.
> + **Chemical knowledge integration.** Reaction Graph integrates domain knowledge such as atomic mapping. This enhances the model's adaptability to chemical tasks, and our design philosophy promotes the development of domain models.
> + **Foundational for reaction modeling.** Reaction Graph is an improvement at the graph representation level. It can be effectively utilized in any future method based on GNNs and enhance performance in chemical reaction tasks. This innovation promises a lasting impact on the field of chemical reaction property prediction.

---

> ### Author Response · Authors · 2024-11-27
>
> Dear Reviewer N9iv,
>
> Thank you very much for your time and effort in reviewing our submission.
> We have tried our best to address the concerns and problems mentioned.
> In particular, we explain the novelty of this work and its significance to the development of interdisciplinary computer science.
> We hope that our responses have addressed your concerns.
> Your support is significantly important to our paper.
> If you need further information, please let us know.
> We look forward to hearing from you.
> Thank you once again.
>
>
> Best wishes,
>
> Paper 4277 Authors

---

> ### Author Response · Authors · 2024-12-03
>
> Dear Reviewer N9iv,
>
> Thank you very much for your efforts in reviewing our paper.
> The discussion phase is nearing its end, and we are looking forward to receiving your feedback on our responses.
> Should you have any further concerns, we are glad to provide additional information.
> Thank you once again.
>
> Best regards,
>
> Paper 4277 Authors

---

### Official Review · Reviewer_dqUg · 2024-11-02

**Soundness:** 4
**Presentation:** 4
**Contribution:** 3
**Rating:** 8
**Confidence:** 3

**Summary:**

This paper proposes the unified 3D Reaction Graph (RG), a novel graph representation for chemical reactions. By conducting various experiments on chemical reactions and comparing RG with existing models that use fingerprints and molecular graphs, the paper demonstrates that RG achieves state-of-the-art performance across multiple tasks. The findings highlight the importance of incorporating reaction edges and 3D structural information, represented as edge lengths and angular edges, rather than relying solely on molecular graphs.

**Strengths:**

- The Reaction Graph (RG) incorporates reaction edge and 3D information into molecular graphs, enhancing the representation of reactants and products as well as improving predictive performance on chemical reactions.
- The authors conduct experiments on a wide range of tasks and compare the proposed method with various models, demonstrating the effectiveness of RG.
- Since the proposed method is independent of the components and architectures of graph neural networks, any model can be used for chemical reaction modeling.

**Weaknesses:**

Although the authors show the importance of 3D information and confirm a slight increase in performance for chemical reaction prediction, the 3D information's utility may depend on the computation method. Thus, performance could deteriorate if adequate computation is not applied. It may be difficult to discern which components (the features obtained from an insufficiently trained GNN or the 3D features themselves) contribute to the performance.

**Questions:**

- According to the ablation studies in Tables 2 and 4, the 3D information obtained from MMFF94 appears to contribute only marginally to performance. Could applying a more accurate method, such as DFT, potentially improve the performance?
- In Figure 3, DFT and other theoretical calculations provide insights into reaction sites by calculating energy potentials and activation energies. How well do the attention weights from the RG model correlate with these theoretical values? Could this correlation potentially validate the model's focus on reaction centers?

---

> ### Author Response · Authors · 2024-11-26
> **Part Ⅰ-1: How Will Accurate Data Improve the Performance (Theoretical Analysis)**
>
> Thank you for acknowledging that our method is **effective**, universally applicable, and can **enhance the representation of reactants and products as well as improve predictive performance on chemical reactions**.
> We also appreciate your valuable and insightful suggestions.
>
>
>
> # How Will Accurate Data Improve the Performance
>
> The accuracy of 3D information does affect model performance, but the factors causing performance bottlenecks are the quality of annotated labels. We illustrate this through theoretical analysis and experimental evidence.
>
>
> ## 1. Theoretical Analysis
>
> We demonstrate that 3D information and label jointly influence model performance. Assume that the neural network is sufficiently trained and perfectly fits the data.
>
> + **Noisy Labels.** For a target function $ f: \\mathbb{X}\\rightarrow \\mathbb{Y} $ of neural network training, the quality of the label $ Y \\in \\mathbb{Y} $ has a significant impact on the neural network performance. If there is noise $ \\Delta Y\_{\\text{error}} $ in $ Y $, then the fitted network $f\_1: \\mathbb{X}\\rightarrow \\mathbb{Y}$ will have output $ f\_{1}(X) = Y + \\Delta Y\_{\\text{error}} $ with input $X$. This results in $f\_1$ being unable to correctly model the relationship between $ X $ and $ Y $.
>
> + **Label Sparsity.** Assume the annotations $ (X, Y) $ are sparse in training set,
> $f_2: \mathbb{X}\rightarrow \mathbb{Y}$ is the fitted neural network on the sparse training set.
> Then for outliers in the test data $ (X + \Delta X, Y + \Delta Y) $, the theoretical upper bound $ |\Delta X| \cdot (L_f + L_{f_2}) $ of the prediction error $ \Delta Y_{outlier} $ will increase, where $X + \Delta X$, $Y + \Delta Y$ are the input and ground truth label of the outlier,
> $ \Delta Y_{outlier} = f_2(X+\Delta X) - (Y+\Delta Y) = [f_2(X+\Delta X)-f_2(X)]-\Delta Y $,
> and $L_f$, $L_{f_2}$ are the Lipschitz constant [1] of $f$ and $f_2$, respectively.
>
> + **Noisy 3D Information.** The accuracy of 3D information also affects the neural network performance. Assume that the error of 3D information is $\Delta X_{error}$, and the network trained with error samples $(X + \Delta X_{error},Y)$ is $f_3: \mathbb{X}\rightarrow \mathbb{Y}$. We have $ f_{3}(X+\Delta X_{error}) = Y $.
> Then $f_3$ will produce bias $\Delta Y_{errorX}$ during inference like $ f_{3}(X) = Y + \Delta Y_{errorX} $
> , where $ |\Delta Y_{errorX}| = |f_{3}(X)-f_{3}(X+\Delta X_{error})| \leq |\Delta X_{error}| \cdot L_{f_3} $, and $L_{f_3}$ is the Lipschitz constant [1] of $f_3$.
>
> The 3D structure is only part of the structural information, and $|\\Delta X\_{error}|$ is controllable when MMFF can converge. However, the distinction $|\\Delta Y\_{\\text{error}}|$ between labels and the distance $|\Delta X|$ between train-test samples can be very large. Therefore, we can conclude that $ |\\Delta Y\_{errorX}|\_{max} \\ll |\\Delta Y\_{\\text{error}}|\_{max}$ and $ |\\Delta Y\_{errorX}|\_{max} \\ll |\\Delta Y\_{outlier}|\_{max}$, where $|\\cdot|\_{max}$ denotes the upper bound.
> This conclusion implies that the quality and sparsity of labels have a greater impact on model's performance, while the 3D accuracy have relatively small influence.
>
> [1] The Lipschitz continuity of a function $ f $ is defined as the existence of a Lipschitz constant $ L $ such that for all $ X_1 $ and $ X_2 $ in the domain of $ f $, the inequality $|f(X_1) - f(X_2)| \leq L |X_1 - X_2|$ holds. This means that the rate of change of the function $ f $ is bounded by $ L $, ensuring that small changes in the input lead to controlled changes in the output. Neural networks can be considered Lipschitz continuous under certain conditions, particularly when they are composed of layers with Lipschitz continuous activation functions and bounded weights.

---

> ### Author Response · Authors · 2024-11-26
> **Part Ⅰ-2: How Will Accurate Data Improve the Performance (Experiment)**
>
> # How Will Accurate Data Improve the Performance
>
> The accuracy of 3D information does affect model performance, but the factors causing performance bottlenecks are the quality of annotated labels. We illustrate this through theoretical analysis and experimental evidence.
>
>
> ## 2. Experiment
>
> We design experiments to explore the impact of 3D accuracy and label quality on model performance.
>
> + **Accuracy of 3D Information.** The computation time of DFT on dataset containing millions of samples is far beyond our processing capacity. We are currently unable to further enhance 3D accuracy. In this case, to evaluate the impact of 3D accuracy on model performance, we adopt a reasonable alternative approach. Specifically, we reduce the 3D accuracy by adding normal noise to original MMFF calculation results. The larger the noise, the lower the accuracy.
>
>   The results are shown in Table 1.
>
>   **Table 1:** The influence of 3D accuracy on model performance in condition prediction task. The noise level reflects the accuracy of 3D information. The smaller the noise, the higher the accuracy. We use 1/8 of the USPTO-Condition dataset.
>   |**Noise Level**|**Top-1↑**|**Top-3↑**|**Top-5↑**|**Top-10↑**|**Top-15↑**|
>   |-|-|-|-|-|-|
>   |0.0|**0.1773**| **0.2749** |**0.3146**| **0.3534** |**0.3670**|
>   |0.05|0.1738| 0.2650 |0.3044| 0.3443| 0.3580|
>   |0.1|0.1695 |0.2633 |0.2990 |0.3420 |0.3570|
>   |0.2|0.1635 |0.2643 |0.2986 |0.3397 |0.3556|
>   |0.4|0.1518 |0.2510 |0.2940 |0.3346 |0.3486|
>
>     The results show that the accuracy of 3D information does affect the performance of the model. As the noise level increases, the model's performance gradually declines.
>
>
> + **Label Quality.** Label sparsity, including insufficient quantity or uneven distribution, is an important aspect of label quality.
> To evaluate the effect of label sparsity on model performance, we conduct experiments on the condition prediction task. Specifically, we separate Pistachio-Condition into multiple scaffolds of different sizes. The size of scaffold can reflect the degree of label sparsity.
>
>     Specifically, the test set is consistent across all scaffolds. From the first to the sixth group, the training set size halves each time, reducing from full size to $1/32$ of the original.
>
>     The result is shown in Table 2. It is observed that the model's performance decreases with the increase of label sparsity. An interesting point is that, in the smallest scaffold, the effect of 3D information is affected by label sparsity. These results demonstrate that label sparsity is the source of bottleneck for the model.
>
>     **Table 2:** The influence of label sparsity on model performance in condition prediction task. The degree of label sparsity can be reflected by the size of the scaffold.
>
>     | **Scaffold Ratio** | 1       | 1/2     | 1/4     | 1/8     | 1/16    | 1/32    |
>     |--------------------|---------|---------|---------|---------|---------|---------|
>     | **w/o 3D**         | 0.3852  | 0.3503  | 0.3116  | 0.2716  | 0.2210 | **0.1907** |
>     | **w 3D**           | **0.3915** | **0.3550** | **0.3146** | **0.2768** | **0.2244** | 0.1857  |
>
> + **Conclusion.** Based on the results in Tables 1 and 2, we find that the accuracy of 3D information has a significant impact on model performance; however, the quality of the labels is even more critical. The quality of the labels acts as a bottleneck for model performance, limiting the 3D information to further enhance model performance.

---

> ### Author Response · Authors · 2024-11-26
> **Part Ⅱ: Discussion on the Correlation Between Attention Weights and Theoretical Values**
>
> # Discussion on the Correlation Between Attention Weights and Theoretical Values
>
> Thank you for your insightful comments, which have provided valuable guidance for our research.
>
> We use def2-svp basis group to calculate theoretical values of molecules and reactions. And we primarily observe the electron density around atoms.
>
> The detailed results are added in Sec.G.1.4. We seem to find a certain correlation within it. For example, for atoms whose attention weights are higher, the visualization of the electron cloud is also more pronounced. Unfortunately, such correlations are not that obvious and not always exist.
>
> Hence, we further attempt the high-precision DFT calculations, and observe other related theoretical values.
> Due to time limitation, these calculations are still ongoing.
> Once completed, we will provide the high-precision results.
>
> To comprehensively validate the model's focus on reaction centers,
> we have also conducted large scale data analysis. We quantitatively test the attention result on dataset scale. The attention weight proportion of the reacting atoms is used reflect the model's focus on the reaction center. The results show that the performance of Reaction Graph exceeds that of the Molecular Graph by 40%. Details can be found in G.1.3.

---

> ### Author Response · Authors · 2024-12-03
> **Part Ⅲ: Further Discussion on the Correlation Between Attention Weights and Theoretical Values**
>
> We further explored the correlation between DFT results and reaction graph attention weights through additional experiments.
> We would like to report our latest findings, which have been uploaded to the link: https://huggingface.co/reactiongraph/ReactionGraph/blob/main/dft.zip
>
>
> ## 1. Electron Density
>
> + **Design.** We analyze the overall and HOMO/LUMO electron density of the molecules in the chemical reactions.
>
> + **Implementation.** For the overall molecular electron density, we use RDKit's MMFF to calculate the conformation, and then employ PySCF to compute the electron density. We use def2-svp basis group. The results are shown in Sec. G.1.4.
>
>     For the HOMO/LUMO electron density, we use Gaussian to simultaneously optimize the geometric structure and calculate the electron density. We use 6-31G basis group. The results can be found in https://huggingface.co/reactiongraph/ReactionGraph/blob/main/dft.zip (including .com input files and .fchk output files).
>
> + **Results and Analysis.** The results show that the overall electron density has a positive correlation with the attention weights of molecular graph. This indicates that the properties of the molecular graph are more related to the properties of individual molecules.
>     The HOMO/LUMO electron density has a positive correlation with the attention weights of both molecular graph and Reaction Graph. This may indicate that, although HOMO/LUMO are molecule-level properties or attributes, they have a strong relationship with reaction-level activities.
>
> ## 2. Activation Energy
>
> + **Design.** We use activation energy analysis to identify the reaction sites of reactants.
>
> + **Implementation.** Given a set of reactants, we locate all candidate reaction sites. Each site corresponds to a possible chemical reaction. Then we calculate the activation energy for each candidate reaction by transition state searching. Ultimately, we determine the most likely reaction sites based on the activation energy of its corresponding reaction. The smaller the activation energy, the higher the likelihood of the reaction site.
>
>     Specifically, we use Gaussian to perform the calculations. First, we identify candidate reaction sites by the electron density. Second, we construct the reaction pathways corresponding to each site. Third, we use QST2 and the 6-31G basis set to search for the transition states. Finally, we use the transition states to calculate the activation energy.
>
> + **Results and Analysis.** For all reaction sites identified by the Reaction Graph, DFT successfully calculates transition states and activation energies. However, for some candidate sites identified by molecular graph, DFT failed to converge. These results validate that the molecular graph has deficiencies in reaction modeling. In contrast, Reaction Graph focuses on correct candidate reaction sites, which demonstrates its effectiveness in reaction modeling.
>
>
> We sincerely appreciate your time and effort in reviewing our submission.
> If there is anything unclear, we are happy to provide additional information.
> Thank you once again.

---

### Official Review · Reviewer_xu23 · 2024-11-04

**Soundness:** 3
**Presentation:** 4
**Contribution:** 4
**Rating:** 6
**Confidence:** 4

**Summary:**

The authors proposed a method (i.e., Reaction Graph, RG) to learn graph representations for chemical reactions. RG conserves atom mapping information by adding edges between atoms in the reactant and product molecules. The authors also included edge distance information to capture the spatial relationship between atoms. The method was evaluated on three tasks: reaction classification, reaction condition recommendation, and reaction yield prediction. Overall, RG showed impressive performance on most dataset splits. This method can be useful for various applications in reaction modeling.

**Strengths:**

- Originality: The idea of conserving atom mapping information and including edge distance information is novel and significant.
- Quality: The method is well-designed, and the experiments are thorough.
- Clarity: The paper is well-written and easy to follow, and the figures are helpful.
- Significance: This method explores a new direction in chemical reaction modeling and can inspire future research in the field.

**Weaknesses:**

- For each table, are the results averaged over multiple runs? It would be better to include the standard deviation or confidence interval.
- The idea of using angular edges to incorporate 3D information does not consider the torsion angle. A torsion angle is defined as the dihedral angle between four connected atoms in a molecule. Molecules with different dihedral angles can show very different properties but are represented the same using the proposed representation method. The authors might need to state this explicitly as a limitation in the current version or provide a solution to this problem in future work.
- Many tools are involved in preparing the dataset, such as MMFF for 3D coordinate generation, RXNMapper for reaction atom mapping, and NameRXN for labeling reactions. The authors might want to briefly comment on the application scope/accuracy of the methods used, as the accuracy of these methods significantly influences the data quality, and consequently, the benchmark results.

**Questions:**

- It might be beneficial to provide general characteristics for each task type so that future papers can informatively discuss the tasks and compare methods. For reaction classification, many methods achieved almost perfect performance. Is it because the task is too simple? For yield prediction, there is a staggering difference between small and large datasets. This is due not only to the noise level in yield records (same reaction but different yield records) but also to the non-smooth surface of the structure-yield relationship (similar reaction but very different yield records). See examples here: https://www.rsc.org/suppdata/d3/sc/d3sc03902a/d3sc03902a2.pdf. As for reaction condition prediction, label canonicalization might be a performance consideration, as there could be multiple different names for a chemical entity. See this paper for an example of canonicalizing names: Figure 2A of https://doi.org/10.1021/jacs.4c00098.
- If I understand correctly, the current reaction global representation is a linear combination of node features. For an LSTM, the last hidden state contains the information about all previous nodes, which might also be a meaningful representation for the global reaction. How did the authors decide to use the linear combination instead of the hidden state?
- Providing the definitions of CEN and MCC might be helpful for readers who are unfamiliar with these metrics. Besides, the F1 score is a common metric for binary classification tasks in science, as it considers label imbalance. Can the authors add the F1 score in Table 1?
- Why did the authors choose to compute angular edges instead of using bond angle information? The latter seems more intuitive and is readily available via RDKit/common GNN models, so there is no need to compute the angular edges.
- Writing can be more concise. For example, "computational and mathematical chemistry" could be "computational chemistry" (line 145).
- What is \( T_1 \) in equation (3)?

---

> ### Author Response · Authors · 2024-11-25
> **Part Ⅰ-1: Standard Deviation and Confidence Interval**
>
> We thank you for acknowledging that our idea is **novel and significant**, the method is **well-designed** and **can be useful for various applications in reaction modeling**, as well as the experiments are **thorough**. We also thank you for your valuable comments.
>
> # Part Ⅰ: Standard Deviation and Confidence Interval
> Thank you for your suggestion. We provide detailed standard deviation and confidence interval values in Sec.G.3.2 and Sec.G.6.
>
> The following (from Table 1 to 14) is a brief summary of our results.
>
> ## 1. Standard Deviation
>
> **Summary:** Based on the results in Tables 1-7, we observe that the standard deviations of the training results is smaller ($≤0.01$) on larger datasets (USPTO, Pistachio). In contrast, we observe a larger training standard deviations ($>0.01$) on some smaller test set.
>
> **Table 1:** Results for leaving group identification on USPTO. The metrics includes mean and standard deviations from multiple trials with different random seeds.
> | Type               | Overall ACC         | Overall CEN         | Overall MCC         | LvG ACC             | LvG CEN             | LvG MCC             |
> |-|-|-|-|-|-|-|
> | Molecular Graph| 0.950±0.001        | 0.037±0.001        | 0.538±0.006        | 0.423±0.004        | 0.209±0.002        | 0.497±0.005        |
> | Reaction Graph | **0.996±0.001**        | **0.002±0.001**        | **0.971±0.001**        | **0.944±0.001**        | **0.035±0.001**       | **0.942±0.001**        |
>
> **Table 2:** Results for reaction condition prediction on USPTO-Condition. The metrics includes mean and standard deviations from multiple trials with different random seeds.
> | Metrics | Top-1       | Top-3       | Top-5       | Top-10      | Top-15      |
> |-|-|-|-|-|-|
> | Mean±Std of Top-K Accuracy                         | 0.322±0.002 | 0.432±0.003 | 0.469±0.003 | 0.504±0.003 | 0.516±0.003 |
>
> **Table 3:** Results for reaction condition prediction on Pistachio-Condition. The metrics includes mean and standard deviations from multiple trials with different random seeds.
> | Metrics | Top-1       | Top-3       | Top-5       | Top-10      | Top-15      |
> |-|-|-|-|-|-|
> | Mean±Std of Top-K Accuracy                    | 0.391±0.001 | 0.556±0.001 | 0.602±0.001 | 0.636±0.001 | 0.641±0.001 |
>
> **Table 4:** The mean and standard deviation of yield prediction on the B-H, B-H-1, B-H-2, B-H-3, B-H-4 and S-M datasets. The evaluation metrics include MAE↓, RMSE↓, and $R^2$↑.
>
> | Models | Metircs | B-H | B-H-1 | B-H-2 | B-H-3 | B-H-4 | S-M |
> |-|-|-|-|-|-|-|-|
> |DRFP            | $R^2$   |   0.95±0.01  		|    0.81±0.01 			|    0.83±0.00  		|    0.71±0.01  		|    0.49±0.00  			|    0.85±0.01	 |
> |YieldBERT		|  $R^2$  | 0.95±0.01			|    **0.84±0.01**			|    0.84±0.03			|    0.75±0.04			|    0.49±0.05				|    0.82±0.01 |
> |YieldBERT-DA	| $R^2$   |   **0.97±0.01**			|    0.81±0.05			|    0.87±0.02	 		|    0.59±0.07			|    0.16±0.03			|    0.86±0.01	|
> |Egret           | $R^2$   |   0.94±0.01 			|    **0.84±0.01** 		|    **0.88±0.03**  		|    0.65±0.06  		|    0.54±0.06  			|    0.85±0.01 |
> |UA-GNN           | $R^2$   |   **0.97±0.01**  		|    0.74±0.04  		|    **0.88±0.03**  		|    0.72±0.02  		|    0.50±0.03  			| **0.89±0.01**	|
>  |D-MPNN             | $R^2$   | 0.94±0.01  | 0.80±0.03  | 0.82±0.03  | 0.73±0.02  | 0.55±0.03 | 0.85±0.01  |
> |Rxn Hypergraph             | $R^2$ | 0.96±0.01  | 0.81±0.01  | 0.83±0.02  | 0.71±0.02  | 0.56±0.02 | 0.85±0.01  |
> | ReaMVP| $R^2$   |  0.92±0.01 		|  0.76±0.02  | 0.83±0.05  	|   0.70±0.05	|    	0.53±0.08	|  0.85±0.01 |
> |Ours             | $R^2$ |**0.97±0.01**|0.80±0.01|**0.88±0.02**|**0.76±0.02**	|**0.68±0.06**|**0.89±0.01**|
>
> **Table 5:** Results for reaction yield prediction on Suzuki-Miyaura and USPTO-Yield dataset. The metrics includes mean and standard deviations from multiple trials with different random seeds.
> |  Dataset         | Gram    |Subgram    |
> |-|-|-|
> | Mean±Std of $R^2$   | 0.125±0.006 |0.211±0.003 |
>
> **Table 6:** Results for reaction classification on USPTO-TPL dataset. The metrics includes mean and standard deviations from multiple trials with different random seeds.
> | Metrics             | ACC   | CEN   | MCC   |
> |-|-|-|-|
> | Mean±Std        | 0.999±0.001 | 0.001±0.001 | 0.999±0.001 |
>
> **Table 7:** Results for reaction classification on Pistachio-Type dataset. The metrics includes mean and standard deviations from multiple trials with different random seeds.
> | Metrics             | ACC   | CEN   | MCC   |
> |-|-|-|-|
> | Mean±Std        | 0.986±0.002 | 0.026±0.004 | 0.985±0.003 |

---

> ### Author Response · Authors · 2024-11-25
> **Part Ⅰ-2: Standard Deviation and Confidence Interval**
>
> ## 2. Confidence Interval
> We use the normal distribution method to calculate the model's confidence interval. Specifically, $CI = x±z · s / \sqrt n$. We take a 95\% confidence level, where the corresponding $z = 1.96$. The result is shown below, with more details in Sec.G.6.4.
>
> **Summary:** According to the results in Tables 8-14, in larger datasets (USPTO, Pistachio), the 95% confidence interval for model performance is small ($≤0.01$). This result indicates that the model can achieve stable performance. However, in smaller datasets (Buchwald-Hartwig and Suzuki-Miyaura), some test sets have wider confidence intervals ($>0.01$). It suggests that the model's training variance cannot be ignored.
>
> **Table 8:** Confidence interval results ([Min,Max]) for leaving group identification on USPTO. The interval is calculated using normal distribution method. The data is from mean and standard deviations of multiple trials with different random seeds.
> | Type  |    Metrics         | Overall ACC         | Overall CEN         | Overall MCC         | LvG ACC             | LvG CEN             | LvG MCC             |
> |-|-|-|-|-|-|-|-|
> |Molecular Graph| Min         | 0.949 | 0.036 | 0.532 | 0.419 | 0.207 | 0.492 |
> || Max         | 0.951 | 0.038 | 0.544 | 0.427 | 0.211 | 0.502 |
> |Reaction Graph| Min         | **0.995** | **0.001** | **0.970** | **0.943** | **0.034** | **0.941** |
> || Max         | **0.997** | **0.003** | **0.972** | **0.945** | **0.036** | **0.943** |
>
> **Table 9:** Confidence interval results ([Min,Max]) for reaction condition prediction on USPTO-Condition dataset. The interval is calculated using normal distribution method. The data is from mean and standard deviations of multiple trials with different random seeds.
> | Metrics | Top-1       | Top-3       | Top-5       | Top-10      | Top-15      |
> |-|-|-|-|-|-|
> | Min         | 0.320 | 0.429 | 0.466 | 0.501 | 0.513 |
> | Max         | 0.324 | 0.435 | 0.472 | 0.507 | 0.519 |
>
> **Table 10:** Confidence interval results ([Min,Max]) for reaction condition prediction on Pistachio-Condition dataset. The interval is calculated using normal distribution method. The data is from mean and standard deviations of multiple trials with different random seeds.
> | Metrics | Top-1       | Top-3       | Top-5       | Top-10      | Top-15      |
> |-|-|-|-|-|-|
> | Min         | 0.390 | 0.555 | 0.601 | 0.635 | 0.640 |
> | Max         | 0.392 | 0.557 | 0.603 | 0.637 | 0.642 |
>
> **Table 11:** Confidence interval results ([Min,Max]) for reaction yield prediction on Buchwald-Hartwig dataset. The interval is calculated using normal distribution method. The data is from mean and standard deviations of multiple trials with different random seeds.
> | $R^2$ Metrics | BH    | BH1   | BH2   | BH3   | BH4   |
> |-|-|-|-|-|-|
> | Min         | 0.96  | 0.79  | 0.87  | 0.75  | 0.64  |
> | Max         | 0.98  | 0.81  | 0.89  | 0.77  | 0.72  |
>
> **Table 12:** Confidence interval results ([Min,Max]) for reaction yield prediction on Suzuki-Miyaura and USPTO-Yield dataset. The interval is calculated using normal distribution method. The data is from mean and standard deviations of multiple trials with different random seeds.
> | $R^2$ Metrics| SM    | Gram  | Subgram |
> |-|-|-|-|
> | Min         | 0.88  | 0.119 | 0.208 |
> | Max         | 0.90  | 0.131 | 0.214 |
>
> **Table 13:** Confidence interval results ([Min,Max]) for reaction classification on USPTO-TPL dataset. The interval is calculated using normal distribution method. The data is from mean and standard deviations of multiple trials with different random seeds.
> | Metrics             | ACC   | CEN   | MCC   |
> |-|-|-|-|
> | Min         | 0.998 | 0.000 | 0.998 |
> | Max         | 1.000 | 0.002 | 1.000 |
>
> **Table 14:** Confidence interval results ([Min,Max]) for reaction classification on Pistachio-Type dataset. The interval is calculated using normal distribution method. The data is from mean and standard deviations of multiple trials with different random seeds.
> | Metrics             | ACC   | CEN   | MCC   |
> |-|-|-|-|
> | Min         | 0.984 | 0.022 | 0.982 |
> | Max         | 0.988 | 0.030 | 0.988 |

---

> ### Author Response · Authors · 2024-11-25
> **Part Ⅱ: Exploring Torsion Angle as a 3D Representation**
>
> ## 1. Experiment
> We try to incorporate torison angles in various ways. However, the performance of model does not improve.
>
> Specifically, we attempt the following methods to incorporate torsion angles:
>
> + **Equivalent Neural Network.** Each edge not only has a length but also a direction represented by a vector. This method can convey torsion angle features through vector operations. Details can be found in Sec.C.2.2.
>
> + **Torsion Angular Edge.** We extend the angular edges to torsion angular edges. Specifically, for any four consecutive nodes ABCD in a molecular graph, we ensure that there are edges between every pair (if there are no edges, we add them). This constructs a shape-stable tetrahedron, thereby determining the dihedral angle (torsion angle) between the planes ABC and BCD. Details can be found in Sec.C.1.1.
>
> + **Atom Coordinate.** We also attempt to directly use atomic coordinates to express the entire conformation, including the torsion angle. Details can be found in Sec.G.5.1.
>
> We conduct experiments on the USPTO-Condition dataset. The results are shown in Table 15. Our current method, excluding torsion angles, outperforms the one that incorporates them. We provide further analysis in the next section.
>
>
>
> **Table 15:** Influence of torsion angle on model's performance in USPTO-Condition dataset. We use three different methods to incorporate torsion angle feature, including Equivalent Neural Network, Torsion Angular Edge and Atom Coordinate. Different from these methods, Reaction Graph do not include torsion angle.
> | **Method**                                        | **Top-1** ↑ | **Top-3** ↑ | **Top-5** ↑ | **Top-10** ↑ | **Top-15** ↑ |
> |-|-|-|-|-|-|
> | w/ Torsion Angle (by Equivalent Neural Network)        | 0.2899       | 0.4026       | 0.4390       | 0.4749        | 0.4879        |
> | w/ Torsion Angle (by Atom Coordinate)            | 0.3123       | 0.4243       | 0.4628       | 0.4987        | 0.5111        |
> | w/ Torsion Angle (by Torsion Angular Edge)   | 0.3022       | 0.4087       | 0.4467       | 0.4821        | 0.4935        |
> | w/o Torsion Angle (ours)       | **0.3246**       | **0.4343**       | **0.4715**       | **0.5061**        | **0.5181**        |
>
>
>
> ## 2. Analysis
> From Table 15, we find that adding torsion angles does not lead to performance improvement. We attribute this to the task characteristics of reaction property prediction.
>
> Specifically, there is a chemical reaction $A + B \xrightarrow{\text{D}} C$. The catalyst $D$ won't change with the conformations of $A$, $B$, or $C$. Therefore, the properties of reactions are invariant to different conformations of the same molecule. However, torsion angles can vary significantly across different conformations. This makes it difficult for the model to learn the relationship between the **varying torsion angles** and the **invariant reaction properties**.
>
>
> In contrast, the bond length and bond angle (implemented by angular edge) in Reaction Graph change only slightly between different conformations. This make Reaction Graph easier for model to learn, and more suitable for representing the property of chemical reaction.
>
>
> ## 3. Future Work
> Based on the results, we find that the torsion angle currently does not demonstrate advanced performance in reaction tasks. However, we recognize that the torsion angle can provide additional structural priors. Therefore, effectively utilizing the torsion angle for reaction property prediction will be the focus of our future research.

---

> ### Author Response · Authors · 2024-11-25
> **Part Ⅲ: More Details on Data Preparation Tools**
>
> Thank you for your comment.
> We summarize data preprocessing tools, detailing their functions, precision, efficiency, application scope, and limitations (as shown in Table 16).
>
> **Table 16:** A summary of the tools used in our methods.
> | **Tools**     | **Function**                             | **Precision**        | **Efficiency** | **Application Scope**       | **Limitation**                |
> |---------------|------------------------------------------|----------------------|----------------|-----------------------------|-------------------------------|
> | ETKDG         | Calculate/Optimize 3D Conformation      | Low                  | High           | Small Organic Molecules     | Fail on Some Metal-Complex    |
> | UFF           | Calculate/Optimize 3D Conformation      | Relatively Low       | High           | Universal                   | Low Accuracy                  |
> | MMFF          | Calculate/Optimize 3D Conformation      | Medium               | High           | Small Organic Molecules     | Fail on Some Metal-Complex    |
> | DFT           | Calculate/Optimize 3D Conformation      | High                 | Low            | Depend on Basis Set         | Slow                          |
> | RXNMapper     | Predict Atom Mapping                     | Relatively High      | High           | Small Molecules             | Lack of Rule Constraints       |
> | NameRXN       | Predict Atom Mapping + Classify Reaction | High                 | High           | Limited by Manual Rule      | Manual Rule Based             |
>
>
> ## 1. Calculate/Optimize 3D conformation
> ETKDG, UFF, MMFF, and DFT are tools for calculating or optimizing 3D conformations. ETKDG and MMFF are accurate for small organic molecules but less effective for larger ones and metal complexes. UFF is more suitable for handling metal complexes. DFT, based on quantum chemistry, provides high precision for various compounds but is inefficient.
>  In this paper, we use large real-world chemical databases, USPTO and Pistachio, with total millions of entries and complex molecular distributions.
> We find that even with the simplest basis sets, DFT is too time-consuming.
>
>  Hence, we initialize conformations using ETKDG and then optimize them with MMFF94. For molecules that MMFF94 cannot handle (e.g., involving heavy metals), we use UFF for conformation optimization.
>
>
> ## 2. Predict Atom Mapping
> RXNMapper and NameRXN are tools for atomic mapping.
> RXNMapper is open-source and based on an unsupervised language model, capable of providing efficient and accurate predictions.
> NameRXN is a commercial rule-based tool, offering highly reliable results, though some reactions fall outside its rule coverage.
>
> After evaluating the mapping performance on the USPTO dataset, we choose RXNMapper because it demonstrates high stability and most of its predictions are accurate.
>
> ## 3. Classify Reaction
> NameRXN is also a tool for reaction classification. In this paper, the labels of the Pistachio dataset are annotated using NameRXN.

---

> ### Author Response · Authors · 2024-11-25
> **Part Ⅳ: Task Descriptions and Characteristics**
>
> Thank you for your suggestion.
> In Sec.B.3 - Sec.B.5, we detail the general characteristics of the three types of chemical reaction tasks.
> A brief summary is provided below.
>
> ## 1. Reaction Condition Prediction
> For the reaction condition prediction task, given the representation of a reaction, the model is designed to predict the necessary conditions for the reaction to proceed, including catalysts, solvents, temperature, pressure, and other relevant factors. Reaction conditions are pivotal for chemical synthesis and drug design. Suitable reaction conditions can significantly enhance the progress of reactions and increase yields, whereas unsuitable conditions can impede reactions and result in the wastage of raw materials.
>
> The challenge of reaction condition prediction mainly lies in the **sparsity** of labeled data.
>
> ## 2. Reaction Yield Prediction
> Reaction yield typically indicates the percentage of reactant molecules converted into the desired product.
> The objective of a reaction condition prediction model is to estimate the yield of a chemical reaction, ranging from 0-100\%, based on the provided reaction representation.
> The significance of this task lies in its ability to provide yield estimates for reactions where yield data is not available in existing databases. Accurate yield predictions are essential for selecting high-quality synthetic routes that minimize material waste and optimize overall efficiency.
>
> The challenge of reaction yield prediction primarily lies in the **non-smooth surface of the structure-yield relationship**, as well as issues with the **low quality of dataset labeling**.
>
> ## 3. Reaction Type Classification
> Reaction type classification refers to the task of categorizing a given chemical reaction into predefined classes based on the nature of the reaction.
> The input is typically a representation of reactions, and the output is the predicted reaction type.
> This task is important for identifying and comparing unknown reactions with those in databases, helping to understand reaction characteristics. It also assesses the model's understanding of reaction mechanisms, enhancing performance analysis and interpretability.
>
> The data quality for reaction classification tasks is high. However, the current annotations are derived from the rule-based NameRXN tool. As a result, the performance of model **struggles to generalize** to reaction categories that NameRXN cannot handle, referred to as "Unknown".

---

> ### Author Response · Authors · 2024-11-25
> **Part Ⅴ: Task Results Analysis**
>
> Thank you for your interest and insights on the tasks in our work. We conduct a detailed analysis of the results on these tasks in Sec.2, Sec.A and Sec.F.
> Here is a brief summary in response to your question.
>
> + **Is The Task of USPTO-TPL is Simple?** USPTO-TPL focuses on reaction templates, while Pistachio-Type targets reaction types. A template corresponds to a single reaction type, whose characteristic is obvious. In contrast, a reaction type can include many templates with diverse characteristics. Therefore, the classification on USPTO-TPL is relatively simple, while Pistachio-Type is harder. According to the experimental results, Reaction Graph has advantages for **both simple and complex tasks**.
>
> + **Complexity of Yield Prediction Tasks.** As you note, the challenge of reaction yield prediction primarily lies in the non-smooth surface of the structure-yield relationship, as well as issues with the low quality of dataset labeling. To overcome this, we need effective **priors beyond molecular structures**. The reaction graph approach explicitly models **atomic-level changes** between reactants and products, linking reaction changes to yields, which improves classification performance on challenging datasets.
>
> + **Label Canonicalization in Condition Prediction.** The USPTO-Condition dataset [1] uses **canonical SMILES** as reaction condition labels. Our Pistachio-Condition does the same. This improves label consistency, making it easier for the model to learn.
>
> [1] Xiaorui Wang, Chang-Yu Hsieh*, Xiaodan Yin, Jike Wang, Yuquan Li, Yafeng Deng, Dejun Jiang, Zhenxing Wu, Hongyan Du, Hongming Chen, Yun Li, Huanxiang Liu, Yuwei Wang, Pei Luo, Tingjun Hou*, Xiaojun Yao*. Generic Interpretable Reaction Condition Predictions with Open Reaction Condition Datasets and Unsupervised Learning of Reaction Center. Research 2023;6:Article 0231. DOI:10.34133/research.0231

---

> ### Author Response · Authors · 2024-11-25
> **Part Ⅵ: Utilization of LSTM Hidden State**
>
> In our implementation, we use the hidden state of the LSTM to obtain the final reaction representation vector.
>
> Recall that after the message passing stage of the GNN, the node's feature is denoted as $\boldsymbol{v}'$.
>
> As detailed in Sec.C.2.1, in our implementation, the process of aggregation is as follows:
> $$
> \alpha_i^t = \frac{\mathrm{exp}(\boldsymbol{v}'\_i \cdot \boldsymbol{h}^{t})}{\sum_{j=1}^{N}\mathrm{exp}(\boldsymbol{v}'\_j \cdot \boldsymbol{h}^{t})},~~~~~~\boldsymbol{q}\^{t+1} = \sum_{i=1}\^{N} \alpha\_i^t \times \boldsymbol{v}'\_i,~~~~~~\boldsymbol{h}^{t+1}, \boldsymbol{c}^{t+1} = \text{LSTM}([\boldsymbol{q}^{t+1}; \boldsymbol{h}^t];~\boldsymbol{h}^t, \boldsymbol{c}^t),
> $$
>
> $$
> \boldsymbol{r} = \boldsymbol{W}\cdot[\boldsymbol{q}^{T}; \boldsymbol{h}^{T-1}] + \boldsymbol{b},
> $$
>
> where $\alpha\_i^t$ represents the attention weight of atom $i$ at the $t$-th iteration, $\boldsymbol{h}$ and $\boldsymbol{c}$ are the hidden state and cell state of LSTM, both initially set to $\boldsymbol{0}$. $\boldsymbol{W}$ and $\boldsymbol{b}$ are learnable parameters of a fully connected layer. After $T$ iterations, we **concatenate** LSTM's hidden state $\boldsymbol{h}^{T-1}$ with the aggregated node features $\boldsymbol{q}^{T}$, and acquire the final reaction representation $\boldsymbol{r}$ by linear projection.
>
> This design helps the model capture the overall changes in the reaction while incorporating the local structural features of each atom.

---

> ### Author Response · Authors · 2024-11-26
> **Part Ⅶ: F1 Score for Leaving Group Identification and Reaction Classification**
>
> ## 1. Explanation of Metrics
> Thank you for your suggestion.
> In this revision, we add definitions of the metrics used, including CEN and MCC, in Sec.G.3.3.
>
>
>
> ## 2. F1 Score for Leaving Group Identification and Reaction Classification
>
> Although MCC considers class imbalance, the F1 Score is a more common metric. Therefore, we add the testing with F1 Score in Sec.4 of the main text. Here's a brief summary.
>
> ### (1) Leaving groups (LvG) identification.
>
> Leaving groups (LvG) identification is a multi-classification problem, as it requires not only locating the position of the leaving group but also identifying its type. Overall performance shows the model's ability of locating the leaving group, while LvG-specific reflects the performance of classification.
>
> The result are shown in Tables 17-18.
>
> **Table 17:** The overall performance of Molecular Graph and Reaction Graph on leaving group identification. Evaluation metrics include accuracy (ACC), confusion entropy (CEN), Matthews Correlation Coefficient (MCC), F1 Macro, and F1 Micro.
> | **Representation**    | **ACC↑** | **CEN↓** | **MCC↑** | **F1 Macro↑** | **F1 Micro↑** |
> |-|-|-|-|-|-|
> | Molecular Graph       | 0.950    | 0.036    | 0.549    | 0.365    | 0.950    |
> | Reaction Graph (ours) | **0.997**| **0.002**| **0.973**| **0.904**| **0.997**|
>
> **Table 18:** The LvG-specific performance of Molecular Graph and Reaction Graph on leaving group identification. Evaluation metrics include accuracy (ACC), confusion entropy (CEN), Matthews Correlation Coefficient (MCC), F1 Macro, and F1 Micro.
> | **Representation**    | **ACC↑** | **CEN↓** | **MCC↑** | **F1 Macro↑** | **F1 Micro↑** |
> |-|-|-|-|-|-|
> | Molecular Graph       | 0.448    | 0.201    | 0.519    | 0.404    | 0.448    |
> | Reaction Graph (ours) | **0.947**| **0.031**| **0.945**| **0.903**| **0.947**|
>
>
> ### (2) Reaction classification.
> We also analyze the results of the reaction classification.
>
> For the existing baseline results on USPTO-TPL, we attempt to find the results of F1 Score in the official open-source repository. Among them, RXNFP provides a complete confusion matrix for calculating the F1 Score, and we successfully reproduce the results of DRFP, providing the corresponding F1 Score. Unfortunately, for T5Chem, we cannot find the checkpoint for USPTO-TPL, so we leave it blank.
>
> The results are shown in Table 19.
>
> **Table 19:** The reaction classification performance of different methods on USPTO-TPL and Pistachio-Type. Evaluation metrics include accuracy (ACC), confusion entropy (CEN), Matthews Correlation Coefficient (MCC), F1 Macro, and F1 Micro.
> | **Method**    | **USPTO-TPL**      |     |      |        |    | **Pistachio-Type**        |        |    |        |      |
> |-|-|-|-|-|-|-|-|-|-|-|
> |         | **ACC** ↑       | **CEN** ↓    | **MCC** ↑     | **F1 Macro** ↑         | **F1 Micro** ↑     | **ACC** ↑       | **CEN** ↓       | **MCC** ↑    | **F1 Macro** ↑                  | **F1 Micro** ↑      |
> | **DRFP**      | 0.977*       | 0.011*      | 0.977*     | 0.972      | 0.977   | 0.899   | 0.149   | 0.890      | 0.898         | 0.899     |
> | **RXNFP**        | 0.989*       | 0.006*                     | 0.989*                     | 0.986                      | 0.989                      | 0.948                      | 0.078                 | 0.944        | 0.946                      | 0.948                      |
> | **T5Chem**                | 0.995*                      | 0.003*                     | 0.995*                     | -                          | -                          | 0.976                      | 0.041                      | 0.974                      | 0.976                      | 0.976                      |
> | **D-MPNN**                | 0.997                       | 0.001                      | 0.997                      | 0.996                      | 0.997                      | 0.982                      | 0.033                      | 0.980                      | 0.982                      | 0.982                      |
> | **Rxn Hypergraph**       | 0.954                       | 0.024                      | 0.953                      | 0.935                      | 0.954                      | 0.911                      | 0.129                      | 0.903                      | 0.910                      | 0.911                      |
> | **Reaction Graph (ours)** | **0.999**                   | **0.001**                  | **0.999**                  | **0.998**                  | **0.999**                  | **0.987**                  | **0.024**                  | **0.986**                  | **0.987**                  | **0.987**                  |
>
> It is observed that the F1 Macro and F1 Micro scores are close on Pistachio-Type. This is due to the uniform data scale of each reaction type in Pistachio-Type. For the F1 Score, Reaction Graph also demonstrates advantages compared to other methods.

---

> ### Author Response · Authors · 2024-11-26
> **Part Ⅷ: Why Choose to Compute Angular Edges Instead of Using Bond Angle Information?**
>
> We detail the reason for using angular edge instead of bond angle in Sec.4.2. Here is a brief summary.
>
> ## 1. Computational Efficiency
> Explicitly introducing bond angles requires **two rounds** of message passing during each single GNN iteration. Specifically, the bond angle information is first aggregated to the edges, and the edge information is then aggregated to the nodes.
>
> In contrast, angular edges only require **one round** of message passing.
> Angular edges perform message passing together with other edges (bond edges and reaction edges), while also conveying bond angle information.
>
> Our experimental results are shown in Table 20. Detailed analysis of the computation time is provided in Sec.E.
>
> **Table 20:** The computational time of the model using Bond Angle and Angular Edge with varying number of graph nodes. The time unit in the table is milliseconds (ms↓).
> | Method        | 2500  | 3000  | 3500  | 4000  | 4500  |
> |---------------|-------|-------|-------|-------|-------|
> | Bond Angle    | 54.18 | 64.22 | 74.28 | 84.90 | 96.06 |
> | Angular Edge  | **37.28** | **43.77** | **50.57** | **58.03** | **65.45** |
>
>
>
> ## 2. Modal Consistency
> The length and angle modalities are inconsistent and **require alignment**. This inconsistency complicates model design and learning. In contrast, using angular edges directly conveys angle information through length. This method **relieves the need for modal alignment**, thereby enhancing the model's understanding of molecular 3D structures.
>
> We conduct experiment on USPTO-Condition. The results in Table 21 show that angular edges achieve better performance, as detailed in Sec.4.2. Implementation details can be found in Sec.C.2.2.
>
> **Table 21:** Comparison between the performance of Bond Angle and Angular Edge on reaction condition prediction. Using USPTO-Condition Dataset.
> | **Method**                                      | **Top-1** ↑ | **Top-3** ↑ | **Top-5** ↑ | **Top-10** ↑ | **Top-15** ↑ |
> |-------------------------------------------------|--------------|--------------|--------------|---------------|---------------|
> | Bond Edge Length + Bond Angle                   | 0.3179       | 0.4290       | 0.4656       | 0.5018        | 0.5146        |
> | Bond Edge Length + Angular Edge Length          | **0.3246**       | **0.4343**       | **0.4715**       | **0.5061**        | **0.5181**        |

---

> ### Author Response · Authors · 2024-11-26
> **Part Ⅸ: Paper Writing Revision**
>
> We appreciate your careful reading and kind feedback. Based on your suggestions, we have made the writing more concise in this revision.
>
> Regarding your question,
> $T_1$ refers to the number of iterations of the GNN. GNN iteratively performs graph convolution on the Reaction Graph to increase the receptive field and capture the neighbor structural information of the atoms. We have added the appropriate explanation in the main text.
>
> Thank you once again for your valuable insights. We hope our explanation can address your concerns.

---

> ### Comment · Reviewer_xu23 · 2024-11-26
>
> Thanks for the detailed clarifications, I appreciate it! I was confused because it says "After $T_2$ iterations, $q^{T_2}$ is used as the reaction global representation" in line 239 in the original pdf. Maybe adding the expression regarding $r = ...$ for clarity in the main text?
>
> Regarding reaction representation $r$, if you make a PCA or UMAP plot of the embedded reactions for the reactions in section 4.5, will the reaction distribution show any meaningful pattern? Please note that this is just an optional question if you are curious.

---

> > ### Author Response · Authors · 2024-11-26
> > **Part Ⅹ: Formula Revision and Reaction Embedding Plotting**
> >
> > We appreciate your prompt response and constructive advice.
> >
> > ## Formula Revision
> >
> > Thank you for your suggestion.
> > We have clarified the expression in the main text.
> > The revision is as follows.
> > After the $T\_2$ iteration, the model outputs the reaction global representation vector $\boldsymbol{r}$, where $\boldsymbol{r} = \boldsymbol{q}^{T\_2}$.
> >
> > ## Reaction Embedding Plotting
> > Thank you for your insightful comments.
> >
> > We perform dimensionality reduction visualization on the reaction representation vector $\boldsymbol{r}$. This experiment aim to observe the distribution of chemical reactions, as well as exploring the relationship between $\boldsymbol{r}$ and chemical properties.
> >
> > We use three methods for dimensionality reduction: TMAP[1], UMAP[2], and t-SNE[3]. The visualization results are in **Sec.G.3.3 (Figure 23)**.
> >
> > The results indicate a strong correlation between $\boldsymbol{r}$ and reaction types. And there's also certain correlation between $\boldsymbol{r}$ and number of hydrogen donors. This indicates that the reaction representation vector $\boldsymbol{r}$ potentially contains high-dimensional features of chemical properties.
> >
> > [1] https://tmap.gdb.tools/
> >
> > [2] https://umap-learn.readthedocs.io/en/latest/
> >
> > [3] https://github.com/scikit-learn/scikit-learn
> >
> >
> > Thank you again for your contributions to improving this paper.
> > If you feel our efforts address your concerns, we would appreciate it if you could kindly consider raising your rating. Your support is significantly important to our work. If you need further information, please let us know. We look forward to hearing from you.

---

> > > ### Comment · Reviewer_xu23 · 2024-11-27
> > >
> > > Thanks for the updates! That looks great!
> > >
> > > In the most recent response you mentioned that the reaction global representation vector $r = q^{T_2}$. But in an earlier response you mentioned that $r = W[q^T; h^{T-1}] + b$. I'm a little confused here, could you clarify? Thanks!

---

> > > > ### Author Response · Authors · 2024-11-27
> > > > **Part ⅩⅠ: Formula Clarification**
> > > >
> > > > Thank you for your prompt feedback and valuable suggestions.
> > > >
> > > > In the original text, we used the simplified expression $ \boldsymbol{r} = \boldsymbol{q}^{T_2} $ for ease of understanding. We omitted the detailed operations of concatenating with the LSTM hidden layer and the linear projection, and left them in the appendix.
> > > >
> > > > However, based on your suggestions, we realized that the LSTM's hidden state also plays a crucial role in aggregation. Additionally, such omissions may lead to difficulties in understanding. Therefore, in the latest version of the revision, we have revised the notation to the complete form $ \boldsymbol{r} = \boldsymbol{W} \cdot [\boldsymbol{q}^T; \boldsymbol{h}^{T-1}] + \boldsymbol{b} $.
> > > >
> > > > The revision is as follows:
> > > > After the $T_2$ iteration, the model outputs the reaction global representation vector $\boldsymbol{r}$, where $\boldsymbol{r} = \boldsymbol{W}_r \cdot [\boldsymbol{q}^{T_2};\boldsymbol{h}^{T_2-1}] + \boldsymbol{b}_r$, with $\boldsymbol{W}_r$ and $\boldsymbol{b}_r$ being the learnable weight matrix and bias, respectively.
> > > >
> > > > Thank you once again for your valuable contribution. We hope this explanation addresses your concerns, and we are looking forward to hearing from you.

---

### Official Review · Reviewer_gd4P · 2024-11-04

**Soundness:** 3
**Presentation:** 2
**Contribution:** 3
**Rating:** 6
**Confidence:** 4

**Summary:**

This paper proposes a method for chemical reaction modeling that, in addition to the molecular graphs includes both a map of correspondences of atoms across the reaction (reaction edges) and the distance matrices of the 3D structures of the reactants and products.  This improved representation helps enhance the ability of the model to accurately handle chemical reactions compared to a baseline method that only uses the molecular graphs.

**Strengths:**

This paper introduces the 3D geometric information into reaction modeling and demonstrates a modest improvement compared to using only the molecular graphs.  Importantly, this improvement is orthogonal to the addition of the correspondence between atoms in reactants and products, which substantially improves reaction modeling compared to using only the graphs.  The paper uses several datasets and demonstrates improved results across a number of metrics and tasks compared to prior efforts.

**Weaknesses:**

Some of the language could benefit from greater clarity.  In the section about the implementation details, could the authors clarify what they mean with the iterative output technique to support beam search?  In page 6, line 295, the experiment results support their hypothesis but do not fully validate it, as Fig 3 only has a single example molecule.  Regarding the construction of Pistachio-Condition, how are the validation and testing splits selected?


Although the addition of 3D features appears to consistently help, the effect is somewhat unclear.  For example, why is the lack of both reaction edge and 3D structure resulting in significantly higher accuracy in Table 4 than past method in Table 3?  In particular, could the authors clarify if and how the total parameter count changes when the 3D information was excluded and whether the hyperparameters were reoptimized for that network?  In addition to ablating and optimal model with 3D, demonstrating that there is room for improvement by simply adding 3D information to a previously optimal model would strengthen the message of this work.  I wonder if the current improvement in accuracy with the addition of the 3D representation in part stems from the reduction in the total parameter count.

**Questions:**

Have the authors considered mixing USPTO and Pistachio (or any of the other datasets) for any of their experiments? I wonder if such mixing might yield valuable insights and lead to better model generalization.  Complementary to this point, have the authors considered separating Pistachio into folds of incremental difficulty (for example by atom count, scaffold size, or other metrics of synthetic difficulty) in order to perform a more detailed analysis of the additional contribution of 3D structure?  It would be interesting to investigate if there are classes of scaffolds where 3D helps and others where it is neutral or detrimental on average.

Have the authors considered alternative modern ways to incorporate 3D information, such as equivariant neural networks?

---

> ### Author Response · Authors · 2024-11-25
> **Part Ⅰ: More Details of Method and Dataset**
>
> Thank you for acknowledging our "**improved representation**" can "**accurately handle chemical reactions**" and "**substantially improve reaction modeling compared to using only the graphs**". We also appreciate your constructive suggestions.
>
> ## 1. Explanation for the Iterative Output Technique to Support Beam Search
>
> Thank you for your comment. We revise the language to clarify the expression of certain concepts.
> We give further explanation for "the Iterative Output Technique to Support Beam Search" in Sec.s C and D. The following is a brief summary.
>
> Reaction condition prediction is a multi-class task with solutions typically divided into one-step and iterative output techniques.
> **One-step output technique** directly predicts all reaction conditions at once.
> **Iterative output technique** autoregressively predicts the next reaction condition, using the reaction representation and previously predicted conditions as inputs.
> The theoretical basis for iterative output is as follows.
> $$
> P(C_1,C_2,\dots,C_n|R) = P(C_1|R) \times P(C_2|R,C_1) \times \dots \times P(C_n|R,C_1,C_2,\dots,C_{n-1}),
> $$
> where $P$ is conditional probability, $R$ is the reaction representation, $C_1$, $C_2$, ..., $C_n$ are reaction conditions.
> The equation shows that the one-step output technique can be converted to an iterative one.
> The iterative output technique can support beam search for the $top$-$k$ reaction conditions by selecting the top $k$ conditions at each intermediate output step and using them as inputs for the next step.
>
> To compare one-step and iterative techniques, we evaluate the $top$-$k$ accuracy for the reaction condition prediction task on USPTO.
> For a clear comparison, we use vanilla molecular graphs as representations and keep irrelevant factors consistent. The experimental results are as follows (Table 1).
> Since the iterative output technique considers the dependencies between reaction conditions, it reduces the model's learning complexity and improves accuracy.
>
> **Table 1:** The $top$-$k$ accuracy for the reaction condition prediction task on USPTO using one-step and iterative output techniques.
>
> | **Method**                 | **Top-1**↑ | **Top-3**↑ | **Top-5**↑ | **Top-10**↑ | **Top-15**↑ |
> | -------------------------- | --------------------- | --------------------- | --------------------- | ---------------------- | ---------------------- |
> | One-step Output Technique  | 0.249                 | 0.305                 | 0.318                 | 0.387                  | 0.422                  |
> | Iterative Output Technique | **0.305**             | **0.417**             | **0.454**             | **0.492**              | **0.506**              |
>
> ## 2. More Details of the Dataset and Split
>
> Thank you for your comment.
> Due to page limitations, we detail dataset information in the initial supplementary material (moved to Sec. F in this revision), including data split, data distribution, data pre-processing, etc.
> As for Pistachio-Condition construction, after data cleaning, we retain $562,471$ samples, which are split into $8:1:1$ for training, validation, and testing.

---

> ### Author Response · Authors · 2024-11-25
> **Part Ⅱ: More Results of Attention Weights**
>
> ## 1. More Visualization Results of Attention Weights
>
> We provide a group of attention-weight visualizations in the main text, which show that the Reaction Graph (RG) helps the model focus on the reaction center.
> We understand your point about needing more results to fully validate the method's effectiveness.
> We give more attention weight visualization results in initial supplementary materials (moved to Sec. G 1.1 in this revision).
>
> ## 2. More Quantitative Results of Attention Weights
> For further validation, we conduct additional quantitative experiments.
> Specifically, we develop an automated algorithm to extract **multiple reaction groups** from the USPTO dataset (detailed in Sec. D.3).
> A reaction group consists of two chemical reactions with reaction centers at different positions on the same molecule.
> We extract $1000$ reaction groups, including $1000$ different molecules and $2000$ chemical reactions.
> We run the reaction condition prediction model on the above reaction groups and extract the attention weights of the reacting atoms and non-reacting atoms at the Set2Set layer (detailed in Sec. D.3).
>
> We use two metrics, **Proportion** and **Ratio** $a/b$, to reflect the model's focus on the reaction center atoms.
> The former is the proportion of the reacting atom's attention weight relative to all attention weights.
> The latter is the ratio of average attention weights $a$ for reacting atoms to $b$ for non-reacting atoms.
>
> The results in Table 2 show that, by incorporating reaction information, our method effectively focuses on reaction center atoms, improving the model's understanding of the reaction mechanism.
>
> **Table 2:** Quantitative statistical results for the attention weights of reacting and non-reacting atoms.
>
> | **Method**      | **Proportion**$\uparrow$ | **Ratio** $a/b\uparrow$ |
> | --------------- | ------------------------ | ----------------------- |
> | Molecular Graph | 0.4326                   | 1.4542                  |
> | Reaction Graph (ours)  | **0.5079**               | **2.0390**              |

---

> ### Author Response · Authors · 2024-11-25
> **Part Ⅲ: Discussion on the effectiveness of adding 3D information**
>
> ## 1. Explaining Enhanced Accuracy Without Reaction Edges and 3D Structures
>
> Thank you for your question.
> In this paper, the main contribution is the introduction of a novel chemical reaction representation, the Reaction Graph (RG), which integrates reaction edges and 3D structure information.
> As a minor contribution, we also design a network structure and training strategy suited for RG.
>
> You mentioned that RG without reaction edges and 3D information still outperforms past methods. This success is due to the minor contribution. Specifically, we implement an **iterative output technique** in output module and a **two-stage training strategy**.
> The iterative output technology (detailed in PartⅠ) considers the dependencies among reaction conditions, reducing the model's learning complexity and increasing accuracy.
> The two-stage training strategy first stabilizes the core network parameters and then enhances the output module for specific tasks.
>
> ## 2. Parameter Count and Hyperparameter Setting When Introducing 3D Features
>
> ### (1) Parameter Count
>
> In the 3D information ablation study, the network architecture and parameter count remain exactly the same.
> It demonstrates that the model's improved performance results from the integration of 3D information, not from an increase in parameters.
> Specifically, to integrate 3D information into the model, we use the Radial Basis Function (RBF) kernel to embed edge lengths.
> During the ablation of 3D information, **we set all edge lengths to $0$** (Setting 2 in Table 3) rather than removing them (Setting 3 in Table 3), thus the neural network's parameter count does not change.
>
> **Table 3:** Comparison of model parameters with and without 3D
>
> | **Setting** | **Representation**                                      | **Parameters** |
> | ----------- | ------------------------------------------------------- | -------------- |
> | 1           | w/ 3D Information (ours)                                      | 189229067      |
> | 2           | w/o 3D Information by setting edge length to $0$ (ours)  | 189229067      |
> | 3           | w/o 3D Information by removing edge length              | 188589035      |
>
>
> ### (2) Hyperparameter Setting
>
> In the 3D information ablation study, the hyperparameter settings remain completely consistent, with no re-optimization performed.
> In this way, we have eliminated the influence of other factors as much as possible, proving that our fusion of 3D information is effective.
>
> ## 3. Adding 3D Information Enhances Previously Optimal Methods
>
> Thank you for suggesting adding 3D information to previously optimal methods to strengthen this work.
> CRM and Parrot achieve optimal results but are unsuited for handling 3D information, due to their 1D neural network implementation.
> Therefore, we incorporate 3D information into the optimal graph neural network-based methods, D-MPNN and Rxn Hypergraph.
>
> Detailed results of reaction condition prediction on USPTO-Condition are shown in Table 4.
> Adding 3D information improves the $Top$-$K$ accuracy of D-MPNN and Rxn Hypergraph, demonstrating the effectiveness of 3D structural priors.
>
> **Table 4:** The results of integrating 3D information into the previously optimal methods.
>
> | **Method**     | **3D Info.** | **Top-$1$**$\uparrow$ | **Top-$3$**$\uparrow$ | **Top-$5$**$\uparrow$ | **Top-$10$**$\uparrow$ | **Top-$15$**$\uparrow$ |
> | -------------- | ------------ | --------------------- | --------------------- | --------------------- | ---------------------- | ---------------------- |
> | D-MPNN         | w/o 3D       | 0.1977                | 0.3000                | 0.3341                | 0.3780                 | 0.3924                 |
> | D-MPNN         | w 3D         | **0.2030**            | **0.3059**            | **0.3410**            | **0.3830**             | **0.3971**             |
> | Rxn Hypergraph | w/o 3D       | 0.2127                | 0.3084                | 0.3447                | 0.3808                 | 0.3927                 |
> | Rxn Hypergraph | w/ 3D        | **0.2149**            | **0.3113**            | **0.3464**            | **0.3825**             | **0.3949**             |

---

> ### Author Response · Authors · 2024-11-25
> **Part Ⅳ: Discussion of Dataset Mixing and Splitting**
>
> ## 1. Impact of Mixing USPTO and Pistachio on a Single Dataset
>
> In this paper, we focus on proposing a novel chemical reaction representation, which can enhance model performance within a fixed data volume.
> Exploring the effect of training with large-scale data is another interesting research topic, which is not the focus of this paper.
> Nevertheless, we conduct additional experiments to explore whether combining the USPTO and Pistachio datasets can provide valuable insights.
>
> In the implementation, since the reaction category labels between the two datasets are not completely consistent, we select reactions from Pistachio that match the categories in USPTO.
> We perform joint training using the merged dataset.
>
> The results on the USPTO test set are as follows (Table 5).
> We find that mixing the two datasets for training does not improve the model's performance on a single dataset.
> This may result from the significant differences in data distribution between the two datasets.
> Although USPTO and Pistachio are large in scale, their sparse annotations still cannot cover the diverse chemical space.
>
> **Table 5:** Reaction classification results before and after mixing USPTO with Pistachio.
>
> | **Method**        | **Top-1**$\uparrow$ | **Top-3**$\uparrow$ | **Top-5**$\uparrow$ | **Top-10**$\uparrow$ | **Top-15**$\uparrow$ |
> | ----------------- | ------------------- | ------------------- | ------------------- | -------------------- | -------------------- |
> | **w/o Pistachio** | **0.325**           | 0.434               | **0.472**           | **0.506**            | **0.518**            |
> | **w Pistachio**   | 0.323               | **0.436**           | 0.470               | 0.500                | 0.511                |
>
> ## 2. Impact of Mixing USPTO and Pistachio on Model Generalization
>
> We further evaluate the effect of mixed data training on model generalization.
> Specifically, we extract a subset of data from Pistachio to serve as the test set. This subset is out-of-distribution (OOD), which can reflect the model's generalization performance.
> The results are as follows (Table 6).
> Training with mixed data significantly improved performance on the OOD test set, indicating a benefit for generalization.
>
> **Table 6:** Comparison of model generalization before and after mixing USPTO with Pistachio.
>
> | **Method**        | **Top-1**$\uparrow$ | **Top-3**$\uparrow$ | **Top-5**$\uparrow$ | **Top-10**$\uparrow$ | **Top-15**$\uparrow$ |
> | ----------------- | ------------------- | ------------------- | ------------------- | -------------------- | -------------------- |
> | **w/o Pistachio** | 0.177               | 0.235               | 0.260               | 0.284                | 0.292                |
> | **w Pistachio**   | **0.275**           | **0.455**           | **0.517**           | **0.567**            | **0.578**            |
>
>
> ## 3. Separate Pistachio into Folds of Incremental Difficulty by Scaffold Size
>
> Based on different scaffold sizes, we design $6$ progressively challenging experiments on Pistachio.
> The test set is consistent across all groups.
> From the first to the sixth group, the training set size halves each time, reducing from full size to $1/32$ of the original.
> For each group, we compare the results with and without 3D information.
> As shown in Table 7, adding 3D information is helpful in the first five groups.
> The performance in the sixth group decreased.
> That is because it is difficult to effectively learn 3D information with limited data.
>
> **Table 7:** The influence of adding 3D information at different scaffold sizes.
>
> | **Setting** | **Scaffold Ratio** | **w 3D**   | **w/o 3D** |
> | ----------- | ------------------ | ---------- | ---------- |
> | 1           | 1                  | **0.3915** | 0.3852     |
> | 2           | 1/2                | **0.3550** | 0.3503     |
> | 3           | 1/4                | **0.3146** | 0.3116     |
> | 4           | 1/8                | **0.2768** | 0.2716     |
> | 5           | 1/16               | **0.2244** | 0.2210     |
> | 6           | 1/32               | 0.1857     | **0.1907** |

---

> ### Author Response · Authors · 2024-11-25
> **Part V: Alternative Modern Ways to Incorporate 3D Information**
>
> ## Compared with equivariant neural networks for incorporating 3D information
>
> Invariant neural networks are effective for extracting conformation-invariant representations, while equivariant neural networks are suited for capturing related representations. Since chemical reaction representations are conformation-invariant, our method, which utilizes invariant neural networks, is well-suited for this purpose.
>
> To validate our approach, we replace our invariant operation with an equivariant module (PaiNN [1]). Specifically, we add the bond direction information into Reaction Graph, making our network equivariant.
> As shown in Table 7, invariant neural network is better than equivariant neural network for chemical reaction modeling.
>
> **Table 7:** Comparison results of equivariant and invariant neural networks.
>
> | **Method**                      | **Top-$1$**$\uparrow$ | **Top-$3$**$\uparrow$ | **Top-$5$**$\uparrow$ | **Top-$10$**$\uparrow$ | **Top-$15$**$\uparrow$ |
> | ------------------------------- | --------------------- | --------------------- | --------------------- | ---------------------- | ---------------------- |
> | Equivariant Nerual Network      | 0.2899                | 0.4026                | 0.4390                | 0.4749                 | 0.4879                 |
> | Invariant Nerual Network (ours) | **0.3246**            | **0.4343**            | **0.4715**            | **0.5061**             | **0.5181**             |
>
> [1] "Equivariant message passing for the prediction of tensorial properties and molecular spectra." International Conference on Machine Learning. PMLR, 2021.

---

> > ### Comment · Reviewer_gd4P · 2024-12-03
> > **Thanks for all the updates**
> >
> > Thanks for the comment.  This particular comparison with equivariant 3D networks reminded me that the 3Dreact paper had also found that using invariant networks was sufficient.  According to their publication the model performed better than RXNmapper in isolation, so perhaps it is useful in your method as well.  https://pubs.acs.org/doi/10.1021/acs.jcim.4c00104
> >
> > I think that the work is slightly improved after all the changes.  I'm still not very clear on the overall practical or methodological importance of this work after reading it again and reading all the QA with the multiple reviewers, but I've updated my recommendation by one click and no longer object to its publication.

---

> > > ### Author Response · Authors · 2024-12-04
> > > **Part Ⅵ:  The Influence of Atom Mapping Accuracy**
> > >
> > > Thank you for your comments. We are glad to have addressed your concerns regarding equivariant and invariant networks.
> > > Further explanations regarding **atom mapping** and **methodological importance** are as follows. Additionally, we also add a comparative experiment for 3DReact [1].
> > >
> > >
> > > ## The Influence of Atom Mapping Accuracy
> > > Thank you for your insightful suggestion. We summarize the applicability of various mapping methods, and design experiments to take a closer look at the impact of atom mapping accuracy.
> > > ### Method
> > > Four methods are used for atom mapping between reactants and products in 3DReact [1].
> > > + **(1) Transition State.** Using transition states to obtain atom mapping. This method is not applicable when the dataset lacks transition state information.
> > > + **(2) Heuristic Search.** Using heuristic methods to search for accurate atom mapping. But it is not applicable to complex reaction data, as mentioned in Sec.2.2 in 3DReact:
> > >
> > >         Accurate atom-maps are not available for all reaction data sets.
> > > + **(3) RXNMapper.** There exist datasets that are both complex and lack transition state information. Hence, 3DReact uses RXNMapper to dynamically predict atom mapping.
> > > + **(4) Cross Attention.** 3DReact also introduces cross attention to substitute for atom mapping. It performs cross attention between atoms of same type in reactants and products. The paper reports that this approach performs worse than methods without atom mapping. Therefore, it is not discussed further.
> > >
> > > Our method uses large-scale real-world datasets. These datasets do not contain transition state information. Meanwhile, they include many complex reactions that cannot be handled by heuristic rules. In this case, RXNMapper is an appropriate choice.
> > >
> > > ### Experiment
> > > + **Design.** It is currently unable to use accurate mapping in our dataset. To evaluate the impact of atom mapping accuracy on model performance, we adopt a reasonable alternative approach.
> > >
> > >     Specifically, we reduce the mapping accuracy by adding noise to RXNMapper's result. This creates reaction datasets with different mapping accuracies.
> > >
> > > + **Implementation.** We first define an error rate $ e $ to represent the proportion of mapping errors. Then, we select $ e\% $ of the reactions from the dataset. For each reaction, we randomly choose a noise level $ n $ between $0-100\%$. Finally, we randomly swap $ n\% $ of the mapping numbers of the same type of atoms. We select error rates $ e $ of $0\%$, $1\%$, $2.5\%$, $5\%$, $10\%$, and $25\%$. We use a 1/8 subset of USPTO-Condition for the experiment.
> > >
> > > + **Result.** According to the results in Table 1, mapping accuracy does have impact on model's performance. When the error rate is low, the decrease in model's performance are manageable. The accuracy of RXNMapper is sufficient, so occasional errors do not significantly impact model performance.
> > >
> > >
> > >     **Table 1: Influence of mapping accuracy on the model's performance. The higher the error rate, the lower the mapping accuracy.**
> > >     | Error Rate |   Top-1↑ |   Top-3↑ |   Top-5↑ |   Top-10↑ |   Top-15↑ |
> > >     |------------|---------|---------|---------|----------|----------|
> > >     | 0.00      | **0.1773**  | **0.2749**  | **0.3146**  | **0.3534**   | **0.3670**   |
> > >     | 0.01       | 0.1766  | 0.2765  | 0.3115  | 0.3519   | 0.3648   |
> > >     | 0.025      | 0.1747  | 0.2723  | 0.3098  | 0.3485   | 0.3629   |
> > >     | 0.05       | 0.1720  | 0.2691  | 0.3048  | 0.3473   | 0.3622   |
> > >     | 0.1        | 0.1693  | 0.2678  | 0.3067  | 0.3465   | 0.3607   |
> > >     | 0.25       | 0.1584  | 0.2631  | 0.3009  | 0.3464   | 0.3602   |
> > >
> > >
> > >
> > > [1] 3DReact: Geometric Deep Learning for Chemical Reactions

---

> ### Author Response · Authors · 2024-11-27
>
> Dear Reviewer gd4P,
>
> Thank you very much for your contributions to improving this paper.
> We have tried our best to address the concerns and problems mentioned.
> If you feel our efforts address your concerns, we would appreciate it if you could kindly consider raising your rating.
> Your support is significantly important to our paper.
> Feel free to let us know if there is anything unclear.
> We are still here and happy to provide additional information.
> Thank you once again.
>
> Best wishes,
>
> Paper 4277 Authors

---

> ### Author Response · Authors · 2024-12-04
> **Part Ⅶ: Comparison with 3DReact and Methodological Importance**
>
> ## Comparison with 3DReact
>
> ### Experiment
>
> We compare 3DReact [1] with our method using the official open-source code in https://github.com/lcmd-epfl/EquiReact. We test the performance of both methods on 1/8 subset of USPTO-Condition, and the results are as follows:
>
> **Table 2:** Comparison between the performance of 3DReact and Reaction Graph methods.
> | Method                 | Top-1↑  | Top-3↑  | Top-5↑  | Top-10↑ | Top-15↑ |
> |------------------------|--------|--------|--------|--------|--------|
> | 3DReact [1]                | 0.1217 | 0.2050 | 0.2412 | 0.2813 | 0.2955 |
> | Reaction Graph (ours)  | **0.1773** | **0.2749** | **0.3146** | **0.3534** | **0.3670** |
>
> ### Analysis
>
> We observe that the Reaction Graph outperforms 3DReact in the reaction property prediction task. We believe the reason is:
>
>
>
> + **Chemical Bond Information.** Reaction Graph contains chemical bond information, while 3DReact lacks prior of chemical bonds.
> + **Neural Network Architecture.** 3DReact adopts TFN [2]. It is suitable for quantum chemistry tasks, but not for predicting reaction properties such as reaction condition.
> + **Utilization of Mapping Information.** Reaction Graph utilize mapping during MPNN feature extraction, while 3DReact does not. Therefore, Reaction Graph enables a deeper integration of molecular structure and reaction change information.
>
> [1] 3DReact: Geometric Deep Learning for Chemical Reactions
>
> [2] Tensor field networks: Rotation-and translation-equivariant neural networks for 3d point clouds
>
> ## Explanation of Overall Practical and Methodological Importance
>
> **(1) Practical Significance**
> The proposed method effectively enhancing performance in tasks such as reaction classification, condition prediction, and yield prediction.
>
> Our **condition prediction advancement** enable efficient reaction condition identification, reducing costly and environmentally harmful high-throughput experiments. This accelerates **new drug development** and promotes **green chemistry**. (Ordinary methods take up to 8 to 12 years and \$2.8 billion to develop a new drug [1]. The proposed method can significantly accelerate the drug discovery process and reduce economic costs.)
>
> Our **contribution in yield prediction** can be used in retrosynthetic evaluating and reaction condition searching. This enables the exploration of high-yield synthetic routes, thereby improving **material utilization** and reducing **costs** in chemical production.
>
> Our **high-precision reaction classification** method provides fundamental tool for reaction data analysis in subsequent research. It can be used for **dataset synthesis** and **mechanism understanding**. They are key to the development of methods like chemical reaction LLMs.
>
> **(2) Methodological Importance**
> **First graph to model atom-level interactions for chemical reaction understanding.** When using deep neural networks for chemical reaction modeling, existing methods lack a mechanism to account for the interactions between atoms during a reaction. Typically, these methods use molecule-level global representations of reactants and products for chemical reaction reasoning, which fails to effectively capture the underlying reaction mechanisms. In contrast, our Reaction Graph is equipped with the ability to model relationships at the atom level, thus enabling effective reasoning of reactions.
>
> **First graph with 3D chemical reaction modeling.** Existing graph-based methods for representing chemical reaction only consider 2D molecular structure, while overlooking 3D structural information. In constrast, our Reaction Graph is capable of modeling 3D structural transformation during chemical reaction, which better captures the dynamics and changes occurring throughout the reaction.
>
> **Data-and-Knowledge-Driven Chemical Reaction Reasoning.** Existing methods for understanding chemical reactions typically employ either rule-based or data-driven approaches. In our work, we successfully integrate chemical knowledge and rules into deep learning to enhance reaction reasoning.
>
> [1] Computer-Aided Retrosynthesis for Greener and Optimal Total Synthesis of a Helicase-Primase Inhibitor Active Pharmaceutical Ingredient
>
> We sincerely appreciate your time and effort in reviewing our submission.
> Thank you once again.

---

### Official Review · Reviewer_MRqU · 2024-11-05

**Soundness:** 2
**Presentation:** 3
**Contribution:** 3
**Rating:** 6
**Confidence:** 3

**Summary:**

The paper proposes a new method for modelling chemical reactions. In particular the paper suggests to incorporate the 3D information of the graph.

**Strengths:**

- The paper is clearly written
- The technical contribution is solid

**Weaknesses:**

The main weakness is the baseline chosen for comparisons.

In the related work you mention [1], but you never compare against it and looking at the results from the paper they seem like they still outperform the method you propose? In fact none of the baselines in [1] are re-used as far as I can tell (why?).

The paper needs a better limitation section.

[1] https://jcheminf.biomedcentral.com/articles/10.1186/s13321-024-00815-2

**Questions:**

Could the others expand on how their method differs from molecular simulation for reaction modelling? It seems that by incorporating the 3D spatial information we already quite close to molecular simulation conceptually.

What is the computation cost of your method, in particular in comparison to the other methods?

---

> ### Author Response · Authors · 2024-11-24
> **Part Ⅰ: Additional Baseline Comparisons and Limitations**
>
> We thank you for acknowledging our work **clearly written** and **technical contribution is solid**. We also thank you for your constructive comments.
>
> ## 1. Additional Baseline Comparisons
>
> In the initial supplementary materials, we compare our method with ReaMVP [1] and all the baselines in [1], more details could be found in Sec.G.3.2.
>
> We did not include its result in the main text because there is a significant difference in the training data scale between ReaMVP and our approach.
> Specifically, ReaMVP was trained on a commercial yield database with 600k samples in a supervised learning manner. Our approach used only 4k samples, which is the same as all other baseline methods.
>
> For completeness, we add results for ReaMVP trained on the same dataset as all other methods. We also consider additional methods such as DFT. The results are as follows (Table 1).
> Based on the results, Reaction Graph achieved the highest performance in yield prediction, demonstrating its effectiveness.
>
> **Table 1:** Yield prediction accuracy (R2↑) for models trained on the same benchmark dataset. Comparison is conducted between the average accuracies of multiple runs.
> | **Models** | **B-H** | **B-H-1** | **B-H-2** | **B-H-3** | **B-H-4** | **S-M** |
> |-|-|-|-|-|-|-|
> | DFT | 0.92 | 0.80 | 0.77 | 0.64 | 0.54 | - |
> | Onehot | 0.89 | 0.69 | 0.67 | 0.49 | 0.49 | - |
> | MFF | 0.93 | 0.85 | 0.71 | 0.64 | 0.18 | - |
> | T5Chem | 0.97 | 0.81 | 0.91 | 0.79 | 0.63 | 0.86 |
> | YieldBERT | 0.95±0.01 | **0.84±0.01** | 0.84±0.03 | 0.75±0.04 | 0.49±0.05 | 0.82±0.01 |
> | YieldBERT-DA | **0.97±0.01** | 0.81±0.05 | 0.87±0.02 | 0.59±0.07 | 0.16±0.03 | 0.86±0.01 |
> | DRFP | 0.95±0.01 | 0.81±0.01 | 0.83±0.00 | 0.71±0.01 | 0.49±0.00 | 0.85±0.01 |
> | Egret | 0.94±0.01 | **0.84±0.01** | **0.88±0.03** | 0.65±0.06 | 0.54±0.06 | 0.85±0.01 |
> | UA-GNN | **0.97±0.01** | 0.74±0.04 | **0.88±0.03** | 0.72±0.02 | 0.50±0.03 | **0.89±0.01** |
> | D-MPNN | 0.94±0.01 | 0.80±0.03 | 0.82±0.03 | 0.73±0.02 | 0.55±0.03 | 0.85±0.01 |
> | Rxn Hypergraph | 0.96±0.01 | 0.81±0.01 | 0.83±0.02 | 0.71±0.02 | 0.56±0.02 | 0.85±0.01 |
> | ReaMVP | 0.92±0.01 | 0.76±0.02 | 0.83±0.05 | 0.70±0.05 | 0.53±0.08 | 0.85±0.01 |
> | Reaction Graph (Ours) | **0.97±0.01** | 0.80±0.01 | **0.88±0.02** | **0.76±0.02** | **0.68±0.06** | **0.89±0.01** |
>
> [1] https://jcheminf.biomedcentral.com/articles/10.1186/s13321-024-00815-2
>
>
> ## 2. Limitations
>
> Thank you for your suggestion. Based on the issues you mentioned, we have revised the limitation section as follows:
>
> ```
> Firstly, like most data-driven methods, the quality of data significantly impacts the performance of our method. Specifically, inaccuracies in the 3D coordinates of atoms can lead to inferior results. Thus, developing accurate and efficient methods for 3D prediction could further enhance our approach.
> Secondly, integrating scientific knowledge and rules into deep learning is crucial for AI applications in science. In RG, we incorporate atom-level reaction relationships into the graph, thereby enhancing our understanding of reactions. However, this is not sufficient. Further exploration of methods to inject chemical laws into deep neural networks could significantly enhance the effectiveness of the proposed method. These potentials can be further investigated in the future.
> ```
>
> The text above has been added in Sec.5.

---

> ### Author Response · Authors · 2024-11-24
> **Part Ⅱ: Differences from molecular simulations**
>
> Thank you for the question. We summarize the main differences between our method and molecular simulation for reaction modeling as follows:
>
> - **Different Output.**
>    The results obtained from molecular dynamics simulations are usually energy and kinetic parameters. These results require further manual analysis or machine learning processing to obtain the desired yield or condition information. In contrast, our method directly outputs specific yields and reaction conditions.
>
> - **Different Algorithm Category.**
>    From an algorithmic perspective, our reaction property prediction method is a Deterministic Algorithm, while molecular simulation for reaction modeling is an Iterative Algorithm.
>    A Deterministic Algorithm follows a fixed sequence of steps to produce results.
>    In contrast, Iterative Algorithms require multiple iterations to approach a result until convergence. Their runtime is uncertain and may fail.
>
> - **Different Information Dependencies.**
>    Molecular simulation is grounded in pre-established rules and knowledge, while our method is data-driven, learning to uncover hidden insights without explicitly defining rules. These rules often struggle to model complex relationships, such as between 3D molecular structure and reaction conditions. In contrast, data-driven methods can learn these relations from data.
>
> - **Different Computation Cost.**
>    Molecular simulations are typically time-intensive for each reaction calculation because they require iterati until convergence is achieved. In contrast, once our model is trained properly, it can perform inferences relatively rapidly.

---

> ### Author Response · Authors · 2024-11-24
> **Part Ⅲ: Computation Cost of Reaction Graph in Comparison to Other Models**
>
> ## Our method
> 1. **Theoretical Analysis**
>
>     We define all graph-based representations of chemical reactions as $ G(V, E) $, where $ V $ represents the set of atoms and $ E $ represents the set of bonds. The computational complexity of the Reaction Graph depends on the number of atoms $ |V| $, the number of bonds $ |E| $, the maximum degree $ D $ of the atoms, and the specific implementation of the graph.
>
>     In Reaction Graph, a reactant atom can match with at most one product atom, so the number of reaction edges does not exceed half the total number of atoms, $ |V|/2 $. Furthermore, as defined for angular edges in Fig. 1, the maximum number of angular edges is $ |V| D^2 $.
>
>     Thus, the computational complexity contributed by bond edges is $ |E| $, by reaction edges is approximately $ |V|/2 $, and by angular edges is around $ |V| D^2 $. Consequently, the total computational complexity of Reaction Graph can be approximated as $ |E| + |V|/2 + |V| D^2 $.
>
> 2. **Experimental Results**
>
>     We assess the impact of reaction edges and angular edges in the Reaction Graph on model inference speed as the number of atoms $ |V| $ increases. The findings are presented in rows 1-3 of Table 2. The time cost associated with reaction edges is lower than that caused by angular edges.
>
> ## Compared with Other Methods
>
> 1. **Theoretical Analysis**
>
>       + Graph-based methods
>
>         When the molecular graph is represented as $G(V,E)$, $\vert V \vert$ indicates the total number of atoms in reactants and products, and $\vert E \vert$ represents the total count of bonds. The computation cost for the molecular graph is $\vert E \vert$. The computation cost of Rxn Hypergraph is at least $\vert E \vert + \vert V \vert$. The edge count in Condensed Graph of Reactions (CGR) is roughly $\vert E \vert /2$. However, when handling the same amount of information, the dimensions of the fully connected layers in CGR far exceed those in the Reaction Graph and Rxn Hypergraph. This leads to greater computational overhead.
>
>       + SMILES-based methods
>
>         SMILES-based methods are generally based on Transformer. When the length of the input SMILES is $\vert L \vert$, the computation complexity is roughly $ \vert L \vert ^2 $. Since there is no direct conversion between the length of SMILES strings $\vert L \vert$ and the number of nodes $\vert V \vert$ and edges $\vert E \vert$ in molecular graphs, we directly measured their computational efficiency through experiments.
>
> 2. **Experimental Results**
>
>       + Compared with graph-based methods
>
>         We compare the inference time of Reaction Graph and other graph-based methods. The results are shown in rows 4-6 of Table 2. Due to incorporating additional reaction information and 3D structures, our method requires slightly more time than molecular graphs and Rxn Hypergraph representations. However, it delivers significant performance improvement, justifying the additional time cost.
>
>
>         **Table2:** Model inference time for different graph representations under varying numbers of graph nodes. Time units are milliseconds (ms).
>
>         | **Representation**           | **Reaction Information** | **3D Structure** | 2500  | 3000  | 3500  | 4000  | 4500  |
>         | ---------------------------- | ------------------------ | ---------------- | ----- | ----- | ----- | ----- | ----- |
>         | Reaction Graph               | √                        | √                | 37.28 | 43.77 | 50.57 | 58.03 | 65.45 |
>         | w/o Reaction Edge            | ×                        | √                | 35.36 | 40.92 | 48.68 | 53.30 | 58.79 |
>         | w/o Angular Edge             | √                        | ×                | 22.51 | 25.96 | 29.82 | 34.21 | 38.77 |
>         | Molecular Graph              | ×                        | ×                | 18.59 | 21.93 | 25.39 | 28.30 | 31.48 |
>         | Rxn Hypergraph               | ×                        | ×                | 31.18 | 37.28 | 43.31 | 48.53 | 54.06 |
>         | Condensed Graph of Reactions | √                        | ×                | 44.55 | 52.48 | 60.75 | 66.32 | 75.34 |
>
>       + Compared with SMILES-based methods
>
>         We compare the inference time of Reaction Graph and SMILES-based methods. The results are shown in the table below. We find that the Reaction Graph, as a graph-based model, exhibits the shortest inference time.
>
>         **Table3:** Inference time comparison for Reaction Graph and other methods.
>
>         | **Methods**           | **Representation** | **Time**     |
>         | --------------------- | ------------------ | ------------ |
>         | T5Chem                | SMILES             | 3min 19s     |
>         | RXNFP                 | SMILES             | 23min 43s    |
>         | Reaction Graph (ours) | Reaction Graph     | **2min 36s** |
>
> The details of the cost analysis can be found in Sec.E.

---

> ### Author Response · Authors · 2024-11-27
>
> Dear Reviewer MRqU,
>
> Thank you for your efforts in reviewing and improving our paper.
> We have tried our best to address the concerns and problems mentioned.
> We hope that our responses have addressed your concerns.
> We would appreciate it if you could kindly consider raising your rating.
> Your support is significantly important to our paper.
> If you need further information, please let us know.
> Thank you once again.
>
> Best regards,
>
> Paper 4277 Authors

---

> > ### Comment · Reviewer_MRqU · 2024-12-02
> >
> > I have raised my score, because I do not have serious concerns for the paper to be accepted, but I do not want to champion the paper for that I am not sufficiently convinced.

---

> > > ### Author Response · Authors · 2024-12-03
> > >
> > > Dear Reviewer MRqU,
> > >
> > > Thank you very much for your time and effort in reviewing our submission.
> > > We appreciate your decision to raise the score and acknowledge the paper's potential for acceptance.
> > > As an early exploration of chemical reaction modeling, our paper still has room for improvement.
> > > We will continue refining and enhancing the proposed method.
> > > If you would like to discuss further, we are happy to provide more information.
> > > Thank you once again.
> > >
> > > Best regards,
> > >
> > > Paper 4277 Authors

---

### Author Response · Authors · 2024-11-26
**Summary of Revision**

Dear Chairs and Reviewers,

We would like to thank the reviewers for their careful and constructive comments. We also thank the reviewers for acknowledging our work is **novel** (xu23, uB2j, L6NH), **solid** (MRqU, dqUg), **effective** (N9iv, dqUg, xQ99), **substantial** (gd4P) and **significant** (xu23). In accordance with the reviewers’ comments and suggestions, **we have conducted approximately 100 new experiments** and revised our paper. Updates and changes are marked by blue color in the revised version. The major changes in this revision lie in:
+ Section 4: Figure revision and more metrics
+ Appendix C: Implementation details of supplementary experiments
+ Appendix E.1: Computational complexity analysis with Big O
+ Appendix E.2: Additional baselines for computational efficiency comparison
+ Appendix G.1: Detailed analysis for attention weight visualization
    + Edge cases
    + Statistical validation on dataset scale
    + Relation between DFT results and attention weight
+ Appendix G.2: Detailed analysis for leaving group identification
    + Edge cases
    + Additional baselines
+ Appendix G.3: Additional experiments on reaction tasks
    + Influence of scaffold size to 3D information utilization
    + Influence of merging dataset on model generalization
    + Dimensionality reduction of reaction representation vector
+ Appendix G.4: Comparative experiment for all network modules
    + Influence of RBF kernel
    + Influence of vertex-edge integration method
    + Influence of attention and LSTM on aggregation
    + Attemptation of using Set Transformer aggregation
+ Appendix G.5: Experiments on potential revisions of graph representations
    + Exploring different methods to utilize 3D information in reaction modeling (This is the first attempt to explore using 3D information for reaction modeling.)
    + Influence of the accuracy of 3D Information
    + Influence of angular edge in message passing perspective
    + Revise Rxn Hypergraph to model reaction
    + Attemptation of adding 3D to existing methods
+ Appendix G.6: Statistical analysis
    + Standard deviations for all metrics
    + Statistical significance tests on all tasks
    + Cross-validation results on all tasks
    + Confidence intervals for all metrics
+ Appendix H: Schematics for Reaction Graph
+ Appendix I: Toolkit analysis and comparison
+ Appendix J: Adding terminology list

Should you need further information, please let us know. We look forward to hearing from you soon.

Yours sincerely,

Authors of Paper “Reaction Graph: Toward Modeling Chemical Reactions with 3D Molecular Structures”

---

### Public Comment · ~Hehe_Fan1 · 2025-02-14
**Thanks & Summary**

# Thanks

We sincerely appreciate the AC and all the reviewers for their valuable comments. We are a bit regretful that our submission was rejected, especially considering the scores of **(8, 8, 6, 6, 6, 6, 5, 3)**. Unfortunately, the two reviewers who provided scores of 5 and 3 did not offer feedback during the rebuttal period.

We are grateful to reviewers such as **Xu23**, **uB2j**, and **L6NH** for recognizing the novelty of our approach.

# Summary

We would like to emphasize two key merits of our work:

## 1. A Unified Reaction Graph Representation for Reactants and Products

Existing methods typically extract separate representations for reactants and products, which fail to capture the reaction-level structure. In contrast, our reaction graph models the entire reaction process by capturing atomic relationships, providing a holistic view. We are pleased that some reviewers highlighted the importance of this unified graph representation:

	Reviewer uB2j: Innovative unified graph representation combining reactants and products.
	Reviewer L6NH: The authors propose a novel unified graph representation that includes a reaction edge.

## 2. A Simple yet Effective Method for 3D Reaction Modeling

We introduced a straightforward angular edge method to model the 3D structure of molecules. Note that, we did **not** claim that this method **fundamentally** provides **new** information beyond existing techniques. **It still focuses on 3D modeling.** However, extensive experiments demonstrate that it outperforms current 3D methods relying on angles. Simplicity and reliability are crucial for deep neural networks, and we believe our angular edge method offers a promising direction for 3D structure modeling.

## Code
We would like to highlight again that we submitted our code with the original submission. It can easily be reproduced.

# Online Platform
We  now offer an online platform **[Molecule Factory](http://47.99.54.221/)** for chemical reaction analysis, which we invite you to explore.

**Once again, we thank all of you for your time and feedback!**

---

### Meta-Review · Area_Chair_s9AZ · 2024-12-18

**Metareview:**

**Summary**:
This work introduces a graph representation 'Reaction Graph' (RG) -  to model chemical reactions. Specifically, the molecular graphs pertaining to the reactants and products are accompanied by atom-mapping information across the reaction via the so-called 'reaction edges'. Furthermore, to account for the 3D structural information, the 'angular edges' are proposed to incorporate features pertaining to bond angles for each molecular graph (together with maintaining the distance information).  Experimental evaluation on three tasks - reaction classification, reaction condition prediction, and yield prediction - was conducted  in support of the proposed approach across several datasets.

**Strengths**:
Reviewers appreciated several strengths of this work, notably,  clarity of presentation, clear motivation, and empirical performance on several datasets

**Weaknesses**:
Some reviewers expressed reservations about the novelty of this work.  In particular, they pointed out that having a unified graph representation for reactants and products is rather incremental given the existing body of work on hypergraph-based representations as well as graph representations that model molecular simulations (e.g., dynamics). Moreover,  3D structural information has already been extensively modeled in prior works on molecular modeling. Questions were also raised about whether the angular edges fundamentally provide any information beyond what can be already captured by some existing prior methods (at the expense of some additional rounds of message passing).

Reviewers also raised concerns about (a) the statistical validity of results (missing standard deviations etc.), (b) issues pertaining to technical clarity (e.g., lack of information about how the molecular conformations were selected for evaluation and how different molecular conformations impacted the results; or whether torsional angles were considered etc.), (c) reproducibility, and (d) ad-hoc design of some components (e.g., using LSTM for attention-based aggregation)

**Recommendation**:
I commend the authors for their detailed response, especially, the effort they invested into providing additional experimental results during the rebuttal period and for all the additional clarifications. While many of the reviewers’ concerns were addressed, those pertaining to novelty of the representation and approach remained largely unresolved.

To make sure that the paper receives fair treatment, I proceeded to carefully reviewing the paper myself. Unfortunately, it turns out that, despite its merits, the authors’ claim that this is the first work for modeling chemical reactions that captures 3D interactions between reactants and products is not correct. Besides, the benefit of incorporating angular edges needs to be ascertained.

For example, [1] infers stereochemistry/chirality (cis/trans conformations, tetrahederal centers) of the molecules from the 3D configurations, transferring this information from the product side to the reactant side. [1] also computes atom mappings between products and reactants, using these mappings to align the two sides of the reaction.  Thus, the claim for novelty in terms of modeling 3D interactions for chemical reactions needs to be revised/revisited. The paper would greatly benefit from appropriately acknowledging, and providing comprehensive empirical comparisons with, such approaches.

I would also point out that interactions between (3D) molecular graphs (with invariances/equivariances) have also been considered extensively in literature; see, e.g., [2], [3], and [4]. I do not see how the approach proposed by the authors here provides any modeling or representational advantages compared to such approaches.  A detailed empirical investigation with at least some of these approaches is also needed.

Finally, works such as [5], [6], and [7] do account for bond distances, angles, etc.. Again, I do not see how the proposed angular edges in this work provide any fundamental representational benefits compared to these methods.

Given that the work in its current form does not seem to propose anything substantially novel compared to the existing works, and absent any theoretical justifications, I do not think it meets the bar for ICLR.

[1] Laabid et al. Applying diffusion models for retrosynthesis, ICML Workshops 2024.

[2] Ganea et al. Independent SE(3) equivariant models for end-to-end rigid protein docking. ICLR 2022.

[3] Stark et al. Equibind: Geometric deep learning for drug binding structure prediction. ICML 2022.

[4] Verma et al. AbODE: Ab initio antibody design using conjoined ODEs. ICML 2023.

[5] Gasteiger et al. GemNet: Universal Directional Graph Neural Networks for Molecules. NeurIPS 2021.

[6] Gasteiger et al. Directional message passing for molecular graphs. ICLR 2020.

[7] Liu et al. Speherical Message Passing for 3D Graph Networks. ICLR 2022.

**Additional Comments On Reviewer Discussion:**

Please see the details above that already cover all the relevant aspects.

---

### Decision · Program_Chairs · 2025-01-22

Reject